# Zero-shot Outlier Detection via Synthetically Pre-trained Transformers: Model Selection Bygone!

## Abstract

Outlier detection (OD) has a vast literature as it finds numerous applications in environmental monitoring, security, manufacturing, and finance to name a few. Being an inherently *unsupervised* task, model selection is a key bottleneck for OD (both algorithm and hyperparameter selection) without label supervision. There is a long list of techniques to choose from – both classical algorithms and deep neural architectures – and while several studies report their hyperparameter sensitivity, the literature remains quite slim on unsupervised model selection—limiting the effective use of OD in practice. In this paper we present FoMo-0D, for zero/0-shot OD exploring a transformative new direction that *bypasses* the hurdle of model selection altogether (!), thus breaking new ground. The fundamental idea behind FoMo-0D is the Prior-data Fitted Networks, recently introduced by Müller et al. (2022), which trains a Transformer model on a large body of *synthetically* generated data from a prior data distribution. In essence, FoMo-0D is a **pretrained Foundation Model for zero/0-shot OD on tabular data**, which can directly predict the (outlier/inlier) label of any test data at inference time, by merely a *single forward pass*—making obsolete the need for choosing an algorithm/architecture and tuning its associated hyperparameters, besides requiring no training of model parameters when given a new OD dataset. Extensive experiments on **57** public benchmark datasets against **26** baseline methods show that FoMo-0D performs statistically no different from the $2nd$ top baseline, while significantly outperforming the majority of the baselines, with an average inference time of **7.7 ms** per test sample.

## 1 Introduction

Outlier detection (OD) finds applications in many domains such as security, environmental monitoring, finance, and so on. This popularity brings along a large literature that offers a plethora of detection algorithms to choose from given a new OD task. These techniques, however, exhibit several hyperparameters (HPs) that need careful tuning to which they are often quite sensitive (Ma et al., 2023). What makes it notoriously difficult to achieve effective OD performance in practice is *model selection* (both algorithm and HPs) in the *absence of any labels*, as most tasks are unsupervised.[1]

In fact, while deep learning and modern architectures have revolutionized many areas of machine learning (ML), it has not quite been the case for OD—mainly because deep OD models (Pang et al., 2021) exhibit many more HPs (for architecture, regularization, and optimization) that detection performance is sensitive to (Ding et al., 2022), as compared to classical methods with only a few HPs.

Large foundation models have stirred up most recent advances in ML, which are (pre-)trained on massive amounts of data. The most notable progress has been in natural languages and vision, thanks to the admirable quantity and quality of public text and image datasets. In contrast, public (benchmark) datasets for OD is minuscule in comparison (Han et al., 2022; Zhao et al., 2021; Steinbuss and Böhm, 2021). Another obstacle for foundation models for tabular OD has been the non-shared feature spaces of different datasets, unlike the shared pixel or word spaces for images and text.

Recently, the introduction of Prior-data Fitted Networks (PFNs) has marked a milestone as a new approach to ML on tabular data (Müller et al., 2022). PFNs are based on Bayesian non-parametrics

---

[1]While semi-/supervised settings of OD exist, unsupervised OD is preferable in most domains for the capacity to detect novel/emergent types of anomalies, beyond just the known types.

Table 1: Comparison of methods across datasets. (top row) Rank w.r.t. AUROC performance avg.'ed over 57 datasets is presented for FoMo-0D (with $D = 100$), **top-10 baselines** with default HPs, and **top-4**[5] baselines with performance **avg.**'ed over varying HPs (denoted w/ $^{avg}$); followed by $p$-values of the pairwise Wilcoxon signed rank test, comparing FoMo-0D to each baseline (from top to bottom) over All (57) datasets, those (42) w/ $d \leq 100$ and (46) w/ $d \leq 500$ dimensions. FoMo-0D performs as well as (**i.e., statistically no different from) the 2nd best model** ($k$NN, w/ $p = 0.106$) across All datasets, while it is **comparable to** ($p > 0.05$) or **better than** ($p > 0.95$) **all baselines** over datasets w/ $d \leq 100$ (aligned w/ pretraining where $D = 100$) *and* $d \leq 500$ (generalizing beyond pretraining).

| | FoMo-0D | DTE-NP | $k$NN | ICL | DTE-C | LOF | CBLOF | Feat.Bag. | SLAD | DDPM | OCSVM | DTE-NP$^{avg}$ | $k$NN$^{avg}$ | ICL$^{avg}$ | DTE-C$^{avg}$ |
|---|---|---|---|---|---|---|---|---|---|---|---|---|---|---|---|
| Rank(avg) | 11.886 | 7.553 | 9.018 | 10.851 | 11.36 | 12.316 | 13.342 | 13.386 | 12.982 | 14.061 | 13.851 | 9.079 | 11.105 | 12.991 | 22.263 |
| All | - | 0.016 | 0.106 | 0.462 | 0.454 | 0.585 | 0.750 | 0.823 | 0.759 | 0.901 | 0.895 | 0.112 | 0.315 | 0.670 | 1.000 |
| $d \leq 100$ | - | 0.415 | 0.700 | 0.949 | **0.953** | **0.970** | **0.971** | **0.996** | 0.876 | **0.980** | **0.978** | 0.752 | 0.860 | **0.958** | **1.000** |
| $d \leq 500$ | - | 0.220 | 0.569 | 0.827 | 0.894 | **0.960** | 0.968 | **0.994** | 0.910 | **0.960** | **0.979** | 0.607 | 0.756 | 0.846 | **1.000** |

and meta-learning on large quantities of *synthetically* simulated data from a data prior. The key idea is to compute a posterior predictive distribution (PPD) for a test point given the training data as input context. To approximate the PPD, a Transformer (Vaswani et al., 2017) is pre-trained to mimic the PPD via simulating numerous training datasets from a (general, complex) data prior. For inference, the fresh training set along with the test samples are passed to the (frozen) pre-trained PFN, which outputs the predictions in a *single forward pass*, requiring no model training or model selection. Variants of PFN are shown to match the performance of tree-based models on small classification datasets (Hollmann et al., 2023) and in time series forecasting with limited data (Dooley et al., 2023).

In this paper, we capitalize on these ideas and introduce FoMo-0D; a prior-data fitted Foundation Model for zero- or 0-shot Outlier Detection (for the "Fear of Missing out"-liers). The implication and "gift" of PFNs for unsupervised OD goes beyond those for supervised learning: it helps *bypass* not only model (parameter) training, but most importantly, the notoriously-hard task of model (hyperparameter) selection altogether. As such, FoMo-0D unlocks zero-shot OD on a new dataset without the need for any algorithm or HP selection. During inference, data is used only as input *context* to FoMo-0D, and *not* for parameter training or HP tuning. Arguably, this is a potential game changer for unsupervised OD, especially for practitioners. Figure 1 illustrates the new FoMo-0D paradigm versus the typical OD setting.

In designing FoMo-0D, we simply use Gaussian mixture models as a simple yet effective tabular data prior, to capture general and diverse inlier data distributions, following current literature (Hollmann et al., 2023; Zhao et al., 2021). We combine these with simulated outlier types common in the real-world; namely local and global subspace outliers (Steinbuss and Böhm, 2021). While the data prior can be extended to comprise more complex data distributions (e.g. through the use of Bayesian Neural Networks (BNNs; (Neal, 2012)) and Structural Causal Models (SCMs; (Pearl, 2009)) as in (Hollmann et al., 2023)), and additional outlier types can be included (e.g. dependency, contextual, etc. outliers), as we show in the experiments, even with the relatively straightforward prior that we employed, FoMo-0D achieves remarkable performance. As shown in Table 1, FoMo-0D pretrained on synthetic datasets with up to 100 dimensions performs *statistically no different* from all 26 state-of-the-art baselines (all $p$-values $> 0.2$) on 46 benchmark datasets with dimensionality $d \leq 500$, while *significantly outperforming the majority* of the baselines (with $p > 0.95$) (see Appendix Tables 12.1 & 12.2). Further, FoMo-0D takes a mere average of 7.7 ms to infer a test sample since a new dataset requires a single forward pass for inference and no training overhead.

**Our contributions:** We summarize the main contributions of our work as follows.

- **A Foundation Model for Tabular OD:** We present FoMo-0D, *the first foundation model for zero-shot OD* on tabular datasets. FoMo-0D is a Prior-data Fitted Network (PFN) (Müller et al., 2022) that is pretrained on many synthetically generated datasets drawn from a novel data prior that we introduce to capture various inlier and outlier distributions. The pretrained FoMo-0D can then directly compute the posterior predictive distribution (PPD) of test points in a new dataset.

- **Unsupervised Outlier Model Selection Made Obsolete:** The most outstanding property of FoMo-0D is its *zero-shot inference* on a new dataset via a single forward pass, fully abolishing the need not only for model training on a new dataset, but importantly also the notorious task of algorithm selection and hyperparameter tuning in the absence of labeled data.

- **Scalable Pre-training Design:** To unlock the premise of large-scale pretraining on numerous large datasets, (1) we implement a new mechanism to speed up sample-to-sample attention from

quadratic to *linear time* complexity—enabling *larger datasets*; and (2) we scale up on-the-fly data synthesis through data transformation—enabling *more datasets* in less time.

- **Fast Inference at Detection Time:** Thanks to a pretrained prior-data fitted Transformer, FoMo-0D bypasses both model (parameter) training and selection, both of which can be slow for modern deep OD models with many hyper/parameters. Rather, it takes *fraction of a second* to label a test point through a single forward pass that can be parallelized across test samples. Such speedy inference also unlocks the potential for deploying FoMo-0D in *real time* on data streams.

- **Effectiveness:** We evaluate FoMo-0D on **57** public benchmark datasets (Han et al., 2022) from diverse domains and compare against **26** baselines from classical to modern (Livernoche et al., 2024), where FoMo-0D significantly outperforms the majority of the baselines while performing statistically no different from the top $2nd$ baseline, at the fraction of the compute cost.

As FoMo-0D proposes a paradigm shift for OD, abolishing model training and selection altogether, while delivering unreasonable effectiveness on benchmark datasets even with a basic data prior, we expect FoMo-0D will trigger further work in both research and practice. To this end, we make all of our codebase for synthetic data generation, model training, and our pretrained FoMo-0D checkpoints, openly available at `https://anonymous.4open.science/r/PFN40D`.

## 2 PROBLEM AND PRELIMINARIES

### 2.1 SEMI-SUPERVISED OUTLIER DETECTION

Outlier detection (OD) methods can be categorized based on the availability of labeled data. In supervised OD, the task is similar to binary classification with imbalanced classes (as outliers typically make up only a small portion of the overall data). The more difficult unsupervised setting assumes the "contaminated" training data contains both inliers and outliers, but without any labels. A semi-supervised or one-class classification approach lies between these two extremes, where only inlier data is available for training, but unknown outliers may appear during inference. Semi-supervised OD is used in practice where it is easy to gather inlier data, but learning from known, labeled outliers is undesirable because outliers are hard to collect and/or new, unknown outlier types are likely to arise in future test data that renders learning only from the known outliers suboptimal/risky.

Note that semi-supervised OD may be a *misnomer* from the supervised ML perspective, where semi-supervised classification assumes the presence of some labeled instances from **all** classes in the training data. As such, model selection continues to be as difficult for semi-supervised OD as unsupervised OD, where no labeled outliers exist in the input/training data in both settings.

We focus on semi-supervised OD. Formally, let $\mathcal{D}_{\text{in}} = \{(\mathbf{x}_1, y_1) \ldots, (\mathbf{x}_n, y_n)\}$ denote the input data containing only inliers $\mathbf{x}_i \in \mathbb{R}^d$, where $y_i = 0 \ \forall i \in [n]$, and $\mathcal{D}_{\text{test}}$ depicts the test data comprising both inliers and outliers. The task is to assign labels to $\mathbf{x}_i \in \mathcal{D}_{\text{test}}$ given the inlier-only input $\mathcal{D}_{\text{in}}$.

### 2.2 BACKGROUND ON PRIOR-DATA FITTED NETWORKS

**Posterior Predictive Distribution (PPD):** In the Bayesian framework for supervised learning, the prior defines a hypotheses space $\Phi$ which expresses our beliefs about the data distribution before seeing any data. Each hypothesis $\phi \in \Phi$ describes a mechanism by which the data is generated. The posterior predictive distribution $p(\cdot|\mathbf{x}_{\text{test}}, \mathcal{D}_{\text{train}})$ provides a framework for making prediction on new, unseen test data $\mathbf{x}_{\text{test}}$, conditioned on observed training data $\mathcal{D}_{\text{train}} = \{(\mathbf{x}_1, y_1), \ldots, (\mathbf{x}_n, y_n)\}$. Based on Bayes' Theorem, the PPD can be derived by the integration over the space of hypotheses $\Phi$:

$$p(y_{\text{test}}|\mathbf{x}_{\text{test}}, \mathcal{D}_{\text{train}}) = \int_{\Phi} p(y_{\text{test}}|\mathbf{x}_{\text{test}}, \phi)p(\mathcal{D}_{\text{train}}|\phi)p(\phi)d\phi, \tag{1}$$

where $p(\phi)$ denotes the prior probability and $p(\mathcal{D}|\phi)$ is the likelihood of the data $\mathcal{D}$ given $\phi$.

**PFNs and PPD Approximation:** As obtaining the above PPD is generally intractable, Prior-data Fitted Networks (PFNs) are proposed to approximate the PPD (Müller et al., 2022). Unlike traditional machine learning models that are trained directly on observed datasets, PFNs are pre-trained offline on simulated datasets that are generated according to a prior distribution. Specifically, it contains the pre-training and inference stages described as the following.

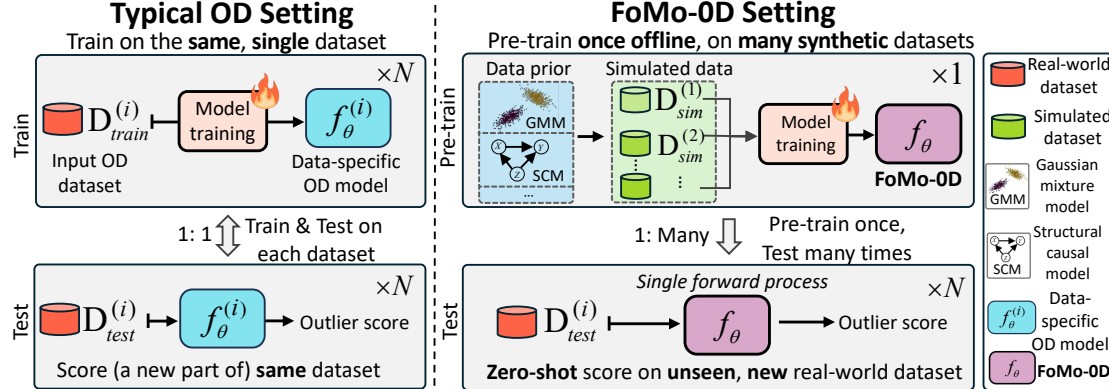

Figure 1: (best in color) Comparison of typical OD vs. the FoMo-0D settings. Given a new un/semi-supervised OD dataset, FoMo-0D not only eliminates the need for model training, but most importantly, also abolishes the onerous task of model selection (algorithm and hyperparameters) w/out labels.

*Pre-training on synthetic data.* At the beginning of the pre-training stage, massive synthetic training datasets are generated, by first sampling a hypothesis (i.e., the generating mechanism) $\phi \sim p(\phi)$, and then sampling a dataset $\mathcal{D} \sim p(\mathcal{D}|\phi)$. For training purposes, each dataset $\mathcal{D}$ can be split as $\mathcal{D}_{\text{test}} \subset \mathcal{D}$ and $\mathcal{D}_{\text{train}} = \mathcal{D} \setminus \mathcal{D}_{\text{test}}$. Thus the PFN with parameters $\theta$ can be optimized by making predictions on data points in $D_{\text{test}}$. For a test point $(\mathbf{x}_{\text{test}}, y_{\text{test}}) \in \mathcal{D}_{\text{test}}$, the training loss is formulated as

$$\mathcal{L} = \mathbb{E}_{(\{(\mathbf{x}_{\text{test}}, y_{\text{test}})\} \cup \mathcal{D}_{\text{train}}) \sim p(\mathcal{D})}[-\log q_\theta(y_{\text{test}}|\mathbf{x}_{\text{test}}, \mathcal{D}_{\text{train}})] \ . \tag{2}$$

The above loss can also be interpreted as minimizing the expected KL divergence between $p(\cdot|\mathbf{x}, \mathcal{D})$ and $q_\theta(\cdot|\mathbf{x}, \mathcal{D})$ (Müller et al., 2022). In practice, a PFN model $q_\theta$ is typically implemented by a Transformer-based architecture (Vaswani et al., 2017), which takes $(\mathbf{x}_{\text{test}}, \mathcal{D}_{\text{train}})$ as input, where $\mathbf{x}_{\text{test}} \in \mathcal{D}_{\text{test}}$ and $\mathcal{D}_{\text{train}}$ contains an arbitrary number of instances. The output is the conditional class probabilities for $\mathbf{x}_{\text{test}}$. As the whole training set $\mathcal{D}_{\text{train}}$ is passed as input/context to the Transformer, it learns to predict class labels through sample-to-sample attention.

*Inference on real-world data.* In the inference stage, a fresh real-word dataset $\mathcal{D}_{\text{train}}$ and some test instance $\mathbf{x}_{\text{test}}$ are fed into the (frozen) pre-trained model, which computes the PPD $q_\theta(\cdot|\mathbf{x}_{\text{test}}, \mathcal{D}_{\text{train}})$ in a single forward process. Importantly, PFNs do not require gradient-based parameter tuning on data observed at inference time, where the training and prediction are delivered through a one-step forward process *in less than a second* (Hollmann et al., 2023).

In summary, PFNs are trained once offline, and can be used many times for zero-shot inference when new datasets with different characteristics are input. The main benefit is that **no training or tuning** is required at the inference stage. This type of learning ability is also termed as in-context learning (ICL) (Xie et al., 2021), which was shown to be an effective paradigm for various tasks in NLP with the stream of large language models (Brown et al., 2020). In fact, ICL with PFNs is recently shown to be a promising paradigm for supervised classification on tabular datasets (Hollmann et al., 2023).

## 3    FoMo-0D: A New PFN for 0-shot OD – Model Selection Bygone!

Inspired by the recent PFNs (Müller et al., 2022) and their successful applications in supervised classification (Hollmann et al., 2023) and time series forecasting (Dooley et al., 2023), we propose FoMo-0D, a prior-data fitted Foundation Model for 0-shot Outlier Detection. FoMo-0D is (pre)trained on a large body of synthetically generated OD datasets toward zero-shot inference on a new dataset. Most notable of our zero-shot FoMo-0D is its elimination of the need not only for model training on a new dataset, but especially also for model selection (both algorithm and HPs), which is notoriously-hard without any labeled data. By breaking such new ground, and its effectiveness on many benchmark datasets compared to classical and modern baselines, we expect FoMo-0D will become a milestone in future research and practice of OD. The new FoMo-0D paradigm (right) versus the typical OD setting (left) is illustrated in Figure 1.

In the following we describe our OD data prior, training of FoMo-0D on prior-simulated datasets, inference on new datasets, and our specific model architecture and improvements for scalable training.

## 3.1 Designing a Data Prior for Outlier Detection

Arguably, what has triggered the recent breakthroughs in NLP and CV is the massive amounts of datasets available for (pre)training, along with high-capacity model architectures. In comparison to the natural language and image domains, the quantity (and quality) of publicly available tabular OD datasets is minuscule. Even in the presence of large quantities of data, in training their Chronos foundation models for time series forecasting, Ansari et al. (2024) show that using synthetic data in combination with real-world data improves the overall zero-shot performance. For these reasons, we design a new data prior from which we simulate numerous OD datasets for pretraining FoMo-0D.

Ideally the data prior should reflect distributions as general and diverse as seen in real-world datasets, however, "*finding a prior supporting a large enough subset of possible [data generating] functions isn't trivial*" (Nagler, 2023). Surprisingly, in contrast, our initial attempt has been sufficient to achieve remarkable performance even with a relatively straightforward and simple-to-implement data prior, which we describe next.

**Inlier synthesis:** We simulate inliers by simply drawing from a Gaussian Mixture Model (GMM) with $m$-clusters in $d$-dimensions, with centers $\boldsymbol{\mu}_{jk} \in [-5, 5]$, $j \in [m]$, $k \in [d]$ and *diagonal*[2] $\boldsymbol{\Sigma}_j$ with entries in $[-5, 5]$. In each step of every epoch during pretraining, we create batch size $B$ different GMMs with varying $m \le M$ and $d \le D$ chosen uniformly at random from $[M]$ and $[D]$, respectively. From each GMM, we draw a set of $S$ inliers, defined as instances within the 90%-ile of the GMM.

**Outlier synthesis:** Following the previous literature on outlier synthesis (Han et al., 2022), we generate *subspace* outliers by first drawing a subset of dimensions $\mathcal{K}$ at random, where $|\mathcal{K}| \le d$, and then generate $S$ points from the corresponding "inflated" GMMs, which share the same centers $\boldsymbol{\mu}_j$'s with the original GMM but with the inflated (diagonal) covariances $5 \times \boldsymbol{\Sigma}_{j,kk}$'s for $k \in \mathcal{K}$. Outliers are defined as points outside the 90%-ile of the original GMM. We label each sample based on its Mahalanobis distance computed analytically (see Property B.2 in the Appendix).

Specifically, we simulate datasets containing $2S = 10,000$ samples (half inlier, half outlier) from the two corresponding GMMs (original and inflated) with up to $M = 5$ clusters and up to $D = 100$ dimensions. Example 2-$d$ synthetic datasets are illustrated in Appendix A.

**Remarks:** We emphasize once again that our model is not trained on **any** real-world data and rather, on purely synthetic data (although future work can combine existing benchmark OD datasets with synthesized data, as was done for Chronos (Ansari et al., 2024)). Notably, our GMM-based data prior can be seen as extremely basic. While it has been our intent to extend our preliminary attempt toward designing a sophisticated data prior for OD, we found to our surprise that even with such an elementary prior, FoMo-0D performs remarkably well against numerous SOTA baselines. Therefore, we present FoMo-0D using this effortless approach for its simplicity to showcase the prowess of PFNs for OD. Future work can employ BNNs and SCMs (Hollmann et al., 2023), and other outlier types (contextual, dependency, etc. (Steinbuss and Böhm, 2021)) toward a more comprehensive data prior.

## 3.2 (Pre)Training and Inference

**Model (Pre)Training (Once, Offline):** FoMo-0D is a Prior-data Fitted Network (PFN, see Section 2.2) based on the Transformer architecture. In the synthetic prior-data fitting phase, it is trained on datasets drawn from our OD data prior for tabular data that we introduced in Section 3.1. Each dataset is simulated from a different GMM configuration based on randomly drawn parameters, and consists of varying number of training samples and dimensions to capture the diversity in real-world tabular datasets. Detailed steps are outlined in Algo. 1 in Appendix C.2, and described as follows.

Each time, we first draw a hypothesis (i.e. GMM configuration) uniformly at random, that is $\phi = \{d \in [D], m \in [M], \{\boldsymbol{\mu}_j\}_{j=1}^m \in [-5, 5]^d, \{\boldsymbol{\Sigma}_j\}_{j=1}^m; diag(\boldsymbol{\Sigma}_j) \in [-5, 5]^d\}$, and then generate a dataset $\mathcal{D} = \{\mathcal{D}_{\text{in}}, \mathcal{D}_{\text{out}}\}$ containing synthetic inlier and outlier samples from the drawn hypothesis and its variance-inflated variant, respectively.

We optimize FoMo-0D's parameters $\theta$ to make predictions on $\mathcal{D}_{\text{test}} = \{\mathcal{D}_{\text{test}}^{\text{in}}, \mathcal{D}_{\text{test}}^{\text{out}}\}$, conditioned on the inlier-only training data $\mathcal{D}_{\text{train}} \subset \mathcal{D}_{\text{in}}$ based on the cross-entropy loss (see Eq. (2)). During

---

[2]In our early experiments, we found no difference in terms of test performance on synthetic datasets between using diagonal and non-diagonal $\boldsymbol{\Sigma}$, however, it is easier to compute the inverse of diagonal $\boldsymbol{\Sigma}$ for generation.

training, $\mathcal{D}_{\text{test}}$ contains a *balanced* number of inlier and outlier samples, where $\mathcal{D}_{\text{test}}^{\text{in}} = \mathcal{D}_{\text{in}} \backslash \mathcal{D}_{\text{train}}$, and $\mathcal{D}_{\text{test}}^{\text{out}} \subset \mathcal{D}_{\text{out}}$ contains an equal number of samples as $\mathcal{D}_{\text{test}}^{\text{in}}$. To vary the training data size, we subsample $\mathcal{D}_{\text{train}}$ of randomly drawn size $n \in [n_L, n_U]$, where $n_L$ and $n_U$ denote the lower and upper bounds. In our current implementation, we set $n_L = 500$, and $n_U = 5,000$.

FoMo-0D is trained on $200,000$ batches ($200$ epochs $\times\ 1,000$ steps/epoch) of $B = 8$ generated datasets in each batch. While this pretraining phase can be expensive, it is done *only once, offline*. Moreover, we introduce several scalability improvements to speed up pretraining, as discussed later in Section 3.3. Full details on the training and implementation of FoMo-0D are given in Appendix C.

**Zero-shot Inference (on Unseen Dataset):** During the inference phase, our pretrained-in-advance FoMo-0D can be employed on any unseen real-world dataset. In fact, we apply the same single pretrained network on all benchmark datasets in our experiments in this paper.

Specifically, for a new semi-supervised OD task with inlier-only training data $\mathcal{D}_{\text{train}}$ and mixed test data $\mathcal{D}_{\text{test}}$, feeding $\langle \mathcal{D}_{\text{train}}, \mathbf{x}_{\text{test}} \rangle$ as input to FoMo-0D (for each $\mathbf{x}_{\text{test}} \in \mathcal{D}_{\text{test}}$ separately) yields the PPD $q_\theta(y|\mathbf{x}_{\text{test}}, \mathcal{D}_{\text{train}})$ in a *single forward pass*. As such, FoMo-0D performs model "training" and prediction *simultaneously* at test time. In fact, as the entire training data is passed as context, FoMo-0D leverages in-context learning (ICL) (Xie et al., 2021; Garg et al., 2022) for inference. The **key** contribution of FoMo-0D goes beyond eliminating gradient-based model training for a new dataset: because no model training is required, one thus neither needs to choose any specific OD model to train, nor grapple with tuning any hyperparameters of the said model—rendering model selection an obsolete concern for the future of OD. Additionally, the speedy, easily parallelizable inference (for *less-than-a-second* per test sample) is then the "icing on the cake".

For a visual summary, Figure 1 (right) illustrates (top) pretrain & (bottom) test phases of FoMo-0D.

### 3.3 ARCHITECTURE AND SCALABILITY

**Architecture and sample-to-sample attention:** Like existing PFNs in the literature, FoMo-0D is based on the Transformer architecture (Vaswani et al., 2017), encoding each sample's feature vector as a token, and allowing token representations to attend to each other, hence enabling *sample-to-sample attention*. We also adopt the three adaptations of TabPFN (Hollmann et al., 2023), which (1) computes self-attention among all the training samples but only *cross*-attention from test samples to the training samples, (2) enables variable feature dimensionality by zero-padding, and (3) randomly rotates input samples while omitting positional encodings to achieve model invariance to sample permutations in the dataset. We defer the architecture details to the original papers.

Given $\mathcal{D}_{\text{train}} = \{\mathbf{x}_1, \ldots, \mathbf{x}_n\}$, each self-attention layer outputs $n$ embeddings $\{\mathbf{z}_i\}_{i=1}^n$; where the $i$-th token is mapped via linear transformations to a key $\mathbf{k}_i$, query $\mathbf{q}_i$ and value $\mathbf{v}_i$ based on which the $i$-th output is computed by weighing all $\mathbf{v}_j$'s by the normalized dot product between $\mathbf{q}_i$ and all the $\mathbf{k}_j$'s (i.e. sample-to-sample dot product similarity) as

$$\mathbf{z}_i = \sum_{j=1}^n \mathtt{softmax}\big(\ \{\langle \mathbf{q}_i, \mathbf{k}_{j\prime} \rangle\}_{j\prime=1}^n\ \big)_j\ \cdot\ \mathbf{v}_j\ . \tag{3}$$

The sample-to-sample attention is intriguing from the perspective of OD: Many classical OD algorithms (Aggarwal, 2013) are based on nonparametrics; in particular, they make use of the distances to the $k$ *nearest* neighbors ($k$NNs) of a point to compute its outlierness (where $k$ is a critical hyperparameter (HP)). One can think of FoMo-0D as mimicking non-parametric models but by using parametric attention mechanisms. Interestingly, PFNs are much more robust and flexible than $k$NN based OD approaches, for (1) sample-to-sample relations are not pre-specified but rather learned through attention weights, and thus (2) they are not limited to just the nearest neighbors but rather can *learn which* training points are worth attending to, and last but not least (3) as attention is dataset-wide across all points, there is no need for specifying a cut-off HP value like $k$, to which most $k$NN based OD techniques are sensitive to (Aggarwal and Sathe, 2015; Campos et al., 2016; Goldstein and Uchida, 2016; Ding et al., 2022)—to reiterate, algorithm & HP selection is bygone with FoMo-0D.

While intuitively beneficial for OD, "vanilla" attention among the training samples incurs quadratic complexity. To be able to seize the benefits with scale, we incorporate a scalable architecture to our design, as we describe next. The scale up also unlocks a larger context (i.e. dataset) size for FoMo-0D, enabling its pretraining on larger datasets for potentially better generalization.

**Scaling up attention with "routers":** The $\mathcal{O}(n^2)$ quadratic sample complexity at pretraining presents an obstacle for achieving high performance at inference. From dataset size perspective, it limits pretraining to relatively small training datasets. From context size perspective, it limits in-context learning that typically benefits from longer context lengths (Xie et al., 2021).

Toward a high-performance pretrained model, we scale up FoMo-0D's attention via the "router mechanism" of Zhang and Yan (2023). As shown in Figure 2, the main idea is to learn a small number ($R \ll n$) of "routers" or representatives, which gather information from all $n$ samples and then distribute the information back to the $n$ output embeddings, creating what-looks-like a "bottleneck" attention mechanism—reducing complexity from $\mathcal{O}(n^2)$ to $\mathcal{O}(2Rn) = \mathcal{O}(n)$. This design allows FoMo-0D training to **scale linearly** with respect to both dimensionality $d$ and also dataset size $n$.

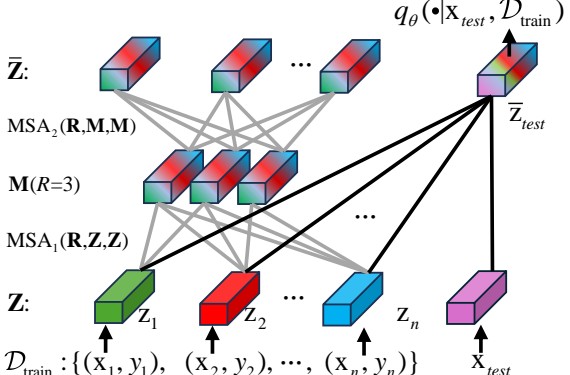

Concretely, the representatives first aggregate information from all samples by serving as query in multi-head self-attention (MSA) and the embedding array of all samples becomes both key and value:

$$\mathbf{M} = \text{MSA}_1(\mathbf{R}, \mathbf{Z}, \mathbf{Z}) , \qquad (4)$$

where $\mathbf{R} \in \mathbb{R}^{R \times d}$ depicts the *learnable* vector array of representatives and $\mathbf{M}$ denotes the aggregated messages. Then, the routers distribute the received information among samples by using the sample embeddings as query and the aggregated messages as both key and value:

$$\hat{\boldsymbol{Z}} = \text{MSA}_2(\mathbf{Z}, \mathbf{M}, \mathbf{M}) . \qquad (5)$$

Figure 2: FoMo-0D architecture employs the "router mechanism" for scalable attention.

Finally, we obtain $\bar{\boldsymbol{Z}} = \text{LayerNorm}(\hat{\boldsymbol{Z}} + \mathbf{Z})$ after layer normalization. Note that the test samples only attend to the training samples' embeddings, computed in the described manner across layers, which finally feed into the prediction head for estimating each test sample's PPD at the output layer.

**Scaling up (pre)training data synthesis with linear transforms:** Besides the scalability challenge associated with architecture/attention, another computational challenge in pretraining FoMo-0D arises from drawing samples from the data prior. That is, generating samples from a pre-specified data distribution requires considerable time, especially in high dimensions[3], provided the large number of datasets we sample (concretely, a batch size of 8 datasets over 1,000 steps each for 200 epochs).

To give an idea, sampling a dataset with $n = 10,000$ points in $d = 100$ dimensions using 10 CPUs in parallel takes $\approx 0.4$ seconds (see Appendix Figure 7). Across 200 training epochs with 1,000 steps each, it adds up to more than 177 hours just to generate 1,6 million datasets on-the-fly. Of course, one can trade storage with compute-time by generating all these datasets apriori via massive parallelism. Nevertheless, synthetic data generation demands considerable time (and/or storage).

To scale up data synthesis, FoMo-0D employs two distinct strategies. **First**, we propose *reuse at epoch level*: that is, one can reuse the same 8K unique datasets at every epoch, or in general, the same 8K$\times P$ datasets periodically at every $P$ epochs. A larger $P$ would lead to more diversity in terms of the overall pretraining data used.

**Second**, and more innovatively, we propose *reuse at dataset level via transformation*: that is, having generated one unique dataset $\mathbf{X} \in \mathbb{R}^{n \times d}$ from a GMM, we propose a linear transform $T(\mathbf{x})$ of the form $\mathbf{W}\mathbf{x} + \mathbf{b}$ for randomly drawn parameters $\mathbf{W} \in \mathbb{R}^{d \times d}$ and $\mathbf{b} \in \mathbb{R}^d$ (see Appendix B.1).[4] This simple yet efficient transformation creates a new dataset, akin to one being drawn from another GMM with centers $T(\boldsymbol{\mu}_j) = \mathbf{W}\boldsymbol{\mu}_j + \mathbf{b}$ and covariance $T(\boldsymbol{\Sigma}_j) = \mathbf{W}\boldsymbol{\Sigma}_j\mathbf{W}^T, \forall j \in [m]$. Note that we do not actually materialize these parameters but only transform the dataset. As we show in the following, such transformations preserve the Mahalanobis distances as well as the percentile thresholds for labeling points as inlier/outlier. Details and proofs are given in Appendix B.

---

[3]This is because the inverse of the $(d \times d)$ covariance matrix plays a crucial role in the process of generating samples from a GMM, which has $\mathcal{O}(d^3)$ time complexity. (It is also partly the reason why we use diagonal $\boldsymbol{\Sigma}_j$'s in our data prior.) In addition, Mahalanobis distance for labeling inliers/outliers also requires the inverse.

[4]In practice, we apply the linear transform on the subspace of inflated features only, wherein inliers and outliers are defined, which remains to be a multi-variate GMM.

**Lemma 1** *Linear transform $T$ with invertible $\mathbf{W}$ on $\mathcal{G}_m^d$ preserves Mahalanobis distances.*

**Lemma 2** *Linear transform $T$ with invertible $\mathbf{W}$ on $\mathcal{G}_m^d$ preserves the percentiles of the GMM.*

The implication of these lemmas is that a linear transformation of a dataset from a GMM retains the identity of the inliers and outliers, i.e. no relabeling is required. Moreover, notice that as a byproduct we obtain a transformed dataset as though it is drawn from a GMM with a *non-diagonal* covariance matrix which, besides the time savings, offers a slightly more complex data prior.

To reach 8K unique datasets for each epoch, we first generate 500 datasets from different GMMs (with varying configurations), and then employ 15 different linear transformations to each unique dataset by varying $\mathbf{W}$ and $\mathbf{b}$. Drawing each $(\mathbf{W}, \mathbf{b})$ takes $\approx 0.02$ seconds, while the matrix-matrix product of $\mathbf{X}$ ($n \times d$) and $\mathbf{W}$ ($d \times d$) takes negligible time (for $d \leq 100$). Thus, obtaining a transformed dataset offers $20\times$ speed-up compared to generating one (0.02 vs. 0.4 seconds).

## 4 EXPERIMENTS

### 4.1 SETUP

We present the experiment setup briefly, including important notes on data synthesis, real-world datasets, baselines, metrics and HPs. For additional details, we refer to Appendix D.

**Pre-training Dataset Synthesis:** During pretraining, we generate unique GMM datasets by first drawing a configuration, including dimensionality $d \in [D]$, number of components $m \in [M]$, centers $\{\boldsymbol{\mu}_j\}_{j=1}^m$ (each $\boldsymbol{\mu}_j \in [-5, 5]^d$) and covariances $\{\boldsymbol{\Sigma}_j\}_{j=1}^m$ ($diag(\boldsymbol{\Sigma}_j) \in [-5, 5]^d$). We set $M = 5$ and vary $D \in \{20, 100\}$ to study pretraining with relatively small and high dimensional datasets, respectively. We synthesize inliers and outliers as described in Section 3.1.

**Real-world Benchmark Datasets:** While pretraining is purely on synthetic datasets, we evaluate FoMo-0D on **57** real-world datasets from the ADBench benchmark (Han et al., 2022) (see Table 15).

We use 5 train/test splits of each dataset via different seeds and report mean performance and standard deviation. Note that the baselines require model re-training and inference for each $\mathcal{D}_{\text{train}}/\mathcal{D}_{\text{test}}$ split, while FoMo-0D uses the splits only for inference as $\mathcal{D}_{\text{train}}$ is merely passed as context.

**Baselines:** We compare FoMo-0D against **26** baselines, from classical/shallow methods to modern/deep models. The baselines are imported from one of the latest papers that proposed the SOTA diffusion-based model DTE (Livernoche et al., 2024), and its three variants; DTE-C, DTE-IG, and DTE-NP. We defer to the original paper for additional details.

**Model Implementation:** We trained our final model for 200,000 steps with a batch size of 8 datasets. That is, our FoMo-0D is trained on 1,600,000 synthetically generated datasets. This training takes about 25 hours on 1 GPU (Nvidia RTX A6000). Each dataset had a fixed size of 10,000 samples, with $|\mathcal{D}_{\text{train}}| \in [n_L = 500, n_U = 5000]$, and the rest used as $\mathcal{D}_{\text{test}}$ with *balanced* number of inliers and outliers. Other implementation details of FoMo-0D, including the training algorithm, model architecture, data synthesis and reuse, and hardware are provided in Appendix C.

**Metrics and Hypothesis Testing:** Detection performance is w.r.t. 3 widely-used metrics for OD: AUROC; area under ROC curve, AUPR; area under Precision-Recall curve, and F1 score; using threshold at the true number of outliers in the test data (varies by dataset).

To compare methods, we compute their rank on each dataset (lower is better), and present average rank across datasets. This is an alternative to the average metric (e.g. AUROC), which is not meaningful when task difficulties and hence metric values vary widely. In addition, we perform significance tests to compare two methods statistically, using the one-sided paired Wilcoxon signed rank test (Demšar, 2006) between FoMo-0D and a baseline based on the performances across all datasets and report the $p$-values. We consider results to be significant at 0.05 following convention.

**Hyperparameters (HPs):** Importantly, Livernoche et al. (2024) picked for each baseline the best-performing set of HPs as recommended by the authors in their original paper. As for their own DTE, which behaves similar to $k$NN, they use $k = 5$ and set the *same $k$* for the $k$NN baseline (Ramaswamy et al., 2000) to be consistent. However, it is well known that $k$NN is sensitive to the value of $k$ (Aggarwal and Sathe, 2015), and so are many other OD models to their respective HPs (Campos et al., 2016; Goldstein and Uchida, 2016; Zhao et al., 2021; Ding et al., 2022).

Therefore, we compare to the performance results of these baselines as imported from DTE's Tables 13, 14 and 15, respectively for AUROC, F1, and AUPR (Livernoche et al., 2024). In addition, we also compare to the **top-4**[5] best performing baselines (in order: DTE-NP, $k$NN, ICL, and DTE-C) on their *average* performance across a list of different HP settings (which reflects their *expected* performance under HP values selected at random, in the absence of any other prior knowledge), which is the recommended approach by Goldstein and Uchida (2016) "*to get a fair evaluation when comparing [OD] algorithms*". We annotate the method name with [avg] for the version with performance averaged over varying HPs. The detailed list of HP values for each top baseline is given in Appendix D.4.

Overall, we compare FoMo-0D to 30 baselines; 26 from Livernoche et al. (2024) and [avg] of the top-4.

## 4.2 RESULTS

**Detection performance:** Table 1 presented the comparison of FoMo-0D w/ $D = 100$ to all baselines w.r.t. average rank across datasets as well as pairwise Wilcoxon signed rank tests based on AUROC (for full results on all datasets and all metrics, see Appendix G). We find that among 30 baselines and 2 variants of FoMo-0D (w/ $D = 100$ and $D = 20$), FoMo-0D w/ $D = 100$ *performs as well as the 2nd best model* ($k$NN with default HP; $k = 5$) on all datasets. While DTE-NP outperforms FoMo-0D with author-recommended $k = 5$, we find that DTE-NP[avg] is on par with FoMo-0D.

Against all other baselines, we obtain notably large $p$-values. Typically, $p > 0.05$ implies no statistical difference between two methods. On the other hand, the large $p$-values we obtain that are often larger than 0.50 suggest that the odds are tilted towards FoMo-0D to outperform.

FoMo-0D w/ $D = 100$ performs statistically no different from **all** baselines on datasets with $d \leq 100$ (i.e., "at its own game" when pretraining data dimensions align with real-world datasets), while it *outperforms the majority of baselines* ($p > 0.95$). These results also hold on datasets with $d \leq 500$.

Table 2 shows similar results for FoMo-0D w/ $D = 20$, which is pretrained on datasets with considerably fewer dimensions. Even in this limited setting, its performance is remarkable: against 30 baselines, it performs on par with the $3rd$ best baseline (ICL, with default HP). The $p$-value is even larger (0.437) when compared to ICL[avg]. Moreover, on datasets with $d \leq 20$ which align with its pretraining data, all $p$-values are larger than 0.5, where it outperforms the top $5th$ baseline and the majority of others. These are outstanding results for a model pretrained purely on synthetic datasets from a simple data prior in small dimensions, showcasing the prowess of PFNs for OD.

Table 2: Comparison of methods across datasets. (top row) Rank w.r.t. AUROC performance avg.'ed over 57 datasets is presented for FoMo-0D (with $D = 20$), **top-10** baselines with default HPs, and **top-4**[5] baselines with performance **avg**.'ed over varying HPs (denoted w/ [avg]); followed by $p$-values of the pairwise Wilcoxon signed rank test, comparing FoMo-0D to each baseline (from top to bottom) over All (57) datasets, those (24) w/ $d \leq 20$ and (38) datasets w/ $d \leq 50$ dimensions, respectively. Even with small $D = 20$, FoMo-0D performs as well as (i.e., statistically no different at 0.05 from) *the top $3rd$ baseline* (ICL, w/ $p = 0.089$) across All datasets, while it *outperforms the top $5th$ (LOF) and onward baselines* over datasets w/ $d \leq 20$ (aligned w/ pretraining where $D = 20$) *and* $d \leq 50$ (generalizing beyond pretraining). (setting: $D = 20$, $P = 50$, $R = 500$, train/inference context size=5K, no data transformation)

| | FoMo-0D | DTE-NP | $k$NN | ICL | DTE-C | LOF | CBLOF | Feat.Bag. | SLAD | DDPM | OCSVM | DTE-NP[avg] | $k$NN[avg] | ICL[avg] | DTE-C[avg] |
|---|---|---|---|---|---|---|---|---|---|---|---|---|---|---|---|
| Rank(avg) | 12.59 | 7.19 | 8.57 | 10.34 | 10.79 | 11.82 | 12.81 | 12.8 | 12.52 | 13.50 | 13.34 | 8.60 | 10.63 | 12.44 | 21.43 |
| All | - | 0.001 | 0.019 | 0.089 | 0.159 | 0.394 | 0.434 | 0.703 | 0.516 | 0.752 | 0.679 | 0.007 | 0.062 | 0.437 | 1.0 |
| $d \leq 20$ | - | 0.572 | 0.789 | 0.968 | 0.616 | **0.993** | **0.989** | **1.0** | **0.978** | 0.906 | **0.992** | 0.813 | 0.924 | **0.999** | **1.0** |
| $d \leq 50$ | - | 0.347 | 0.794 | 0.893 | 0.946 | **0.997** | **0.988** | **1.0** | **0.963** | **0.994** | **0.986** | 0.574 | 0.847 | **0.995** | **1.0** |

Figure 3 shows the distribution of ranks across datasets for each of the 32 methods. While paired significant tests are the most conclusive, FoMo-0D achieves relatively small average rank as well as notably low ranks across datasets that is also visually better than the majority of the baselines.

**Running time:** Table 3 presents the total training time and the average inference time per test sample, as measured on our largest benchmark dataset, for FoMo-0D and the top-3 baselines. Given

---

[5]To rank the baselines, we compute the $26 \times 26$ pairwise $p$-values based on the Wilcoxon signed rank test, as shown in Appendix Figure 16, and rank the baselines w.r.t. their mean $p$-value.

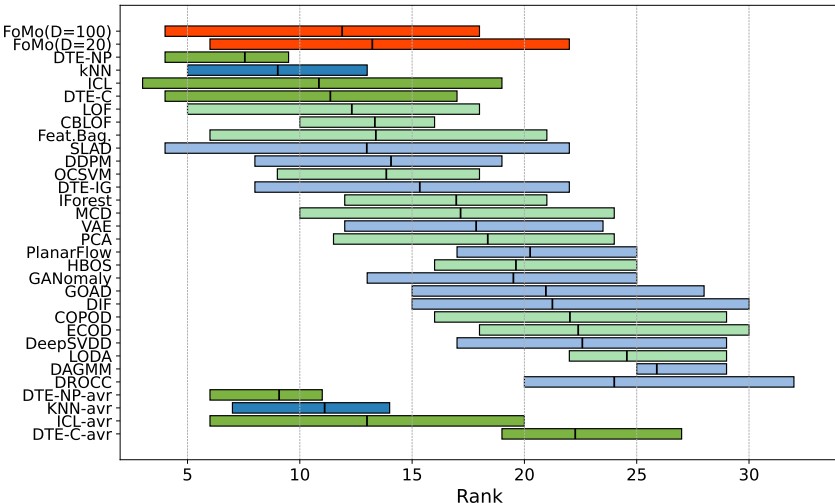

Figure 3: (best in color) Rank (w.r.t. AUROC performance, lower is better) distribution across datasets shown via boxplots for (from top to bottom) FoMo-0D in red, all 26 baselines as ordered by mean $p$-value[5] (shallow and deep baselines resp. in green and blue), and top-4 baselines' $^{avg}$ variants.

a new dataset, FoMo-0D bypasses model training (and HP tuning) and directly performs inference, with an average of 7.7 ms per sample (see Appendix Figure 6). In comparison, all baseline methods need to train on each individual dataset preceding inference. This training time can be high for deep learning based models like ICL, and further compounded with training *multiple* models for hyperparameter tuning purposes. Even for non-parametric and/or shallow models like $k$NN and DTE-NP (which queries $k$ nearest neighbors), the training involves various data pre-processing steps such as constructing a tree-like data structure for fast (often approximate) $k$NN distance querying.

Table 3: Training and inference time (in milliseconds) comparison between FoMo-0D and the top-3[5] baselines (w/ *default* HPs, *excluding* the time for model selection/hyperparameter optimization) on our largest dataset (namely, `donors`, see Appendix Table 15).

| Method | FoMo-0D | DTE-NP | $k$NN | ICL |
|---|---|---|---|---|
| Training time (total) | none | 56.83 | 1433.74 | 186461.48 |
| Inference time (per sample) | 7.7 | 0.76 | 0.17 | 0.01 |

### 4.3 ABLATION ANALYSES

Due to space limits, we present the detailed ablation analyses in Appendix E. We discuss the effect of $D$ in E.1, the cost and performance of varying $R$ in E.2 and E.3, the context size in E.4, the reuse periodicity $P$ in E.5, the effect of data transformation $T$ on performance and speed up in E.6 and E.7, data diversity and prolonged training in E.8, and quantile transformation on ADBench in E.9.

## 5 RELATED WORK

Due to space limits, we present the detailed related work in Appendix J.

## 6 CONCLUSION

We introduced FoMo-0D, **the first foundation model for outlier detection** (OD) on tabular data. It capitalizes on the in-context learning ability of a Transformer model pretrained on a large number of synthetic datasets that can then perform zero-shot inference on a new dataset by directly passing it as input context. FoMo-0D breaks new ground by fully abolishing notoriously-hard model selection. Further, FoMo-0D offers extremely fast inference thanks to a mere single forward pass. Against **26** baselines on **57** public datasets from diverse domains, FoMo-0D performs on par with the $2nd$ best baseline, while significantly outperforming the majority of baselines. We leave improving the data prior and extending beyond tabular OD, among others, as future directions. For a detailed discussion on limitations and future directions, we refer to Appendix K.

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

# A    ILLUSTRATION OF SYNTHETIC DATA IN $2$-$d$

We visualize our synthetic data in Figure 4, with 3 randomly created 2-$d$ GMMs with the number of clusters ($N = 1, 2, 3$). We choose the $80th$ percentile as the criterion, such that inliers are samples drawn from the GMM and within the $80th$ percentile, and outliers are samples drawn from the inflated GMMs and outside of the $80th$ percentile.

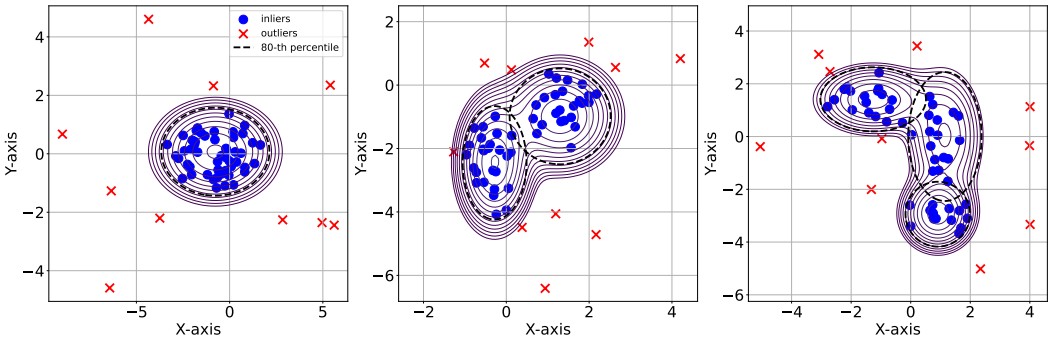

Figure 4: Illustration of synthetic data in 2D with $80th$ percentile as the criterion.

# B    LINEAR TRANSFORM FOR SCALABLE GMM DATA SYNTHESIS

## B.1    DEFINITIONS

**Definition 1 (Gaussian Mixture Model)** We denote an $m$-cluster $d$-dimension Gaussian Mixture Model as $\mathcal{G}_m^d = \{(w_j, \boldsymbol{\mu}_j, \boldsymbol{\Sigma}_j)\}_{j=1}^m$, which is the weighted sum of $m$ Gaussian distributions:

$$p(\mathbf{x}) = \sum_{j=1}^m w_i \cdot g(\mathbf{x}|\boldsymbol{\mu}_j, \boldsymbol{\Sigma}_j) \ , \tag{6}$$

where $w_j \in \mathbb{R}^+$ is the weight for the $j$-th Gaussian $\mathcal{N}(\boldsymbol{\mu}_j, \boldsymbol{\Sigma}_j)$ with $\sum_{j=1}^m w_j = 1$, and $g(\cdot|\boldsymbol{\mu}_j, \boldsymbol{\Sigma}_j)$ is the density of the $j$-th component/cluster, with mean/center $\boldsymbol{\mu}_j \in \mathbb{R}^d$ and covariance $\boldsymbol{\Sigma}_j \in \mathbb{R}^{d \times d}$ being positive semi-definite, such that $\mathbf{x}^T \boldsymbol{\Sigma}_i \mathbf{x} \geq 0$, for all $\mathbf{x} \in \mathbb{R}^d$.

**Definition 2 (Linear Transform)** We denote a linear transformation $T$ in $\mathbb{R}^d$ as:

$$T(\mathbf{x}) = \mathbf{W}\mathbf{x} + \mathbf{b} \ , \tag{7}$$

where $\mathbf{x} \in \mathbb{R}^d$, and $\mathbf{W} \in \mathbb{R}^{d \times d}, \mathbf{b} \in \mathbb{R}^d$ are the parameters of $T$.

**Definition 3 (Mahalanobis Distance)** The Mahalanobis distance $\text{dist}_M$ between a point $\mathbf{x} \in \mathbb{R}^d$ and a Gaussian distribution $\mathcal{N}(\boldsymbol{\mu}, \boldsymbol{\Sigma})$ is defined as:

$$\text{dist}_M(\mathbf{x}) = \sqrt{(\mathbf{x} - \boldsymbol{\mu})^T \boldsymbol{\Sigma}^{-1} (\mathbf{x} - \boldsymbol{\mu})} \ . \tag{8}$$

**Definition 4 ($\chi_d^2$-distribution)** The Chi-squared distribution $\chi_d^2$ with $d$ degrees of freedom is the distribution of the sum of squares of $d$ independent standard Normal random variables.

## B.2    PROPERTIES

**Property B.1 (Lemma 5.3.2 (Casella and Berger, 2024))** *If* $Z \sim \mathcal{N}(0,1)$, *then* $Z^2 \sim \chi_1^2$; *If* $X_1, ..., X_d$ *are independent and* $X_i \sim \chi_1^2$, *then* $\sum_{i=1}^d X_i \sim \chi_d^2$.

**Property B.2** *The squared Mahalanobis distance* $\text{dist}_M^2(\mathbf{x}) \sim \chi_d^2$, *with* $\mathbf{x} \sim \mathcal{N}(\boldsymbol{\mu}, \boldsymbol{\Sigma})$.

*Proof*: If $\mathbf{x} \sim \mathcal{N}(\boldsymbol{\mu}, \boldsymbol{\Sigma})$, then we have $\mathbf{z} = \boldsymbol{\Sigma}^{-\frac{1}{2}}(\mathbf{x} - \boldsymbol{\mu}) \sim \mathcal{N}(\mathbf{0}, \mathbf{I}_d)$ (Gut, 2009), such that:

$$\text{dist}_M^2(\mathbf{x}) = \mathbf{z}^T \mathbf{z} = \sum_{i=1}^{d} z_i^2 \tag{9}$$

where $z_i$ are independent standard Normal random variables. We have $\sum_{i=1}^{d} z_i^2 \sim \chi_d^2$ from Property B.1, which completes the proof.

### B.3 LEMMAS

**Lemma 1** *Linear transform $T$ with invertible $\mathbf{W}$ on $\mathcal{G}_m^d$ preserves Mahalanobis distances.*

*Proof*: We denote the transformed GMM as $T(\mathcal{G}_m^d) = \{(w_j, \mathbf{W}\boldsymbol{\mu}_j + \mathbf{b}, \mathbf{W}\boldsymbol{\Sigma}_j \mathbf{W}^T)\}_{j=1}^m$, then with $\mathbf{x} \sim \mathcal{N}(\boldsymbol{\mu}_j, \boldsymbol{\Sigma}_j)$, for the transformed point $T(\mathbf{x})$ we have:

$$\text{dist}_M(T(\mathbf{x})) = \sqrt{(T(\mathbf{x}) - (\mathbf{W}\boldsymbol{\mu}_j + \mathbf{b}))^T (\mathbf{W}\boldsymbol{\Sigma}\mathbf{W}^T)^{-1}(T(\mathbf{x}) - (\mathbf{W}\boldsymbol{\mu}_j + \mathbf{b}))} \tag{10}$$

$$= \sqrt{(\mathbf{W}(\mathbf{x} - \boldsymbol{\mu}_j))^T (\mathbf{W}\boldsymbol{\Sigma}\mathbf{W}^T)^{-1}(\mathbf{W}(\mathbf{x} - \boldsymbol{\mu}_j))} \tag{11}$$

$$= \sqrt{(\mathbf{x} - \boldsymbol{\mu}_j)^T \mathbf{W}^T (\mathbf{W}^T)^{-1} \boldsymbol{\Sigma}^{-1} \mathbf{W}^{-1} \mathbf{W}(\mathbf{x} - \boldsymbol{\mu}_j)} \tag{12}$$

$$= \sqrt{(\mathbf{x} - \boldsymbol{\mu}_j)^T \boldsymbol{\Sigma}^{-1}(\mathbf{x} - \boldsymbol{\mu}_j)} = \text{dist}_M(\mathbf{x}) . \tag{13}$$

$\blacksquare$

**Lemma 2** *Linear transform $T$ with invertible $\mathbf{W}$ on $\mathcal{G}_m^d$ preserves the percentiles of the GMM.*

*Proof*: Let $\chi_d^2(\alpha)$ denote the $\alpha$-th percentile of $\chi_d^2$, such that for $X \sim \chi_d^2$:

$$\text{Prob}(X \leq \chi_d^2(n)) = \frac{\alpha}{100} . \tag{14}$$

Based on Property B.2, we have $\text{Prob}(\text{dist}_M^2(\mathbf{x}) \leq \chi_d^2(\alpha)) = \frac{\alpha}{100}$.

Let $\mathbf{x} \sim \mathcal{G}_m^d$, such that $\text{dist}_M^2(\mathbf{x}) > \chi_d^2(\alpha)$ for all $\mathcal{N}_j(\boldsymbol{\mu}_j, \boldsymbol{\Sigma}_j)$, which indicates that $\mathbf{x}$ is outside the $\alpha$-th percentile of $\mathcal{G}_m^d$. Since $\text{dist}_M(\mathbf{x})$ is preserved under $T$ (see Lemma 1), then we conclude that the linear transform $T$ with invertible $\mathbf{W}$ preserves the percentiles of the GMM. $\blacksquare$

## C IMPLEMENTATION DETAILS

### C.1 HARDWARE

We base our experiments on a NVIDIA RTX A6000 GPU with AMD EPYC 7742 64-Core Processors.

### C.2 TRAINING AND INFERENCE

We train our models for 200 epochs with the Adam optimizer (Kingma and Ba, 2017) and a `learning_rate = 0.001`, and test with the model corresponding to the lowest training loss. The size of our $D = \{20, 100\}$ model is 4.87M and 4.89M parameters, respectively. We show the training process of PFNs and our model in Algorithm 1.

**Dealing with varying dimensions and dataset size** For an input with $d$ features, we follow Müller et al. (2022) and deal with $d < D$ by rescaling the input with $\frac{D}{d}$ and padding the features to size $D$ with 0, and randomly sample $D$ features out of $d$ if $d > D$. In addition, FoMo-0D uses context size of 5K at inference, where we randomly sample (5K$-1$) points as $\mathcal{D}_{\text{train}}$ from datasets with $n > $ 5K for each test sample $\mathbf{x} \in \mathcal{D}_{\text{test}}$.

**Model architecture** We use a 4-layer Transformer with hidden dimension `h_dim` $= 256$, a linear layer ($\mathbb{R}^D \rightarrow \mathbb{R}^{\text{h\_dim}}$) as the embedding layer and a 2-layer MLP ($\mathbb{R}^{\text{h\_dim}} \rightarrow \mathbb{R}^2$) as the classification layer for inlier vs. outlier. For each Transformer layer, we use `num_head` $= 4$ for each attention module and $R = 500$ for the router-based attention (Figure 2).

---

**Algorithm 1:** Prior-fitting of a PFN (Müller et al., 2022) and ours

---

**Input** : A prior distribution over datasets $p(\mathcal{D})$, from which samples can be drawn and the number of datasets $Q$ to draw for one epoch, the number of training epochs $E$, the periodicity $P$, the number of unique datasets $q$, linear transformation $T$.

**Output** : A model $q_\theta$ that will approximate the PPD

1 Initialize the neural network $q_\theta$;
2 Initialize the epoch-level collection $\mathcal{C}_E = [\ ]$;
3 **for** $i \leftarrow 1$ **to** $E$ **do**
4     **if** $i \leq P$ **then**
5        Initialize an empty buffer $\mathcal{B}_i = [\ ]$;
6        Initialize the dataset-level collection $\mathcal{C}_q = [\ ]$;
7        **for** $j \leftarrow 1$ **to** $Q$ **do**
8           **if** $j \leq q$ **then**
9              **Step 1**: sample $D_j := \mathcal{D}_{\text{train}} \cup \{(\mathbf{x}_k, y_k)\}_{i=k}^{|\mathcal{D}_{\text{test}}|} \sim p(\mathcal{D})$;
10              $\mathcal{C}_q \leftarrow \mathcal{C}_q + [D_j]$
11           **end**
12           **else**
13              $j \leftarrow j \mod q$
14              $D_j \leftarrow T(\mathcal{C}_q[j])$
15           **end**
16           **Step 2**: compute stochastic loss approximation $\bar{\ell}_\theta = \sum_{k=1}^{|\mathcal{D}_{\text{test}}|}(-\log q_\theta(y_k|\mathbf{x}_k, \mathcal{D}_{\text{train}}))$;
17           **Step 3**: update parameters $\theta$ with stochastic gradient descent on $\nabla_\theta \bar{\ell}_\theta$;
18           $\mathcal{B}_i \leftarrow \mathcal{B}_i + [D_j]$
19        **end**
20        $\mathcal{C}_E \leftarrow \mathcal{C}_E + [\mathcal{B}_i]$
21     **end**
22     **else**
23        $i \leftarrow i \mod P$
24        $\mathcal{B}_i \leftarrow \mathcal{C}_E[i]$
25        **for** $j \leftarrow 1$ **to** $Q$ **do**
26           $D_j \leftarrow T(\mathcal{B}_i[j])$
27           Perform **Step 2** and **Step 3**
28        **end**
29     **end**
30 **end**

---

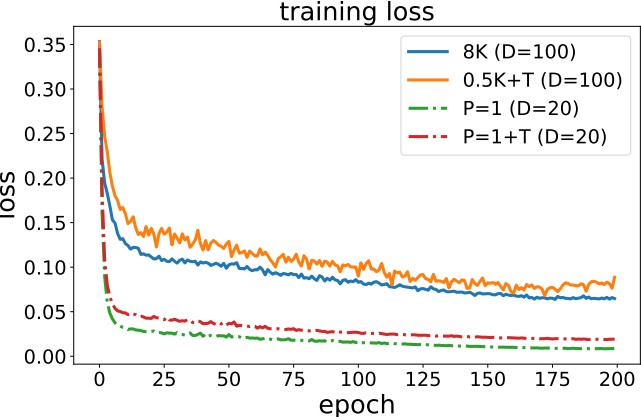

Figure 5: (best in color) Training loss of FoMo-0D ($D = 100$) with 8K unique datasets/epoch (in blue) and using 0.5K unique + 7.5K transformed datasets/epoch (in orange), and FoMo-0D ($D = 20$) with $P = 1$ (in green) and $P = 1$ with transformation (in red) over 200 epochs.

**Training loss** In Figure 5, we plot the training loss of our $D = 100$ model trained with 8K unique datasets/epoch (denoted as "8K") versus 0.5K unique + 7.5K transformed datasets/epoch (denoted as "0.5K+T"), together with the $D = 20$ model trained with reuse periodicity $P = 1$ (denoted as "P=1", reusing the same 8K datasets across epochs) and $P = 1$ with transformation (denoted as

"P=1+T", transforming the 8K datasets across epochs). Notice that the loss with transformation is slightly higher than no transformation (i.e., $D = 100$, "0.5K+T" vs. "8K", and $D = 20$, "P=1+T" vs. "P=1") across all 200 epochs, which is reasonable since the transformed datasets have non-diagonal covariances that make the learning task harder and thus result in a higher training loss. The training losses of FoMo-0D with $D = 100$ are also higher than with $D = 20$ since the subspace OD tasks are harder in higher dimensions.

**Inference time**   Figure 8 (left) showed the inference time of FoMo-0D on CPU, comparing typical attention versus the router-based attention (with $R = 500$ routers) under varying context sizes from 1K to 10K. The time is measured on CPU to clearly showcase the scalability trends; *quadratic* without routers and *linear* with routers.

Figure 6 shows the inference time on GPU. Notice that the time is much lower (in milliseconds), thanks to the Transformer architecture taking advantage of GPU parallelism, while the compute time for attention without routers continues to grow faster than that with routers.

In implementation, FoMo-0D (with $R = 500$ routers) uses inference context size of 5K by default, which takes about 7.7 ms per test sample on average.

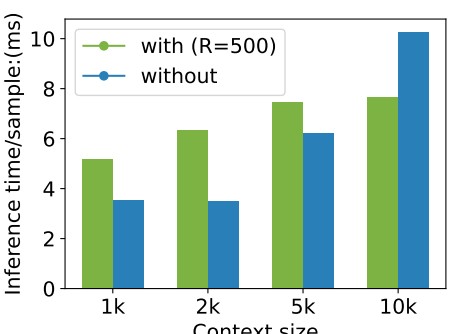

Figure 6: Inference time of FoMo-0D on *GPU* with vs. w/out router-based attention under varying context size.

## D   DETAILED EXPERIMENT SETUP

### D.1   PRE-TRAINING DATASET SYNTHESIS

During pretraining, we generate unique GMM datasets by first drawing a configuration, including dimensionality $d \in [D]$, number of components $m \in [M]$, centers $\{\boldsymbol{\mu}_j\}_{j=1}^m$ (each $\boldsymbol{\mu}_j \in [-5,5]^d$) and covariances $\{\boldsymbol{\Sigma}_j\}_{j=1}^m$ ($diag(\boldsymbol{\Sigma}_j) \in [-5,5]^d$). We set $M = 5$ and vary $D \in \{20, 100\}$ to study pretraining with relatively small and high dimensional datasets, respectively. We synthesize inliers and outliers as described in Section 3.1.

We then sample $S = 5,000$ points that are within the $90th$ percentile of the GMM. To synthesize outliers, we "inflate" a *subset* of dimensions by randomly choosing $|\mathcal{K}| \in [D]$ dimensions and multiplying the corresponding variances by $\times 5$ (following (Han et al., 2022)), i.e. $5 \times \boldsymbol{\Sigma}_{j,kk}$'s for $k \in \mathcal{K}$, and then draw $S = 5,000$ samples from the inflated GMM that are outside the $90th$ percentile of the original GMM.

To speed up data synthesis via linear transformations, we first draw 500 unique datasets using $m \in [5]$ and $d \in \{1, 2, \ldots, 100\}$ (i.e. $5 \times 100$) and transform each one $15 \times$ using varying parameters $(\mathbf{W}, \mathbf{b})$ as described in Section 3.3.[6] This yields 8K unique datasets (500 original and 7,500 transformed) to use at one training epoch (over 1,000 steps with batch size $B = 8$). We repeat this process at each epoch, drawing 500 new datasets and transforming them to reach 8K datasets per epoch.

### D.2   REAL-WORLD BENCHMARK DATASETS

While pretraining is purely on synthetic datasets, we evaluate FoMo-0D on **57** real-world datasets from the ADBench benchmark (Han et al., 2022) (see Table 15). They consist of 47 popular tabular outlier detection datasets, as well as 10 newly-constructed tabular datasets created from images and natural language tasks by using pretrained models to extract embeddings. We defer to the original paper for the details on these benchmark datasets.

---

[6]It is important to ensure that the eigenvalues of $\mathbf{W}$ (i.e. variances) are not too small such that the dataset does not flatten in any direction. To this end, we draw a random orthonormal basis $\mathbf{U} \in [-1, 1]^{d \times d}$ and a diagonal $\boldsymbol{\Lambda}$ with eigenvalues $\lambda_{kk} \in ([-1, -0.1] \cup [0.1, 1])^d$, and obtain $\mathbf{W} = \mathbf{U}\boldsymbol{\Lambda}\mathbf{U}^T$. We also use $\mathbf{b} \in [-1, 1]^d$.

We compare to DTE (Livernoche et al., 2024) and baselines therein as described next, thus, following their semi-supervised OD setup we split each dataset five times into train/test using five different seeds and report the mean performance and its standard deviation. In particular, each random split designates 50% of the inliers as $\mathcal{D}_{\text{train}}$, while $\mathcal{D}_{\text{test}}$ contains the rest of the inliers and all the outlier samples. Note that while the baseline methods require model re-training and inference for each $\mathcal{D}_{\text{train}}/\mathcal{D}_{\text{test}}$ split, FoMo-0D uses the splits only for inference as $\mathcal{D}_{\text{train}}$ is merely passed as context.

Before passing the datasets as input to FoMo-0D, we perform a quantile transform such that the features follow a Normal distribution, to better align with the pretraining data from GMMs.

## D.3 BASELINES

We compare FoMo-0D against **26** baselines, from classical/shallow methods to modern/deep models. Our baselines include all the baselines imported from one of the latest papers that proposed the SOTA diffusion-based model DTE (Livernoche et al., 2024), and its three variants; DTE-C, DTE-IG, and DTE-NP. Their baselines comprise all those in ADBench (Han et al., 2022); both classical ones ($k$NN (Ramaswamy et al., 2000), LOF (Breunig et al., 2000), iForest (Liu et al., 2008), HBOS (Goldstein and Dengel, 2012), etc.) and deep models (DeepSVDD (Ruff et al., 2018), DAGMM (Zong et al., 2018), DROCC (Goyal et al., 2020), etc.). They also include more recent approaches based on self-supervised learning (GOAD (Bergman and Hoshen, 2020), ICL (Shenkar and Wolf, 2022), SLAD (Xu et al., 2023), etc.), besides the four additional generative baselines: normalizing planar flows (Rezende and Mohamed, 2015), DDPM (Ho et al., 2020), VAE (Kingma, 2013) and GANomaly (Akcay et al., 2019). We defer to the original paper for additional details. Overall, our 26 baselines consist of the most recent, SOTA approaches for OD that span a diverse family (nonparametric, self-supervised, generative, etc.).

## D.4 HYPERPARAMETERS FOR BASELINES

Table 4 gives the list of HP values we used to study the HP sensitivity/performance variability of the (from top to bottom) top-4 baselines.

Table 4: Top-4 baselines (from top to bottom) and hyperparameter (HP) configurations.

| Baseline | Hyperparameters |
|---|---|
| DTE-NP | $k \in \{5, 10, 20, 40, 50\}$ |
| $k$NN | $k \in \{5, 10, 20, 40, 50\}$ |
| ICL | `learning_rate` $\in \{10^{-1}, 10^{-2}, 10^{-3}, 10^{-4}, 10^{-5}\}$ |
| DTE-C | $k \in \{5, 10, 20, 40, 50\}$ |

## D.5 RANKING THE 26 BASELINES

Figure 16 presents the visualization of the $p$-values of the pairwise Wilcoxon signed rank test w.r.t. AUROC among the baseline methods used by Livernoche et al. (2024). We rank these 26 baselines based on their mean $p$-value (i.e., row-wise average) against the other baselines.

## D.6 COMPARISON OF TOP-4 BASELINE VARIANTS WITH VARYING HP CONFIGURATIONS

Figure 17, 18, 19, 20 give the $p$-values, respectively comparing the variants of the top-4 baselines (DTE-NP, $k$NN, ICL, DTE-C) among themselves using different HP configurations, as well as the $^{\text{avg}}$ model with the average performance across HPs. (Specifically for ICL, `learning_rate` (`lr`) $\in \{10^{-1}, 10^{-2}, 10^{-3}, 10^{-4}, 10^{-5}\}$; and for others, #nearest-neighbors $k \in \{5, 10, 20, 40, 50\}$). We find that for ICL, `lr`$= 10^{-3}$ or $10^{-4}$ are preferable while those that are too small or too large perform poorly. For others, small $k \in \{5, 10\}$ tend to outperform larger $k \in \{40, 50\}$. Note that Livernoche et al. (2024) used $k = 5$ in their paper that proposed DTE (and variants) as well as the $k$NN baseline for fair comparison, while the DTE$^{\text{avg}}$ and $k$NN$^{\text{avg}}$ models across HP configurations perform subpar.

### D.7  Sampling time of $d$-dimensional GMM

Figure 7 shows the sampling time of drawing 10,000 points from different GMMs with increasing dimensionality $d = \{10, 20, ..., 200\}$. We parallelize the sampling process over 10 CPUs, where each CPU draws 1000 samples.

We observe that the sampling time grows nonlinearly as the number of dimensions increases, which suggests that it may incur considerable computational overhead to directly draw from the data prior over hundreds of thousands of training steps, motivating the use of our proposed on-the-fly linear transformation $T$ for scalability.

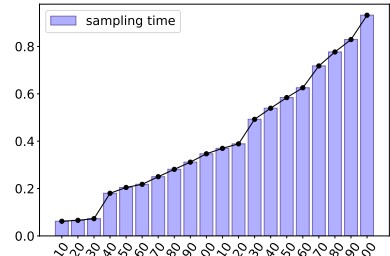

Figure 7: Sampling time (in seconds) of 10,000 points from GMMs with varying number of dimensions.

## E  Ablation Analyses

In this section, we perform various ablations to study the effect of different design choices in FoMo-0D; namely, **E.1** maximum pretraining data dimensionality $D$, the number of routers $R$ on **E.2** cost and **E.3** performance, **E.4** context size (both for training and inference), **E.5** number of unique datasets used for pretraining (i.e., reuse periodicity $P$), data transformation $T$ during synthesis on **E.6** performance and **E.7** speed up, **E.8** data diversity and prolonged training, and finally, **E.9** quantile transforming the benchmark datasets preceding inference.

Unless stated otherwise, most ablation results are performed using FoMo-0D with $D = 20$, as it is faster to pretrain under these many varying settings.

### E.1  Effect of pretraining dimensionality $D$

***How does** FoMo-0D**'s generalization performance change by increasing dimensionality of the pretraining data**?*

We start by comparing FoMo-0D pretrained on datasets with up to $D = 20$ versus $D = 100$ dimensions. Note that learning on higher dimensional datasets is harder, as evident from the relatively larger pretraining loss as shown in Appendix Figure 5. While the statement is accurate in general, it is also partly because subspace outliers "hide" better in higher dimensions.

Comparing Table 1 ($D = 100$) with Table 2 ($D = 20$) w.r.t. $p$-values over All datasets, we find that FoMo-0D at larger scale does better, where **all** $p$-values are larger for $D = 100$ than $D = 20$. We find that FoMo-0D with $D = 20$ performs well on datasets with $d \leq 20$ (i.e., "on its own game"), however beyond its pretraining setting, e.g. on datasets with $d \leq 50$, $D = 100$ is superior to $D = 20$ as shown in Appendix Table 8.

### E.2  Effect of routers on cost

***What is the running time and memory cost of** FoMo-0D **with & w/out router-based attention**?*

Figure 8(left) shows the average inference time per test sample, comparing FoMo-0D using a router-based attention mechanism with $R = 500$ routers (in green) versus FoMo-0D using typical attention without any routers (in blue). As inference context size increases, running time for traditional attention grows quadratically while router mechanism scales linearly.[7]

Similarly, memory cost with routers is considerably lower when using routers, especially for larger context sizes, as shown in Figure 8(middle).

---

[7]Note that the inference time is reported on CPUs to show scalability. On GPUs, w/ 5K context size, see Appendix Figure 6, where typical attention takes advantage of parallelism (6.5ms), while router-based attention is slightly slower (7.7 ms w/ 500 routers) due to its **two** sequential self-attentions; see Eq.s (4) and (5).

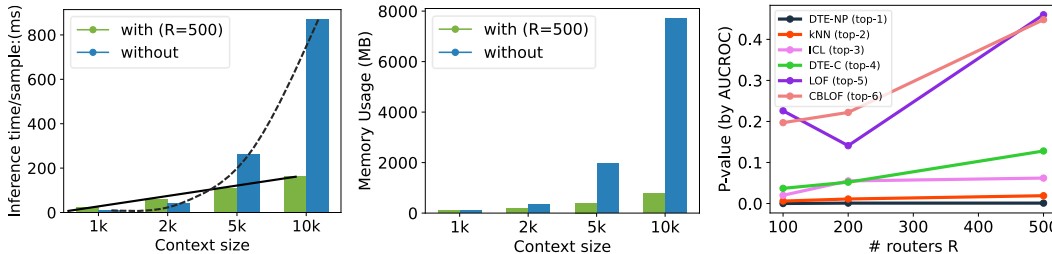

Figure 8: FoMo-0D w/ *router mechanism saves time and memory* while *more #routers perform better*, offering a cost-performance trade-off: (left) inference-time (ms) per sample and (middle) memory cost (MB) with & w/out routers by varying context size; (right) performance (based on $p$-value against top baselines, higher is better) vs. number of routers. (setting: $D = 20$, $P = 1$)

### E.3 EFFECT OF ROUTERS ON PERFORMANCE

***What is the impact of the number $R$ of routers (or representatives) on performance***?

Router-based mechanism allows to trade-off running time with expressiveness of the attention and hence performance. Figure 8(right) shows the $p$-values of the Wilcoxon signed rank test as the number of routers $R$ is increased from 100 to 200 and 500, comparing FoMo-0D to each of the top-6 baselines. We notice that FoMo-0D performance tends to increase monotonically with more routers.

### E.4 EFFECT OF CONTEXT SIZE

***What is the impact of context size, both during model pretraining as well as during inference***?

To study how performance changes by context size, we train FoMo-0D with varying context size in {1K,2K,5K} and employ each pretrained model for inference with varying context size in {1K,2K,5K,10K}. Table 5 shows the results, where performance is depicted by the average rank of FoMo-0D (the lower, the better).

Table 5: Average rank (based on comparison to 30 baselines w.r.t. AUROC) of FoMo-0D across datasets under *different context sizes* for training and inference. Smaller ranks imply better performance. (setting: $D = 20$, $R = 500$, $P = 1$)

|          | Infer:1K | Infer:2K | Infer:5K | Infer:10K |
|----------|----------|----------|----------|-----------|
| Train:1K | 13.816   | 14.623   | 15.193   | 15.439    |
| Train:2K | 13.079   | 13.219   | 13.439   | 13.561    |
| Train:5K | 13.088   | 13.211   | 13.307   | 13.430    |

We find that training with a larger context improves performance at any inference context size. On the other hand, perhaps counter-intuitively, FoMo-0D with smaller inference context size does better. We conjecture that is because the #routers-to-context size ratio increases with a larger context size at inference, limiting the expressive power of the "bottleneck" attention mechanism. The pairwise statistical tests among the $3 \times 4 = 12$ models support these observations, as shown in Figure 9. Interestingly, when training context size is large enough at 5K, inference with 10K samples generalizes beyond training with no significant difference (at 0.05) from other inference context sizes.

### E.5 EFFECT OF NUMBER OF UNIQUE DATASETS

***How do FoMo-0D performances compare when pretrained on unique vs. reused datasets, via varying periodicity $P$***?

Next we study the effect of dataset *reuse at epoch level* (w/out transformation) on performance as presented in Section 3.3. We vary reuse periodicity $P$ in {1, 50, 100}, and accordingly, increase the number of unique datasets used for pretraining across epochs. As shown in Table 6, FoMo-0D (w/ $D = 20$) performs similarly with varying dataset reuse. In fact, it is competitive even with $P = 1$, remaining no different from the $3rd$ best baseline (ICL) across All (57) datasets, while significantly outperforming the top $5th$ (LOF) across (24) datasets with $d \leq 20$ as well as (38) with $d \leq 50$.

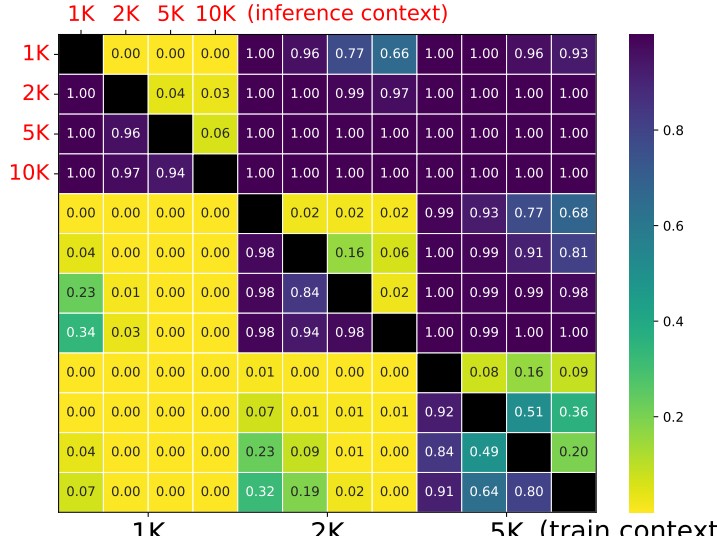

Figure 9: $p$-values of the pairwise Wilcoxon signed rank test between models (larger $p$ implies col-method is better than row-method) w/ different context sizes for **training** (1K/2K/5K, $1st$/$2nd$/$3rd$ four grids, in **black**) and inference (1K/2K/5K/10K, every $1st$/$2nd$/$3rd$/$4th$ grid, in red): Larger training context improves overall performance, while smaller inference context is preferable.

Table 6: Ablation results on dataset reuse across epochs with varying $P \in \{1, 50, 100\}$ show stable $p$-values against the top-5 baselines, where FoMo-0D with $D = 20$ remains no different from the top $3rd$ baseline at 0.05 w.r.t. pairwise Wilcoxon signed rank test comparisons, while it continues to significantly outperform the top $5th$ baseline (LOF) when $d \leq 50$. (setting: $D$=20, $R$=500, context size=5K, w/out transformation $T$)

| top-5 | $P = 1$ (#unique datasets: 8K) | | | | | $P = 50$ (#unique datasets: $8 \times 50 = 400$K) | | | | | $P = 100$ (#unique datasets: $8 \times 100 = 800$K) | | | | |
|---|---|---|---|---|---|---|---|---|---|---|---|---|---|---|---|
| | DTE-NP | kNN | ICL | DTE-C | LOF | DTE-NP | kNN | ICL | DTE-C | LOF | DTE-NP | kNN | ICL | DTE-C | LOF |
| All | 0.001 | 0.019 | 0.062 | 0.128 | 0.460 | 0.001 | 0.019 | 0.089 | 0.159 | 0.394 | 0.001 | 0.015 | 0.072 | 0.121 | 0.290 |
| $d \leq 20$ | 0.583 | 0.755 | 0.943 | 0.736 | **0.998** | 0.572 | 0.789 | 0.968 | 0.616 | **0.993** | 0.439 | 0.678 | 0.953 | 0.550 | **0.972** |
| $d \leq 50$ | 0.415 | 0.750 | 0.869 | **0.962** | **0.999** | 0.347 | 0.794 | 0.893 | 0.946 | **0.997** | 0.293 | 0.697 | 0.890 | 0.924 | **0.994** |

### E.6 EFFECT OF TRANSFORMATION $T$ FOR SYNTHESIS

***How do* FoMo-0D *performances compare when pretrained on datasets with vs. w/out linear transformation*?**

Setting $P = 1$, we next study the impact of linear transformation $T$. Table 7 presents the results, where we compare reuse of the *same* 8K unique datasets across epochs (w/out $T$), versus *transforming* these datasets with $T$ at every epoch with different parameters (w/ $T$). FoMo-0D performance remains stable; no different from the top $3rd$ model on All datasets, while significantly outperforming the top $5th$ across those with $d \leq 20$ and $d \leq 50$. This suggests that $T$ can be employed without sacrificing performance to save time during pretraining.

Table 7: Ablation results on performance w/ & w/out linear transformation $T$ show stable $p$-values against the top-5 baselines, where FoMo-0D with $D = 20$ remains no different from the top $3rd$ baseline at 0.05 w.r.t. pairwise Wilcoxon signed rank test comparisons. (setting: $D = 20$, $R = 500$, context size=5K, $P = 1$)

| top-5 | w/out transformation $T$ | | | | | w/ transformation $T$ | | | | |
|---|---|---|---|---|---|---|---|---|---|---|
| | DTE-NP | kNN | ICL | DTE-C | LOF | DTE-NP | kNN | ICL | DTE-C | LOF |
| All | 0.001 | 0.019 | 0.062 | 0.128 | 0.460 | 0.002 | 0.015 | 0.226 | 0.210 | 0.280 |
| $d \leq 20$ | 0.583 | 0.755 | 0.943 | 0.736 | **0.998** | 0.648 | 0.708 | 0.988 | 0.718 | **0.955** |
| $d \leq 50$ | 0.415 | 0.750 | 0.869 | **0.962** | **0.999** | 0.264 | 0.382 | 0.971 | 0.900 | **0.963** |

### E.7 SPEED UP BY $T$

***What is the time saving on data synthesis with linear transformation?***

Figure 10 shows the distribution of pretraining running-time per epoch with and w/out data transformation. Specifically, we compare (left) generating 8K unique datasets/epoch on-the-fly and (right) first generating 500 unique datasets on-the-fly and then transforming each one 15 times using $T$ with different parameters to reach 8K datasets at each epoch.

Notice that pretraining with $T$ takes about 450 sec./epoch on average, while without $T$ it requires 1200 sec./epoch to generate 8K unique datasets and gradient descent across 1000 steps. Different from other ablation results, which are based on the $D = 20$ model, here we report the running times for our $D = 100$ model. Overall, our final FoMo-0D took ≈**25 hours** for pre-training (450 sec. ×200 epochs). Importantly, this is a one-time cost that amortizes across many downstream tasks with as low as **7.7 ms inference time** per test sample (see Table 3 and Appendix Figure 6).

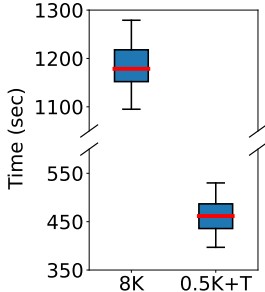

Figure 10: Runtime/epoch dist.n over 100 epochs for FoMo-0D ($D$=100) with (left) $P$=100, i.e. 8K unique datasets/epoch vs. (right) 0.5K unique+7.5K transformed datasets/epoch.

### E.8 EFFECT OF DATA DIVERSITY AND PROLONGED TRAINING

***How does* FoMo-0D*'s performance change by increasing pretraining data diversity and number of training epochs?***

Originally we have trained FoMo-0D w/ $D = 100$ using 0.5K unique + 7.5K transformed datasets over 200 epochs. As mentioned earlier, learning in higher dimensions tends to incur a larger loss in general but also specifically here, as subspace outliers are harder to detect in high dimensions.

Toward reducing the loss further, we resume the pretraining for another 100 epochs. Further, to simplify the tasks and thereby increase data diversity, we also decrease the inlier/outlier labeling percentile threshold from 90% to 80% during on-the-fly data generation in the last 100 epochs. In Figure 12, we present the training loss of FoMo-0D ($D = 100$) trained with 0.5K unique + 7.5K transformed datasets/epoch over 200 epochs ($90th$ percentile as labeling threshold) and then 100 additional

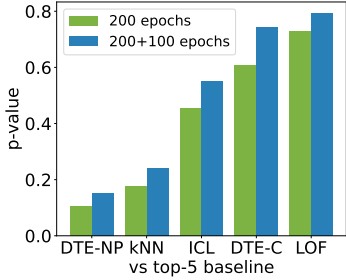

Figure 11: $p$-values increase with additional 100 epochs of pretraining, i.e. FoMo-0D w/ $D = 100$ performs better against top-5 baselines on datasets w/ $d \leq 100$.

epochs ($80th$ percentile as the threshold) to show how data diversity and amount affect model performance. Figure 11 compares FoMo-0D's performance (w/ $D = 100$) to top-5 baselines w.r.t. $p$-values of the paired Wilcoxon signed rank test on datasets with $d \leq 100$, after the first 200 epochs versus after 300 epochs. The increase in all the $p$-values showcases the benefit of additional training.

### E.9 EFFECT OF APPLYING QUANTILE TRANSFORM ON BENCHMARK DATASETS

***What is the impact of quantile data transform preceding inference on performance?***

We pretrain FoMo-0D on synthetic datasets from a simple data prior based on GMMs. The real-world benchmark datasets, on the other hand, may exhibit features with distributions different from Gaussians. To close the gap, we apply a quantile transform (denoted QT) on the benchmark datasets prior to feeding them to FoMo-0D for inference, which transforms the features to exhibit a more Gaussian-like probability distribution.

Figure 13 compares the performance of three FoMo-0D w/ $D = 100$ variants with and w/out QT against the top-5 baselines w.r.t. the $p$-values of the paired Wilcoxon signed rank test. FoMo-0D tends to perform better as suggested by larger $p$-values when QT is applied.

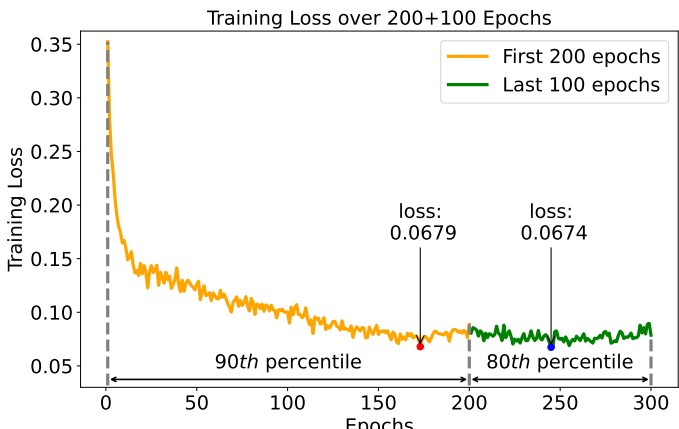

Figure 12: (best in color) Training loss of FoMo-0D ($D = 100$) with 0.5K unique + 7.5K transformed datasets/epoch for 200 epochs (in orange), followed with additional 100 epochs of training (in green). For the first 200 epochs we train with $90th$ percentile as the inlier/outlier threshold, which we reduce to $80th$ in the subsequent 100 epochs.

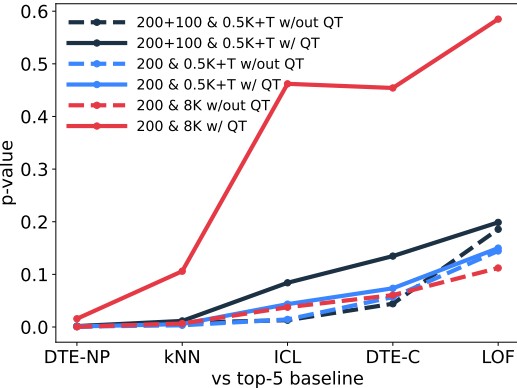

Figure 13: $p$-values increase, i.e. FoMo-0D performance improves, against top-5 baselines with quantile transform (QT) preceding inference, for 3 different settings of FoMo-0D w/ $D = 100$.

Besides the ablation studies, we provide a qualitative case study of sample-to-sample attention in Appendix F, showing that an outlier attends to the points in context that are within a short distance significantly more than random points, suggesting that PFNs tend to mimic non-parametrics.

## F    QUALITATIVE ANALYSIS ON SAMPLE-TO-SAMPLE ATTENTION

We sample 50 inliers as context and 100 outliers from a 2-$d$ GMM using the $80th$ percentile as the labeling threshold, and visualize the top 5 inliers most attended by the 100 outliers based on the average (cross) attention weights over 4 heads from the last layer of FoMo-0D ($D = 100$), which accurately labeled all the 100 outliers. In Figure 14, the most frequently attended inliers are close to either the center of a Gaussian (e.g., $1st, 5th$) or the criterion (e.g., $3rd, 4th$), suggesting FoMo-0D tends to learn decision boundaries that reflect the prior data generation process. For each outlier, we compute the sum of L2 distances to its top-5 attended inliers (att), the sum of L2 distances to 5 randomly chosen inliers (rdm), and the sum of L2 distances to top-5 inliers with highest likelihood under the GMM (prob). We perform

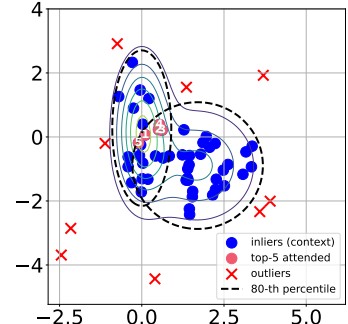

Figure 14: Top-5 attended inliers (all 50 inliers and only part of the outliers are shown for better visualization).

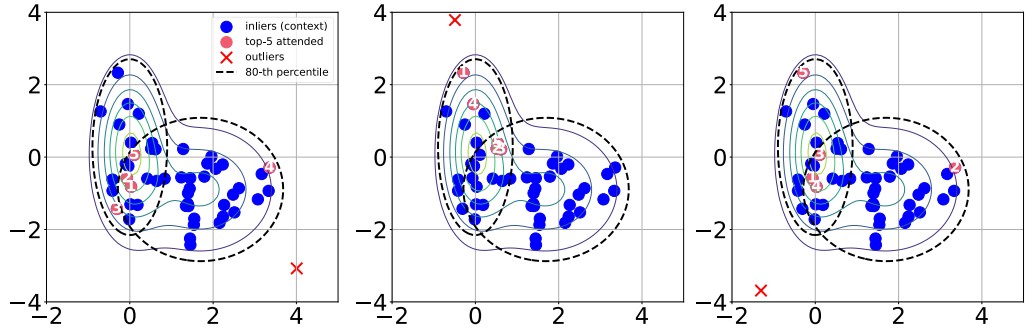

Figure 15: Top-5 attended inliers of 3 outliers at different position of the GMM

Wilcoxon signed rank test between `att` and `rdm` (alternative "less"), `att` and `prob` (alternative "greater") over all the outliers, with a $p$-value of $4.4 \times 10^{-4}$ and 0.99, respectively, suggesting the distances based on attention weights are significantly less than the random distances, and **not** significantly greater than the distances to inliers in high probability region.

We visualize the top-5 attended inliers for 3 outliers at different position of the 2-$d$ GMM in Figure 15. For a specific outlier, there is a similar trend of attending to the center of a Gaussian (as shown in Figure 14), besides, inliers that reflect the criterion boundary or are close to the outlier are actively attended (e.g., $3rd, 4th$ in the left, $1st$ in the middle, $2nd, 5th$ in the right), suggesting FoMo-0D is incorporating both boundary and nearest neighbor information dynamically for each outlier.

# G    FULL RESULTS

Tables 9.1 & 9.2, 10.1 & 10.2, and 11.1 & 11.2 respectively show the AUROC, AUPR and F1 scores of the top-4 baselines, DTE-NP, $k$NN, ICL, and DTE-C as well as their corresponding $^{\text{avg}}$ model with the average performance across HPs, as listed in Table 4.

Tables 12.1 & 12.2, 13.1 & 13.2, and 14.1 & 14.2 respectively show the AUROC, AUPR and F1 scores of all methods across all benchmark datasets. In all these tables, the last four rows show the avg_rank of methods across datasets, and $p$-values of the Wilcoxon signed rank test comparing FoMo-0D w/ $D = 100$ with other baselines. The preceding four rows are the same for FoMo-0D w/ $D = 20$, when ranking 31 models (26 baselines + 4 $^{\text{avg}}$ variants of top-4 baselines + FoMo-0D w/ $D = 20$).

Table 8: Comparison of methods across datasets. (top row) Rank w.r.t. AUROC performance avg.'ed over 57 datasets is presented for FoMo-0D (with $D = 100$), **top-10 baselines** with default HPs, and **top-4**[5] baselines with performance **avg.**'ed over varying HPs (denoted w/ $^{\text{avg}}$); followed by $p$-values of the pairwise Wilcoxon signed rank test, comparing FoMo-0D to each baseline (from top to bottom) over All (57) datasets, those (24) w/ $d \leq 20$, (38) w/ $d \leq 50$, (42) w/ $d \leq 100$ and (46) datasets w/ $d \leq 500$ dimensions. FoMo-0D performs as well as **(i.e., statistically no different from) the 2$nd$ best model** ($k$NN, w/ $p = 0.106$) across All datasets, while it is **comparable to** ($p > 0.05$) **or better than** ($p > 0.95$) **all baselines** over datasets w/ $d \leq 100$ (aligned w/ pretraining where $D = 100$) *and* $d \leq 500$ (generalizing beyond pretraining).

| | FoMo-0D | DTE-NP | $k$NN | ICL | DTE-C | LOF | CBLOF | Feat.Bag. | SLAD | DDPM | OCSVM | DTE-NP$^{\text{avg}}$ | $k$NN$^{\text{avg}}$ | ICL$^{\text{avg}}$ | DTE-C$^{\text{avg}}$ |
|---|---|---|---|---|---|---|---|---|---|---|---|---|---|---|---|
| Rank(avg) | 11.886 | 7.553 | 9.018 | 10.851 | 11.36 | 12.316 | 13.342 | 13.386 | 12.982 | 14.061 | 13.851 | 9.079 | 11.105 | 12.991 | 22.263 |
| All | - | 0.016 | 0.106 | 0.462 | 0.454 | 0.585 | 0.750 | 0.823 | 0.759 | 0.901 | 0.895 | 0.112 | 0.315 | 0.670 | 1.000 |
| $d \leq 20$ | - | 0.428 | 0.665 | 0.987 | 0.727 | 0.911 | 0.940 | 0.987 | 0.868 | 0.758 | 0.968 | 0.781 | 0.868 | 0.990 | 1.000 |
| $d \leq 50$ | - | 0.734 | 0.923 | 0.992 | 0.973 | 0.989 | 0.987 | 0.999 | 0.948 | 0.985 | 0.986 | 0.948 | 0.967 | 0.989 | 1.000 |
| $d \leq 100$ | - | 0.415 | 0.700 | 0.949 | 0.953 | 0.970 | 0.971 | 0.996 | 0.876 | 0.980 | 0.978 | 0.752 | 0.860 | 0.958 | 1.000 |
| $d \leq 200$ | - | 0.315 | 0.605 | 0.923 | 0.919 | 0.944 | 0.977 | 0.990 | 0.904 | 0.970 | 0.983 | 0.663 | 0.789 | 0.937 | 1.000 |
| $d \leq 500$ | - | 0.220 | 0.569 | 0.827 | 0.894 | 0.960 | 0.968 | 0.994 | 0.910 | 0.960 | 0.979 | 0.607 | 0.756 | 0.846 | 1.000 |

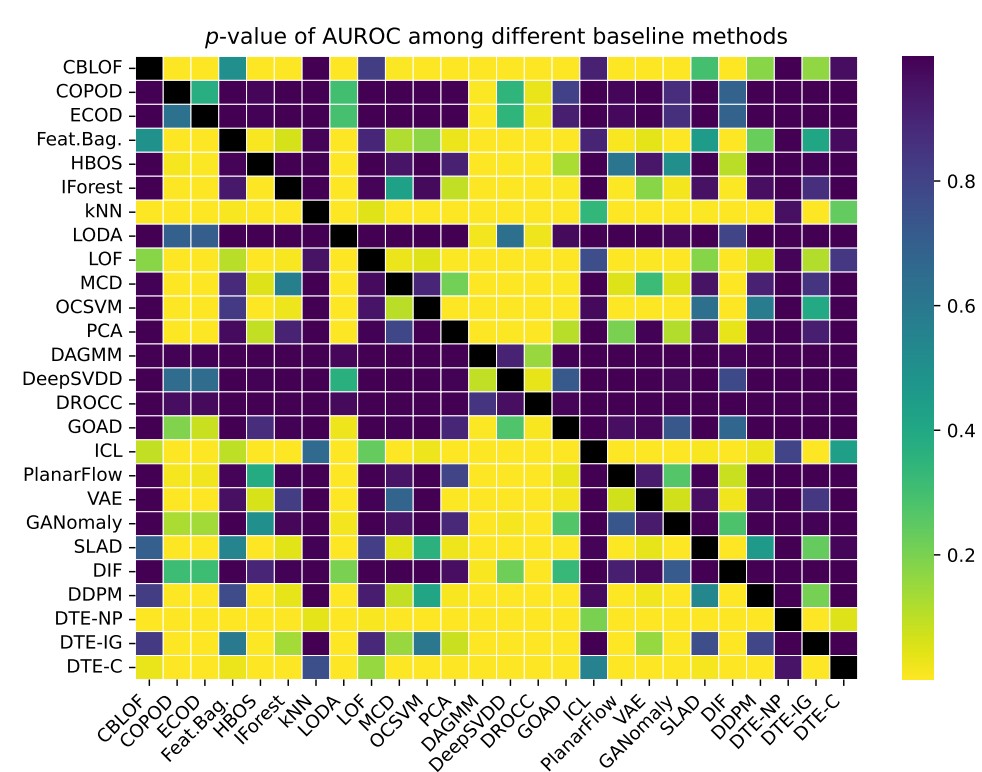

Figure 16: Pairwise $p$-values among baseline methods based on the Wilcoxon signed rank test w.r.t. AUROC performances across datasets.

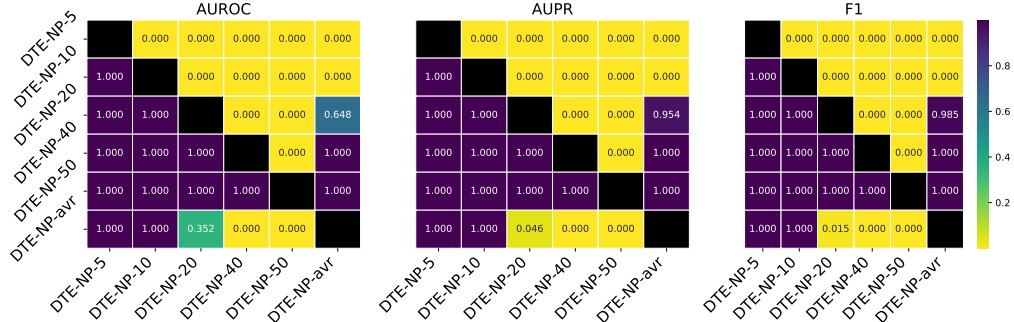

Figure 17: $p$-values w.r.t. AUROC/AUPR/F1 among different HP configurations of **DTE-NP** (i.e., $k \in \{5, 10, 20, 40, 50\}$), along with the $^{\mathrm{avg}}$ model with the average performance across HPs.

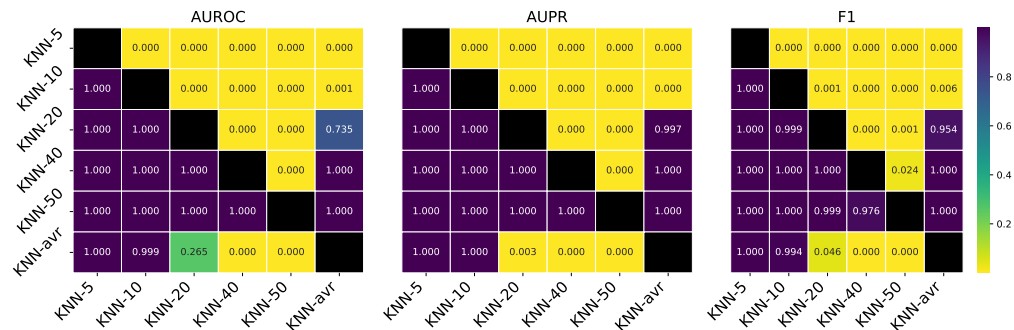

Figure 18: $p$-values w.r.t. AUROC/AUPR/F1 among different HP configurations of $k$**NN** (i.e., $k \in \{5, 10, 20, 40, 50\}$), along with the $^{\mathrm{avg}}$ model with the average performance across HPs.

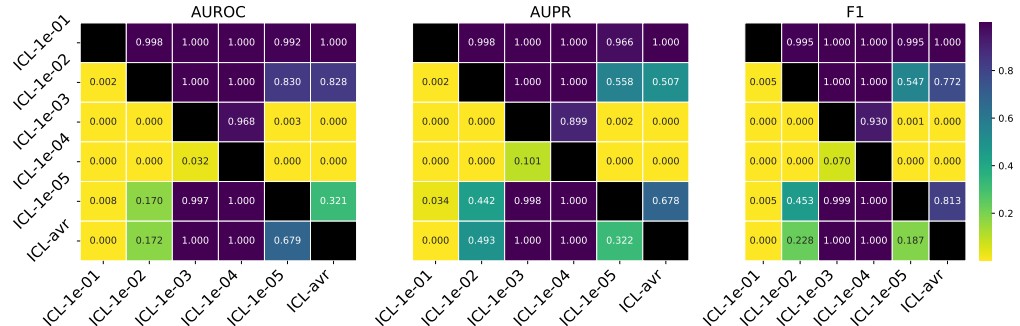

Figure 19: $p$-values w.r.t. AUROC/AUPR/F1 among different HP configurations of **ICL** (i.e., learning_rate $\in \{10^{-1}, 10^{-2}, 10^{-3}, 10^{-4}, 10^{-5}\}$), along with the $^{\mathrm{avg}}$ model with the average performance across HPs.

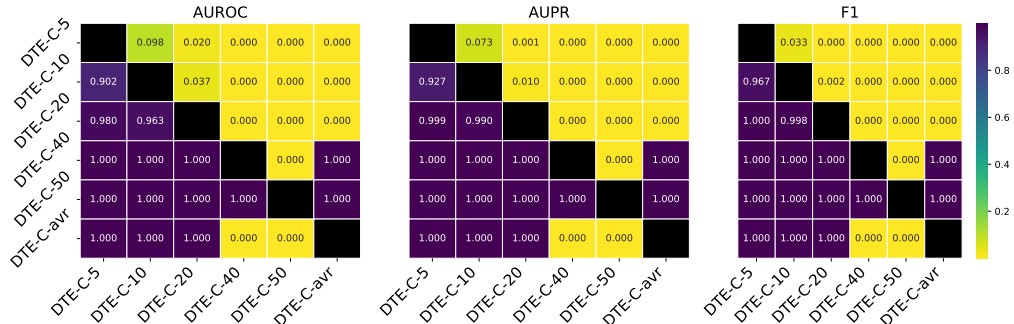

Figure 20: $p$-values w.r.t. AUROC/AUPR/F1 among different HP configurations of **DTE-C** (i.e., $k \in \{5, 10, 20, 40, 50\}$), along with the $^{\mathrm{avg}}$ model with the average performance across HPs.

Table 9.1: Average AUROC ± standard dev. over five seeds for the semi-supervised setting of DTE-NP, $k$NN with varying hyperparameter (HP) values; $k \in \{5, 10, 20, 40, 50\}$. Also reported is the avg model. We use **bold** and underline respectively to mark the **best** and the worst performance of each model to showcase the variability of performance across different HP settings.

| dataset | DTE-NP-5 | DTE-NP-10 | DTE-NP-20 | DTE-NP-40 | DTE-NP-50 | DTE-NP-avr | KNN-5 | KNN-10 | KNN-20 | KNN-40 | KNN-50 | KNN-avr |
|---|---|---|---|---|---|---|---|---|---|---|---|---|
| aloi | 50.69±0.00 | 51.02±0.00 | 51.26±0.00 | 51.58±0.00 | 51.69±0.00 | 51.25±0.00 | 51.04±0.00 | 51.33±0.00 | 51.63±0.00 | 51.97±0.00 | 52.08±0.00 | 51.61±0.00 |
| amazon | 60.76±0.00 | 60.69±0.00 | 60.53±0.00 | 60.17±0.00 | 60.22±0.00 | 60.47±0.00 | 60.58±0.00 | 60.52±0.00 | 60.23±0.00 | 60.02±0.00 | 59.91±0.00 | 60.25±0.00 |
| annthyroid | 93.01±0.00 | 92.89±0.00 | 92.66±0.00 | 92.38±0.00 | 92.26±0.00 | 92.64±0.00 | 92.81±0.00 | 92.60±0.00 | 92.34±0.00 | 91.98±0.00 | 91.79±0.00 | 92.30±0.00 |
| backdoor | 94.48±0.42 | 93.72±0.46 | 92.67±0.46 | 91.20±0.45 | 90.68±0.45 | 92.55±0.44 | 93.71±0.46 | 92.58±0.46 | 91.14±0.46 | 89.18±0.47 | 88.39±0.51 | 91.00±0.47 |
| breastw | 99.10±0.28 | 98.91±0.35 | 98.59±0.34 | 98.40±0.35 | 98.36±0.28 | 98.67±0.28 | 99.09±0.24 | 99.11±0.27 | 99.16±0.22 | 99.21±0.18 | 99.22±0.17 | 99.16±0.21 |
| campaign | 78.34±0.00 | 78.71±0.00 | 78.91±0.00 | 78.93±0.00 | 78.90±0.00 | 78.76±0.00 | 78.48±0.00 | 78.74±0.00 | 78.65±0.00 | 78.60±0.00 | 78.60±0.00 | 78.66±0.21 |
| cardio | 91.53±0.00 | 92.03±0.00 | 92.46±0.00 | 93.06±0.00 | 93.28±0.00 | 92.47±0.00 | 92.00±0.00 | 92.44±0.00 | 92.99±0.00 | 93.85±0.00 | 94.08±0.00 | 93.07±0.00 |
| cardiotocography | 60.40±0.00 | 61.63±0.00 | 63.14±0.00 | 65.05±0.00 | 65.81±0.00 | 63.21±0.00 | 62.11±0.00 | 63.39±0.00 | 65.12±0.00 | 67.85±0.00 | 68.91±0.00 | 65.48±0.00 |
| celeba | 70.39±0.33 | 72.58±0.26 | 74.81±0.34 | 76.87±0.38 | 77.47±0.37 | 74.42±0.28 | 72.9±0.29 | 75.24±0.40 | 77.50±0.47 | 79.14±0.38 | 79.68±0.37 | 76.90±0.35 |
| census | 72.18±0.34 | 72.34±0.17 | 72.28±0.10 | 71.93±0.17 | 71.80±0.17 | 72.11±0.16 | 72.23±0.29 | 72.36±0.12 | 71.94±0.19 | 71.37±0.21 | 71.28±0.16 | 71.84±0.15 |
| cover | 97.90±0.17 | 97.72±0.14 | 97.40±0.18 | 96.99±0.23 | 96.84±0.24 | 97.37±0.19 | 97.51±0.15 | 97.19±0.15 | 96.75±0.22 | 96.21±0.28 | 96.00±0.31 | 96.73±0.22 |
| donors | 99.72±0.03 | 99.61±0.03 | 99.43±0.06 | 99.14±0.09 | 99.02±0.10 | 99.38±0.06 | 99.51±0.06 | 99.24±0.08 | 98.85±0.10 | 98.20±0.13 | 97.90±0.14 | 98.74±0.09 |
| fault | 58.34±0.00 | 58.37±0.00 | 58.70±0.00 | 60.00±0.00 | 60.43±0.00 | 59.17±0.00 | 58.73±0.00 | 58.76±0.00 | 60.12±0.00 | 61.71±0.00 | 61.79±0.00 | 60.22±0.00 |
| fraud | 95.70±0.90 | 95.67±0.93 | 95.64±0.93 | 95.60±0.92 | 95.60±0.92 | 95.64±0.92 | 95.59±0.97 | 95.55±0.99 | 95.55±0.89 | 95.54±0.92 | 95.62±0.88 | 95.57±0.93 |
| glass | 96.08±0.39 | 93.04±1.06 | 89.82±1.12 | 87.89±1.10 | 87.31±1.40 | 90.83±0.91 | 92.13±0.94 | 88.67±0.98 | 87.24±1.18 | 84.93±2.92 | 83.55±2.61 | 87.30±1.59 |
| hepatitis | 99.84±0.20 | 99.27±0.51 | 96.89±0.96 | 93.15±1.69 | 91.97±1.76 | 96.22±0.88 | 96.77±1.47 | 86.88±2.21 | 85.50±2.34 | 85.46±1.92 | 84.88±2.09 | 87.90±1.75 |
| http | 99.99±0.00 | 99.98±0.01 | 99.95±0.01 | 99.91±0.01 | 99.91±0.02 | 99.95±0.01 | 100.00±0.00 | 99.99±0.02 | 99.95±0.01 | 99.95±0.01 | 99.95±0.01 | 99.96±0.01 |
| imdb | 50.48±0.00 | 50.38±0.00 | 50.32±0.00 | 50.28±0.00 | 50.27±0.00 | 50.35±0.00 | 50.08±0.00 | 50.04±0.00 | 50.29±0.00 | 50.23±0.00 | 50.23±0.00 | 50.18±0.00 |
| internetads | 70.96±0.05 | 68.65±0.00 | 66.86±0.00 | 65.97±0.00 | 65.82±0.00 | 67.65±0.00 | 68.08±0.00 | 65.48±0.00 | 65.02±0.00 | 65.04±0.00 | 65.04±0.00 | 65.73±0.00 |
| ionosphere | 98.48±0.60 | 98.13±0.74 | 97.84±0.64 | 96.83±0.71 | 96.21±0.79 | 97.50±0.63 | 97.32±0.85 | 97.62±0.81 | 96.33±0.76 | 92.80±1.64 | 91.53±1.65 | 95.12±0.92 |
| landsat | 68.99±0.00 | 68.02±0.00 | 66.46±0.00 | 64.73±0.00 | 64.16±0.00 | 66.47±0.00 | 68.25±0.00 | 66.48±0.00 | 64.36±0.00 | 62.49±0.00 | 61.93±0.00 | 64.70±0.00 |
| letter | 36.12±0.00 | 35.66±0.00 | 34.78±0.00 | 33.72±0.00 | 33.40±0.00 | 34.74±0.00 | 35.43±0.00 | 34.54±0.00 | 33.17±0.00 | 32.11±0.00 | 31.69±0.00 | 33.39±0.00 |
| lymphography | 99.88±0.25 | 99.79±0.32 | 99.79±0.32 | 99.76±0.31 | 99.76±0.31 | 99.80±0.30 | 99.87±0.10 | 99.85±0.05 | 99.85±0.05 | 99.88±0.08 | 99.88±0.08 | 99.87±0.06 |
| magic.gamma | 83.91±0.00 | 83.49±0.00 | 82.87±0.00 | 82.05±0.00 | 81.73±0.00 | 82.81±0.00 | 83.27±0.00 | 82.81±0.00 | 81.85±0.00 | 80.76±0.00 | 80.30±0.00 | 81.76±0.00 |
| mammography | 87.65±0.00 | 87.73±0.00 | 87.68±0.00 | 87.42±0.00 | 87.29±0.00 | 87.55±0.00 | 87.58±0.00 | 87.75±0.00 | 87.38±0.00 | 86.97±0.00 | 86.78±0.00 | 87.29±0.00 |
| mnist | 94.22±0.00 | 93.93±0.00 | 93.57±0.00 | 93.20±0.00 | 93.08±0.00 | 93.60±0.00 | 93.85±0.00 | 93.45±0.00 | 93.00±0.00 | 92.55±0.00 | 92.36±0.00 | 93.04±0.00 |
| musk | 100.00±0.00 | 100.00±0.00 | 100.00±0.00 | 100.00±0.00 | 100.00±0.00 | 100.00±0.00 | 100.00±0.00 | 100.00±0.00 | 100.00±0.00 | 100.00±0.00 | 100.00±0.00 | 100.00±0.00 |
| optdigits | 95.00±0.00 | 93.97±0.00 | 92.53±0.00 | 90.87±0.00 | 90.28±0.00 | 92.53±0.00 | 93.72±0.00 | 92.09±0.00 | 90.20±0.00 | 87.66±0.00 | 86.67±0.00 | 90.07±0.00 |
| pageblocks | 89.04±0.00 | 89.40±0.00 | 89.56±0.00 | 89.37±0.00 | 89.23±0.00 | 89.32±0.00 | 89.65±0.00 | 89.86±0.00 | 89.88±0.00 | 89.30±0.00 | 89.18±0.00 | 89.57±0.00 |
| pendigits | 99.90±0.00 | 99.88±0.00 | 99.83±0.00 | 99.51±0.00 | 99.38±0.00 | 99.70±0.00 | 99.87±0.00 | 99.79±0.00 | 99.21±0.00 | 99.21±0.00 | 98.39±0.00 | 99.17±0.00 |
| pima | 82.21±1.82 | 79.74±1.61 | 77.98±1.38 | 77.28±1.35 | 77.14±1.33 | 78.87±1.43 | 77.44±2.07 | 76.14±1.36 | 76.39±1.21 | 76.55±1.25 | 76.38±1.29 | 76.58±1.39 |
| satellite | 82.40±0.00 | 82.09±0.00 | 81.55±0.00 | 80.71±0.00 | 80.38±0.00 | 81.43±0.00 | 82.24±0.00 | 81.56±0.00 | 80.56±0.00 | 79.22±0.00 | 78.76±0.00 | 80.47±0.00 |
| satimage-2 | 99.68±0.00 | 99.73±0.00 | 99.79±0.00 | 99.82±0.00 | 99.80±0.00 | 99.77±0.00 | 99.71±0.00 | 99.80±0.00 | 99.79±0.00 | 99.78±0.00 | 99.77±0.00 | 99.77±0.00 |
| shuttle | 99.94±0.00 | 99.92±0.00 | 99.91±0.00 | 99.91±0.00 | 99.91±0.00 | 99.92±0.00 | 99.91±0.00 | 99.90±0.00 | 99.89±0.00 | 99.89±0.00 | 99.89±0.00 | 99.89±0.00 |
| skin | 99.66±0.05 | 99.52±0.04 | 99.28±0.02 | 98.77±0.09 | 98.54±0.09 | 99.15±0.05 | 99.51±0.06 | 99.17±0.05 | 98.70±0.12 | 97.43±0.05 | 96.90±0.09 | 98.34±0.04 |
| smtp | 92.94±2.55 | 92.84±2.56 | 93.14±2.20 | 93.14±2.15 | 93.14±2.20 | 93.04±2.32 | 92.90±2.48 | 92.97±2.52 | 93.25±2.06 | 93.11±2.28 | 93.24±2.25 | 93.10±2.30 |
| spambase | 84.06±0.00 | 83.49±0.00 | 82.97±0.00 | 82.50±0.00 | 82.36±0.00 | 83.07±0.00 | 83.36±0.00 | 82.68±0.00 | 82.22±0.00 | 81.83±0.00 | 81.72±0.00 | 82.36±0.00 |
| speech | 41.12±0.00 | 39.00±0.00 | 37.59±0.00 | 37.37±0.00 | 37.01±0.00 | 38.42±0.00 | 36.36±0.00 | 36.15±0.00 | 36.37±0.00 | 36.40±0.00 | 36.25±0.00 | 36.31±0.00 |
| stamps | 97.88±0.33 | 97.04±0.33 | 95.60±0.78 | 94.59±1.07 | 94.29±1.08 | 95.88±0.66 | 96.19±1.24 | 94.35±1.35 | 93.67±1.09 | 93.44±1.21 | 93.33±1.25 | 94.20±1.17 |
| thyroid | 98.58±0.00 | 98.63±0.00 | 98.64±0.00 | 98.63±0.00 | 98.59±0.00 | 98.61±0.00 | 98.68±0.00 | 98.67±0.00 | 98.69±0.00 | 98.70±0.00 | 98.69±0.00 | 98.68±0.00 |
| vertebral | 79.59±2.23 | 67.05±2.09 | 56.99±2.07 | 49.07±1.60 | 47.08±1.43 | 59.96±1.83 | 57.33±3.80 | 45.06±1.20 | 41.08±1.52 | 41.08±1.23 | 40.08±1.23 | 46.59±1.67 |
| vowels | 82.11±0.00 | 81.62±0.00 | 80.46±0.00 | 78.11±0.00 | 76.87±0.00 | 79.83±0.00 | 82.21±0.00 | 80.20±0.00 | 76.04±0.00 | 72.26±0.00 | 69.03±0.00 | 76.30±0.00 |
| waveform | 74.42±0.00 | 75.40±0.00 | 76.12±0.00 | 76.79±0.00 | 76.90±0.00 | 75.93±0.00 | 75.21±0.00 | 76.04±0.00 | 76.62±0.00 | 77.47±0.00 | 77.60±0.00 | 76.62±0.00 |
| wbc | 99.62±0.27 | 99.36±0.33 | 99.17±0.41 | 99.12±0.37 | 99.15±0.20 | 99.27±0.34 | 99.23±0.27 | 99.09±0.29 | 98.86±0.38 | 99.08±0.22 | 99.10±0.34 | 98.95±0.35 |
| wdbc | 99.62±0.26 | 99.47±0.29 | 99.26±0.25 | 99.18±0.22 | 99.15±0.20 | 99.34±0.23 | 99.09±0.38 | 99.17±0.05 | 99.08±0.21 | 99.08±0.22 | 99.08±0.17 | 99.11±0.22 |
| wilt | 66.98±0.00 | 63.87±0.00 | 60.18±0.00 | 56.26±0.00 | 55.02±0.00 | 60.46±0.00 | 63.66±0.00 | 59.33±0.00 | 55.21±0.00 | 51.13±0.00 | 49.77±0.00 | 55.82±0.00 |
| wine | 99.90±0.14 | 99.73±0.09 | 99.33±0.18 | 98.88±0.19 | 98.57±0.30 | 99.28±0.12 | 99.17±0.18 | 97.96±0.77 | 98.15±0.47 | 97.65±0.43 | 97.68±0.41 | 98.12±0.40 |
| wpbc | 89.29±1.07 | 79.39±1.04 | 69.61±1.14 | 63.21±1.50 | 61.86±1.64 | 72.67±0.81 | 64.27±2.37 | 59.33±1.67 | 57.03±2.29 | 55.89±1.96 | 55.32±1.96 | 58.14±1.82 |
| yeast | 44.73±0.00 | 44.61±0.00 | 44.33±0.00 | 43.88±0.00 | 43.68±0.00 | 44.25±0.00 | 44.74±0.00 | 44.64±0.00 | 44.23±0.00 | 43.31±0.00 | 42.85±0.00 | 43.95±0.00 |
| yelp | 68.67±0.00 | 68.22±0.00 | 67.75±0.00 | 67.22±0.00 | 66.98±0.00 | 67.57±0.00 | 68.07±0.00 | 67.74±0.00 | 67.12±0.00 | 66.53±0.00 | 66.36±0.00 | 67.16±0.00 |
| MNIST-C | 84.73±0.00 | 84.21±0.00 | 83.58±0.00 | 82.87±0.00 | 82.64±0.00 | 83.60±0.00 | 84.11±0.00 | 83.39±0.00 | 82.64±0.00 | 81.81±0.00 | 81.53±0.00 | 82.70±0.00 |
| FashionMNIST | 90.13±0.00 | 89.91±0.00 | 89.65±0.00 | 89.37±0.00 | 89.27±0.00 | 89.67±0.00 | 89.87±0.00 | 89.55±0.00 | 89.27±0.00 | 88.96±0.00 | 88.86±0.00 | 89.30±0.00 |
| CIFAR10 | 67.81±0.00 | 67.63±0.00 | 67.47±0.00 | 67.32±0.00 | 67.26±0.00 | 67.50±0.00 | 67.53±0.00 | 67.39±0.00 | 67.21±0.00 | 67.09±0.00 | 67.06±0.00 | 67.26±0.00 |
| SVHN | 62.12±0.00 | 61.80±0.00 | 61.48±0.00 | 61.19±0.00 | 61.11±0.00 | 61.54±0.00 | 61.69±0.00 | 61.37±0.00 | 61.06±0.00 | 60.82±0.00 | 60.74±0.00 | 61.14±0.00 |
| MVTec-AD | 89.65±0.57 | 85.94±0.49 | 82.74±0.00 | 80.49±0.49 | 79.94±0.46 | 83.75±0.49 | 81.33±0.47 | 79.41±0.45 | 78.35±0.51 | 77.55±0.42 | 77.39±0.43 | 78.81±0.45 |
| 20news | 60.08±0.67 | 58.74±0.78 | 57.43±0.82 | 56.90±0.88 | 56.97±0.87 | 58.02±0.80 | 57.53±0.84 | 56.74±0.86 | 56.28±0.90 | 56.07±0.92 | 55.62±0.88 | 56.45±0.88 |
| agnews | 67.87±0.00 | 67.26±0.00 | 66.42±0.00 | 65.52±0.00 | 65.21±0.00 | 66.46±0.00 | 67.05±0.00 | 66.15±0.00 | 65.24±0.00 | 64.32±0.00 | 64.00±0.00 | 65.35±0.00 |

Table 9.2: Average AUROC ± standard dev. over five seeds for the semi-supervised setting of ICL and DTE-C baselines with varying hyperparameter (HP) values; For ICL, the learning rate $\in \{0.1, 0.02, 0.001, 0.0001, 1e-05\}$, for DTE-C, $k \in \{5, 10, 20, 40, 50\}$. Also reported is the $^{avg}$ model. We use **bold** and underline respectively to mark the **best** and the underline worst performance of each model to showcase the variability of performance across different HP settings.

| dataset | ICL-0.1 | ICL-0.01 | ICL-0.001 | ICL-0.0001 | ICL-1e-05 | ICL-avr | DTE-C-5 | DTE-C-10 | DTE-C-20 | DTE-C-40 | DTE-C-50 | DTE-C-avr |
|---|---|---|---|---|---|---|---|---|---|---|---|---|
| aloi | 47.74±0.28 | 47.12±0.51 | 46.81±0.47 | 48.42±0.24 | 48.06±0.23 | 47.63±0.15 | 50.20±0.21 | 50.84±0.10 | 50.16±0.34 | 50.26±0.33 | 50.00±0.00 | 50.29±0.09 |
| amazon | 53.07±0.19 | 53.44±0.19 | 52.75±0.68 | 53.31±0.21 | 53.18±0.15 | 53.15±0.19 | 56.20±1.62 | 55.07±4.20 | 56.12±2.73 | 50.00±0.00 | 50.00±0.00 | 53.48±1.33 |
| annthyroid | 84.02±9.46 | 72.68±3.79 | 87.29±2.69 | 88.84±2.35 | 88.52±1.40 | 84.27±2.31 | 97.47±0.10 | 97.65±0.11 | 97.73±0.15 | 97.40±0.22 | 50.00±0.00 | 88.05±0.05 |
| backdoor | 93.03±0.66 | 92.91±0.70 | 93.30±0.44 | 93.93±0.58 | 93.32±0.71 | 93.30±0.48 | 88.65±1.08 | 92.06±1.04 | 92.64±0.84 | 50.00±0.00 | 50.00±0.00 | 74.67±0.36 |
| breastw | 98.96±0.47 | 98.87±0.20 | 99.19±0.34 | 99.11±0.18 | 97.61±0.65 | 98.75±0.18 | 93.45±1.34 | 94.34±1.25 | 96.71±0.91 | 98.85±0.45 | 99.26±0.07 | 96.52±0.53 |
| campaign | 76.07±0.87 | 74.61±1.97 | 78.88±8.42 | 78.68±2.18 | 82.30±0.38 | 78.33±0.52 | 79.18±1.09 | 78.24±2.09 | 78.49±1.13 | 50.00±0.00 | 50.00±0.00 | 67.18±0.74 |
| cardio | 73.99±7.79 | 84.98±4.75 | 82.30±3.36 | 82.30±0.38 | 68.59±0.64 | 77.71±1.76 | 88.07±0.51 | 87.54±0.69 | 87.66±0.63 | 50.00±0.54 | 50.00±0.00 | 72.66±0.21 |
| cardiotocography | 48.99±2.02 | 54.18±2.77 | 53.24±3.25 | 50.67±2.55 | 47.20±1.33 | 50.85±1.24 | 60.09±2.19 | 60.36±1.54 | 59.05±1.34 | 50.00±0.00 | 50.00±0.00 | 55.90±0.23 |
| celeba | 79.38±2.17 | 79.15±2.47 | 76.13±1.88 | 78.39±1.78 | 79.43±1.44 | 78.49±0.93 | 82.95±1.26 | 81.59±0.79 | 80.16±1.26 | 50.00±0.00 | 50.00±0.00 | 68.94±0.44 |
| census | 70.41±2.27 | 67.52±8.79 | 73.93±0.78 | 74.37±0.53 | 74.37±0.53 | 72.25±1.63 | 70.95±0.91 | 68.04±0.85 | 68.05±2.58 | 50.00±0.00 | 50.00±0.00 | 61.41±0.71 |
| cover | 93.59±3.09 | 82.32±9.14 | 91.92±4.58 | 94.97±2.91 | 94.27±3.40 | 91.42±1.74 | 97.57±0.86 | 97.81±0.75 | 96.61±0.63 | 50.00±0.00 | 50.00±0.00 | 78.40±0.14 |
| donors | 86.20±10.29 | 97.76±1.57 | 99.37±0.30 | 99.37±0.30 | 96.30±1.95 | 95.64±0.27 | 98.68±0.15 | 97.56±0.53 | 95.64±0.27 | 58.94±17.89 | 50.00±0.00 | 80.17±3.55 |
| fault | 63.37±3.29 | 63.04±1.93 | 61.65±0.61 | 61.44±0.87 | 62.83±1.01 | 62.47±1.14 | 59.16±1.69 | 58.41±1.41 | 59.04±1.53 | 50.21±0.25 | 50.21±0.42 | 55.41±0.63 |
| fraud | 93.24±1.45 | 93.21±1.25 | 94.97±0.71 | 95.84±0.87 | 93.90±1.05 | 94.23±0.87 | 94.49±1.56 | 93.08±2.20 | 93.50±2.01 | 50.00±0.00 | 50.00±0.00 | 76.21±1.08 |
| glass | 84.02±8.38 | 74.66±3.83 | 99.25±0.37 | 99.30±0.35 | 99.12±0.55 | 95.27±1.45 | 93.46±0.91 | 89.96±2.70 | 84.89±2.89 | 66.42±8.60 | 50.00±0.00 | 76.95±2.58 |
| hepatitis | 99.95±0.11 | 98.76±1.84 | 99.93±0.14 | 99.86±0.29 | 99.97±0.06 | 99.69±0.48 | 99.54±0.58 | 98.90±1.01 | 98.90±1.08 | 99.39±0.07 | 50.00±0.00 | 89.59±0.09 |
| http | 99.96±0.07 | 99.97±0.02 | 99.98±0.01 | 100.00±0.00 | 100.00±0.00 | 99.98±0.02 | 99.54±0.08 | 99.64±0.28 | 99.39±0.06 | 50.00±0.00 | 50.00±0.00 | 79.71±0.05 |
| imdb | 52.16±0.31 | 51.88±0.40 | 52.28±0.15 | 52.35±0.12 | 53.06±0.15 | 52.35±0.12 | 47.68±3.79 | 50.89±1.90 | 47.81±2.08 | 50.00±0.00 | 50.00±0.00 | 49.27±1.12 |
| internetads | 72.61±1.19 | 73.97±0.44 | 74.09±1.75 | 71.23±0.45 | 68.51±0.74 | 72.53±0.65 | 79.02±1.49 | 77.71±0.82 | 77.23±2.48 | 50.00±0.00 | 50.00±0.00 | 66.79±0.73 |
| ionosphere | 96.81±2.22 | 96.13±3.45 | 98.90±0.41 | 98.91±0.31 | 98.14±0.79 | 97.78±1.23 | 94.67±1.52 | 94.49±2.41 | 95.23±1.37 | 85.77±2.62 | 50.00±0.00 | 83.01±5.81 |
| landsat | 65.71±2.05 | 60.63±1.79 | 65.95±0.76 | 67.85±0.68 | 61.82±1.06 | 64.39±0.59 | 50.31±2.61 | 50.31±2.61 | 48.67±2.62 | 50.22±0.45 | 50.00±0.00 | 55.20±0.98 |
| letter | 48.14±3.53 | 42.74±3.41 | 40.70±2.13 | 40.32±1.11 | 47.20±2.50 | 43.82±1.97 | 36.25±1.19 | 37.72±1.31 | 37.40±0.89 | 48.67±2.62 | 44.90±23.57 | 42.38±0.25 |
| lymphography | 100.00±0.00 | 100.00±0.00 | 100.00±0.00 | 100.00±0.00 | 100.00±0.00 | 100.00±0.00 | 98.97±0.31 | 99.40±0.22 | 99.78±0.66 | 50.00±0.00 | 50.00±0.00 | 88.19±1.78 |
| magic.gamma | 69.73±3.02 | 76.94±3.25 | 77.61±0.70 | 77.81±0.88 | 78.36±1.34 | 76.09±1.17 | 86.42±0.59 | 87.21±0.73 | 87.21±0.73 | 93.82±8.49 | 50.00±0.00 | 75.19±3.50 |
| mammography | 79.90±5.19 | 85.08±3.05 | 79.26±2.14 | 80.75±1.40 | 77.35±0.77 | 80.47±0.62 | 83.00±3.62 | 86.02±1.23 | 84.85±2.49 | 64.88±18.23 | 50.00±0.00 | 77.85±0.91 |
| mnist | 75.53±1.43 | 76.05±3.81 | 79.13±1.20 | 87.05±1.12 | 89.10±0.77 | 81.37±0.93 | 90.21±0.70 | 86.98±1.70 | 85.58±2.12 | 85.40±1.03 | 50.00±0.00 | 72.55±0.64 |
| musk | 100.00±0.00 | 100.00±0.00 | 100.00±0.00 | 100.00±0.00 | 85.92±1.84 | 97.18±0.37 | 100.00±0.00 | 100.00±0.00 | 100.00±0.00 | 50.00±0.00 | 50.00±0.00 | 80.00±0.00 |
| optdigits | 91.22±1.88 | 94.15±0.97 | 95.17±0.85 | 88.47±0.03 | 95.08±1.06 | 94.82±0.49 | 86.29±2.19 | 74.81±6.84 | 74.81±6.84 | 50.00±0.00 | 50.00±0.00 | 67.73±1.71 |
| pageblocks | 89.21±1.10 | 90.77±1.73 | 88.70±1.01 | 88.58±0.56 | 88.53±0.27 | 89.16±0.29 | 89.96±0.39 | 90.29±0.81 | 89.23±0.26 | 88.80±0.13 | 50.00±0.00 | 81.66±0.20 |
| pendigits | 87.01±9.78 | 96.45±2.40 | 96.17±2.68 | 98.28±0.77 | 95.10±0.89 | 94.60±2.36 | 98.24±0.47 | 97.42±0.83 | 96.89±0.68 | 50.00±0.00 | 50.00±0.00 | 78.51±0.15 |
| pima | 74.66±3.27 | 73.04±1.77 | 79.77±1.92 | 78.06±0.74 | 75.09±2.65 | 76.12±1.04 | 70.12±1.89 | 67.22±3.15 | 66.78±2.78 | 71.26±2.38 | 69.39±3.15 | 68.95±0.96 |
| satellite | 74.62±9.67 | 83.23±2.89 | 85.43±0.52 | 89.24±0.57 | 83.35±0.48 | 83.57±2.48 | 80.21±1.18 | 79.01±0.76 | 78.54±0.62 | 56.11±12.22 | 50.00±0.00 | 68.77±2.26 |
| satimage-2 | 94.41±2.46 | 93.04±12.69 | 99.80±0.06 | 99.55±0.09 | 98.15±0.57 | 96.99±2.56 | 99.61±0.13 | 99.17±0.15 | 99.38±0.56 | 69.34±23.69 | 50.00±0.00 | 83.30±4.71 |
| shuttle | 99.43±0.50 | 99.89±0.05 | 99.99±0.00 | 99.99±0.00 | 99.97±0.01 | 99.85±0.11 | 99.76±0.00 | 99.74±0.01 | 99.70±0.01 | 91.63±0.28 | 50.00±0.00 | 89.70±0.05 |
| skin | 53.71±32.40 | 86.42±5.89 | 86.94±5.22 | 71.25±17.43 | 70.72±13.69 | 73.81±10.45 | 91.69±0.28 | 92.12±0.27 | 91.95±0.22 | 50.00±0.00 | 50.00±0.00 | 83.48±0.19 |
| smtp | 81.23±10.57 | 87.97±4.57 | 90.53±5.25 | 89.48±4.52 | 92.63±4.06 | 88.37±5.41 | 95.06±1.20 | 95.61±1.24 | 95.84±1.23 | 95.84±1.22 | 50.00±0.00 | 86.47±0.96 |
| spambase | 79.20±2.89 | 79.36±5.07 | 83.16±0.34 | 83.39±0.40 | 78.42±0.49 | 80.71±0.86 | 92.98±0.37 | 88.76±0.26 | 83.74±0.38 | 50.00±0.00 | 50.00±0.00 | 70.00±0.04 |
| speech | 50.00±3.03 | 50.96±2.48 | 73.52±1.46 | 51.29±2.33 | 46.64±1.97 | 50.53±1.10 | 38.02±1.49 | 38.55±1.07 | 38.72±1.79 | 50.00±0.00 | 50.00±0.00 | 43.06±0.55 |
| stamps | 77.62±9.15 | 84.33±6.20 | 96.20±0.74 | 96.70±0.85 | 93.73±0.80 | 95.35±0.60 | 93.01±1.38 | 91.27±2.53 | 86.48±2.57 | 50.10±26.50 | 50.10±26.50 | 82.21±5.22 |
| thyroid | 94.79±1.60 | 95.24±1.04 | 75.60±6.18 | 96.78±1.19 | 81.92±1.33 | 90.11±2.02 | 98.75±0.07 | 98.92±0.02 | 98.92±0.04 | 98.94±0.05 | 50.00±0.00 | 89.11±0.02 |
| vertebral | 53.80±4.97 | 58.76±7.63 | 84.59±3.49 | 82.96±2.86 | 83.99±1.93 | 95.35±0.60 | 67.07±2.73 | 86.66±0.72 | 62.93±4.76 | 56.35±4.78 | 48.40±5.99 | 59.97±2.61 |
| vowels | 73.59±6.94 | 79.11±2.66 | 84.59±3.49 | 84.81±5.61 | 64.69±1.66 | 81.22±1.57 | 87.25±1.51 | 63.97±2.75 | 87.49±0.93 | 50.00±0.00 | 50.00±0.00 | 72.28±0.42 |
| waveform | 70.94±1.78 | 73.85±6.07 | 70.23±2.82 | 61.96±1.35 | 99.29±0.25 | 88.33±2.15 | 65.16±1.34 | 66.24±1.45 | 66.24±1.45 | 50.00±0.00 | 50.00±0.00 | 59.07±0.68 |
| wbc | 98.13±1.33 | 99.08±0.62 | 99.89±0.19 | 99.90±0.16 | 99.90±0.22 | 99.34±0.42 | 86.07±4.41 | 85.96±4.40 | 99.01±0.41 | 97.03±1.02 | 45.17±9.66 | 78.42±3.73 |
| wdbc | 96.66±2.87 | 99.05±0.72 | 99.51±0.25 | 99.67±0.15 | 99.29±0.25 | 98.84±0.43 | 98.96±0.46 | 88.76±0.26 | 99.81±0.29 | 84.90±1.24 | 49.98±0.04 | 77.44±0.57 |
| wilt | 52.75±14.46 | 61.09±5.94 | 83.81±3.04 | 84.63±1.21 | 79.81±0.93 | 72.42±2.57 | 84.17±0.27 | 81.09±1.72 | 81.09±1.72 | 66.04±29.91 | 49.98±0.04 | 83.09±5.95 |
| wine | 99.64±0.73 | 99.73±0.29 | 99.92±0.12 | 99.81±0.39 | 99.84±0.26 | 99.79±0.33 | 99.91±0.11 | 99.70±0.33 | 99.81±0.29 | 51.85±2.56 | 49.96±5.03 | 61.89±2.06 |
| wpbc | 91.76±2.53 | 76.46±9.54 | 95.40±1.42 | 95.51±1.23 | 46.32±0.91 | 90.32±2.53 | 68.39±4.15 | 70.67±1.94 | 68.60±3.44 | 49.96±5.03 | 50.00±0.00 | 47.74±0.50 |
| yeast | 45.12±3.69 | 47.02±1.77 | 47.25±0.62 | 47.38±1.07 | 52.73±0.21 | 46.62±1.16 | 46.82±1.26 | 48.44±1.44 | 49.01±2.16 | 44.44±2.06 | 50.00±0.00 | 56.47±0.84 |
| yelp | 53.40±0.49 | 54.26±0.22 | 54.08±0.78 | 54.07±0.26 | 83.37±0.12 | 53.71±0.22 | 62.00±1.65 | 60.99±2.26 | 60.99±2.26 | 50.00±0.00 | 50.00±0.00 | 71.28±0.16 |
| MNIST-C | 80.78±0.30 | 83.24±0.51 | 85.02±0.14 | 84.91±0.11 | 88.57±0.10 | 83.46±0.11 | 85.01±0.43 | 86.12±0.43 | 85.25±0.30 | 50.00±0.00 | 50.00±0.00 | 74.14±0.06 |
| FashionMNIST | 87.80±0.21 | 90.44±0.09 | 90.91±0.06 | 90.95±0.03 | 59.25±0.13 | 89.74±0.07 | 90.13±0.19 | 90.30±0.13 | 90.27±0.15 | 50.00±0.00 | 50.00±0.00 | 61.25±0.12 |
| CIFAR10 | 59.53±0.31 | 64.12±0.37 | 65.72±0.09 | 65.89±0.27 | 60.61±0.08 | 62.90±0.15 | 68.78±0.24 | 68.59±0.36 | 68.87±0.31 | 97.03±1.02 | 50.00±0.00 | 57.80±0.05 |
| SVHN | 59.44±0.27 | 61.25±0.11 | 61.73±0.07 | 61.54±0.06 | 89.82±0.44 | 60.91±0.03 | 63.02±0.30 | 63.02±0.36 | 62.97±0.38 | 50.75±1.11 | 50.13±0.45 | 73.32±0.42 |
| MVTec-AD | 93.31±0.36 | 93.73±0.30 | 94.26±0.35 | 94.27±0.32 | 55.58±0.17 | 93.08±0.33 | 86.78±0.49 | 89.56±0.58 | 89.38±0.93 | 89.56±0.58 | 50.13±0.45 | 58.00±0.60 |
| 20news | 57.95±0.57 | 59.21±0.72 | 59.52±0.72 | 60.25±0.48 | 55.58±0.17 | 58.50±0.44 | 64.14±1.95 | 64.01±0.79 | 61.86±0.96 | 50.00±0.28 | 50.00±0.28 | 58.00±0.60 |
| agnews | 56.84±0.12 | 57.43±0.39 | 57.59±0.12 | 57.89±0.09 | 56.85±0.16 | 57.32±0.11 | 68.75±0.81 | 65.60±1.74 | 65.95±1.66 | 50.00±0.00 | 50.00±0.00 | 60.06±0.42 |

Table 10.1: Average AUPR ± standard dev. over five seeds for the semi-supervised setting of DTE-NP, $k$NN baselines with varying hyperparameter (HP) values; $k \in \{5, 10, 20, 40, 50\}$. Also reported is the $^{avg}$ model. We use **bold** and underline respectively to mark the **best** and the worst performance of each model to showcase the variability of performance across different HP settings.

| dataset | DTE-NP-5 | DTE-NP-10 | DTE-NP-20 | DTE-NP-40 | DTE-NP-50 | DTE-NP-avr | KNN-5 | KNN-10 | KNN-20 | KNN-40 | KNN-50 | KNN-avr |
|---|---|---|---|---|---|---|---|---|---|---|---|---|
| aloi | 5.95±0.00 | 5.99±0.00 | 6.02±0.00 | 6.06±0.00 | **6.07**±0.00 | 6.02±0.00 | 6.02±0.00 | 6.07±0.00 | 6.09±0.00 | 6.13±0.00 | **6.15**±0.00 | 6.09±0.00 |
| amazon | **11.68**±0.00 | 11.68±0.00 | 11.68±0.00 | 11.61±0.00 | 11.62±0.00 | 11.65±0.00 | 11.69±0.00 | **11.70**±0.00 | 11.65±0.00 | 11.60±0.00 | 11.59±0.00 | 11.65±0.00 |
| annthyroid | 67.49±0.00 | 66.73±0.00 | 66.04±0.00 | 65.39±0.00 | 64.87±0.00 | 66.11±0.00 | **68.07**±0.00 | 67.90±0.00 | 67.26±0.00 | 66.27±0.00 | 65.79±0.00 | 67.06±0.00 |
| backdoor | 55.90±0.99 | 47.16±1.45 | 38.31±1.02 | 31.44±0.47 | 29.58±0.37 | 40.48±0.81 | 46.70±1.22 | 37.36±1.35 | 29.58±0.58 | 24.34±0.41 | 22.34±0.53 | 32.06±0.76 |
| breastw | 98.51±0.56 | 98.19±0.58 | 97.56±0.51 | 97.13±0.62 | 97.05±0.45 | 97.69±0.40 | 98.97±0.28 | 99.01±0.31 | 99.08±0.23 | 99.15±0.17 | **99.16**±0.16 | 99.08±0.22 |
| campaign | 48.48±0.00 | 49.05±0.00 | 49.77±0.00 | 49.77±0.00 | 49.51±0.00 | 49.31±0.00 | 49.04±0.00 | 49.89±0.00 | **50.45**±0.00 | 49.47±0.00 | 49.33±0.00 | 49.64±0.00 |
| cardio | 76.90±0.00 | 77.73±0.00 | 78.30±0.00 | 79.19±0.00 | **79.53**±0.00 | 78.33±0.00 | 77.22±0.00 | 78.33±0.00 | 79.14±0.00 | 80.67±0.00 | **81.15**±0.00 | 79.30±0.00 |
| cardiotocography | 56.55±0.00 | 57.18±0.00 | 58.19±0.00 | 59.42±0.00 | **59.95**±0.00 | 58.26±0.00 | 57.43±0.00 | 58.37±0.00 | 59.44±0.00 | 61.41±0.00 | **62.19**±0.00 | 59.77±0.00 |
| celeba | 10.56±0.44 | 11.63±0.49 | 12.74±0.52 | 13.92±0.58 | **14.30**±0.59 | 12.63±0.51 | 11.99±0.57 | 13.26±0.61 | 14.50±0.58 | 15.70±0.65 | **16.10**±0.68 | 14.31±0.60 |
| census | 21.14±0.39 | **21.38**±0.54 | 21.16±0.43 | 20.67±0.41 | 20.52±0.43 | 20.97±0.43 | **21.36**±0.76 | 21.22±0.39 | 20.59±0.33 | 20.00±0.42 | 19.94±0.44 | 20.62±0.44 |
| cover | 63.67±3.21 | 57.85±3.52 | 51.55±3.10 | 44.58±2.49 | 42.11±2.29 | 51.95±2.90 | 55.15±3.45 | 48.67±2.84 | 41.44±2.04 | 33.72±1.51 | 31.69±1.35 | 42.14±2.20 |
| donors | 93.23±0.80 | 91.25±0.72 | 87.17±0.95 | 83.92±1.29 | 82.34±1.32 | 87.78±0.99 | 89.44±0.96 | 85.33±1.15 | 80.15±1.33 | 73.68±1.32 | 71.00±1.30 | 79.92±1.13 |
| fault | 62.03±0.00 | 61.58±0.00 | 61.29±0.00 | 61.98±0.00 | **62.31**±0.00 | 61.84±0.00 | 61.98±0.00 | 61.16±0.00 | 61.92±0.00 | 63.67±0.00 | **64.06**±0.00 | 62.56±0.00 |
| fraud | 40.60±6.67 | **43.77**±5.38 | 43.03±4.92 | 39.91±4.75 | 38.80±4.94 | 41.22±5.02 | 42.35±5.61 | **44.96**±3.90 | 41.19±3.73 | 37.33±4.15 | 36.42±4.12 | 40.45±4.07 |
| glass | 60.15±6.89 | 47.75±5.62 | 37.27±4.92 | 31.23±3.12 | 30.48±2.85 | 41.38±4.46 | 44.05±6.38 | 32.96±3.74 | 29.87±3.75 | 26.62±2.65 | 26.04±3.61 | 31.91±3.73 |
| hepatitis | 99.47±0.73 | 97.98±1.43 | 91.71±2.26 | 81.65±3.68 | 78.95±3.45 | 89.95±1.80 | 91.10±4.41 | 69.28±4.16 | 64.12±5.59 | 64.29±5.08 | 64.33±6.16 | 70.62±4.48 |
| http | 98.52±0.37 | 95.38±2.26 | 88.66±1.01 | 84.43±2.65 | 80.40±4.55 | 89.48±1.90 | **100.00**±0.00 | 98.01±3.98 | 91.24±1.42 | 91.44±1.25 | 91.28±1.36 | 94.39±1.31 |
| imdb | 9.11±0.00 | 9.09±0.00 | 9.06±0.00 | 9.07±0.00 | 9.06±0.00 | 9.08±0.00 | 8.92±0.00 | 8.94±0.00 | **8.99**±0.00 | 8.98±0.00 | 8.99±0.00 | 8.96±0.00 |
| internetads | 52.20±0.00 | 49.76±0.00 | 48.19±0.00 | 47.56±0.00 | 47.45±0.00 | 49.03±0.00 | **49.22**±0.00 | 47.29±0.00 | 46.93±0.00 | 46.95±0.00 | 46.94±0.00 | 47.47±0.00 |
| ionosphere | 98.72±0.48 | 98.46±0.54 | 98.27±0.42 | 97.44±0.50 | 96.93±0.61 | 97.96±0.46 | 97.86±0.60 | **98.11**±0.52 | 97.04±0.60 | 94.12±1.45 | 92.90±1.65 | 96.01±0.78 |
| landsat | 56.14±0.00 | 54.25±0.00 | 50.75±0.00 | 46.43±0.00 | 45.17±0.00 | 50.55±0.00 | **54.85**±0.00 | 50.62±0.00 | 45.18±0.00 | 41.32±0.00 | 40.50±0.00 | 46.49±0.00 |
| letter | 8.86±0.00 | 8.78±0.00 | 8.67±0.00 | 8.54±0.00 | 8.50±0.00 | 8.67±0.00 | **8.70**±0.00 | 8.58±0.00 | 8.41±0.00 | 8.27±0.00 | 8.22±0.00 | 8.44±0.00 |
| lymphography | 97.27±5.45 | 96.07±6.79 | 96.07±6.79 | 95.68±6.60 | 95.68±6.60 | 96.16±6.43 | **98.61**±1.02 | 98.43±0.52 | 98.43±0.52 | **98.70**±0.83 | 98.70±0.83 | 98.57±0.65 |
| magic.gamma | 86.30±0.00 | 85.86±0.00 | 85.28±0.00 | 84.56±0.00 | 84.29±0.00 | 85.26±0.00 | **85.86**±0.00 | 85.25±0.00 | 84.51±0.00 | 83.61±0.00 | 83.25±0.00 | 84.50±0.00 |
| mammography | 42.14±0.00 | 41.51±0.00 | 40.67±0.00 | 40.37±0.00 | 40.50±0.00 | 41.04±0.00 | **41.27**±0.00 | 40.55±0.00 | 40.24±0.00 | 38.97±0.00 | 38.10±0.00 | 39.83±0.00 |
| mnist | 74.43±0.00 | 73.09±0.00 | 71.84±0.00 | 70.69±0.00 | 70.36±0.00 | 72.08±0.00 | **72.72**±0.00 | 71.40±0.00 | 70.09±0.00 | 69.02±0.00 | 68.60±0.00 | 70.36±0.00 |
| musk | **100.00**±0.00 | 100.00±0.00 | 100.00±0.00 | 100.00±0.00 | 100.00±0.00 | 100.00±0.00 | **100.00**±0.00 | 100.00±0.00 | 100.00±0.00 | 100.00±0.00 | 100.00±0.00 | 100.00±0.00 |
| optdigits | 34.44±0.00 | 30.53±0.00 | 26.28±0.00 | 22.67±0.00 | 21.61±0.00 | 27.11±0.00 | **29.11**±0.00 | 24.76±0.00 | 21.10±0.00 | 17.68±0.00 | 16.62±0.00 | 21.85±0.00 |
| pageblocks | 62.78±0.00 | 62.52±0.00 | 62.20±0.00 | 61.02±0.00 | 60.30±0.00 | 61.76±0.00 | 67.60±0.00 | 67.74±0.00 | **67.87**±0.00 | 66.41±0.00 | 66.13±0.00 | 67.15±0.00 |
| pendigits | 97.68±0.00 | 97.31±0.00 | 96.28±0.00 | 90.01±0.00 | 86.69±0.00 | 93.59±0.00 | 96.99±0.00 | 95.65±0.00 | 81.40±0.00 | 70.28±0.00 | 67.39±0.00 | 82.34±0.00 |
| pima | 80.27±1.65 | 78.05±2.13 | 75.87±2.40 | 74.73±2.71 | 74.49±2.71 | 76.68±2.22 | 75.66±2.91 | 73.62±2.59 | 73.42±2.98 | 73.71±2.79 | 73.63±2.86 | 74.01±2.75 |
| satellite | 85.98±0.00 | 85.74±0.00 | 85.17±0.00 | 84.15±0.00 | 83.72±0.00 | 84.95±0.00 | **86.01**±0.00 | 85.31±0.00 | 84.02±0.00 | 82.19±0.00 | 81.56±0.00 | 83.82±0.00 |
| satimage-2 | 96.10±0.00 | 96.64±0.00 | 97.02±0.00 | 97.39±0.00 | **97.42**±0.00 | 96.92±0.00 | 96.69±0.00 | 97.21±0.00 | 97.39±0.00 | **97.42**±0.00 | 97.42±0.00 | 97.22±0.00 |
| shuttle | 99.16±0.00 | 98.76±0.00 | 98.72±0.00 | 98.78±0.00 | 98.77±0.00 | 98.84±0.00 | 97.86±0.00 | 97.34±0.00 | 97.28±0.00 | 97.22±0.00 | 97.20±0.00 | 97.38±0.00 |
| skin | 98.92±0.23 | 93.20±4.25 | 90.32±6.04 | 90.14±5.60 | 88.96±6.03 | 92.31±4.46 | 89.48±5.58 | 89.07±3.41 | 88.67±4.37 | 88.39±0.08 | 86.43±0.46 | 92.71±0.16 |
| smtp | 56.70±7.16 | 54.77±7.80 | 54.75±7.81 | 54.76±7.81 | 48.74±10.23 | 53.94±7.83 | 50.26±5.73 | 50.20±5.74 | 50.18±5.75 | 50.33±5.85 | 50.41±5.72 | 50.27±5.76 |
| spambase | 83.93±0.00 | 83.42±0.00 | 83.03±0.00 | 82.73±0.00 | 82.63±0.00 | 83.15±0.00 | **83.32**±0.00 | 82.70±0.00 | 82.41±0.00 | 82.17±0.00 | 82.11±0.00 | 82.54±0.00 |
| speech | 3.02±0.00 | 2.89±0.00 | 2.70±0.00 | 2.76±0.00 | 2.70±0.00 | 2.82±0.00 | **2.80**±0.00 | 2.73±0.00 | 2.74±0.00 | 2.76±0.00 | 2.74±0.00 | 2.75±0.00 |
| stamps | 82.50±3.71 | 77.11±4.30 | 69.99±5.29 | 65.85±6.16 | 64.64±6.16 | 72.02±4.82 | 73.26±7.70 | 65.58±7.71 | 63.12±6.69 | 62.09±7.20 | 61.57±7.18 | 65.12±7.10 |
| thyroid | 77.22±0.00 | **77.53**±0.00 | 77.26±0.00 | 76.43±0.00 | 74.75±0.00 | 76.64±0.00 | 80.94±0.00 | 81.09±0.00 | 81.50±0.00 | 81.90±0.00 | **81.93**±0.00 | 81.47±0.00 |
| vertebral | 43.77±5.50 | 31.72±3.49 | 24.99±2.97 | 21.10±2.37 | 20.29±2.40 | 28.38±3.31 | 25.07±3.19 | 21.55±2.53 | 19.65±2.31 | 18.09±1.91 | 17.76±2.02 | 20.42±2.34 |
| vowels | 31.68±0.00 | 30.30±0.00 | 29.54±0.00 | 27.85±0.00 | 27.32±0.00 | 29.34±0.00 | **30.21**±0.00 | 28.75±0.00 | 27.41±0.00 | 24.27±0.00 | 22.44±0.00 | 26.62±0.00 |
| waveform | 26.96±0.00 | 26.71±0.00 | 25.68±0.00 | 24.67±0.00 | 24.49±0.00 | 25.70±0.00 | **27.00**±0.00 | 25.82±0.00 | 23.77±0.00 | 24.13±0.00 | 23.87±0.00 | 24.92±0.00 |
| wbc | 96.59±2.18 | 89.03±6.77 | 86.03±5.81 | 83.79±5.59 | 83.41±5.19 | 86.87±5.84 | 85.35±5.46 | 83.72±5.56 | 82.05±5.59 | 82.37±4.91 | 82.33±4.34 | 83.17±5.03 |
| wdbc | 92.08±6.52 | 89.03±6.77 | 86.03±5.81 | 83.79±5.59 | 83.41±5.19 | 91.84±4.57 | 98.31±0.34 | 96.30±0.30 | 94.09±0.51 | 88.39±0.08 | 86.43±0.46 | 90.15±3.97 |
| wilt | 13.43±0.00 | 12.36±0.00 | 11.30±0.00 | 10.40±0.00 | 10.12±0.00 | 11.52±0.00 | **12.25**±0.00 | 11.04±0.00 | 10.11±0.00 | 9.33±0.00 | 9.09±0.00 | 10.36±0.00 |
| wine | 99.42±0.77 | 98.32±0.52 | 96.09±1.31 | 93.12±1.51 | 91.59±2.00 | 95.71±0.92 | **95.18**±1.66 | 88.85±3.38 | 88.36±2.97 | 85.43±3.02 | 85.79±1.56 | 88.72±2.32 |
| wpbc | 75.30±1.88 | 61.19±1.49 | 51.53±2.17 | 46.62±2.23 | 45.63±1.44 | 56.05±1.44 | **47.07**±2.57 | 43.16±2.41 | 42.43±2.53 | 42.28±2.60 | 42.11±2.47 | 43.41±2.44 |
| yeast | 48.37±0.00 | 47.91±0.00 | 47.52±0.00 | 47.26±0.00 | 47.20±0.00 | 47.65±0.00 | **48.26**±0.00 | 47.48±0.00 | 47.24±0.00 | 46.74±0.00 | 46.48±0.00 | 47.24±0.00 |
| yelp | 16.05±0.00 | 15.78±0.00 | 15.40±0.00 | 15.01±0.00 | 14.89±0.00 | 15.43±0.00 | **16.03**±0.00 | 15.63±0.00 | 15.17±0.00 | 14.77±0.00 | 14.66±0.00 | 15.25±0.00 |
| MNIST-C | 47.21±0.00 | 46.26±0.00 | 45.35±0.00 | 44.50±0.00 | 44.24±0.00 | 45.51±0.00 | **46.20**±0.00 | 45.18±0.00 | 44.32±0.00 | 43.50±0.00 | 43.23±0.00 | 44.49±0.00 |
| FashionMNIST | 59.52±0.00 | 59.05±0.00 | 58.57±0.00 | 58.09±0.00 | 57.96±0.00 | 58.64±0.00 | **59.15**±0.00 | 58.64±0.00 | 58.19±0.00 | 57.74±0.00 | 57.60±0.00 | 58.26±0.00 |
| CIFAR10 | 19.77±0.00 | 19.59±0.00 | 19.43±0.00 | 19.31±0.00 | 19.28±0.00 | 19.48±0.00 | **19.62**±0.00 | 19.50±0.00 | 19.37±0.00 | 19.27±0.00 | 19.24±0.00 | 19.40±0.00 |
| SVHN | 15.44±0.00 | 15.30±0.00 | 15.18±0.00 | 15.08±0.00 | 15.05±0.00 | 15.21±0.00 | **15.34**±0.00 | 15.22±0.00 | 15.11±0.00 | 15.02±0.00 | 14.99±0.00 | 15.13±0.00 |
| MVTec-AD | 82.66±1.11 | 79.02±0.97 | 76.24±0.90 | 74.41±0.89 | 73.96±0.86 | 77.26±0.94 | 75.38±0.86 | 73.88±0.83 | 73.13±0.87 | 72.56±0.79 | 72.46±0.78 | 73.48±0.82 |
| 20news | 15.45±1.12 | 14.32±1.09 | 13.25±0.77 | 12.71±0.65 | 12.59±0.66 | 13.66±0.85 | **13.76**±0.93 | 12.83±0.58 | 12.50±0.63 | 12.15±0.62 | 11.95±0.62 | 12.64±0.66 |
| agnews | 17.03±0.00 | 16.50±0.00 | 15.92±0.00 | 15.29±0.00 | **15.07**±0.00 | 15.96±0.00 | **16.68**±0.00 | 15.98±0.00 | 15.35±0.00 | 14.71±0.00 | 14.50±0.00 | 15.45±0.00 |

Table 10.2: Average AUPR ± standard dev. over five seeds for the semi-supervised setting of ICL and DTE-C baselines with varying hyperparameter (HP) values; For ICL, the learning rate ∈ {0.1, 0.02, 0.001, 0.0001, 1e − 05}, for DTE-C, $k$ ∈ {5, 10, 20, 40, 50}. Also reported is the $^{avg}$ model. We use **bold** and underline respectively to mark the **best** and the worst performance of each model to showcase the variability of performance across different HP settings.

| dataset | ICL-0.1 | ICL-0.01 | ICL-0.001 | ICL-0.0001 | ICL-1e-05 | ICL-avr | DTE-C-5 | DTE-C-10 | DTE-C-20 | DTE-C-40 | DTE-C-50 | DTE-C-avr |
|---|---|---|---|---|---|---|---|---|---|---|---|---|
| aloi | 5.50±0.09 | 5.39±0.07 | 5.50±0.08 | **5.59±0.04** | 5.46±0.01 | 5.49±0.02 | 5.76±0.03 | 5.82±0.02 | _5.72±0.05_ | 5.73±0.05 | **5.91±0.00** | 5.79±0.01 |
| amazon | 10.06±0.10 | **10.08±0.03** | 9.91±0.13 | 10.01±0.05 | 9.99±0.02 | 10.01±0.05 | 11.01±0.43 | 10.99±1.11 | **11.05±0.93** | 9.52±0.00 | 9.52±0.00 | 10.42±0.40 |
| annthyroid | **58.53±13.33** | 39.69±5.23 | 53.66±3.08 | 53.85±5.44 | 55.94±2.70 | 52.34±3.72 | 82.46±6.50 | 83.25±0.34 | **83.27±0.63** | 81.52±1.18 | _13.81±0.00_ | 68.86±0.20 |
| backdoor | 85.81±2.45 | 88.14±1.38 | **88.99±1.12** | 88.96±1.20 | 86.24±0.75 | 87.63±1.03 | 43.90±3.90 | 61.21±2.58 | **63.64±1.61** | _4.83±0.09_ | _4.83±0.09_ | 35.68±0.79 |
| breastw | 98.65±0.81 | 98.46±0.51 | **98.98±0.57** | 98.50±0.42 | 95.32±1.11 | 97.98±0.31 | 88.69±2.54 | 89.49±2.00 | 94.04±1.49 | 98.56±0.68 | **99.22±0.11** | 94.00±0.69 |
| campaign | 47.46±0.41 | 44.03±2.05 | 47.94±0.25 | 49.22±0.91 | **51.41±0.51** | 48.01±0.47 | 49.90±2.01 | 46.77±1.41 | 48.40±1.60 | 17.78±0.59 | 20.25±0.00 | 37.11±0.64 |
| cardio | 48.81±14.30 | **65.70±11.26** | 63.55±5.07 | 61.75±2.55 | 29.67±1.62 | 53.90±3.02 | 69.32±0.43 | 70.19±0.63 | 70.25±0.47 | 36.12±0.59 | 17.55±0.00 | 49.02±0.29 |
| cardiotocography | 43.21±5.03 | **52.56±1.14** | 50.69±2.12 | 49.01±1.29 | 39.79±1.23 | 47.05±1.34 | 53.83±0.94 | 53.56±0.73 | 49.12±4.17 | 4.31±0.09 | 36.12±0.00 | 45.75±0.91 |
| celeba | 12.89±1.17 | **13.90±1.49** | 13.09±1.73 | 13.50±1.15 | 12.97±0.55 | 13.27±0.47 | 15.16±1.05 | 13.87±0.59 | 12.60±1.22 | 11.66±0.20 | 4.31±0.09 | 10.05±0.49 |
| census | 20.29±1.27 | 20.38±2.24 | 23.32±0.50 | **23.68±0.70** | 22.37±0.59 | 22.01±0.42 | 18.39±0.83 | 17.28±0.52 | 53.74±2.07 | 1.94±0.07 | 11.66±0.20 | 15.29±0.31 |
| cover | 22.58±13.63 | 35.90±22.12 | 35.90±22.12 | **47.50±19.76** | 40.81±22.18 | 31.42±3.97 | 71.28±4.54 | 61.74±4.63 | 63.62±1.06 | 18.86±15.32 | 1.94±0.07 | 35.03±0.82 |
| donors | 39.48±10.00 | 77.81±8.28 | 82.59±7.03 | **92.24±2.17** | 91.01±0.36 | 76.75±3.27 | 76.07±1.84 | 66.66±3.73 | 24.97±2.19 | 51.69±0.25 | 11.20±0.14 | 45.30±2.85 |
| fault | **65.37±3.40** | 64.58±1.81 | 63.83±0.74 | 64.32±0.96 | 63.54±1.39 | 64.41±1.09 | 63.92±1.40 | 62.97±0.60 | 27.63±1.55 | 0.34±0.03 | 51.74±0.32 | 58.79±0.41 |
| fraud | 51.45±12.34 | 49.97±9.27 | 56.24±9.47 | 72.24±6.51 | **87.79±8.75** | 58.46±4.58 | **68.85±0.35** | 57.17±4.68 | 4.76±0.13 | 20.53±3.60 | 0.34±0.03 | 30.33±4.25 |
| glass | 49.70±15.55 | 69.98±13.15 | 88.19±6.00 | 87.25±7.44 | **99.93±0.14** | 76.58±1.38 | 46.09±0.97 | 36.80±5.13 | 50.00±7.08 | 54.16±11.62 | 7.83±1.03 | 27.78±2.62 |
| hepatitis | **99.85±0.30** | 97.89±3.11 | 99.79±0.41 | 98.94±2.13 | 99.55±0.61 | 99.28±1.20 | **98.64±1.76** | 95.49±4.81 | 85.00±3.00 | 48.53±7.22 | 27.45±1.71 | 74.38±1.79 |
| http | 94.85±9.03 | 95.76±3.25 | 97.86±1.72 | **99.77±0.25** | 99.72±0.12 | 97.56±2.20 | 56.93±7.39 | 69.18±22.25 | 89.03±4.93 | 9.52±0.00 | 0.74±0.04 | 45.08±7.07 |
| imdb | 10.09±0.09 | 10.02±0.07 | 10.16±0.04 | 10.18±0.05 | **10.41±0.04** | 10.17±0.03 | 8.71±0.63 | 9.49±0.47 | 8.79±0.30 | 31.53±0.00 | 9.52±0.00 | 9.21±0.20 |
| internetads | 57.32±2.19 | 60.94±1.44 | 62.86±1.88 | **63.83±0.88** | 47.64±2.20 | 58.52±1.04 | 60.45±4.51 | 56.24±2.74 | 38.26±3.29 | 86.50±4.57 | 31.53±0.00 | 47.51±1.92 |
| ionosphere | 96.94±2.27 | 97.20±2.57 | **99.00±0.36** | 98.91±0.48 | 97.54±1.39 | 97.92±1.11 | 96.52±0.77 | 96.49±1.17 | 83.45±12.01 | 34.27±0.09 | **56.41±15.94** | 86.53±4.28 |
| landsat | **58.66±1.95** | 56.49±1.07 | 55.72±1.20 | 55.69±0.77 | 52.73±0.98 | 55.86±0.60 | 36.22±0.76 | 36.06±2.33 | 89.15±0.44 | 11.76±0.00 | 34.32±0.00 | 34.98±0.81 |
| letter | 11.22±1.58 | 10.84±0.96 | 9.41±0.34 | 9.71±0.55 | **15.87±1.71** | 11.41±0.43 | 8.92±0.10 | 9.01±0.23 | 32.22±4.57 | 72.50±16.37 | 11.76±0.00 | 10.09±0.04 |
| lymphography | 100.00±0.00 | 100.00±0.00 | 100.00±0.00 | 100.00±0.00 | 100.00±0.00 | 100.00±0.00 | 87.03±7.20 | **93.30±2.48** | 55.17±3.05 | 66.79±18.08 | 7.80±0.50 | 68.82±5.96 |
| magic.gamma | 73.92±3.66 | 81.23±2.83 | 82.64±0.71 | 82.47±0.98 | **83.45±0.84** | 80.74±1.03 | 89.01±0.45 | **89.46±0.29** | 100.00±0.00 | 35.30±2.06 | 52.03±0.00 | 77.29±3.52 |
| mammography | 33.51±8.47 | **37.72±7.54** | 27.06±2.69 | 26.94±1.92 | 17.30±1.13 | 28.51±2.57 | 35.02±0.35 | 37.47±2.68 | 12.82±3.07 | 16.86±0.00 | 4.54±0.00 | 28.91±1.92 |
| mnist | 49.27±1.39 | 50.40±3.02 | 54.46±1.34 | 63.49±1.45 | **65.20±1.41** | 56.56±0.65 | 60.64±1.13 | 56.20±2.31 | 63.78±4.40 | 6.14±0.00 | 16.86±0.00 | 41.15±1.11 |
| musk | 100.00±0.00 | 100.00±0.00 | 100.00±0.00 | 100.00±0.00 | 48.40±11.06 | 99.48±1.94 | 100.00±0.00 | 100.00±0.00 | 40.53±4.47 | 5.59±0.00 | 6.14±0.00 | 62.46±0.00 |
| optdigits | 27.13±3.47 | 36.36±3.03 | 38.29±3.65 | **61.20±0.68** | 42.39±5.22 | 41.08±1.34 | 18.29±2.20 | 11.55±3.63 | 66.04±3.31 | 59.09±4.25 | 5.59±0.00 | 10.77±1.02 |
| pageblocks | 64.80±3.36 | 66.26±4.89 | 63.86±2.44 | 61.26±3.91 | **68.37±1.14** | 64.91±1.13 | 64.41±1.43 | **68.08±1.72** | 84.72±0.41 | 4.44±0.00 | 17.28±0.00 | 54.53±1.01 |
| pendigits | 39.64±20.69 | 66.20±12.32 | 63.38±8.59 | **73.08±7.23** | 50.36±5.65 | 58.53±6.17 | 53.63±4.87 | 45.92±5.63 | 46.25±5.02 | 70.01±2.92 | 4.44±0.00 | 29.79±0.96 |
| pima | 74.29±3.85 | 71.44±2.85 | **78.53±2.86** | 76.72±1.68 | 75.58±3.60 | 75.31±2.15 | 67.82±2.03 | 65.53±3.71 | 93.19±9.08 | 55.55±14.95 | **68.66±8.30** | 67.61±2.81 |
| satellite | 76.76±8.77 | **86.99±1.74** | 86.43±0.25 | 90.38±0.44 | 50.77±13.44 | 85.84±2.16 | 85.28±0.60 | 85.11±0.52 | 41.15±7.61 | 24.56±27.32 | 48.08±0.00 | 71.75±2.86 |
| satimage-2 | 37.06±8.31 | 77.53±35.36 | **96.77±0.59** | 95.65±0.53 | 22.16±3.78 | 71.56±7.98 | 83.45±5.57 | 62.80±5.31 | 84.12±0.39 | 88.77±1.78 | 2.42±0.00 | 43.90±6.54 |
| shuttle | **97.77±1.43** | 99.19±0.23 | 99.82±0.12 | **99.91±0.05** | 99.72±0.12 | 99.28±0.28 | 94.26±0.06 | 94.06±0.16 | 2.85±0.16 | 69.53±0.66 | 13.35±0.00 | 76.73±0.36 |
| skin | 44.77±20.35 | **73.00±9.87** | 68.00±9.89 | 37.40±6.86 | 65.20±1.41 | 56.71±7.09 | 68.88±0.57 | 70.08±0.63 | 53.34±4.09 | 15.36±9.94 | 34.42±0.19 | 62.54±0.43 |
| smtp | 38.06±20.67 | **42.88±7.84** | 38.77±14.99 | 85.42±0.58 | 36.22±4.31 | 38.67±3.38 | 50.08±8.85 | 52.14±8.82 | 82.86±1.27 | 57.05±0.00 | 57.05±0.00 | 31.76±4.31 |
| spambase | 80.80±1.90 | 81.18±2.88 | 85.16±0.70 | **85.42±0.58** | 82.51±0.72 | 83.01±0.50 | 83.44±0.47 | 83.59±0.51 | 28.95±5.65 | 3.26±0.00 | 3.26±0.00 | 73.05±0.13 |
| speech | 3.74±1.02 | 3.79±0.38 | 3.92±0.39 | 3.46±0.20 | 3.41±0.23 | 3.66±0.23 | 2.78±0.12 | 3.32±0.58 | 44.91±4.07 | 56.82±10.61 | **24.85±19.23** | 3.09±0.15 |
| stamps | 45.19±8.30 | 52.05±12.46 | **80.41±3.57** | 80.49±4.25 | 74.72±8.62 | 66.57±4.58 | 61.33±7.30 | 57.64±7.65 | 10.41±0.55 | **83.14±1.08** | 4.81±0.00 | 50.80±7.62 |
| thyroid | **72.79±3.97** | 56.56±5.89 | 56.38±5.72 | 60.13±6.20 | 30.60±3.56 | 55.29±1.31 | 80.34±1.28 | 81.93±2.54 | 40.09±7.65 | 25.00±4.75 | 21.17±4.01 | 66.61±0.60 |
| vertebral | 25.76±4.65 | 31.37±8.67 | 50.83±13.40 | **59.35±3.61** | 54.90±6.08 | 44.44±4.60 | 34.31±5.07 | 31.45±6.44 | 76.35±9.33 | 25.00±4.75 | 25.00±0.00 | 28.18±4.88 |
| vowels | 24.00±15.67 | 21.57±7.57 | 28.45±5.52 | 28.35±6.64 | 22.16±3.78 | 24.91±3.58 | 39.58±4.05 | **41.45±3.27** | 98.70±1.77 | 5.65±0.00 | 5.65±0.00 | 27.84±1.66 |
| waveform | **51.44±2.31** | 34.95±18.32 | 28.41±9.48 | 9.51±0.97 | 20.22±2.30 | 28.91±5.68 | 9.88±0.96 | 9.59±0.83 | 59.53±3.80 | **67.56±7.06** | 8.72±1.13 | 8.24±0.18 |
| wbc | **82.98±11.29** | 91.66±5.01 | 99.04±1.73 | 99.16±1.35 | 96.58±2.18 | 93.88±3.76 | 36.71±3.31 | 73.63±7.98 | 27.94±1.52 | 15.20±19.59 | 5.21±0.94 | 37.32±1.43 |
| wdbc | 66.53±18.38 | 85.94±10.45 | 89.36±6.27 | **92.02±6.00** | 89.28±3.59 | 84.63±3.40 | 76.31±10.20 | 33.53±4.49 | 76.35±9.33 | 25.25±1.80 | 10.13±0.00 | 49.34±5.19 |
| wilt | 11.08±3.73 | 12.91±2.67 | 32.19±7.32 | **97.94±4.12** | 98.11±3.43 | 23.88±1.71 | 25.16±0.41 | 21.04±1.52 | 98.70±1.77 | 45.36±23.65 | 14.95±1.08 | 71.37±4.46 |
| wine | 98.06±3.88 | 98.33±1.88 | 99.48±0.75 | 85.99±2.53 | 83.48±3.70 | 38.38±2.75 | **99.47±0.66** | 98.39±1.77 | 56.31±0.95 | 40.41±3.04 | 38.36±1.82 | 52.00±2.61 |
| wpbc | 82.28±2.27 | 66.72±11.51 | 85.45±3.99 | **85.99±2.53** | 47.21±0.41 | 80.78±4.52 | 60.51±5.10 | 61.17±3.87 | 19.79±0.18 | 47.04±1.19 | **50.90±0.00** | 49.58±0.27 |
| yeast | 48.32±2.44 | 49.17±1.53 | 49.01±0.03 | **49.32±0.70** | 9.84±0.06 | 48.61±0.81 | 49.65±0.87 | 50.38±1.14 | 15.55±0.11 | 9.52±0.00 | 9.52±0.00 | 11.86±0.62 |
| yelp | 9.79±0.11 | **9.91±0.05** | 9.82±0.17 | 9.84±0.06 | 9.59±0.03 | 9.79±0.05 | 14.09±1.54 | 12.96±1.75 | 13.22±0.32 | 9.50±0.00 | 9.50±0.00 | 31.85±0.20 |
| MNIST-C | 44.47±0.54 | 48.19±0.57 | 50.18±0.26 | **50.34±0.31** | 45.98±0.37 | 47.83±0.12 | 46.68±0.62 | 47.39±0.52 | 46.12±0.32 | 9.52±0.00 | 9.52±0.00 | 31.85±0.17 |
| FashionMNIST | 58.47±0.45 | 64.11±0.18 | 64.96±0.32 | **65.22±0.32** | 55.84±0.54 | 61.72±0.18 | 54.92±0.36 | 56.00±0.23 | 56.31±0.95 | 9.52±0.00 | 9.52±0.00 | 37.25±0.17 |
| CIFAR10 | 14.30±0.27 | 17.49±0.41 | **19.10±0.21** | 18.89±0.21 | 13.77±0.14 | 16.71±0.06 | 13.77±0.14 | 19.61±0.15 | 19.79±0.18 | 9.52±0.00 | 9.52±0.00 | 15.68±0.07 |
| SVHN | 13.92±0.24 | 15.32±0.05 | **15.82±0.01** | 15.74±0.09 | 15.01±0.13 | 15.16±0.06 | **19.95±0.09** | 15.62±0.19 | 15.55±0.11 | 9.52±0.00 | 9.52±0.00 | 13.16±0.04 |
| MVTec-AD | 86.94±0.92 | 88.30±0.67 | 88.91±0.75 | 88.97±0.75 | 83.64±0.91 | 87.35±0.76 | 83.04±1.01 | **84.66±1.01** | 84.41±1.22 | 38.89±1.52 | 38.21±0.62 | 65.84±0.58 |
| 20news | 12.37±0.20 | 13.20±0.34 | 13.52±0.34 | **13.98±0.35** | 11.83±0.29 | 12.98±0.23 | 17.28±1.40 | 15.75±0.40 | 14.55±0.94 | 9.44±0.39 | 9.44±0.39 | 13.29±0.47 |
| agnews | _12.45±0.12_ | 12.84±0.25 | 13.04±0.08 | **13.05±0.04** | 12.55±0.05 | 12.79±0.06 | 18.40±0.64 | 16.51±0.99 | 16.18±1.53 | 9.52±0.00 | 9.52±0.00 | 14.03±0.43 |

Table 11.1: Average F1 score ± standard dev. over five seeds for the semi-supervised setting of DTE-NP, $k$NN baselines with varying hyperparameter (HP) values; $k \in \{5, 10, 20, 40, 50\}$. Also reported is the $^{avg}$ model. We use **bold** and underline respectively to mark the **best** and the worst performance of each model to showcase the variability of performance across different HP settings.

| dataset | DTE-NP-5 | DTE-NP-10 | DTE-NP-20 | DTE-NP-40 | DTE-NP-50 | DTE-NP-avr | KNN-5 | KNN-10 | KNN-20 | KNN-40 | KNN-50 | KNN-avr |
|---|---|---|---|---|---|---|---|---|---|---|---|---|
| aloi | 5.90±0.00 | 5.84±0.00 | 5.20±0.00 | 5.90±0.00 | 5.97±0.00 | 5.86±0.00 | 5.90±0.00 | 5.64±0.00 | 5.97±0.00 | 6.17±0.00 | 6.37±0.00 | 6.01±0.00 |
| amazon | 10.80±0.00 | 10.80±0.00 | 10.20±0.00 | 10.20±0.00 | 11.00±0.00 | 10.60±0.00 | 11.40±0.00 | 10.20±0.00 | 10.60±0.00 | 11.20±0.00 | 11.20±0.00 | 10.92±0.00 |
| annthyroid | 62.55±0.00 | 61.80±0.00 | 60.67±0.00 | 58.99±0.00 | 58.80±0.00 | 60.56±0.00 | 61.99±0.00 | 60.49±0.00 | 58.43±0.00 | 58.24±0.00 | 56.74±0.00 | 59.18±0.00 |
| backdoor | 64.15±1.04 | 52.30±1.87 | 40.62±1.46 | 30.25±1.34 | 26.96±1.20 | 42.86±1.32 | 52.53±1.63 | 40.37±2.04 | 28.71±1.50 | 20.21±0.78 | 17.52±0.83 | 31.87±1.22 |
| breastw | 96.72±0.64 | 96.23±0.39 | 96.17±0.47 | 95.99±0.39 | 95.99±0.39 | 96.22±0.43 | 96.00±0.44 | 96.05±0.33 | 95.87±0.28 | 95.99±0.39 | 95.93±0.32 | 95.97±0.31 |
| campaign | 49.94±0.00 | 50.62±0.00 | 51.14±0.00 | 51.38±0.00 | 51.57±0.00 | 50.93±0.00 | 50.37±0.00 | 50.86±0.00 | 51.27±0.00 | 51.70±0.00 | 51.29±0.00 | 51.10±0.00 |
| cardio | 63.64±0.00 | 61.36±0.00 | 61.93±0.00 | 63.64±0.00 | 64.20±0.00 | 62.95±0.00 | 61.93±0.00 | 61.93±0.00 | 64.20±0.00 | 67.61±0.00 | 69.32±0.00 | 65.00±0.00 |
| cardiotocography | 44.64±0.00 | 45.71±0.00 | 47.00±0.00 | 48.50±0.00 | 49.14±0.00 | 47.00±0.00 | 46.35±0.00 | 46.78±0.00 | 47.85±0.00 | 50.86±0.00 | 51.93±0.00 | 48.76±0.00 |
| celeba | 15.83±0.69 | 17.05±0.43 | 18.17±0.61 | 19.02±0.69 | 19.30±0.60 | 17.87±0.57 | 17.08±0.58 | 18.41±0.65 | 19.30±0.81 | 20.27±0.68 | 20.48±0.68 | 19.11±0.61 |
| census | 22.22±0.54 | 21.93±0.52 | 21.46±0.25 | 21.38±0.48 | 21.12±0.29 | 21.62±0.29 | 22.23±0.42 | 21.48±0.40 | 21.47±0.57 | 21.33±0.65 | 21.26±0.50 | 21.55±0.24 |
| cover | 69.15±2.12 | 66.87±2.35 | 63.15±2.07 | 55.99±2.08 | 53.06±2.23 | 61.65±2.14 | 65.04±1.92 | 60.56±2.04 | 52.76±1.83 | 42.69±1.94 | 39.92±1.99 | 52.19±1.92 |
| donors | 97.27±0.36 | 96.20±0.45 | 94.49±0.55 | 91.70±0.90 | 90.57±0.86 | 94.05±0.60 | 94.98±0.60 | 92.36±0.59 | 88.71±0.99 | 80.36±1.90 | 76.62±1.50 | 86.60±0.98 |
| fault | 56.02±0.00 | 55.72±0.00 | 55.57±0.00 | 56.91±0.00 | 57.34±0.00 | 56.32±0.00 | 55.57±0.00 | 55.87±0.00 | 57.06±0.00 | 57.50±0.00 | 58.25±0.00 | 56.85±0.00 |
| fraud | 48.18±4.56 | 49.60±3.28 | 49.00±3.95 | 46.66±3.52 | 45.46±3.60 | 47.78±3.53 | 47.78±3.09 | 49.39±5.24 | 46.64±4.18 | 42.58±3.16 | 41.64±3.36 | 45.61±3.48 |
| glass | 47.81±5.78 | 35.14±2.75 | 27.98±4.21 | 18.37±2.40 | 17.81±2.98 | 29.42±2.61 | 29.87±9.62 | 22.55±6.87 | 18.58±4.19 | 17.23±3.73 | 17.23±3.73 | 21.09±5.16 |
| hepatitis | 98.94±1.41 | 94.16±2.13 | 81.68±4.18 | 75.95±4.78 | 71.99±4.14 | 84.54±2.23 | 81.98±4.50 | 66.04±4.53 | 62.31±6.09 | 60.39±6.21 | 60.04±7.30 | 66.15±4.71 |
| http | 98.50±0.38 | 95.10±2.50 | 95.10±2.50 | 82.85±3.80 | 78.96±6.40 | 88.73±2.54 | 100.00±0.00 | 98.57±2.86 | 98.57±2.86 | 92.67±0.91 | 92.67±0.91 | 95.32±0.75 |
| imdb | 5.20±0.00 | 5.40±0.00 | 5.40±0.00 | 5.20±0.00 | 5.20±0.00 | 5.28±0.00 | 5.40±0.00 | 5.40±0.00 | 5.40±0.00 | 5.00±0.00 | 5.00±0.00 | 5.24±0.00 |
| internetads | 55.16±0.00 | 51.63±0.00 | 48.37±0.00 | 46.47±0.00 | 46.20±0.00 | 49.57±0.00 | 51.90±0.00 | 46.20±0.00 | 45.11±0.00 | 45.11±0.00 | 45.11±0.00 | 46.68±0.00 |
| ionosphere | 92.33±1.17 | 92.05±1.63 | 91.63±1.09 | 90.41±1.35 | 89.19±1.72 | 91.12±1.24 | 90.23±1.86 | 91.81±1.87 | 89.45±1.91 | 85.06±3.37 | 82.75±3.12 | 87.86±1.86 |
| landsat | 52.29±0.00 | 51.24±0.00 | 49.06±0.00 | 45.99±0.00 | 45.39±0.00 | 48.79±0.00 | 51.46±0.00 | 49.29±0.00 | 45.76±0.00 | 42.69±0.00 | 41.34±0.00 | 46.11±0.00 |
| letter | 1.00±0.00 | 1.00±0.00 | 1.00±0.00 | 1.00±0.00 | 1.00±0.00 | 1.00±0.00 | 1.00±0.00 | 1.00±0.00 | 1.00±0.00 | 1.00±0.00 | 1.00±0.00 | 1.00±0.00 |
| lymphography | 97.89±4.21 | 93.61±7.97 | 94.67±6.53 | 92.71±6.09 | 91.66±7.34 | 94.11±5.93 | 91.57±5.96 | 90.69±3.65 | 90.69±3.65 | 92.96±4.97 | 92.96±4.97 | 91.77±3.69 |
| magic.gamma | 76.79±0.00 | 76.20±0.00 | 75.49±0.00 | 74.84±0.00 | 74.75±0.00 | 75.61±0.00 | 76.17±0.00 | 75.13±0.00 | 74.60±0.00 | 73.49±0.00 | 73.00±0.00 | 74.48±0.00 |
| mammography | 41.92±0.00 | 41.15±0.00 | 44.23±0.00 | 44.23±0.00 | 44.23±0.00 | 43.15±0.00 | 40.38±0.00 | 43.46±0.00 | 43.85±0.00 | 43.85±0.00 | 43.46±0.00 | 43.00±0.00 |
| mnist | 72.71±0.00 | 72.29±0.00 | 71.57±0.00 | 70.43±0.00 | 69.86±0.00 | 71.37±0.00 | 71.86±0.00 | 71.29±0.00 | 69.86±0.00 | 69.71±0.00 | 69.71±0.00 | 70.49±0.00 |
| musk | 100.00±0.00 | 100.00±0.00 | 100.00±0.00 | 100.00±0.00 | 100.00±0.00 | 100.00±0.00 | 100.00±0.00 | 100.00±0.00 | 100.00±0.00 | 100.00±0.00 | 100.00±0.00 | 100.00±0.00 |
| optdigits | 30.00±0.00 | 24.00±0.00 | 12.00±0.00 | 7.33±0.00 | 6.67±0.00 | 16.00±0.00 | 21.33±0.00 | 12.00±0.00 | 6.67±0.00 | 2.67±0.00 | 2.00±0.00 | 8.93±0.00 |
| pageblocks | 59.41±0.00 | 59.22±0.00 | 59.61±0.00 | 58.43±0.00 | 58.43±0.00 | 59.02±0.00 | 59.02±0.00 | 59.22±0.00 | 60.20±0.00 | 56.08±0.00 | 56.27±0.00 | 58.16±0.00 |
| pendigits | 94.23±0.00 | 92.31±0.00 | 91.03±0.00 | 80.13±0.00 | 78.21±0.00 | 87.18±0.00 | 90.38±0.00 | 90.38±0.00 | 73.72±0.00 | 64.74±0.00 | 62.82±0.00 | 76.41±0.00 |
| pima | 74.73±2.13 | 72.94±2.46 | 71.48±1.96 | 71.44±2.36 | 71.93±1.99 | 72.50±1.97 | 71.03±2.53 | 70.18±2.12 | 71.58±2.16 | 71.67±2.10 | 72.03±2.02 | 71.30±2.00 |
| satellite | 72.20±0.00 | 71.76±0.00 | 70.68±0.00 | 69.79±0.00 | 69.20±0.00 | 70.73±0.00 | 71.81±0.00 | 70.73±0.00 | 69.84±0.00 | 69.93±0.00 | 69.29±0.00 | 69.52±0.00 |
| satimage-2 | 90.14±0.00 | 90.14±0.00 | 90.14±0.00 | 92.96±0.00 | 92.96±0.00 | 91.27±0.00 | 90.14±0.00 | 91.55±0.00 | 92.96±0.00 | 92.96±0.00 | 92.96±0.00 | 92.11±0.00 |
| shuttle | 98.35±0.00 | 98.23±0.00 | 98.15±0.00 | 98.12±0.00 | 98.12±0.00 | 98.19±0.00 | 98.23±0.00 | 98.06±0.00 | 98.06±0.00 | 98.09±0.00 | 98.09±0.00 | 98.11±0.00 |
| skin | 97.12±0.46 | 96.83±0.38 | 96.14±0.17 | 94.73±0.23 | 94.30±0.27 | 95.82±0.14 | 96.71±0.41 | 95.85±0.17 | 95.85±0.17 | 91.73±0.16 | 90.60±0.28 | 93.89±0.13 |
| smtp | 68.05±5.12 | 68.05±5.12 | 68.05±5.35 | 69.59±3.95 | 63.34±8.31 | 67.73±4.99 | 83.65±4.53 | 84.82±3.86 | 69.59±3.95 | 79.49±5.02 | 78.80±0.00 | 69.59±3.95 |
| spambase | 80.88±0.00 | 80.29±0.00 | 80.23±0.00 | 79.81±0.00 | 79.45±0.00 | 80.13±0.00 | 80.52±0.00 | 79.93±0.00 | 79.15±0.00 | 78.98±0.00 | 79.48±0.00 | 79.48±0.00 |
| speech | 3.28±0.00 | 3.28±0.00 | 1.64±0.00 | 1.64±0.00 | 3.28±0.00 | 2.62±0.00 | 3.28±0.00 | 3.28±0.00 | 3.28±0.00 | 3.28±0.00 | 3.28±0.00 | 3.28±0.00 |
| stamps | 85.93±2.66 | 80.63±2.92 | 74.04±4.45 | 68.12±7.14 | 66.98±6.79 | 75.14±4.45 | 78.57±8.41 | 68.89±8.23 | 62.67±6.19 | 60.52±7.08 | 61.78±6.58 | 66.49±7.15 |
| thyroid | 75.27±0.00 | 75.27±0.00 | 74.19±0.00 | 74.19±0.00 | 74.19±0.00 | 74.62±0.00 | 75.27±0.00 | 73.12±0.00 | 72.04±0.00 | 73.12±0.00 | 74.19±0.00 | 73.55±0.00 |
| vertebral | 49.03±4.93 | 32.73±5.11 | 24.06±3.90 | 15.07±1.64 | 12.51±2.44 | 26.68±3.24 | 23.02±5.17 | 17.18±2.01 | 12.19±3.52 | 9.07±3.30 | 8.51±2.65 | 13.99±2.95 |
| vowels | 28.00±0.00 | 28.00±0.00 | 28.00±0.00 | 30.00±0.00 | 30.00±0.00 | 28.80±0.00 | 26.00±0.00 | 26.00±0.00 | 30.00±0.00 | 26.00±0.00 | 26.00±0.00 | 26.80±0.00 |
| waveform | 26.00±0.00 | 26.00±0.00 | 27.00±0.00 | 28.00±0.00 | 28.00±0.00 | 27.00±0.00 | 27.00±0.00 | 27.00±0.00 | 27.00±0.00 | 27.00±0.00 | 27.00±0.00 | 27.00±0.00 |
| wbc | 89.35±3.08 | 88.05±5.35 | 88.05±5.35 | 89.16±2.38 | 89.16±2.38 | 88.75±2.48 | 83.65±4.53 | 84.82±3.86 | 79.49±5.02 | 89.16±2.38 | 89.16±2.38 | 86.10±2.43 |
| wdbc | 85.05±6.11 | 83.36±7.16 | 81.00±0.00 | 81.00±4.08 | 81.00±4.08 | 82.28±4.91 | 80.92±4.47 | 78.72±5.61 | 78.70±0.00 | 77.95±6.52 | 76.84±4.81 | 78.79±5.12 |
| wilt | 3.50±0.00 | 2.33±0.00 | 1.56±0.00 | 0.78±0.00 | 0.78±0.00 | 1.79±0.00 | 2.33±0.00 | 1.56±0.00 | 0.78±0.00 | 0.78±0.00 | 0.78±0.00 | 1.25±0.00 |
| wine | 97.73±3.52 | 93.65±2.20 | 86.39±3.51 | 83.56±1.93 | 80.50±3.68 | 88.37±1.61 | 85.91±3.49 | 80.47±4.23 | 80.89±5.12 | 78.83±4.40 | 78.83±4.40 | 80.99±2.84 |
| wpbc | 75.11±2.68 | 63.70±1.39 | 53.31±2.33 | 45.02±2.91 | 43.40±2.66 | 56.11±1.23 | 50.17±2.76 | 42.79±2.99 | 40.15±1.26 | 37.87±3.42 | 39.00±3.24 | 41.99±2.22 |
| yeast | 45.76±0.00 | 46.35±0.00 | 46.35±0.00 | 46.75±0.00 | 46.75±0.00 | 46.39±0.00 | 46.75±0.00 | 45.96±0.00 | 47.14±0.00 | 46.75±0.00 | 46.94±0.00 | 46.71±0.00 |
| yelp | 19.40±0.00 | 18.80±0.00 | 17.40±0.00 | 16.40±0.00 | 16.60±0.00 | 17.72±0.00 | 18.80±0.00 | 17.60±0.00 | 17.40±0.00 | 16.20±0.00 | 16.80±0.00 | 17.36±0.00 |
| MNIST-C | 47.51±0.00 | 46.59±0.00 | 45.57±0.00 | 44.74±0.00 | 44.61±0.00 | 45.80±0.00 | 46.30±0.00 | 45.24±0.00 | 44.46±0.00 | 43.87±0.00 | 43.70±0.00 | 44.72±0.00 |
| FashionMNIST | 59.75±0.00 | 59.17±0.00 | 58.76±0.00 | 58.16±0.00 | 57.94±0.00 | 58.76±0.00 | 59.05±0.00 | 58.48±0.00 | 57.87±0.00 | 57.27±0.00 | 57.21±0.00 | 57.97±0.00 |
| CIFAR10 | 23.23±0.00 | 23.00±0.00 | 22.89±0.00 | 22.70±0.00 | 22.59±0.00 | 22.88±0.00 | 22.85±0.00 | 22.89±0.00 | 22.78±0.00 | 22.47±0.00 | 22.47±0.00 | 22.69±0.00 |
| SVHN | 19.19±0.00 | 18.99±0.00 | 18.90±0.00 | 18.80±0.00 | 18.58±0.00 | 18.89±0.00 | 18.95±0.00 | 18.79±0.00 | 18.70±0.00 | 18.42±0.00 | 18.38±0.00 | 18.65±0.00 |
| MVTec-AD | 75.68±0.99 | 71.31±0.96 | 68.50±0.99 | 66.57±0.72 | 66.05±0.61 | 69.62±0.83 | 67.06±1.03 | 65.80±0.80 | 65.00±0.73 | 64.20±0.60 | 64.09±0.54 | 65.23±0.73 |
| 20news | 18.08±1.41 | 16.56±1.63 | 14.56±1.60 | 12.95±1.47 | 12.86±1.43 | 15.00±1.49 | 15.11±1.90 | 13.63±1.52 | 12.70±1.43 | 11.85±1.43 | 10.81±1.54 | 12.82±1.49 |
| agnews | 20.80±0.00 | 20.30±0.00 | 19.60±0.00 | 18.35±0.00 | 17.95±0.00 | 19.40±0.00 | 20.10±0.00 | 19.25±0.00 | 18.10±0.00 | 16.85±0.00 | 16.40±0.00 | 18.14±0.00 |

Table 11.2: Average F1 score $\pm$ standard dev. over five seeds for the semi-supervised setting of ICL and DTE-C baselines with varying hyperparameter (HP) values; For ICL, the learning rate $\in \{0.1, 0.02, 0.001, 0.0001, 1e-05\}$, for DTE-C, $k \in \{5, 10, 20, 40, 50\}$. Also reported is the $^{\text{avg}}$ model. We use **bold** and underline respectively to mark the **best** and the worst performance of each model to showcase the variability of performance across different HP settings.

| dataset | ICL-0.1 | ICL-0.01 | ICL-0.001 | ICL-0.0001 | ICL-1e-05 | ICL-avr | DTE-C-5 | DTE-C-10 | DTE-C-20 | DTE-C-40 | DTE-C-50 | DTE-C-avr |
|---|---|---|---|---|---|---|---|---|---|---|---|---|
| aloi | 4.51±0.69 | 4.34±0.42 | **5.28±0.47** | 4.68±0.30 | 4.16±0.38 | 4.59±0.07 | **4.75±0.22** | 4.27±0.19 | 4.28±0.10 | 4.51±0.17 | 0.00±0.00 | 3.56±0.03 |
| amazon | **10.44±0.46** | 9.76±0.34 | 9.92±0.84 | 10.08±0.35 | 9.52±0.43 | 9.94±0.32 | 11.48±0.97 | **11.96±1.68** | 11.60±2.07 | 0.00±0.00 | 0.00±0.00 | 7.01±0.72 |
| annthyroid | 54.87±13.24 | 42.25±3.55 | 53.45±4.13 | 54.72±5.45 | **57.53±2.97** | 52.56±3.29 | 77.23±0.25 | **77.94±0.28** | 77.53±0.85 | 75.43±0.93 | 0.00±0.00 | 61.63±0.20 |
| backdoor | 87.17±0.98 | **87.32±0.99** | 87.11±1.09 | 86.85±0.95 | 85.37±1.01 | 86.76±1.00 | 46.19±8.39 | 83.03±2.14 | **84.50±0.60** | 0.00±0.00 | 0.00±0.00 | 42.75±1.54 |
| breastw | 95.98±0.34 | 96.07±0.94 | 96.80±0.40 | **97.44±0.55** | 96.11±0.75 | 96.48±0.28 | 88.50±1.59 | 90.10±1.35 | 92.46±1.78 | 95.31±0.70 | **96.11±0.44** | 92.50±0.79 |
| campaign | 48.12±0.36 | 46.81±1.72 | 50.68±0.66 | **51.37±0.85** | 50.07±0.49 | 50.07±0.49 | **52.45±1.07** | 52.33±1.00 | 58.07±0.23 | 0.34±0.68 | 0.00±0.00 | 31.35±0.44 |
| cardio | 49.09±11.28 | **61.93±5.57** | 58.86±1.59 | 57.95±2.30 | 40.57±4.28 | 53.68±2.83 | **58.30±0.58** | 57.84±0.43 | 58.07±0.23 | **62.02±0.00** | 62.02±0.00 | 34.91±0.09 |
| cardiotocography | 36.14±1.28 | 41.07±1.73 | 39.18±4.49 | 35.36±2.14 | 32.66±1.59 | 36.88±1.27 | 39.91±1.05 | 37.73±1.73 | 14.31±2.28 | 0.00±0.00 | 0.00±0.00 | 48.23±0.39 |
| celeba | 15.42±2.29 | 17.97±2.55 | 17.20±1.92 | 17.46±1.17 | 16.17±0.65 | 16.84±0.88 | 19.18±2.74 | 17.12±1.45 | 14.31±2.28 | 0.00±0.00 | 0.00±0.00 | 10.12±1.04 |
| census | 22.72±1.73 | 24.06±2.05 | 25.80±1.34 | 26.80±1.34 | 24.06±1.31 | 24.76±0.50 | 17.58±1.42 | 17.54±1.79 | 16.44±1.41 | 0.00±0.00 | 0.00±0.00 | 10.31±0.52 |
| cover | 26.77±15.24 | 15.53±8.17 | 42.68±17.00 | 53.70±16.88 | 44.34±20.24 | 36.61±2.59 | 76.51±2.37 | 68.92±4.22 | 46.54±3.93 | 0.00±0.00 | 0.00±0.00 | 38.39±0.62 |
| donors | 43.71±11.52 | 81.85±3.56 | 83.59±4.47 | 89.28±2.66 | 92.77±1.39 | 88.24±2.61 | 87.99±1.87 | 75.05±7.18 | 55.33±1.66 | 11.80±23.60 | 0.00±0.00 | 47.58±4.61 |
| fault | **60.33±3.36** | 59.52±2.09 | 58.57±1.11 | 58.37±0.79 | 58.87±1.51 | 59.13±1.36 | 55.57±1.57 | 55.16±0.81 | 55.33±1.66 | **97.03±0.00** | **97.03±0.00** | 72.00±0.47 |
| fraud | 57.54±10.13 | 48.97±8.02 | 58.90±6.77 | 66.88±4.88 | 79.18±3.21 | 62.30±4.91 | 75.61±7.76 | 54.25±5.90 | 22.55±24.73 | 0.00±0.00 | 0.00±0.00 | 30.48±4.66 |
| glass | 43.53±20.21 | 57.05±16.03 | 84.05±6.11 | 87.24±5.04 | 82.50±5.41 | 70.87±3.12 | 35.43±5.07 | 34.48±4.93 | 31.36±0.69 | 19.45±5.66 | 0.00±0.00 | 24.15±2.73 |
| hepatitis | 99.64±0.71 | 94.69±7.81 | 99.64±0.71 | 99.64±0.71 | 99.64±0.71 | 98.65±2.11 | 96.40±3.01 | 94.63±4.90 | 92.51±3.12 | 51.68±9.78 | 18.97±10.60 | 70.84±3.40 |
| http | 93.91±10.35 | 96.07±3.07 | 97.69±1.51 | 99.36±0.19 | 99.14±0.43 | 97.23±2.45 | 38.08±11.97 | 18.33±10.68 | 18.33±10.68 | 16.75±12.06 | 0.00±0.00 | 24.79±12.88 |
| imdb | 10.52±0.65 | 10.44±0.54 | 9.84±0.37 | 9.76±0.15 | 10.40±0.33 | 10.19±0.23 | 6.64±0.89 | 8.40±1.23 | 7.32±1.04 | 0.00±0.00 | 0.00±0.00 | 4.47±0.43 |
| internetads | 55.92±2.66 | 57.45±0.61 | 57.77±1.10 | 58.26±0.50 | 49.02±1.45 | 55.68±0.94 | 67.99±1.98 | 65.87±0.95 | 64.95±2.24 | 41.85±0.00 | 41.85±0.00 | 56.50±0.87 |
| ionosphere | 92.64±4.66 | 91.41±4.67 | 93.86±1.63 | 94.48±0.71 | 94.49±1.56 | 93.38±2.21 | 89.67±1.44 | 89.41±1.53 | 89.52±1.13 | 78.12±2.07 | 49.44±16.82 | 79.23±4.14 |
| landsat | 49.50±1.24 | 47.94±1.88 | 54.51±0.52 | 54.25±0.74 | 47.97±1.09 | 50.83±0.68 | 35.45±0.79 | 35.47±3.76 | 31.67±3.98 | 50.29±8.34 | **54.46±0.00** | 41.47±2.39 |
| letter | 6.80±4.21 | 4.00±1.55 | 3.60±1.36 | 3.20±0.75 | 11.60±2.06 | 5.84±1.11 | 2.40±1.50 | 3.00±1.55 | **3.20±1.17** | 0.00±0.00 | 0.00±0.00 | 1.72±0.37 |
| lymphography | 100.00±0.00 | 100.00±0.00 | 100.00±0.00 | 100.00±0.00 | 100.00±0.00 | 100.00±0.00 | 77.11±6.79 | **79.84±1.53** | 74.75±5.79 | 60.02±26.09 | 0.00±0.00 | 58.34±6.75 |
| magic.gamma | 64.88±2.50 | 69.99±2.98 | 69.99±0.87 | 70.21±0.50 | 71.64±0.93 | 69.34±1.08 | 79.82±0.48 | 80.94±0.33 | 80.19±0.73 | 89.73±7.81 | **96.10±0.00** | 85.36±1.69 |
| mammography | 36.62±8.76 | 38.15±8.41 | 27.69±1.80 | 29.08±2.63 | 18.15±1.23 | 29.94±2.59 | 32.69±5.73 | 34.62±2.19 | 35.08±2.68 | **39.69±2.19** | 0.00±0.00 | 28.42±1.40 |
| mnist | 45.37±1.84 | 46.20±3.18 | 50.54±1.26 | 59.20±1.30 | 62.66±2.09 | 52.79±0.77 | 63.86±1.77 | 56.74±2.71 | 53.74±3.75 | 0.00±0.00 | 0.00±0.00 | 34.87±1.48 |
| musk | 100.00±0.00 | 100.00±0.00 | 100.00±0.00 | 100.00±0.00 | 46.80±8.26 | 89.36±1.65 | 100.00±0.00 | 100.00±0.00 | 100.00±0.00 | 0.00±0.00 | 0.00±0.00 | 60.00±0.00 |
| optdigits | 29.87±5.68 | 41.20±3.49 | 44.67±5.56 | 71.73±0.53 | 46.00±5.11 | 46.00±5.11 | 14.80±1.81 | 6.40±3.88 | 9.33±5.72 | 0.00±0.00 | 0.00±0.00 | 6.11±1.52 |
| pageblocks | 62.16±2.84 | 64.08±2.96 | 62.31±2.65 | 63.88±1.08 | 62.20±1.05 | 62.93±0.29 | 62.24±0.98 | **62.71±1.24** | 61.25±0.32 | 60.47±0.69 | 0.00±0.00 | 49.33±0.37 |
| pendigits | 46.03±17.81 | 60.51±10.58 | 59.23±5.15 | 66.03±5.18 | 51.03±3.57 | 56.56±5.52 | **63.97±6.36** | 54.36±7.40 | 43.46±7.50 | 43.46±7.50 | 0.00±0.00 | 32.36±1.43 |
| pima | 68.77±4.96 | 68.29±2.03 | 74.95±1.27 | 71.40±1.67 | 68.83±2.08 | 70.45±1.78 | 66.14±2.35 | 63.89±4.28 | 64.71±3.39 | **67.99±2.34** | 65.96±4.44 | 65.74±2.10 |
| satellite | 65.94±10.44 | 72.95±2.96 | 76.27±0.81 | 78.70±0.90 | 74.22±0.57 | 73.62±2.69 | 72.06±0.60 | 72.97±0.77 | 72.63±0.32 | 91.51±9.02 | **96.02±0.00** | 81.04±1.96 |
| satimage-2 | 38.03±8.59 | 72.68±32.26 | **91.83±0.56** | 89.58±1.13 | 56.34±12.38 | 69.69±7.51 | 78.31±3.03 | 60.85±5.87 | 46.20±5.07 | 26.20±32.11 | 0.00±0.00 | 42.31±7.14 |
| shuttle | 97.17±1.11 | 88.27±0.10 | 88.83±0.14 | **98.91±0.12** | 98.38±0.21 | 98.31±0.24 | 98.00±0.01 | 97.98±0.00 | 97.73±0.10 | 92.83±2.84 | 0.00±0.00 | 77.31±0.58 |
| skin | 38.03±28.70 | **72.09±8.03** | 67.99±9.92 | 95.71±4.90 | 78.33±5.25 | 90.32±5.34 | 78.62±5.69 | **82.23±0.37** | 82.15±0.39 | 80.48±0.35 | **54.74±0.80** | 76.25±0.32 |
| smtp | 39.28±18.81 | 59.49±7.84 | 54.93±13.50 | 54.29±12.30 | 57.59±2.50 | 26.35±2.35 | 19.53±2.50 | 69.59±3.95 | 69.07±5.05 | 27.32±14.61 | 87.61±0.00 | 43.11±3.67 |
| spambase | 76.97±2.12 | 76.88±3.76 | 80.35±0.48 | 80.23±0.62 | 74.93±0.44 | 77.86±2.03 | **98.86±1.44** | 80.19±0.18 | 80.56±0.29 | **87.61±0.00** | 87.61±0.00 | 83.25±0.07 |
| speech | 2.95±1.23 | 3.61±1.23 | 3.28±1.04 | 2.95±2.41 | **4.59±1.61** | 3.48±0.92 | 3.93±0.80 | 2.95±1.61 | 3.28±1.80 | 0.00±0.00 | 0.00±0.00 | 2.03±0.25 |
| stamps | 34.84±12.18 | 42.45±13.97 | **83.03±3.34** | 76.24±2.92 | 33.76±5.95 | 62.00±6.23 | 63.62±10.79 | 58.37±9.83 | 75.27±0.00 | 56.93±11.83 | 14.60±29.20 | 50.23±10.89 |
| thyroid | **68.30±4.17** | 61.29±1.80 | 61.08±4.63 | 63.87±3.57 | 54.89±6.76 | 57.68±1.74 | 73.76±2.21 | 76.13±1.72 | 75.27±0.00 | **78.28±0.80** | 0.00±0.00 | 60.69±0.57 |
| vertebral | 23.17±6.96 | 29.15±9.05 | 52.53±11.60 | **64.81±1.19** | 54.89±6.76 | 44.91±2.82 | 43.88±5.22 | 36.26±11.73 | 45.20±0.98 | 24.68±8.71 | 3.48±6.96 | 28.18±7.79 |
| vowels | 22.40±17.59 | 18.80±6.52 | 30.80±7.44 | 30.80±12.43 | 24.00±7.04 | 25.36±2.94 | 40.00±5.51 | 40.80±3.25 | **45.20±0.98** | 0.00±0.00 | 0.00±0.00 | 25.20±1.36 |
| waveform | **47.80±3.06** | 38.20±17.57 | 35.40±7.47 | 11.60±1.62 | 28.40±4.22 | 32.28±5.79 | 11.20±1.72 | 11.80±2.14 | 12.20±1.60 | **67.85±5.79** | 0.00±0.00 | 7.04±0.23 |
| wbc | **78.86±7.89** | 84.03±7.23 | **96.89±4.35** | 87.13±4.26 | 90.33±5.34 | 89.16±3.80 | 40.59±7.11 | 70.31±4.12 | 97.73±0.10 | 10.43±20.87 | **67.85±5.79** | 36.84±4.92 |
| wdbc | 64.32±18.21 | 80.42±5.68 | 84.10±6.72 | 39.61±4.50 | 78.35±5.25 | 26.35±2.35 | 34.81±8.65 | **95.05±5.74** | 82.80±8.... | 16.96±5.74 | 16.96±5.74 | 47.39±3.91 |
| wilt | 6.15±7.00 | 11.36±4.93 | 37.04±8.60 | 39.61±4.50 | 37.59±2.50 | 97.76±2.03 | **19.92±4.34** | 19.92±4.34 | 49.07±5.05 | 41.73±22.73 | 41.73±22.73 | 12.36±2.38 |
| wine | 97.95±4.09 | 96.72±3.06 | 88.98±2.23 | 98.18±3.64 | 97.67±3.23 | 93.38±2.50 | **59.07±3.42** | 97.27±3.64 | 97.27±3.64 | 56.98±5.14 | 58.96±14.97 | 66.58±4.38 |
| wpbc | 81.91±5.11 | 63.78±10.33 | 63.78±2.23 | 88.64±3.69 | 93.46±2.71 | 81.35±3.33 | 49.23±1.21 | 50.81±1.76 | 50.81±1.76 | 36.48±3.67 | 36.48±3.67 | 53.91±5.76 |
| yeast | 46.31±2.91 | 48.95±1.78 | 49.35±0.54 | 49.31±0.62 | 48.13±0.93 | 48.41±0.87 | 49.23±1.21 | 49.55±1.27 | 50.81±1.76 | 46.82±1.91 | **98.22±0.00** | 58.93±0.44 |
| yelp | 7.64±0.70 | 8.20±0.38 | 7.52±0.65 | 7.56±0.29 | 7.56±0.37 | 7.70±0.20 | 16.12±2.67 | 14.76±3.88 | 14.68±3.19 | 3.48±6.96 | 0.00±0.00 | 9.11±1.45 |
| MNIST-C | 45.87±0.49 | 50.05±0.46 | 51.49±0.27 | 51.46±0.39 | 47.26±0.49 | 49.22±0.12 | 50.07±0.51 | 48.78±0.62 | 47.93±0.48 | 48.78±0.62 | 0.00±0.00 | 29.35±0.11 |
| FashionMNIST | 56.83±0.32 | 63.23±4.00 | 64.48±0.20 | 64.41±0.16 | 58.13±0.29 | 61.42±0.13 | **59.78±0.30** | 59.24±0.30 | 58.65±0.27 | 58.65±0.27 | 0.00±0.00 | 35.54±0.10 |
| CIFAR10 | 16.33±0.58 | 20.51±0.66 | 22.84±0.26 | 22.87±0.57 | 15.56±0.45 | 19.62±0.19 | 24.23±0.19 | 23.86±0.44 | 23.92±0.30 | 23.92±0.30 | 0.00±0.00 | 14.40±0.12 |
| SVHN | 16.44±0.90 | 19.22±0.23 | 20.00±0.14 | 20.07±0.30 | 18.85±0.22 | 18.92±0.20 | 19.80±0.22 | 19.61±0.10 | 19.47±0.17 | 19.47±0.17 | 0.00±0.00 | 11.78±0.08 |
| MVTec-AD | 79.81±0.59 | 81.35±0.82 | 82.39±0.82 | 82.20±0.46 | 77.51±0.66 | 80.65±0.56 | 76.79±0.49 | 78.35±0.88 | 78.60±1.12 | **78.60±1.12** | 63.09±1.51 | 71.97±0.92 |
| 20news | 12.82±0.91 | 14.12±1.10 | 15.05±1.36 | 16.54±1.42 | 12.79±0.95 | 14.26±0.73 | 19.22±2.15 | 16.99±0.93 | 15.62±1.07 | 63.03±1.80 | 0.18±0.35 | 10.44±0.45 |
| agnews | 14.17±0.19 | 14.40±0.24 | 14.47±0.12 | 14.53±0.20 | 13.69±0.12 | 14.25±0.08 | 22.14±0.86 | 20.77±1.57 | 19.76±2.41 | 0.18±0.35 | 0.00±0.00 | 12.53±0.63 |

Table 12.1: Average AUROC ± standard dev. over five seeds for the semi-supervised setting on ADBench. Rank of each model among 32 models (26 baselines + 4 avg variants of top-4 baselines + 2 FoMo-0D variants w/ $D = 100$ and $D = 20$) per dataset is provided (in parentheses) (the lower, the better). We use blue and green respectively to mark the top-1 and the top-2 method. Last four rows show avg_rank of methods across datasets, and $p$-values of the Wilcoxon signed rank test comparing FoMo-0D ($D = 100$) with other baselines. The previous four rows are the same for FoMo-0D ($D = 20$), when ranking 31 models (26 baselines + 4 avg variants of top-4 baselines + FoMo-0D w/ $D = 20$).

| Dataset | FoMo-0D(100) | FoMo-0D(20) | DTE-NP | kNN | ICL | DTE-C | LOF | CBLOF | FeatureBagging | SLAD | DDPM | OCSVM | DTE-IG | IForest | MCD |
|---|---|---|---|---|---|---|---|---|---|---|---|---|---|---|---|
| aloi | 54.05 (±0.32) | 51.48 (±1.5)(10) | 51.19 (±0.14)(12) | 51.04 (±0.04)(14) | 47.5 (±0.98)(32) | 50.44 (±0.19)(20) | 48.76 (±0.02)(27) | 53.69 (±0.15)(6) | 49.07 (±0.53)(26) | 50.76 (±0.29)(18) | 49.91 (±0.35)(23) | 54.29 (±0.01) | 50.87 (±1.17)(16) | 50.74 (±0.68)(19) | 48.54 (±0.35)(28) |
| amazon | 54.74 (±4.18)(18) | 56.76 (±1.2)(17) | 60.83 (±0.01) | 60.58 (±0.02) | 54.21 (±0.22)(19) | 56.71 (±2.15)(10) | 57.88 (±0.08) | 58.17 (±0.12)(6) | 57.94 (±0.04)(7) | 55.01 (±0.06)(25) | 55.09 (±0.1)(5) | 56.48 (±0.05)(13) | 51.93 (±0.35)(26) | 56.4 (±0.95)(12) | 63.96 (±0.09)(4) |
| annthyroid | 85.11 (±1.6)(18) | 87.76 (±0.47)(8) | 93.74 (±0.5) | 91.75 (±0.01)(5) | 81.21 (±0.07)(26) | 67.26 (±0.15)(31) | 86.61 (±0.01)(10) | 88.05 (±0.16)(9) | 88.87 (±0.16)(10) | 88.82 (±1.56)(12) | 88.29 (±0.28)(13) | 94.75 (±0.36)(3) | 87.02 (±0.06)(9) | 90.28 (±0.33)(8) | 90.16 (±0.02)(9) |
| backdoor | 78.9 (±1.71)(17) | 55.76 (±4.88)(26) | 93.32 (±0.5)(5) | 93.71 (±0.3)(3) | 93.62 (±0.09)(6) | 93.1 (±0.07)(24) | 95.33 (±0.25) | 60.65 (±0.23)(22) | 94.83 (±0.62)(2) | 50.0 (±0.30) | 80.93 (±0.56)(16) | 62.53 (±0.64)(25) | 94.02 (±1.4)(66) | 74.89 (±2.67)(19) | 85.13 (±8.97)(14) |
| breastw | 99.15 (±0.21)(10) | 98.88 (±0.21)(4) | 98.28 (±0.12)(5) | 99.05 (±0.26)(3) | 98.28 (±0.45)(20) | 92.78 (±1.75)(26) | 89.91 (±6.8)(28) | 99.11 (±0.23)(12) | 50.07 (±1.54)(30) | 99.53 (±0.1)(1) | 98.7 (±0.43)(17) | 99.53 (±0.06)(2) | 78.65 (±1.1)(29) | 99.5 (±0.08)(2) | 98.66 (±0.65)(19) |
| campaign | 65.3 (±0.7)(25) | 63.98 (±0.3)(26) | 78.79 (±0.25)(2) | 78.48 (±0.06) | 81.92 (±0.79)(1) | 77.05 (±1.11)(9) | 70.55 (±0.02)(20) | 77.05 (±0.31)(4) | 69.1 (±3.8)(23) | 76.75 (±0.16)(16) | 74.51 (±0.42)(18) | 77.67 (±0.01)(10) | 74.81 (±1.69)(17) | 73.64 (±1.48)(19) | 78.51 (±0.81)(5) |
| cardio | 94.27 (±0.0)(6) | 91.86 (±0.2)(9) | 91.86 (±0.43)(15) | 92.0 (±0.04)(4) | 80.01 (±2.1)(25) | 87.26 (±1.49)(9) | 92.21 (±0.01)(12) | 93.49 (±1.4)(7) | 92.12 (±0.56)(13) | 83.05 (±1.1)(22) | 86.94 (±1.96)(20) | 95.61 (±0.04)(1) | 73.79 (±14.09)(28) | 93.52 (±1.4)(8) | 82.82 (±9.85)(23) |
| cardiotocography | 72.1 (±0.0)(8) | 62.51 (±0.3)(19) | 63.78 (±1.08)(15) | 62.01 (±0.01)(20) | 54.22 (±1.8)(29) | 60.11 (±2.5)(22) | 60.4 (±3.7)(23) | 67.61 (±2.1)(16) | 63.61 (±2.4)(19) | 47.32 (±3.15)(30) | 54.54 (±2.9)(25) | 75.22 (±0.0)(5) | 52.44 (±3.68)(6) | 74.24 (±2.8)(6) | 57.11 (±1.41)(23) |
| celeba | 74.81 (±0.6)(13) | 60.4 (±6.7)(27) | 97.73 (±0.5)(7) | 97.54 (±0.15)(8) | 98.34 (±4.0)(20) | 97.76 (±1.2)(6) | 43.71 (±0.75)(32) | 79.26 (±1.3)(6) | 46.89 (±1.8)(30) | 67.43 (±1.6)(23) | 78.56 (±1.9)(17) | 79.79 (±0.7)(15) | 74.51 (±2.3)(14) | 71.23 (±2.7)(19) | 84.73 (±2.3)(41) |
| census | 60.54 (±4.96)(20) | 54.24 (±7.8)(26) | 72.1 (±0.4)(5) | 99.49 (±0.06)(3) | 70.56 (±0.35)(8) | 69.62 (±0.9)(13) | 58.46 (±1.06)(22) | 70.84 (±0.2)(7) | 55.92 (±1.0)(24) | 57.91 (±0.8)(23) | 70.02 (±0.2)(12) | 79.79 (±0.7)(15) | 61.79 (±4.9)(17) | 62.55 (±2.3)(15) | 74.14 (±1.94)(1) |
| cover | 94.12 (±0.2)(3) | 97.87 (±0.1)(5) | 97.73 (±0.5)(7) | 97.54 (±0.15)(8) | 93.34 (±4.0)(20) | 97.76 (±1.2)(6) | 43.71 (±0.75)(32) | 94.04 (±0.2)(7) | 93.16 (±0.6)(32) | 73.97 (±1.8)(26) | 94.35 (±0.6)(4) | 96.17 (±0.1)(11) | 95.83 (±1.5)(12) | 86.31 (±2.2)(17) | 70.02 (±0.6)(28) |
| donors | 99.91 (±0.0)(21) | 95.91 (±3.1)(11) | 99.26 (±0.2)(9) | 99.49 (±0.06)(3) | 99.9 (±0.0)(2) | 98.15 (±0.3)(18) | 96.97 (±0.2)(9) | 93.47 (±0.2)(13) | 88.51 (±5.5)(20) | 82.5 (±1.8)(22) | 92.09 (±0.2)(14) | 92.09 (±0.6)(6) | 90.79 (±3.2)(24) | 89.44 (±2.2)(17) | 81.93 (±10.7)(23) |
| fault | 61.82 (±0.0)(4) | 56.72 (±3.9)(19) | 58.64 (±0.65)(16) | 58.73 (±0.01)(5) | 60.63 (±0.5)(6) | 59.46 (±1.4)(9) | 47.42 (±0.0)(32) | 59.0 (±1.2)(3) | 48.32 (±0.9)(31) | 63.93 (±0.19)(1) | 61.09 (±1.1)(6) | 57.21 (±0.0)(18) | 59.42 (±1.4)(11) | 55.86 (±2.0)(22) | 59.44 (±3.7)(10) |
| fraud | 93.91 (±2.8)(18) | 95.91 (±3.1)(11) | 95.91 (±0.3)(14) | 95.45 (±1.0)(6) | 62.78 (±1.4)(22) | 93.52 (±1.5)(20) | 94.35 (±1.4)(15) | 94.91 (±1.0)(14) | 94.83 (±1.5)(12) | 94.38 (±1.1)(14) | 93.65 (±0.9)(19) | 90.79 (±3.2)(24) | 94.73 (±1.2)(13) | 94.73 (±1.2)(13) | 91.1 (±1.7)(23) |
| glass | 98.3 (±0.5)(13) | 97.09 (±0.8)(64) | 95.91 (±0.3)(14) | 92.04 (±1.1)(28) | 99.44 (±0.58)(1) | 92.42 (±2.37) | 88.82 (±4.98)(12) | 89.35 (±1.48)(11) | 88.49 (±0.8)(10) | 86.64 (±1.5)(9) | 69.73 (±5.0)(27) | 69.73 (±0.0)(3) | 79.81 (±9.5)(18) | 81.09 (±2.3)(6) | 79.71 (±1.37)(21) |
| hepatitis | 99.88 (±0.0)(4) | 98.81 (±0.3)(5) | 93.22 (±3.9)(14) | 96.46 (±1.46)(10) | 96.17 (±0.0)(5) | 98.78 (±0.89)(8) | 66.92 (±7.0)(13) | 86.32 (±2.35)(18) | 67.76 (±6.5)(30) | 99.03 (±0.6)(15) | 61.11 (±1.9)(9) | 93.03 (±0.9)(5) | 59.19 (±2.3)(19) | 82.69 (±2.7)(24) | 80.64 (±4.2)(26) |
| http | 99.92 (±0.0)(5) | 99.98 (±0.02)(5) | 99.98 (±0.02)(5) | 100.0 (±0.0)(2) | 98.2 (±4.5)(23) | 99.45 (±0.1)(7) | 99.98 (±0.0)(5) | 99.93 (±0.01)(5) | 92.1 (±0.5)(25) | 99.91 (±0.0)(15) | 100.0 (±0.02) | 100.0 (±0.1)(2) | 99.93 (±1.0)(5) | 93.55 (±0.2)(20) | 99.95 (±0.0)(10) |
| imdb | 50.84 (±6.2)(10) | 49.03 (±2.4)(23) | 50.43 (±0.01)(11) | 50.08 (±0.01)(4) | 52.34 (±0.49)(2) | 48.05 (±2.2)(27) | 49.57 (±0.0)(18) | 49.94 (±0.0)(16.5) | 49.53 (±0.1)(19.5) | 51.26 (±0.1)(6) | 47.91 (±0.1)(30) | 48.72 (±0.0)(24) | 50.97 (±3.4)(9) | 49.53 (±0.78)(19.5) | 51.24 (±0.1)(87) |
| internetads | 55.64 (±7.1)(25) | 55.18 (±5.2)(26) | 59.91 (±2.2)(10) | 68.08 (±0.01)(2) | 72.2 (±0.5)(5) | 77.57 (±1.5)(41) | 71.72 (±0.0)(6) | 65.16 (±0.08)(21) | 71.38 (±2.3)(8) | 75.94 (±0.1)(2) | 65.63 (±0.0)(20) | 65.63 (±0.01)(9) | 71.52 (±3.8)(7) | 47.87 (±2.1)(31) | 47.73 (±0.0)(32) |
| ionosphere | 96.59 (±1.2)(11) | 97.14 (±0.5)(8) | 97.77 (±1.3)(94) | 97.44 (±0.98)(6) | 98.98 (±0.32)(1) | 95.42 (±0.58)(13) | 94.29 (±2.2)(19) | 96.78 (±1.5)(10) | 94.47 (±2.1)(18) | 98.21 (±0.6)(2) | 94.6 (±0.8)(17) | 96.32 (±0.03)(12) | 95.15 (±3.6)(15) | 91.21 (±1.3)(23) | 95.4 (±0.6)(14) |
| landsat | 69.53 (±0.0)(2) | 64.76 (±1.7)(10) | 68.22 (±1.7)(94) | 68.25 (±0.0)(3) | 65.13 (±0.44)(8) | 52.79 (±1.6)(22) | 66.58 (±0.0)(5) | 57.21 (±0.26)(15) | 66.38 (±0.1)(7) | 72.0 (±0.0)(22) | 51.37 (±1.0)(23) | 47.98 (±0.0)(27) | 44.72 (±5.5)(28) | 58.8 (±2.2)(14) | 56.78 (±6.0)(16) |
| letter | 40.72 (±0.0)(9) | 39.53 (±4.4)(11) | 34.38 (±0.9)(22) | 35.43 (±0.02)(20) | 42.68 (±1.1)(7) | 36.72 (±0.59)(16) | 44.82 (±0.6)(5) | 33.24 (±0.06)(25) | 44.48 (±1.1)(4) | 36.81 (±0.7)(17) | 38.05 (±0.2)(10) | 38.05 (±1.2)(14) | 39.86 (±2.2)(10) | 32.04 (±1.6)(27) | 31.47 (±4.16)(28) |
| lymphography | 99.93 (±0.1)(19) | 99.93 (±0.0)(9) | 99.93 (±0.0)(9) | 99.93 (±0.0)(9) | 100.0 (±0.0)(3) | 98.59 (±0.42)(5) | 98.21 (±0.75)(27) | 99.83 (±0.02)(16) | 99.61 (±2.3)(28) | 99.94 (±0.0)(7) | 99.94 (±0.0)(7) | 99.88 (±0.05)(6) | 99.88 (±0.05)(6) | 77.09 (±1.2)(33) | 98.88 (±5.5)(26) |
| magic.gamma | 84.75 (±0.1)(45) | 85.94 (±0.1)(4) | 83.57 (±0.76)(7) | 83.27 (±0.09) | 75.56 (±4.2)(16) | 87.5 (±0.9)(1) | 83.4 (±0.08)(9) | 75.81 (±0.01)(15) | 84.19 (±0.7)(26) | 84.19 (±0.04)(6) | 85.97 (±1.0)(3) | 74.25 (±0.0)(19) | 86.46 (±1.1)(2) | 77.09 (±1.2)(33) | 73.67 (±0.1)(21) |
| mammography | 69.11 (±1.2)(32) | 84.08 (±1.6)(19) | 87.62 (±0.0)(98) | 87.58 (±0.0)(9) | 71.87 (±9.1)(29) | 86.42 (±1.7)(23) | 85.52 (±0.0)(15) | 84.74 (±0.0)(17) | 86.31 (±0.3)(13) | 74.51 (±0.07)(26) | 81.01 (±2.0)(21) | 88.63 (±0.06) | 84.64 (±3.4)(18) | 89.42 (±0.5)(12) | 72.87 (±0.6)(28) |
| mnist | 91.23 (±0.0)(7) | 90.95 (±0.0)(4) | 94.0 (±0.4)(1) | 90.01 (±0.0)(8) | 90.11 (±1.3)(12) | 93.17 (±0.8)(23) | 100.0 (±0.08)(5) | 100.0 (±0.08)(5) | 92.55 (±0.4)(6) | 100.0 (±0.08)(5) | 100.0 (±0.08)(5) | 100.0 (±0.03) | 80.78 (±5.9)(22) | 86.6 (±1.9)(18) | 88.3 (±1.0)(15) |
| optdigits | 99.62 (±0.4)(23) | 93.52 (±0.7)(22) | 89.32 (±0.26)(8.5) | 93.72 (±0.7) | 79.34 (±1.6)(16) | 92.38 (±2.6)(16) | 99.65 (±0.2) | 83.52 (±1.06)(5) | 96.27 (±0.49)(3) | 95.28 (±0.1)(4) | 90.76 (±1.2)(9) | 98.03 (±0.0)(10) | 79.81 (±9.5)(18) | 89.42 (±0.9)(12) | 64.86 (±9.2)(22) |
| pageblocks | 86.58 (±0.0)(19) | 83.45 (±0.0)(25) | 89.32 (±2.0)(8.5) | 89.65 (±0.0)(6) | 88.39 (±7.7)(12) | 89.89 (±0.5)(5) | 91.23 (±0.1)(3) | 96.67 (±0.0)(14) | 91.11 (±0.3)(4) | 87.86 (±0.0)(15) | 86.93 (±0.4)(18) | 88.58 (±0.0)(11) | 85.66 (±1.7)(22) | 82.64 (±0.8)(27) | 87.07 (±0.0)(17) |
| pendigits | 98.11 (±0.0)(8.5) | 97.98 (±1.8)(8.5) | 99.61 (±0.1)(3) | 99.87 (±0.0)(1) | 96.71 (±0.8)(13) | 97.79 (±0.6)(10) | 99.05 (±0.0)(7) | 96.67 (±0.07)(14) | 99.5 (±0.1)(4) | 98.11 (±0.2)(8.5) | 98.11 (±0.2)(8.5) | 96.36 (±0.0)(15) | 96.96 (±1.6)(12) | 97.22 (±0.4)(11) | 83.69 (±0.0)(26) |
| pima | 66.84 (±0.4)(6) | 77.98 (±1.8)(5) | 81.15 (±2.5)(1) | 76.94 (±1.8)(6) | 79.68 (±3.1)(6) | 58.74 (±0.1)(42) | 72.94 (±1.0)(13) | 72.94 (±1.0)(13) | 71.93 (±2.2)(16) | 60.62 (±4.2)(26) | 97.95 (±0.2)(14) | 71.53 (±1.7)(17) | 68.59 (±3.9)(22) | 74.26 (±1.6)(10) | 73.64 (±1.3)(11) |
| satellite | 84.98 (±0.0)(4) | 83.02 (±1.0)(6) | 82.11 (±0.6)(18) | 82.24 (±0.0)(7) | 85.15 (±0.4)(3) | 78.61 (±0.7)(14) | 80.3 (±0.0)(11) | 73.2 (±0.9)(22) | 80.11 (±0.1)(12) | 87.49 (±0.1) | 77.73 (±0.5)(15) | 73.91 (±0.0)(20) | 76.52 (±2.6)(17) | 77.46 (±1.4)(16) | 72.76 (±3.6)(24) |
| satimage-2 | 98.4 (±0.0)(20) | 99.46 (±0.38)(11) | 98.03 (±0.0)(64) | 99.71 (±0.0)(5) | 98.98 (±0.03)(1) | 98.74 (±0.0)(14) | 99.38 (±0.0)(18) | 99.42 (±0.0)(12) | 99.47 (±0.03)(10) | 99.77 (±0.1)(1) | 99.62 (±0.1)(6) | 99.61 (±0.0)(8) | 95.34 (±1.8)(28) | 99.12 (±0.2)(16) | 99.92 (±0.0)(1) |
| shuttle | 99.62 (±0.0)(12) | 99.95 (±0.0)(42) | 99.93 (±0.0)(23) | 99.91 (±0.0)(6.5) | 99.96 (±0.0)(4.5) | 99.5 (±0.0)(15) | 99.98 (±0.0)(1) | 99.72 (±0.02)(16) | 86.89 (±2.2)(27) | 99.9 (±0.01)(6.5) | 99.9 (±0.01)(6.5) | 99.62 (±0.0)(18) | 99.86 (±1.3)(10) | 99.65 (±0.0)(17) | 98.98 (±0.0)(13) |
| skin | 70.6 (±0.8)(23) | 87.79 (±2.9)(15) | 98.86 (±0.4)(5) | 98.56 (±0.3)(6) | 74.36 (±7.0)(29) | 86.34 (±1.7)(17) | 86.34 (±1.7)(17) | 91.82 (±0.2)(16) | 78.78 (±0.9)(19) | 83.38 (±3.2)(17) | 83.34 (±4.1)(17) | 84.65 (±4.4)(22) | 98.71 (±1.1)(34) | 89.42 (±0.5)(12) | 94.87 (±0.8)(4) |
| smtp | 83.04 (±4.69)(23) | 92.08 (±2.9)(19) | 92.98 (±2.9)(10) | 92.43 (±2.7)(10) | 88.39 (±7.7)(12) | 38.17 (±0.57)(17) | 87.28 (±5.6)(16) | 88.4 (±1.3)(66)(20) | 69.64 (±2.0)(20) | 84.81 (±3.6)(62) | 86.93 (±0.4)(18) | 95.42 (±1.2)(61) | 39.57 (±1.5)(13) | 93.47 (±2.1)(31) | 38.81 (±0.36)(14) |
| spambase | 78.21 (±0.0)(18) | 73.72 (±0.84)(22) | 83.74 (±0.69)(3) | 83.36 (±0.0)(5) | 83.53 (±0.4)(4) | 83.01 (±0.4)(17) | 73.23 (±0.02)(23) | 81.52 (±0.55)(13) | 69.64 (±2.0)(20) | 81.97 (±3.8)(29)(2)(7) | 81.84 (±4.1)(18) | 79.72 (±1.4)(00) | 93.38 (±3.3)(13) | 93.47 (±1.4)(21) | 84.93 (±2.0)(12) |
| speech | 40.1 (±1.7)(10) | 39.62 (±4.6)(12) | 41.37 (±0.0)(9.5) | 41.37 (±0.0)(9.5) | 48.86 (±2.7)(6) | 38.17 (±0.57)(17) | 37.53 (±0.0)(20) | 35.88 (±0.1)(32) | 37.48 (±0.37)(21) | 37.48 (±0.37)(21) | 36.96 (±3.86)(23) | 40.25 (±0.24)(40) | 39.57 (±1.5)(13) | 43.47 (±1.4)(15) | 38.11 (±0.36)(14) |
| stamps | 97.52 (±0.3)(9) | 97.98 (±0.67)(1) | 97.87 (±0.37)(2) | 95.89 (±1.4)(45) | 96.68 (±1.0)(9) | 91.6 (±2.0)(42)(20) | 93.74 (±2.3)(60) | 93.41 (±1.6)(12) | 94.21 (±2.2)(67) | 81.97 (±8.2)(97)(2)(7) | 97.95 (±0.2)(14) | 75.91 (±0.0)(17) | 79.92 (±3.9)(14) | 88.96 (±0.3)(1) | 41.85 (±0.7)(30) |
| thyroid | 96.64 (±0.0)(16) | 98.85 (±0.1)(4) | 98.68 (±0.0)(4) | 98.08 (±0.0)(3.5) | 95.4 (±1.98)(20) | 98.71 (±0.4)(2) | 97.22 (±0.0)(12) | 98.56 (±0.03)(7) | 98.54 (±0.0)(8) | 91.45 (±2.2)(13) | 97.95 (±0.2)(14) | 98.56 (±0.08) | 99.43 (±1.1)(73)(30) | 98.49 (±0.0)(12) | 98.49 (±0.0)(12) |
| vertebral | 97.2 (±2.5)(1) | 82.27 (±2.5)(2) | 81.42 (±4.4)(1) | 57.67 (±3.58)(12) | 85.1 (±2.1)(7) | 56.87 (±1.6)(04) | 64.41 (±1.7)(7) | 87.22 (±1.5)(1) | 83.5 (±2.1)(66)(6) | 83.51 (±0.1)(8) | 62.17 (±2.6)(24) | 76.04 (±0.0)(14) | 73.68 (±2.6)(80) | 72.29 (±1.5)(21)(2) | 27.66 (±2.8)(32) |
| vowels | 81.29 (±0.0)(12) | 84.53 (±0.0)(9) | 81.42 (±1.4)(11) | 82.21 (±0.01)(0) | 85.1 (±2.1)(07) | 86.93 (±2.2)(2) | 86.3 (±0.0)(44) | 78.72 (±4.0)(15) | 85.32 (±1.6)(66) | 85.02 (±0.0)(28) | 86.38 (±1.9)(43) | 75.91 (±0.0)(17) | 85.68 (±3.1)(65) | 61.8 (±6.6)(22) | 27.66 (±2.8)(32) |
| waveform | 71.02 (±0.0)(13) | 75.24 (±0.6)(6) | 74.48 (±0.0)(8) | 75.21 (±0.0)(7) | 66.68 (±3.66)(6) | 65.31 (±1.0)(19) | 76.0 (±0.0)(3) | 80.52 (±2.4)(7.30) | 55.08 (±0.0)(8.4) | 48.92 (±0.0)(32) | 62.17 (±2.0)(24) | 70.44 (±0.0)(14) | 73.68 (±2.6)(80) | 72.29 (±1.5)(21)(2) | 58.39 (±0.0)(29) |
| wbc | 99.61 (±0.3)(14) | 99.3 (±0.5)(09) | 99.54 (±0.2)(87) | 99.12 (±0.9)(16) | 99.06 (±2.0)(9) | 98.48 (±0.7)(24) | 99.62 (±0.2)(7.5) | 98.31 (±1.3)(20) | 99.65 (±0.1)(8.4) | 99.49 (±2.5)(8) | 92.2 (±0.3)(9.4) | 99.63 (±0.9)(3) | 99.6 (±0.3)(16) | 99.73 (±0.6)(21.5) | 98.88 (±1.0)(19) |
| wdbc | 99.77 (±0.2)(2) | 99.71 (±0.2)(3) | 99.54 (±0.3)(87) | 99.05 (±0.27)(17) | 99.78 (±0.2)(11) | 98.46 (±0.7)(24) | 99.62 (±0.0)(12) | 98.73 (±0.4)(12.5) | 73.35 (±0.5)(9) | 61.76 (±0.8)(11) | 93.34 (±2.0)(9.5) | 93.34 (±0.2)(9) | 99.6 (±0.3)(16) | 98.73 (±0.6)(21.5) | 97.03 (±0.4)(25) |
| wilt | 97.13 (±0.0)(1) | 85.33 (±0.0)(3) | 62.91 (±5.9)(14) | 63.66 (±0.0)(13) | 76.42 (±3.4)(6.7) | 85.1 (±1.3)(4) | 68.81 (±0.0)(12) | 42.9 (±1.1)(4.23) | 73.35 (±0.5)(9) | 61.76 (±0.8)(11) | 71.66 (±0.8)(11) | 34.81 (±0.0)(29) | 93.75 (±3.3)(2) | 47.97 (±3.3)(22) | 81.72 (±0.0)(15) |
| wine | 99.94 (±0.2)(4) | 99.44 (±0.7)(9) | 99.44 (±0.7)(9) | 99.19 (±0.22)(11) | 99.87 (±0.04)(2) | 99.97 (±0.04)(2) | 98.36 (±3.8)(12) | 97.75 (±0.5)(16) | 97.93 (±0.8)(48) | 97.93 (±0.8)(15) | 96.91 (±0.9)(8) | 97.82 (±0.4)(15) | 99.95 (±0.1)(3) | 93.92 (±1.7)(22) | 97.28 (±1.0)(17) |
| wpbc | 76.12 (±1.0)(2) | 82.57 (±4.4)(09) | 83.16 (±3.14)(64) | 61.69 (±0.06)(6) | 59.16 (±1.1)(67) | 63.12 (±1.1)(13) | 63.82 (±0.0)(12) | 59.51 (±1.96)(16) | 56.79 (±1.1)(86)(22) | 56.79 (±0.3)(25) | 76.01 (±1.9)(10) | 77.44 (±2.0)(18) | 50.71 (±9.0)(10) | 56.33 (±2.7)(23) | 63.36 (±0.9)(14) |
| yeast | 49.91 (±0.0)(4) | 46.39 (±0.0)(17) | 44.58 (±0.3)(23) | 44.74 (±0.2)(1) | 48.58 (±2.3)(96) | 47.08 (±1.1)(13) | 45.79 (±0.0)(18) | 50.44 (±0.0)(86)(3) | 46.4 (±1.3)(16) | 48.69 (±0.1)(17) | 49.13 (±2.8)(5) | 44.84 (±0.0)(20) | 48.58 (±2.8)(98) | 41.8 (±0.7)(30) | 43.05 (±0.1)(27) |
| yelp | 60.29 (±1.4)(13) | 62.07 (±1.4)(10) | 68.66 (±0.0)(1) | 68.07 (±0.0)(2) | 55.79 (±0.5)(25) | 61.36 (±1.1)(24) | 67.2 (±0.0)(4) | 63.8 (±0.0)(8) | 67.06 (±0.1)(26) | 62.08 (±0.0)(9) | 59.3 (±0.9)(17) | 59.55 (±0.0)(15) | 57.32 (±4.2)(1) | 61.07 (±0.5)(11.5) | 66.15 (±0.0)(47) |
| MNIST-C | 86.66 (±0.6)(18) | 72.68 (±4.6)(22) | 84.74 (±0.0)(5) | 84.11 (±0.0)(6) | 85.62 (±0.5)(24) | 86.1 (±0.8)(43) | 87.21 (±0.0)(2) | 81.11 (±0.0)(11) | 87.33 (±0.15)(1) | 83.51 (±0.1)(8) | 80.12 (±0.1)(12) | 79.55 (±0.15) | 79.92 (±3.9)(14) | 76.75 (±1.4)(20) | 75.26 (±1.2)(21) |
| FashionMNIST | 64.53 (±0.6)(17) | 70.04 (±4.3)(24) | 67.82 (±0.06) | 90.56 (±0.2)(3) | 90.21 (±0.5)(4) | 86.53 (±1.9)(3) | 91.6 (±0.0)(2) | 89.1 (±0.1)(17) | 91.67 (±0.1)(11) | 88.16 (±0.0)(14) | 88.52 (±0.0)(13) | 88.16 (±0.0)(14) | 84.32 (±2.4)(20) | 84.15 (±1.0)(21) | 84.37 (±1.1)(19) |
| CIFAR10 | 59.91 (±1.4)(25) | 59.91 (±1.4)(25) | 67.82 (±0.0)(6) | 67.53 (±0.03)(5) | 68.53 (±1.3)(93) | 70.3 (±0.0)(2) | 70.3 (±0.0)(2) | 67.87 (±0.2)(15) | 70.31 (±0.2)(11) | 66.6 (±0.1)(15) | 67.91 (±0.1)(14) | 67.79 (±0.0)(7) | 62.4 (±3.3)(22) | 64.04 (±0.9)(18) | 63.15 (±0.5)(16) |
| SVHN | 73.31 (±1.2)(25) | 75.75 (±2.0)(24) | 59.07 (±0.74)(6.5) | 81.51 (±1.82)(1) | 81.46 (±1.7)(2) | 89.39 (±2.0)(6) | 80.36 (±2.1)(4.4) | 79.96 (±2.0)(15) | 80.45 (±2.2)(13) | 93.24 (±1.0)(2) | 78.0 (±1.9)(17) | 77.44 (±2.0)(18) | 85.88 (±3.6)(38) | 77.38 (±1.9)(19) | 86.75 (±2.1)(27) |
| MVTec-AD | 56.93 (±1.4)(16) | 59.07 (±1.8)(76.5) | 59.07 (±10.74)(6.5) | 57.36 (±0.66)(14) | 61.3 (±1.0)(63) | 64.3 (±3.1)(71) | 60.19 (±0.62)(5) | 57.1 (±1.2)(1.5) | 60.25 (±0.7)(4) | 59.47 (±0.8)(9) | 57.82 (±0.12)(19) | 56.25 (±0.62)(18) | 58.28 (±6.1)(11) | 54.91 (±1.1)(21) | 62.85 (±1.6)(2) |
| 20news | 60.96 (±3.69)(12) | 64.31 (±1.4)(19) | 67.98 (±0.04)(5) | 67.05 (±0.0)(6) | 62.55 (±0.47)(11) | 68.16 (±3.2)(3) | 74.58 (±0.0)(2) | 62.82 (±0.08)(10) | 74.64 (±0.09)(1) | 58.43 (±0.07)(17.5) | 57.82 (±0.12)(19) | 60.64 (±0.0)(13) | 56.55 (±4.15)(24) | 58.43 (±1.09)(17.5) | 67.98 (±0.25)(45) |
| Rank(avg) | 12.596 | 13.228 | 7.553 | 8.57 | 10.342 | 10.798 | 11.825 | 12.816 | 12.86 | 12.526 | 13.599 | 13.342 | 14.675 | 16.246 | 16.509 |
| All | - | - | 0.001 | 0.019 | 0.089 | 0.159 | 0.394 | 0.434 | 0.703 | 0.516 | 0.752 | 0.679 | 0.761 | 0.971 | 0.959 |
| $d \le 20$ | - | - | 0.003 | 0.034 | 0.068 | 0.016 | 0.394 | 0.389 | 0.380 | 0.389 | 0.906 | 0.987 | 0.789 | 0.849 | 1.000 |
| $d \le 50$ | - | - | 0.547 | 0.794 | 0.893 | 0.046 | 0.997 | 0.988 | 1.000 | 0.963 | 0.986 | 0.986 | 0.986 | 1.000 | 1.000 |
| Rank(avg) | 11.886 | 13.228 | 7.553 | 9.018 | 10.851 | 11.36 | 12.316 | 13.342 | 13.386 | 12.982 | 14.061 | 13.851 | 15.342 | 16.965 | 17.158 |
| All | - | - | 0.016 | 0.019 | 0.089 | 0.454 | 0.585 | 0.750 | 0.823 | 0.516 | 0.901 | 0.895 | 0.996 | 1.000 | 0.996 |
| $d \le 100$ | - | - | 0.415 | 0.700 | 0.462 | 0.953 | 0.970 | 0.971 | 0.996 | 0.876 | 0.980 | 0.978 | 0.999 | 0.998 | 1.000 |
| $d \le 50$ | - | - | 0.220 | 0.569 | 0.827 | 0.394 | 0.960 | 0.968 | 0.994 | 0.910 | 0.960 | 0.979 | 0.998 | 0.999 | 0.999 |

Table 12.2: Average AUROC ± standard dev. over five seeds for the semi-supervised setting on ADBench. Rank of each model per dataset is provided (in parentheses) (the lower, the better). We use blue and green respectively to mark the top-1 and the top-2 method.

| Dataset | VAE | PCA | PlanarFlow | HBOS | GANomaly | GOAD | DIF | COPOD | ECOD | DeepSVDD | LODA | DAGMM | DROCC | DTE-NP^avg | KNN^avg | ICL^avg | DTE-C^avg |
|---|---|---|---|---|---|---|---|---|---|---|---|---|---|---|---|---|---|
| aloi | 54.04±0.0(3.5) | 54.04±0.0(3.5) | 48.52±2.3(4.29) | 52.23±0.0(7) | 53.94±1.6(5) | 48.01±0.92(30) | 51.1±0.0(13) | 49.51±0.0(24) | 51.73±0.0(8) | 50.89±2.05(15) | 49.24±2.75(25) | 50.84±2.97(17) | 50.0±0.0(22) | 51.25±0.0(11) | 51.61±0.0(9) | 47.63±0.15(31) | 50.29±0.09(21) |
| amazon | 5.94±0.0(32) | 54.9±0.0(16) | 49.94±2.1(2.31) | 56.32±0.0(13) | 53.42±0.04(22) | 56.07±0.9(14) | 51.43±0.34(27) | 56.78±0.0(9) | 53.79±0.20(20) | 51.2±4.4(28) | 52.23±2.84(24) | 50.47±2.01(29) | 50.0±0.0(30) | 60.47±0.0(3) | 60.25±0.0(5) | 53.15±0.19(23) | 53.48±1.13(21) |
| annthyroid | 85.44±0.0(20) | 85.19±0.0(21) | 93.19±2.1(4.3) | 66.02±0.0(31) | 67.52±5.5(54.30) | 81.01±5.1(6.25) | 88.36±0.0(16) | 76.77±0.0(28) | 78.45±0.0(26) | 55.01±3.6(2(32) | 77.35±7.51(27) | 72.23±15.13(29) | 88.9±2.3(1.12) | 92.64±0.0(6) | 92.3±0.0(7) | 84.27±2.31(23) | 88.05±0.05(17) |
| backdoor | 64.57±0.65(24) | 64.57±0.65(24) | 76.03±11.56(18) | 70.81±0.89(21) | 87.17±1.4(3.13) | 52.9±14.48(28) | 83.73±0.76(15) | 50.0±0.0(30) | 50.0±0.0(30) | 91.44±2.6(2(11) | 47.62±22.63(32) | 54.36±19.91(27) | 94.25±0.73(3) | 92.55±0.44(9) | 91.0±0.47(12) | 93.3±0.48(8) | 74.67±0.36(20) |
| breastw | 99.22±0.1(8) | 99.21±0.1(8) | 97.93±0.82(22) | 99.23±0.17(6) | 94.75±2.73(25) | 98.86±0.3(15) | 56.34±1.19(31) | 99.46±0.09(23) | 99.14±0.22(11) | 96.96±0.92(23) | 89.53±0.3(52(7) | 89.53±0.2(27) | 47.32±31.95(32) | 98.67±0.28(18) | 99.16±0.21(9) | 98.75±0.18(16) | 96.52±0.53(24) |
| campaign | 77.07±0.0(11.5) | 77.07±0.0(11.5) | 89.75±3.75(21) | 77.06±0.0(13) | 69.21±3.28(22) | 47.89±12.32(32) | 57.87±0.0(30) | 78.15±0.0(8) | 76.86±0.0(15) | 62.22±12.88(27) | 88.88±4.48(29) | 61.47±2.73(28) | 50.0±0.0(31) | 78.76±0.0(3) | 78.66±0.0(4) | 78.33±0.52(7) | 67.18±0.74(24) |
| cardio | 96.55±0.0(1) | 88.9±0.93(18) | 88.9±0.93(18) | 80.7±0.0(24) | 68.25±0.0(30) | 96.01±0.0(2) | 68.25±0.0(30) | 93.16±0.0(9) | 94.95±0.0(5) | 65.43±4.37(31) | 91.34±2.93(17) | 77.92±8.85(26) | 62.14±23.76(32) | 92.47±0.0(11) | 93.07±0.0(10) | 77.71±1.76(27) | 72.66±0.21(29) |
| cardiotocography | 78.9±0.0(2) | 80.53±0.0(3) | 69.88±5.65(9) | 61.24±0.0(27) | 62.77±8.62(18) | 76.06±1.4(4) | 41.79±0.0(32) | 66.35±0.0(12) | 79.3±0.0(1) | 47.75±8.59(29) | 72.79±7.6(7) | 67.11±9.03(11) | 45.98±16.75(31) | 63.21±0.0(17) | 65.48±0.0(13) | 50.85±1.24(28) | 55.9±0.23(24) |
| celeba | 80.52±0.84(4) | 70.51±0.21(10) | 71.64±7.91(18) | 76.68±0.66(10) | 43.8±10.9(30) | 43.8±10.9(30) | 66.69±3.55(24) | 75.72±0.59(12) | 76.31±0.63(11) | 56.17±22.46(28) | 62.46±13.17(26) | 63.81±4.28(25) | 64.89±1.11(22) | 74.42±0.28(15) | 76.9±0.35(9) | 78.49±0.93(8) | 68.94±0.44(21) |
| census | 70.52±0.22(9) | 70.51±0.21(10) | 59.33±2.89(21) | 62.5±0.47(16) | 68.07±3.66(14) | 35.24±4.19(32) | 41.45±2.04(18) | 50.0±0.0(30.5) | 50.0±0.0(30.5) | 54.16±4.33(27) | 51.12±11.19(29) | 52.24±1.19(28) | 55.36±3.62(25) | 72.11±0.16(4) | 71.84±0.15(6) | 72.25±1.63(3) | 61.41±0.71(19) |
| cover | 94.35±0.15(16) | 94.41±0.14(15) | 47.52±8.02(31) | 71.11±0.82(27) | 76.35±9.11(24) | 13.83±13.34(32) | 57.69±5.24(30) | 88.2±0.27(21) | 91.86±0.21(18) | 49.12±14.74(30) | 94.93±3.07(14) | 75.94±14.06(25) | 95.79±0.69(13) | 97.37±0.19(9) | 96.73±0.22(10) | 91.42±1.74(19) | 78.4±0.14(23) |
| donors | 88.6±0.25(19) | 88.12±0.57(21) | 91.64±3.52(15) | 81.19±0.61(25) | 73.34±11.69(27) | 33.57±16.0(32) | 90.04±1.79(16) | 81.5±0.21(24) | 88.74±0.38(18) | 72.95±17.81(29) | 63.52±27.27(30) | 62.15±16.19(31) | 74.18±22.09(28) | 99.38±0.06(4) | 98.74±0.09(7) | 96.3±1.95(10) | 80.17±3.55(26) |
| fault | 55.87±0.0(20.5) | 55.87±0.0(20.5) | 57.51±5.15(17) | 53.06±0.0(26) | 59.5±1.5(48) | 58.89±0.61(14) | 62.31±0.0(3) | 49.14±0.0(30) | 50.37±0.0(28) | 54.31±1.62(25) | 56.85±7.19(27) | 52.85±7.19(27) | 50.0±0.0(32) | 59.17±0.0(12) | 60.22±0.0(7) | 64.23±0.87(17) | 55.41±0.63(34) |
| fraud | 95.47±0.75(5) | 95.38±0.68(7) | 90.72±2.26(25) | 95.02±0.67(9) | 93.25±2.85(21) | 69.75±21.3(31) | 92.58±2.27(29) | 94.89±1.27(11) | 94.95±1.51(8) | 83.13±6.6(28) | 89.05±8.12(26) | 85.33±6.76(27) | 50.0±0.0(32) | 95.64±0.92(1.5) | 95.57±0.93(4) | 94.23±0.87(17) | 76.21±1.08(30) |
| glass | 72.55±1.16(25) | 73.44±4.22(24) | 85.33±6.2(16) | 82.59±3.23(18) | 79.77±8.72(20) | 59.03±12.64(32) | 86.45±1.39(5) | 76.0±1.94(23) | 83.67±16.2(17) | 71.14±3.54(26) | 67.34±5.39(28) | 65.33±15.53(30) | 64.89±23.64(31) | 90.83±1.0(19) | 87.3±1.45(6) | 95.27±1.45(6) | 76.95±2.58(22) |
| hepatitis | 84.84±2.27(19.5) | 84.48±2.29(22) | 95.8±1.67(13) | 84.84±6.78(19.5) | 87.39±7.55(17) | 84.5±3.25(21) | 96.07±2.32(12) | 80.9±1.22(25) | 73.84±1.99(27) | 99.57±0.24(7) | 68.98±3.97(29) | 70.22±6.66(28) | 51.8±17.93(32) | 96.22±0.88(11) | 87.9±1.75(16) | 99.69±0.48(6) | 84.42±1.2(23) |
| http | 99.94±0.01(12) | 99.95±0.01(10) | 98.93±5.71(23) | 98.58±1.04(22) | 50.14±14.85(30) | 99.68±0.0(13.6) | 99.36±0.07(19) | 99.19±0.09(17) | 97.95±0.12(24) | 61.31±15.49(29) | 42.78±17.91(26) | 91.78±17.91(26) | 50.0±0.0(31) | 99.95±0.01(10) | 99.96±0.01(8) | 99.98±0.02(5.5) | 89.59±0.09(27) |
| imdb | 47.97±0.0(28.5) | 47.97±0.0(28.5) | 49.23±2.75(22) | 49.44±0.0(16.5) | 51.58±0.7(13) | 48.46±0.65(26) | 51.4±0.0(4) | 46.88±0.0(32) | 46.88±0.0(32) | 49.97±5.71(15) | 47.23±2.24(31) | 48.6±0.38(25) | 51.35±2.0(5) | 50.35±0.0(13) | 50.18±0.0(13) | 52.35±0.0(1) | 49.27±1.12(21) |
| internetads | 65.12±0.0(22.5) | 65.12±0.0(22.5) | 70.87±0.84(9) | 49.18±0.0(30) | 69.86±2.3(11) | 65.65±0.21(19) | 49.33±0.34(29) | 65.94±0.0(16) | 66.0±0.0(15) | 65.24±6.6(19) | 58.73±3.84(24) | 49.47±5.05(28) | 53.4±7.55(27) | 67.65±0.0(13) | 65.73±0.0(14) | 72.5±0.65(4) | 66.79±0.73(14) |
| ionosphere | 89.76±1.2(24) | 89.11±1.31(25) | 96.86±1.2(19) | 70.68±2.85(31) | 93.89±2.03(20) | 91.54±3.05(22) | 93.59±2.06(21) | 78.32±2.13(28) | 71.77±1.43(30) | 97.2±1.2(67) | 85.56±3.59(26) | 73.95±1.59(8(29) | 61.14±28.54(32) | 97.5±0.63(5) | 95.12±0.92(16) | 97.78±1.2(3) | 83.01±5.81(27) |
| landsat | 54.22±9.45(20) | 43.9±0.0(30) | 50.85±2.13(24) | 73.21±0.0(1) | 55.34±10.05(19) | 40.52±2.32(32) | 56.56±0.0(17) | 49.29±0.0(26) | 42.01±0.0(31) | 36.4±3.05(18) | 44.65±3.3(29) | 56.27±3.46(18) | 61.14±28.54(32) | 97.5±0.63(5) | 64.7±0.0(11) | 64.39±0.59(12) | 50.2±0.98(25) |
| letter | 30.3±0.0(30) | 30.3±0.0(30) | 38.73±3.53(13) | 35.91±0.0(19) | 34.02±0.74(23) | 31.08±0.55(29) | 74.07±0.0(1) | 36.53±0.0(17) | 45.37±0.0(3) | 36.4±3.05(18) | 30.2±0.94(32) | 38.97±8.38(12) | 55.26±11.04(2) | 34.74±0.0(21) | 33.39±0.0(24) | 43.82±1.97(6) | 42.38±0.25(8) |
| lymphography | 99.88±0.09(12) | 99.86±0.05(14) | 99.58±0.51(20) | 99.69±0.17(19) | 99.89±0.08(11) | 99.89±0.08(11) | 99.84±0.22(15) | 99.52±0.15(22) | 99.52±0.15(22) | 99.73±0.3(18) | 67.04±13.87(31) | 94.94±3.85(29) | 75.91±13.38(29) | 99.8±0.3(17) | 99.87±0.06(13) | 94.6±2.36(17) | 88.19±1.78(30) |
| magic.gamma | 73.18±1.93(12) | 72.28±1.99(14) | 72.17±2.71(15) | 74.53±0.0(18) | 69.18±1.6(32) | 62.31±13.7(25) | 63.86±0.0(28) | 68.0±0.0(27) | 63.58±0.0(29) | 62.97±1.07(30) | 70.53±1.36(25) | 59.23±4.32(31) | 47.55±16.52(32) | 82.81±0.01(10) | 81.76±0.01(11) | 76.09±1.17(14) | 75.19±3.5(17) |
| mammography | 89.58±0.17(5) | 89.93±0.0(3) | 78.93±5.71(23) | 85.01±0.0(16) | 85.54±7.43(14) | 69.94±8.59(31) | 73.87±0.2(27) | 80.9±0.0(27) | 71.5±7.4(30) | 71.5±7.4(30) | 89.62±0.87(4) | 76.03±14.65(25) | 78.83±0.66(12) | 87.55±0.0(10) | 87.29±0.01(11) | 80.47±0.62(22) | 77.85±0.91(24) |
| mnist | 90.21±0.0(10.5) | 90.21±0.0(10.5) | 81.9±2.67(20) | 62.34±0.0(29) | 79.96±6.44(23) | 90.07±0.35(13) | 50.21±0.0(30) | 50.0±0.0(31.5) | 50.0±0.0(31.5) | 66.37±11.03(27) | 64.74±7.74(28) | 72.19±7.16(25) | 83.13±1.64(19) | 93.6±0.0(3) | 93.04±0.0(4) | 81.37±0.93(21) | 72.55±0.64(24) |
| musk | 100.0±0.0(8.5) | 100.0±0.0(8.5) | 76.65±18.72(31) | 89.92±0.0(11) | 74.3±12.58(19) | 100.0±0.0(8.5) | 48.64±0.0(28) | 78.32±2.13(28) | 99.87±0.0(18) | 99.99±0.01(17) | 99.67±0.35(20) | 95.01±4.27(26) | 33.99±33.18(32) | 100.0±0.0(8.5) | 100.0±0.0(8.5) | 97.18±0.37(25) | 80.0±0.0(30) |
| optdigits | 58.17±0.0(24.5) | 58.17±0.0(24.5) | 34.12±8.29(31) | 92.15±5.51(24) | 74.3±12.58(19) | 67.46±4.94(21) | 48.64±0.0(28) | 50.0±0.0(26.5) | 50.0±0.0(26.5) | 39.45±18.65(30) | 32.77±8.62(32) | 40.04±20.45(29) | 55.25±2.9(13) | 92.5±3.0(8) | 94.82±0.49(5) | 94.82±0.49(5) | 67.73±1.71(20) |
| pageblocks | 86.16±0.0(20) | 86.12±0.0(21) | 84.85±1.19(23) | 65.62±0.0(32) | 72.83±9.94(31) | 88.05±1.19(13) | 87.39±0.0(16) | 80.85±0.0(29) | 87.95±0.0(14) | 78.39±1.53(30) | 83.62±2.6(24) | 82.8±10.44(26) | 89.32±0.0(8.5) | 89.32±0.0(8.5) | 89.57±0.0(7) | 89.16±0.29(10) | 81.66±0.2(28) |
| pendigits | 94.5±0.0(18) | 94.37±0.0(19) | 83.45±6.13(27) | 93.55±0.0(20) | 67.93±21.6(30) | 89.97±2.03(24) | 89.71±0.0(25) | 90.74±0.0(23) | 92.95±0.0(21) | 46.29±11.35(32) | 92.13±1.02(22) | 56.89±21.83(31) | 75.91±13.38(29) | 99.7±0.0(6) | 99.17±0.0(6) | 94.6±2.36(17) | 81.66±0.2(28) |
| pima | 73.18±1.93(12) | 72.28±1.99(14) | 72.17±2.71(15) | 74.76±1.6(49) | 60.45±5.01(28) | 62.31±13.7(25) | 55.19±6.24(30) | 66.59±1.51(22) | 60.56±1.57(27) | 57.99±2.85(29) | 62.68±7.55(24) | 54.54±6.0(31) | 47.55±16.52(32) | 78.87±1.43(4) | 76.58±1.39(7) | 76.12±1.04(8) | 68.95±0.96(21) |
| satellite | 74.14±0.25(19) | 66.63±0.0(31) | 72.3±1.73(25) | 85.5±0.0(2) | 79.87±0.56(13) | 78.76±0.92(38) | 66.91±0.0(30) | 68.34±0.0(29) | 62.22±0.0(32) | 76.19±2.66(18) | 69.73±1.15(26) | 72.99±2.01(23) | 73.38±4.44(21) | 81.43±0.0(9) | 80.47±0.0(10) | 83.57±2.48(5) | 68.77±2.26(27) |
| satimage-2 | 98.98±0.04(18) | 98.17±0.0(21) | 96.07±0.57(27) | 97.95±0.0(22) | 97.57±1.19(24) | 98.99±0.06(17) | 86.9±0.0(31) | 92.95±0.0(21) | 97.09±0.0(25) | 92.94±2.84(29) | 86.67±0.52(19) | 91.82±4.43(30) | 92.92±0.04.5) | 99.77±0.0(3) | 99.77±0.0(3) | 96.99±2.56(26) | 83.3±4.71(32) |
| shuttle | 99.35±0.0(27) | 99.36±0.0(20) | 65.66±5.68(28) | 88.64±0.0(24) | 95.53±0.74(25) | 70.44±16.28(31) | 98.96±0.0(20) | 99.47±0.0(19) | 99.33±0.0(22) | 99.79±0.07(13) | 71.68±3.88(30) | 73.78±25.27(30) | 47.55±16.23(9.5) | 99.92±0.04.5) | 99.89±0.0(9) | 99.85±0.11(11) | 89.7±0.05(26) |
| skin | 66.05±0.19(25) | 59.73±0.31(28) | 91.27±7.09(8) | 86.92±0.35(20) | 48.48±2.87(30) | 64.95±2.25(26) | 87.55±0.76(16) | 47.21±0.18(31) | 49.14±0.18(29) | 59.95±4.47(27) | 75.51±5.43(21) | 67.91±30.02(24) | 43.8±32.33(32) | 43.8±32.33(32) | 98.34±0.04(5) | 73.81±10.45(22) | 83.48±0.19(18) |
| smtp | 81.93±5.07(25) | 81.81±7.32(26) | 84.22±6.88(22) | 82.75±5.51(24) | 54.55±5.98(33) | 78.78±13.12(28) | 71.15±1.56(13) | 91.15±1.56(13) | 88.26±2.46(15) | 85.24±6.6(19) | 73.03±6.65(30) | 77.13±15.93(31) | 43.8±32.33(32) | 43.8±32.33(32) | 93.1±2.3(7) | 88.75±5.41(14) | 86.47±0.96(18) |
| spambase | 81.4±0.0(14.5) | 81.4±0.0(14.5) | 82.26±3.55(10) | 77.88±0.0(19) | 82.5±2.1.3(48) | 81.78±0.36(11) | 41.13±0.0(30) | 72.09±0.0(25) | 68.83±0.0(30) | 70.24±5.02(26) | 72.39±6.9(24) | 69.4±4.42(29) | 75.37±4.45(21) | 83.07±0.0(6) | 82.36±0.0(9) | 80.71±0.86(16) | 70.0±0.04(27) |
| speech | 36.36±0.00(27.5) | 36.36±0.00(27.5) | 48.56±4.54(7) | 36.66±0.0(24) | 38.7±3.4(15) | 36.6±3.1(14(25) | 36.27±7.84(33) | 48.88±2.88(5) | 35.86±0.0(31) | 37.09±3.68(30) | 38.02±2.67(18) | 50.66±1.9(2) | 98.96±2.2(14) | 38.42±0.01(16) | 36.31±0.0(30) | 46.62±1.16(14) | 43.06±0.5(58) |
| stamps | 93.08±1.28(14) | 92.71±1.68(16) | 87.3±1.69(23) | 91.8±1.19(10) | 66.32±7.84(31) | 81.46±3.76(24) | 94.32±2.84(29) | 81.46±3.76(24) | 87.62±2.01(22) | 71.69±3.68(30) | 91.84±2.94(15) | 91.08±1.68(29) | 91.53±0.0(15) | 94.5±0.0(15) | 94.23±0.17(18) | 88.17±5.41(14) | 82.21±5.72(26) |
| thyroid | 98.55±0.0(9.5) | 98.55±0.0(9.5) | 98.42±0.51(13) | 98.6±0.0(5) | 97.38±3.22(25) | 95.14±0.8.5(23) | 96.29±0.0(18) | 93.81±0.26(26) | 97.55±0.01(15) | 88.77±3.69(32) | 96.06±1.65(19) | 94.96±1.55(24) | 94.96±1.55(24) | 98.69±0.04.5) | 98.68±0.0(3.5) | 95.35±0.6(21) | 89.11±0.02(31) |
| vertebral | 42.63±2.75(27) | 42.08±3.49(28) | 49.78±7.82(19) | 40.09±3.97(30) | 50.67±10.71(16) | 46.73±8.74(21) | 57.21±3.04(13) | 26.34±2.49(32) | 41.95±4.78(29) | 44.79±1.71(25) | 31.66±5.28(31) | 50.6±11.93(17) | 43.84±24.28(26) | 59.96±1.83(11) | 46.59±1.67(22) | 70.61±1.1(16) | 59.97±2.61(10) |
| vowels | 52.12±0.02(30) | 52.29±0.0(29) | 54.59±10.55(26) | 53.31±0.0(27) | 63.11±9.82(20) | 68.49±3.71(19) | 88.48±0.0(1) | 82.82±0.0(28) | 61.47±0.0(22) | 55.73±4.43(23) | 55.52±8.1(24) | 42.55±11.56(31) | 54.74±21.32(25) | 79.83±0.0(14) | 76.3±0.0(16) | 81.22±1.57(13) | 72.28±0.42(18) |
| waveform | 64.84±0.0(21) | 64.84±0.0(21) | 64.8±2.42(22) | 69.28±0.0(15) | 75.95±7.58(4) | 64.99±3.1(20) | 50.63±0.0(31) | 72.36±0.0(11) | 59.44±0.0(27) | 59.94±2.58(26) | 60.96±5.08(25) | 51.89±7.39(30) | 61.7±5.0(18) | 75.93±0.0(5) | 76.62±0.0(2) | 68.33±2.15(17) | 59.07±0.68(28) |
| wbc | 99.35±0.13(10) | 99.32±0.14(11) | 95.97±10.13(23) | 90.0±4.1(17) | 66.22±5.1(28) | 99.14±0.0(14) | 99.14±0.0(14) | 94.9±0.25(18) | 93.81±0.62(20) | 97.0±4.1(7.27) | 97.71±4.0(17) | 86.76±15.13(27) | 64.22±28.21(32) | 95.95±0.35(11) | 82.7±0.0(10) | 68.33±2.15(17) | 71.28±0.16(23) |
| wdbc | 99.14±0.36(14.5) | 99.14±0.22(14.5) | 98.86±0.76(19) | 98.55±0.28(23) | 96.22±3.1(28) | 98.96±0.2(18) | 98.96±0.2(18) | 99.18±0.21(13) | 56.88±0.0(27) | 56.88±0.0(27) | 96.98±2.04(26) | 77.75±25.27(30) | 40.08±33.97(32) | 99.11±0.22(16) | 67.26±0.0(13) | 62.9±0.15(20) | 74.14±0.06(27) |
| wilt | 35.41±0.0(28) | 26.07±0.0(32) | 74.62±4.1(8) | 39.1±0.0(26) | 44.01±5.48(22) | 51.38±5.33(19) | 55.02±0.0(18) | 32.09±0.0(31) | 37.48±0.0(27) | 34.41±1.7(30) | 41.1±7.0(25) | 41.81±7.29(24) | 49.49±12.62(20) | 55.82±0.0(17) | 55.82±0.0(17) | 72.42±2.57(10) | 77.44±0.57(6) |
| wine | 94.25±1.47(20) | 93.79±1.58(23) | 95.38±2.4(19) | 95.63±2.63(18) | 74.32±28.64(29) | 94.11±1.86(21) | 94.11±1.86(21) | 86.37±4.37(27) | 73.86±5.42(30) | 92.16±4.34(24) | 90.94±4.75(26) | 66.17±39.61(31) | 43.8±32.3(32) | 98.12±0.4(13) | 98.12±0.4(13) | 93.08±0.33(3) | 83.09±5.95(28) |
| wpbc | 54.42±2.76(24) | 52.54±2.3(26) | 57.46±2.93(20) | 60.91±2.67(16) | 60.11±5.48(17) | 51.39±5.74(28) | 82.5±1.3(267) | 52.33±2.93(27) | 49.5±2.47(30) | 82.67±5.47(5) | 51.32±3.74(29) | 46.99±2.56(31) | 43.78±4.34(32) | 72.67±0.81(9) | 58.14±1.82(19) | 90.3±2.5(3) | 61.89±2.06(15) |
| yeast | 42.39±0.01(29) | 43.24±0.0(26) | 45.07±3.33(19) | 42.88±0.0(28) | 47.64±6.7(11) | 47.64±6.7(11) | 38.44±0.0(32) | 38.88±0.0(31) | 44.64±0.0(22) | 47.62±5.96(12) | 46.51±5.8(15) | 51.03±3.92(2) | 48.42±5.46(9) | 44.25±0.0(24) | 43.95±0.0(25) | 47.74±0.5(10) | 47.74±0.5(10) |
| yelp | 59.14±0.04(19) | 59.16±0.0(18) | 53.6±2.03(28) | 59.95±0.0(15) | 56.11±1.6(24) | 61.07±0.98(11.5) | 60.21±0.0(14) | 60.21±0.0(14) | 57.39±0.0(20) | 49.9±3.3(30) | 56.26±3.78(23) | 49.87±1.25(31) | 50.67±1.15(29) | 67.77±0.0(3) | 67.16±0.0(5) | 53.71±0.22(27) | 56.47±0.84(22) |
| MNIST-C | 78.35±0.01(17.5) | 78.35±0.01(17.5) | 71.21±1.5(24) | 70.43±0.0(25) | 80.08±1.1(6(13) | 79.34±0.41(16) | 55.03±1.77(29) | 50.0±0.0(31.5) | 50.0±0.0(31.5) | 49.9±3.3(30) | 69.39±5.31(26) | 63.71±6.37(28) | 57.2±11.1(29) | 83.6±0.0(7) | 82.7±0.0(10) | 83.46±0.11(9) | 71.28±0.16(23) |
| FashionMNIST | 87.6±0.0(16.5) | 87.6±0.0(16.5) | 82.19±0.96(22) | 75.42±0.0(26) | 67.19±0.84(14) | 88.03±0.0(24) | 65.53±1.77(29) | 50.0±0.0(31.5) | 56.88±0.0(27) | 75.45±2.22(25) | 79.28±4.1(23) | 70.8±4.89(28) | 51.58±14.44(30) | 89.67±0.0(9) | 89.3±0.0(11) | 89.74±0.67(8) | 74.14±0.06(27) |
| CIFAR10 | 67.42±0.0(11.5) | 67.42±0.0(11.5) | 62.75±1.09(21) | 57.89±0.0(26) | 67.19±0.84(14) | 67.53±0.72(8.5) | 52.06±1.59(31) | 56.88±0.0(27) | 50.0±0.0(30.5) | 62.15±2.22(28) | 61.62±4.35(23) | 53.97±2.96(30) | 53.36±2.12(28) | 67.5±0.0(10) | 67.26±0.0(13) | 62.9±0.15(20) | 61.25±0.12(24) |
| SVHN | 60.79±0.0(15.5) | 60.79±0.0(15.5) | 58.87±0.87(2.15) | 54.65±0.0(25) | 61.25±0.77(9.5) | 60.75±0.49(17) | 52.98±0.67(29) | 50.0±0.0(30.5) | 50.0±0.0(30.5) | 53.92±3.05(27) | 54.5±4.0(2(26) | 53.36±2.12(28) | 49.99±2.47(32) | 61.54±0.0(17) | 61.14±0.01(11) | 60.91±0.03(13) | 57.8±0.0(57.6) |
| MVTec-AD | 76.21±1.81(22) | 76.37±1.91(21) | 72.28±2.39(27) | 75.97±1.84(23) | 81.1±1.8(12) | 77.06±2.1(20) | 77.06±2.1(20) | 81.99±2.78(10) | 50.0±0.0(31.5) | 72.31±3.37(28) | 72.21±2.12(28) | 64.69±5.63(29) | 60.52±1.12(30) | 83.75±0.49(9) | 78.81±0.45(16) | 93.08±0.33(3) | 73.32±0.42(26) |
| 20news | 54.59±0.74(23) | 54.39±0.41(24) | 51.59±3.0(31) | 53.57±0.35(26) | 59.68±1.8(78) | 54.92±1.04(20) | 54.92±1.04(20) | 55.92±0.43(29) | 54.11±0.24(25) | 55.64±4.17(19) | 72.21±3.37(28) | 51.48±4.08(32) | 58.02±0.8(12) | 58.02±0.8(12) | 58.14±1.82(19) | 58.5±0.44(10) | 58.0±0.6(13) |
| agnews | 56.9±0.02(2.5) | 56.9±0.02(2.5) | 50.15±1.19(30) | 55.69±0.0(25) | 58.63±1.14(16) | 59.87±0.86(15) | 59.87±0.86(15) | 55.05±0.0(27) | 55.11±0.0(26) | 49.83±3.63(31) | 56.95±3.54(21) | 51.03±3.6(29) | 49.65±0.76(32) | 66.46±0.0(7) | 65.35±0.0(8) | 57.32±0.11(20) | 60.06±0.42(14) |
| **Rank(avg)** | | | | | | | | | | | | | | | | | |
| All | 17,062 | 17,719 | 19,43 | 18,868 | 18,772 | 20,237 | 20,491 | 21,219 | 21,605 | 21,757 | 23,667 | 24,965 | 23,158 | 8,605 | 10,632 | 12,447 | 21,439 |
| $d \le 20$ | 1.000 | 0.971 | 0.997 | 1.000 | 0.999 | 1.000 | 1.000 | 1.000 | 1.000 | 1.000 | 1.000 | 1.000 | 1.000 | 0.007 | 0.062 | 0.437 | 1.000 |
| $d \le 50$ | 1.000 | 1.000 | 1.000 | 1.000 | 1.000 | 1.000 | 1.000 | 1.000 | 1.000 | 1.000 | 1.000 | 1.000 | 1.000 | 0.813 | 0.924 | 0.999 | 1.000 |
| | 1.000 | 1.000 | 1.000 | 1.000 | 1.000 | 1.000 | 1.000 | 1.000 | 1.000 | 1.000 | 1.000 | 1.000 | 1.000 | 0.574 | 0.847 | 0.995 | 1.000 |
| **Rank(avg)** | | | | | | | | | | | | | | | | | |
| All | 17,851 | 18,368 | 20,254 | 19,623 | 19,509 | 20,956 | 21,246 | 22,026 | 22,395 | 21,737 | 24,561 | 25,895 | 24.0 | 9,079 | 11,105 | 12,991 | 22,263 |
| $d \le 100$ | 0.998 | 0.999 | 1.000 | 1.000 | 1.000 | 1.000 | 1.000 | 1.000 | 1.000 | 1.000 | 1.000 | 1.000 | 1.000 | 0.112 | 0.315 | 0.670 | 0.846 |
| $d \le 500$ | 1.000 | 1.000 | 1.000 | 1.000 | 1.000 | 1.000 | 1.000 | 1.000 | 1.000 | 1.000 | 1.000 | 1.000 | 1.000 | 0.752 | 0.860 | 0.958 | |
| | 1.000 | 1.000 | 1.000 | 1.000 | 1.000 | 1.000 | 1.000 | 1.000 | 1.000 | 1.000 | 1.000 | 1.000 | 1.000 | 0.607 | 0.756 | 0.846 | |

Table 13.1: Average AUPR ± standard dev. over five seeds for the semi-supervised setting on ADBench. Rank of each model per dataset is provided (in parentheses) (the lower, the better). We use blue and green respectively to mark the top-1 and the top-2 method.

| Dataset | FoMo(D=100) | FoMo(D=20) | DTE-NP | kNN | ICL | DTE-C | LOF | CBLOF | FeatureBagging | SLAD | DDPM | OCSVM | DTE-IG | IForest | MCD |
|---|---|---|---|---|---|---|---|---|---|---|---|---|---|---|---|
| aloi | 6.93±0.05(2) | 6.23±0.22(10.5) | 6.05±0.06(15) | 6.02±0.06(16.5) | 5.5±0.13(30) | 5.76±0.04(26) | 6.54±0.05(5) | 6.4±0.02(9) | 6.8±0.19(3) | 5.99±0.04(18) | 5.97±0.06(20) | 6.52±0.07(7) | 5.98±0.14(19) | 5.82±0.06(1)(23) | 5.55±0.07(29) |
| amazon | 11.1±1.46(9.5) | 10.68±0.14(19) | 11.73±0.0(1) | 11.69±0.0(3) | 10.19±0.08(23) | 11.15±0.65(7.5) | 11.04±0.0(14) | 11.46±0.02(10.5) | 11.06±0.02(12.5) | 9.74±0.02(29) | 10.76±0.02(16) | 11.06±0.0(12.5) | 10.16±1.08(25) | 11.07±0.0(19)(11) | 11.7±0.03(2) |
| annthyroid | 48.81±0.0(24) | 49.52±0.0(22) | 68.15±0.38(4) | 68.07±0.0(5) | 45.83±2.16(27) | 62.44±2.39(6) | 53.53±0.0(19) | 63.62±4.4(10) | 48.49±8.07(25) | 70.58±0.9(2)(2) | 62.88±2.59(11) | 60.11±0.0(13) | 49.87±9.05(21) | 59.02±5.39(15) | 59.75±0.05(14) |
| backdoor | 43.86±14.97(11) | 14.76±10.63(19) | 45.7±12.5(10) | 46.54±1.4(9) | 89.18±1.04(1) | 62.43±2.39(6) | 53.47±2.6(7) | 9.07±1.37(22) | 49.52±8.43(8) | 4.83±0.1(32) | 14.2±0.63(20) | 7.66±0.06(26) | 81.99±4.2(4)(5) | 9.37±1.48(21) | 22.16±13.73(17) |
| breastw | 99.03±0.34(12) | 98.15±0.29(17) | 99.19±0.14(5.5) | 98.92±0.32(13) | 96.79±1.61(21) | 88.25±1.06(27) | 80.01±0.09(29) | 99.06±0.24(11) | 52.39±0.09(31) | 99.47±0.2(1)(2) | 98.6±0.51(15) | 99.35±0.21(4) | 81.4±7.86(28) | 99.49±0.1(2)(1) | 98.27±1.23(16) |
| campaign | 34.24±0.47(26) | 37.78±8.11(23) | 49.95±0.67(2) | 49.04±0.08(8) | 48.9±0.98(9) | 46.9±0.71(17) | 40.24±0.02(31) | 48.56±0.59(13) | 33.31±6.49(27) | 48.11±0.12(14) | 48.87±0.29(10) | 49.43±0.06(6) | 46.17±2.19(18) | 45.73±1.85(19) | 47.91±1.49(16) |
| cardio | 78.82±0.0(7) | 58.63±1.79(19) | 71.66±3.85(11) | 67.01±0.85(11) | 47.91±1.14(7)(29) | 47.91±1.14(29) | 78.53±2.0(4)(9) | 59.59±0.0(17) | 73.1±1.14(16) | 68.3±0.0(21) | 69.22±0.91(20) | 75.13±0.0(10) | 78.55±1.8(2)(10) | 85.59±0.9(2)(5) | 67.0±0.62(23) |
| cardiotocography | 63.55±0.0(6) | 55.67±1.79(19) | 55.88±1.14(13) | 57.43±0.0(15) | 46.86±3.84(26) | 53.34±1.26(21) | 57.32±0.0(16) | 61.71±1.92(8) | 57.04±2.15(17) | 49.37±0.19(25) | 51.31±1.98(23) | 66.19±0.0(5) | 39.59±5.95(31) | 62.85±3.42(7) | 52.83±0.7(22) |
| celeba | 8.39±0.38(23) | 6.04±1.19(28) | 10.65±0.49(18) | 11.92±0.5(16) | 9.74±0.68(20) | 14.19±2.31(11) | 3.61±0.15(31) | 18.5±4.43(5) | 3.87±0.26(30) | 2.9±0.92(32) | 18.03±2.6(6) | 20.27±0.9(3) | 13.4±2.37(12) | 11.7±1.35(17) | 19.02±3.4(4) |
| census | 16.01±2.97(16) | 14.2±1.57(23) | 21.05±0.67(5) | 21.05±0.67(5) | 20.34±0.6(14) | 17.94±1.0(15) | 13.71±0.42(26) | 15.96±0.6(28) | 78.1±13.95(3) | 6.96±5.18(26) | 73.27±3.36(4) | 20.32±0.66(9) | 16.3±1.77(17) | 14.19±0.73(24) | 28.98±1.47(1) |
| cover | 72.57±8.38(5) | 68.27±2.5(6) | 59.97±10.57(8) | 55.79±3.74(9) | 34.48±16.39(13) | 63.73±1.2(33)(7) | 82.92±2.1(9)(1) | 65.25±0.25(11) | 65.25±0.25(11) | 45.1±0.17(14) | 73.27±3.36(4) | 22.28±1.01(18) | 80.3±4.6(2) | 8.66±1.53(25) | 3.14±0.18(28) |
| donors | 98.43±0.48(1) | 75.72±5.93(9) | 85.55±4.56(6) | 89.09±0.94(4) | 98.35±0.8(7)(2) | 71.33±3.89(10) | 63.39±1.89(12) | 46.46±1.12(14) | 46.18±9.8(15) | 8.93±0.08(16.5) | 26.66±2.91(28) | 42.71±0.86(18) | 95.77±2.6(3) | 40.51±3.59(20) | 31.24±13.62(26) |
| fault | 63.08±0.0(8) | 60.86±3.58(17) | 62.17±0.0(14)(11) | 61.98±0.0(13) | 63.18±0.0(7) | 62.14±0.92(4) | 50.84±0.0(32) | 61.29±1.82(15) | 50.84±0.0(32) | 66.69±0.19(1) | 23.63±0.06(29) | 61.12±0.0(16) | 63.83±1.23(5) | 59.19±2.0(22) | 73.24±13.62(26) |
| fraud | 55.13±3.77(8) | 34.75±7.84(20) | 42.17±63(14) | 34.68±7.84(10)(9) | 12.8±1.17(3) | 41.51±5.91(9) | 55.09±8.19(9) | 27.77±2.14(27) | 36.09±8.99(14) | 44.97±5.43(13) | 95.14±1.3(4)(9) | 26.76±7.68(21) | 51.14±8.64(11) | 18.22±3.66(29) | 60.06±3.89(6) |
| glass | 83.6±4.6(3)(2) | 70.36±9.67(5) | 37.38±15.51(13) | 42.32±8.47(8) | 92.35±8.32(1) | 41.51±5.91(9) | 38.12±9.91(12) | 31.7±2.74(16) | 36.09±8.99(14) | 41.15±3.98(11) | 31.21±1.13(17) | 26.76±7.68(21) | 80.57±12.8(3)(3) | 21.37±3.58(25) | 20.29±3.22(27) |
| hepatitis | 99.45±1.14(5) | 99.45±0.99(4.5) | 82.32±10.26(14) | 90.31±4.36(10) | 99.83±0.38(1)(8) | 99.83±0.38(8) | 43.67±0.79(31) | 63.36±6.8(23) | 44.6±9.85(30) | 99.79±0.47(3) | 95.14±1.3(4) | 77.63±3.49(15) | 99.8±0.45(2) | 35.36±6.09(26) | 56.8±8.0(24) |
| http | 90.58±5.01(11) | 96.39±2.57(7) | 97.1±0.04(6) | 100.0±0.0(1) | 70.82±42.06(17) | 55.45±5.61(20) | 97.12±3.92(5) | 90.31±1.45(13) | 8.21±1.15(30) | 88.09±7.46(15) | 99.96±0.06(2) | 99.88±0.26(3) | 78.8±2.53(16) | 53.43±12.08(21) | 92.16±1.88(9) |
| imdb | 10.53 ±2.36(2) | 9.08±0.51(15.5) | 8.97±0.02(20.5) | 8.97±0.02(20.5) | 8.9±0.15(27) | 8.9±0.15(27) | 8.95±0.05(23) | 9.03±0.02(17.5) | 9.03±0.02(17.5) | 9.79±0.03(8) | 8.7±0.03(31) | 8.8±5.0.0(26) | 10.06±1.49(6) | 8.97±0.17(20.5) | 9.45±0.04(11) |
| internetads | 38.29±5.22(27) | 39.5±7.81(25) | 51.3±2.02(10) | 49.22±0.0(13) | 60.03±1.39(4) | 55.22±3.65(7) | 50.43±0.0(11) | 47.04±0.0(82)(1) | 49.26±1.85(12) | 60.52±1.05(3) | 47.7±0.16(16) | 48.15±0.0(15) | 58.68±6.22(5) | 29.2±1.74(32) | 34.36±0.0(28) |
| ionosphere | 97.75±0.76(8) | 97.77±0.44(7) | 98.22±1.02(2) | 97.95±0.69(5) | 99.00±0.32(1) | 96.83±0.38(13) | 94.58±1.57(20) | 97.26±1.05(9) | 94.94±1.4(19) | 95.38±0.51(17.5) | 96.43±0.5(15) | 97.45±0.51(10) | 96.91±2.1(12) | 91.7±1.89(23) | 96.66±0.39(14) |
| landsat | 58.24±0.0(4) | 54.75±2.34(7) | 54.52±4.05(8) | 54.85±0.0(6) | 53.12±1.57(9) | 36.75±1.23(23) | 61.37 +0.0(2) | 36.89±0.24(22) | 61.49 +0.23(1) | 45.1±0.17(14) | 34.83±0.91(26) | 37.01±0.0(21) | 32.65±2.84(30) | 47.31±3.5(12) | 39.68±4.51(17) |
| letter | 9.33±0.0(12) | 9.13±0.39(14) | 8.57±0.0(13.22) | 8.7±0.0(20) | 12.8±1.17(3) | 8.95±0.1(31.5) | 11.26±0.0(6) | 8.33±0.08(25) | 11.66±0.51(4) | 8.93±0.08(16.5) | 9.53±0.5(11) | 8.26±0.0(26) | 10.23±1.12(9) | 8.22±0.0(25.27) | 8.1±0.4(29) |
| lymphography | 99.15±1.26(10) | 100.0±0.0(2.5) | 99.34±0.93(6.5) | 99.17±0.94(9) | 100.0±0.0(2.5) | 86.77±9.24(25) | 84.16±4.27(27) | 98.26±0.34(15) | 72.73±15.5(29) | 99.24±0.12(6.5) | 99.3±1.0(28) | 100.0 +0.0(2.5) | 99.76±0.55(5) | 94.38±3.28(21.5) | 86.76±6.31(26) |
| magic.gamma | 87.54±0.18(5) | 87.37±0.12(35) | 86.15±0.74(8) | 85.86±0.0(9) | 81.33±0.57(13) | 83.95±0.5(11) | 86.34±0.0(7) | 80.24±0.0(16) | 86.89±0.5(6) | 81.95±0.84(11.5) | 87.95±0.84(11.5) | 79.16±0.0(17) | 83.11±0.7(8)(2) | 80.27±1.07(15) | 77.21±0.09(21) |
| mammography | 36.28±1.26(16) | 45.99±0.99(3) | 42.09±0.86(5) | 41.27±0.0(6)(8) | 17.11±3.75(30) | 39.85±2.6(13) | 34.07±0.0(17) | 41.08±0.1(39) | 29.34±1.55(19) | 18.98±1.05(28) | 19.93±3.92(27) | 40.52±0.0(11) | 33.36±9.9(18) | 37.94±3.21(14) | 7.96±0.2(32) |
| mnist | 57.97±0.0(16) | 32.15±3.35(28) | 73.68 ±1.3(1) | 72.72 +0.02(2) | 68.45±0.6(37) | 56.26±3.26(18) | 70.97±0.0(4) | 66.49±0.3(6)(9) | 69.29±1.07(6) | 68.39±0.46(8) | 62.42±4.39(14) | 66.2±0.0(10) | 56.1±8.07(19) | 54.15±6.53(22) | 55.75±6.4(20) |
| musk | 97.0±2.27(19) | 94.34±6.15(21) | 100.0 +0.0(8.5) | 100.0 +0.0(8.5) | 92.21±6.32(22) | 100.0 +0.0(8.5) | 100.0 +0.0(8.5) | 100.0 +0.0(8.5) | 100.0 +0.0(8.5) | 100.0 +0.0(8.5) | 100.0 +0.0(8.5) | 100.0 +0.0(8.5) | 88.87±24.88(25) | 40.99±26(30) | 66.32±12.08(28) |
| optdigits | 31.88±0.0(7) | 20.6±5.31(14) | 31.75±8.66(8) | 29.11±0.0(9) | 50.94±8.59(1) | 15.34±2.17(17) | 43.6 +0.0(2) | 13.97±1.2(18) | 41.23±2.99(4) | 36.3±0.94(6) | 25.56±4.25(11) | 6.92±0.0(23) | 22.06±15.09(12) | 15.41±3.21(16) | 7.1±0.1(22) |
| pageblocks | 62.67±0.0(15) | 60.24±0.0(19) | 91.89±3.3(3) | 96.99 +0.0(1) | 68.11±2.3(5) | 66.42±1.23(9) | 71.07 +0.0(2) | 70.6±0.1(13) | 70.16±1.16(4) | 64.7±0.79(11) | 62.1±0.95(16) | 64.25±0.0(12) | 57.46±2.99(25) | 43.42±2.0(30) | 63.17±0.04(14) |
| pendigits | 66.33±0.0(9) | 86.15±0.0(4) | 79.7 +2.44(1) | 89.17±0.94(9) | 66.41±7.58(8) | 48.44±5.92(16) | 78.5±5.0(7) | 51.24±0.47(15) | 85.67±2.53(5) | 35.35±0.15(22) | 61.14±4.73(10) | 51.78±0.0(14) | 59.2±0.53(11) | 58.79±5.21(12) | 68.64±3.1(20) |
| pima | 79.27 ±1.74(2) | 76.92±2.16(4) | 85.83±0.72(9) | 86.01±0.0(6) | 78.63±1.94(3) | 84.79±0.35(13) | 68.4±3.79(21) | 77.28±0.37(27) | 85.82±0.05(10) | 69.54±3.79(18) | 71.18±2.06(15.5) | 80.9±0.0(20) | 81.71±1.91(17) | 73.65±2.07(10) | 79.93±2.95(21) |
| satellite | 88.05 +0.02(3) | 86.85±0.0(5)(4) | 85.83±0.72(9) | 86.69±0.0(6) | 87.62±0.24(3) | 68.21±3.75(28) | 84.5±5.0(7) | 96.76±0.0(15) | 90.65±0.96(17) | 95.44±0.23(9) | 88.05±5.1(19) | 92.6±2.0(3.5) | 83.3±4.52(22) | 82.35±0.88(15) | 83.13±9.43(17) |
| satimage-2 | 92.78±0.0(14) | 92.32±1.27(15) | 96.16±0.0(7) | 97.86±0.0(6) | 94.7±1.19(10) | 94.03±0.0(11.23) | 96.96±0.0(15) | 96.77±0.12(18) | 46.35±2.5(9)(7) | 98.04±0.0(11) | 97.91±0.26(13) | 96.92±0.0(3.5) | 99.35±0.09(5) | 98.61±0.34(8) | 90.9±0.0(25) |
| shuttle | 99.36±0.17(4) | 99.59±0.0(15) | 98.14±0.48(9) | 97.86±0.0(14) | 99.72 +0.1(42) | 94.03±0.0(11.23) | 99.75 +0.0(1) | 49.7±6.0(4.10) | 49.21±1.1(24) | 98.73±7.88(6) | 76.37±5.79(7) | 97.67±0.0(15) | 99.35±0.09(5) | 64.58±1.09(14) | 62.39±0.42(17) |
| skin | 51.04±1.72(22) | 56.78±1.84(3) | 50.2±6.39(9) | 50.53±5.92(6) | 32.46±1.0(29) | 50.37±6.15(7) | 48.09±7.38(14) | 69.47±0.5(9) | 0.38±0.25(31) | 50.6±6.3(5) | 40.81±13.36(15) | 66.31±0.51(12) | 33.6±2.33.3(18) | 1.1±0.12(28) | 1.18±0.08(26) |
| smtp | 38.18±9.9(17) | 2.82±0.39(22.5) | 48.12±0.49(22) | 83.32±0.08(8) | 3.8±1.83(25) | 72.71±0.0(9) | 72.71±0.0(9) | 82.03±0.41(14) | 68.4±2.58(31) | 83.64±0.14(3) | 72.89±0.42(28) | 82.19±0.0(12) | 80.99±2.36(18.5) | 83.26±1.32(1) | 81.78±2.94(17) |
| spambase | 80.99±0.0(18.5) | 2.82±0.39(22.5) | 3.17±0.0(10) | 2.8±0.0(25) | 3.3±0.5(5.5) | 2.85±0.1(20) | 3.15±0.0(11) | 2.7±0.02(32) | 2.98±0.0(15) | 3.1±0.06(12) | 3.0±0.29(14) | 2.78±0.0(27) | 2.88±0.48(18) | 3.25±1.0(8) | 2.83±0.0(7.21) |
| speech | 2.94±0.31(17) | 4.678 +3.5(1) | 82.47±4.08(3) | 80.94±0.0(10) | 51.51±12.75(29) | 81.67±0.97(12) | 60.57±0.05(25) | 86.83±4.29(20) | 36.49±17.33(31) | 50.61±1.32(6)(24) | 64.74±12.87(13) | 64.91±7.95(11) | 45.67±16.28(30) | 79.66±5.62(11) | 41.7±6.16(30) |
| stamps | 89.37 +3.5(1) | 59.93±0.0(26) | 81.03±0.31(7) | 58.58±8.9(15) | 26.11±2.49(13) | 58.75 +7.3(2) | 33.87±4.3(8) | 1.51±0.0(16.3) | 32.89±4.98(9) | 19.87±4.37(27) | 35.84±9.27(6) | 22.23±2.0(21) | 51.5±10.55(4) | 20.75±1.98(24) | 20.96±2.2(23) |
| thyroid | 67.0±0.0(19) | 57.33±6.69(3) | 31.59±1.59(8) | 30.21±0.0(9) | 27.39±5.75(13) | 38.1±4.89(4) | 33.0±0.6(6) | 23.85±4.5(3)(7) | 32.73±5.31(7) | 39.23±1.65(3) | 42.72 +4.8(2) | 27.43±0.0(12) | 33.63±6.25(5) | 11.97±1.05(24) | 4.36±0.0(32) |
| vertebral | 69.39 +3.97(1) | 19.5±1.38(12) | 27.87±0.0(4) | 27.0±0.0(5) | 9.99±1.13(18) | 58.75 +7.3(2) | 30.66 +0.0(1) | 22.49±1.4(9) | 28.73±3.9(3) | 5.31±0.0(32) | 9.31±1.05(21) | 10.91±0.0(16) | 19.61±0.0(17) | 10.53±0.76(17) | 7.83±0.02(27) |
| vowels | 24.1±0.0(16) | 23.31±0.0(18) | 16.33 +0.0(1) | 16.03±0.0(4) | 10.4±0.09(26) | 13.0±1.57(16) | 16.14 +0.0(2) | 13.72±0.03(10) | 16.68±0.07(3) | 9.96±0.05(29) | 4.78±0.42(45)(10) | 13.42±0.0(11) | 12.25±1.58(20) | 13.15±0.29(13) | 13.81±0.02(9) |
| waveform | 9.95±0.0(9) | 19.5±1.38(12) | 96.1±2.17(4) | 46.2±0.0(8) | 51.47±1.1(3) | 51.47±1.1(3) | 51.89 +0.0(2) | 42.54±0.1(2)(13) | 52.15 +0.36(1) | 46.89±0.1(47) | 41.78±0.12(14) | 41.57±0.0(15) | 44.08±4.87(11) | 32.8±2.4(22) | 25.81±4.48(27) |
| wbc | 96.46±2.49(3) | 94.29±3.42(7) | 59.79±0.0(5) | 59.15±0.08(8) | 63.08±0.96(3) | 55.01±1.04(17) | 63.6±0.0(2) | 57.55±0.25(11) | 63.94 +0.43(1) | 59.6±0.17(7) | 57.14±0.12(12) | 56.53±0.0(14) | 53.73±2.28(18) | 44.73±1.88(23) | 37.41±6.46(25) |
| wdbc | 93.82 ±8.3(2) | 92.75±5.82(5) | 96.64±5.58(9) | 95.11±1.81(11) | 17.39±0.59(17) | 17.39±0.59(17) | 21.17 +0.0(2) | 19.73±1.8(6) | 22.2 +0.34(1) | 19.98±0.07(4) | 19.55±0.08(9) | 19.42±0.0(11) | 16.69±1.63(20) | 16.46±0.64(21) | 15.92±1.04(22) |
| wilt | 77.46 ±0.0(1) | 27.35±0.0(4) | 12.2±1.6(14) | 12.25±0.0(13) | 15.6±0.0(13) | 15.47±0.47(5) | 15.97 +0.0(2) | 15.06±0.0(13) | 16.06 +0.0(12) | 15.4±0.26(6) | 15.08±0.04(12) | 15.0±0.0(14) | 14.21±0.95(20) | 13.85±0.49(21) | 14.62±0.97(19) |
| wine | 99.02±0.77(4) | 98.48±3.04(5) | 96.8±5.58(9) | 19.91±0.05(6) | 89.31 +5.37(1) | 60.35±4.88(10) | 91.9±5.2.64(12) | 44.8±1.47(18) | 40.97±2.44(22) | 87.45 +6.2(12) | 97.65±0.72(8) | 88.68±3.04(23) | 65.78±8.55(9) | 67.12±7.62(25) | 45.16±1.42(17) |
| wpbc | 67.36±4.34(8) | 75.17±5.73(4) | 69.02±13.9(17) | 46.11±2.74(15) | 89.31 +5.37(1) | 55.01±1.04(17) | 4.12±2.62(21) | 13.72±0.03(10) | 49.89±0.68(9) | 50.74±0.02(26) | 54.61±3.75(12) | 40.88±3.04(23) | 40.73±3.15(24) | 46.78±0.73(28.5) | 45.67±0.08(31) |
| yeast | 51.0±0.0(4) | 50.24±0.08(8) | 16.33 +0.0(1) | 16.03±0.0(4) | 10.4±0.09(26) | 13.0±1.57(16) | 48.94 +0.0(9) | 13.72±0.03(10) | 49.89±0.68(9) | 50.74±0.02(26) | 12.8±0.0(21)(17) | 13.42±0.0(11) | 12.25±1.58(20) | 13.15±0.29(13) | 13.81±0.02(9) |
| yelp | 13.9±3.0(8) | 14.35±0.69(7) | 16.33 +0.0(1) | 16.03±0.0(4) | 10.4±0.09(26) | 13.0±1.57(16) | 16.14 +0.0(2) | 13.72±0.03(10) | 16.68±0.07(3) | 9.96±0.05(29) | 12.8±0.0(17) | 13.42±0.0(11) | 12.25±1.58(20) | 13.15±0.29(13) | 13.81±0.02(9) |
| MNIST-C | 38.4±3.34(19) | 33.32±6.3(21) | 59.79±0.0(5) | 59.15±0.08(8) | 63.08±0.96(3) | 55.01±1.04(17) | 51.89 +0.0(2) | 42.54±0.12(13) | 52.15 +0.36(1) | 46.89±0.17(7) | 41.78±0.12(14) | 41.57±0.0(15) | 44.08±4.87(11) | 44.73±1.88(23) | 37.41±6.46(25) |
| FashionMNIST | 52.19±2.78(19) | 40.62±3.93(24) | 59.79±0.0(5) | 59.15±0.08(8) | 63.08±0.96(3) | 55.01±1.04(17) | 63.6±0.0(2) | 57.55±0.25(11) | 63.94 +0.43(1) | 59.6±0.17(7) | 57.14±0.12(12) | 56.53±0.0(14) | 53.73±2.28(18) | 44.73±1.88(23) | 37.41±6.46(25) |
| CIFAR10 | 17.45±0.52(16) | 15.34±0.69(25) | 15.53±0.0(4) | 15.34±0.0(7) | 17.39±0.59(17) | 17.39±0.59(17) | 21.17 +0.0(2) | 19.73±1.8(6) | 22.2 +0.34(1) | 19.98±0.07(4) | 19.55±0.08(9) | 19.42±0.0(11) | 16.69±1.63(20) | 16.46±0.64(21) | 15.92±1.04(22) |
| SVHN | 14.67±0.31(18) | 13.21±0.45(22) | 15.53±0.0(4) | 15.34±0.0(7) | 15.6±0.0(13) | 15.47±0.47(5) | 15.97 +0.0(2) | 15.06±0.0(13) | 16.06 +0.0(12) | 15.4±0.26(6) | 15.08±0.04(12) | 15.0±0.0(14) | 14.21±0.95(20) | 13.85±0.49(21) | 14.62±0.97(19) |
| MVTec-AD | 73.25±1.79(17) | 71.96±2.43(21) | 82.94±2.68(6) | 75.76±0.27(10)(12) | 85.11±2.96(4) | 17.33 +2.36(1) | 75.79±2.95(10) | 74.88±2.84(14) | 75.77±3.0(13) | 87.91 +2.3(12) | 73.66±2.72(15) | 73.03±2.82(18) | 82.85±3.49(7) | 70.01±3.07(23) | 80.51±2.92(8) |
| 20news | 13.51±0.34(12) | 15.55±1.39(3) | 15.61 +0.29(2) | 13.47±0.52(13) | 14.62±0.58(7) | 17.33 +2.36(1) | 15.04±0.67(5) | 12.63±0.64(18) | 25.9 +0.0(21) | 13.59±0.34(11) | 11.48±0.44(24) | 11.83±0.52(20) | 14.07±2.82(8) | 11.56±0.43(21) | 15.45±1.0(64) |
| agnews | 15.16±3.17(10) | 17.54±0.6(14) | 17.55±0.0(5) | 15.42±0.36(9) | 15.42±0.36(9) | 19.22±2.9(7.3) | 13.78±0.0(12) | 13.78±0.0(12) | 25.9 +0.0(21) | 12.5±0.06(19) | 11.85±0.03(22) | 12.82±0.0(14) | 12.81±2.3(115) | 11.94±0.32(21) | 14.62±0.14(11) |
| Rank(avg) | | | | | | | | | | | | | | | |
| All | 7.561 | 12.132 | 7.561 | 9.263 | 9.877 | 12.088 | 12.0 | 13.605 | 13.509 | 12.439 | 12.763 | 13.377 | 12.754 | 17.991 | 18.193 |
| d ≤ 20 | 0.006 | | 0.006 | 0.250 | 0.118 | 0.769 | 0.065 | 0.769 | 0.939 | 0.495 | 0.735 | 0.848 | 0.476 | 0.999 | 0.999 |
| d ≤ 50 | 0.605 | | 0.605 | 0.708 | 0.914 | 0.987 | 0.998 | 0.997 | 1.000 | 0.963 | 0.924 | 0.983 | 0.755 | 1.000 | 1.000 |
| | 0.371 | | 0.371 | 0.651 | 0.729 | 0.997 | 0.995 | 0.995 | 1.000 | 0.934 | 0.969 | 0.986 | 0.900 | 1.000 | 1.000 |
| Rank(avg) | | | | | | | | | | | | | | | |
| All | 10.868 | 12.825 | 7.965 | 10.368 | 12.614 | 12.614 | 12.526 | 14.202 | 14.123 | 12.947 | 13.325 | 13.921 | 13.377 | 18.798 | 18.93 |
| d ≤ 100 | | | 0.082 | 0.347 | 0.568 | 0.883 | 0.720 | 0.953 | 0.978 | 0.753 | 0.926 | 0.873 | 0.984 | 1.000 | 1.000 |
| d ≤ 500 | | | 0.485 | 0.728 | 0.898 | 0.993 | 0.967 | 0.989 | 1.000 | 0.871 | 0.960 | 0.944 | 0.997 | 1.000 | 1.000 |
| | | | 0.340 | 0.627 | 0.816 | 0.980 | 0.946 | 0.988 | 0.999 | 0.894 | 0.957 | 0.941 | 0.993 | 1.000 | 1.000 |

Table 13.2: Average AUPR ± standard dev. over five seeds for the semi-supervised setting on ADBench. Rank of each model per dataset is provided (in parentheses) (the lower, the better). We use `blue` and `green` respectively to mark the `top-1` and the `top-2` method.

| Dataset | VAE | PCA | PlanarFlow | HBOS | GANomaly | GOAD | DIF | COPOD | ECOD | DeepSVDD | LODA | DAGMM | DROCC | DTE-NP[avg] | KNN[avg] | ICL[avg] | DTE-C[avg] |
|---|---|---|---|---|---|---|---|---|---|---|---|---|---|---|---|---|---|
| aloi | 6.54±0.05(5) | 6.54±0.05(6) | 5.48±0.3(32) | 6.42±0.08(8) | 8.09±1.27(1) | 5.7±0.24(28) | 5.8±0.02(24) | 5.72±0.00(27) | 6.06±0.00(14) | 6.23±0.35(10.5) | 5.93±0.5(21) | 6.07±0.55(13) | 5.91±0.00(22) | 6.02±0.00(16.5) | 6.09±0.00(12) | 5.49±0.02(31) | 5.79±0.00(25) |
| amazon | 10.72±0.04(7.5) | 10.72±0.04(7.5) | 9.56±0.59(30) | 11.1±0.00(5) | 9.93±0.19(27) | 10.94±0.21(15) | 9.89±0.41(28) | 11.15±0.00(7.5) | 10.4±0.00(21) | 10.24±1.27(22) | 10.18±0.78(24) | 9.53±0.46(31) | 9.52±0.00(32) | 11.65±0.04(4.5) | 11.65±0.04(4.5) | 10.01±0.05(26) | 10.42±0.4(20) |
| annthyroid | 56.57±0.00(18) | 56.57±0.00(18) | 65.15±8.63(8) | 39.03±0.029(29) | 34.3±10.62(30) | 58.74±5.0(16) | 61.12±0.0(12) | 29.61±0.00(31) | 40.02±0.00(28) | 27.83±5.93(32) | 49.01±6.73(23) | 48.03±17.56(26) | 63.72±3.08(9) | 66.11±0.0(7) | 67.06±0.0(6) | 52.34±4.37(2(20) | 68.86±0.2(3) |
| backdoor | 7.97±0.24(24) | 7.9±0.13(25) | 32.15±23.78(14) | 8.56±0.26(23) | 27.87±6.63(16) | 6.31±1.94(28) | 17.81±1.03(18) | 11.35±0.00(19.5) | 4.84±0.1(30.5) | 84.77±2.79(3) | 5.96±3.84(29) | 90.95±8.37(26) | 84.59±1.93(4) | 40.48±3.81(12) | 32.06±0.76(15) | 87.63±1.03(2) | 35.68±0.79(13) |
| breastw | 99.17±0.17(7) | 99.19±0.15(5.5) | 97.47±1.08(20) | 99.08±0.27(9.5) | 93.78±2.46(25) | 98.77±0.37(14) | 52.05±5.33(32) | 99.44±0.12(3) | 99.16±0.2(8) | 96.01±1.28(23) | 96.76±0.62(22) | 90.95±8.37(26) | 63.19±22.35(30) | 97.69±0.4(19) | 99.08±0.22(9.5) | 97.98±0.31(18) | 94.0±0.69(24) |
| campaign | 48.84±0.01(11.5) | 48.84±0.01(11.5) | 42.77±2.92(20) | 49.69±0.0(3) | 39.17±4.25(22) | 23.09±7.18(31) | 24.99±0.0(30) | 51.05±0.00(1) | 49.51±0.0(5) | 36.95±12.7(25) | 29.75±5.8(29) | 32.35±4.60(28) | 20.25±0.0(32) | 49.31±0.0(7) | 49.64±0.0(4) | 48.01±0.47(15) | 37.11±0.64(24) |
| cardio | 86.25(+0.01) | 86.17(+0.02) | 68.92±1.77(21) | 58.87±0.0(24) | 67.71±4.44(22) | 84.79±0.57(3) | 29.26±0.0(32) | 74.88±0.0(14) | 78.55±0.0(9) | 38.89±5.52(31) | 72.47±5.87(15) | 55.86±7.98(25) | 51.15±24.55(27) | 58.26±0.0(14) | 79.3±0.0(6) | 53.9±3.02(26) | 49.02±0.29(28) |
| cardiotocography | 69.69(+0.01) | 69.68(+0.02) | 59.27±4.03(12) | 50.7±0.0(24) | 67.52±0.82(4) | 67.52±0.82(4) | 33.51±0.0(32) | 56.07±0.0(19) | 68.98±0.0(3) | 45.78±5.1(28) | 60.56±7.01(9) | 59.77±0.0(19) | 43.91±12.55(30) | 58.26±0.0(14) | 59.77±0.0(19) | 47.05±1.34(27) | 45.75±0.91(29) |
| celeba | 20.95±1.1(1.5) | 20.95±1.1(1.5) | 12.85±4.44(14) | 16.77±0.78(8) | 7.47±5.6(26) | 4.01±1.21(29) | 7.99±1.06(24) | 16.48±0.82(9) | 16.9±0.79(7) | 7.09±4.32(27) | 9.46±6.75(21) | 9.04±2.73(22) | 7.65±0.16(25) | 12.63±0.51(15) | 14.31±0.6(10) | 13.27±0.47(13) | 10.05±0.49(19) |
| census | 16.05±0.88(21) | 12.76±0.01(19) | 14.68±1.56(20) | 14.01±0.32(25) | 17.49±2.11(14) | 8.69±0.99(32) | 14.66±1.25(21) | 33.5±0.8(25) | 15.35±1.07(17) | 13.42±3.92(27) | 13.62±0.55(11) | 31.78±3.63(29) | 14.25±1.1(22) | 49.03±0.0(14) | 47.47±0.0(19) | 58.52±1.04(6) | 47.51±1.92(18) |
| cover | 35.22±1.21(24) | 16.17±0.86(20) | 1.98±0.6(31) | 5.42±0.6(27) | 25.03±38.24(16) | 9.0±1.94(32) | 2.23±0.32(30) | 12.26±0.85(23) | 19.22±1.54(19) | 22.56±9.16(17) | 22.56±9.16(17) | 77.5±4.94(29) | 31.33±5.85(15) | 97.96±0.46(4) | 96.01±0.78(16) | 97.92±1.11(6) | 86.53±4.28(26) |
| donors | 36.01±0.72(23) | 32.72±0.0(29) | 49.31±14.82(13) | 36.33±1.85(22) | 23.9±11.25(30) | 6.14±0.82(12) | 37.26±4.14(21) | 53.19±0.02(9) | 41.27±0.97(19) | 25.39±21.33(29) | 54.49±2.63(27) | 40.28±2.2(16) | 30.2±17.77(27) | 50.55±0.0(10) | 62.56±0.0(9) | 64.41±1.09(3) | 58.79±0.41(23) |
| fault | 28.74±5.54(26) | 28.51±3.73(30) | 60.35±2.93(20) | 53.89±0.0(28) | 62.47±4.52(10) | 2.08±0.82(31) | 60.8±0.0(18) | 53.19±0.02(9) | 51.71±0.0(30) | 55.46±1.48(26) | 55.46±1.48(26) | 76.75±6.65(25) | 57.81±4.16(24) | 41.22±5.02(15) | 40.45±4.07(16) | 46.5±3.2.85(16) | 30.33±4.25(23) |
| fraud | 18.51±3.73(30) | 26.93±1.91(28) | 62.81±9.37(3) | 32.25±5.42(22) | 26.04±10.3(22) | 60.24±1.15(5) | 67.02±1.3(6) | 38.43±3.99(18) | 33.2±4.68(21) | 48.33±17.13(12) | 36.59±15.17(19) | 15.57±20.11(30) | 0.33±0.03(32) | 41.38±4.46(10) | 31.91±3.73(15) | 58.46±7.58(7) | 27.78±2.62(19) |
| glass | 64.48±5.1(21) | 64.85±5.06(20) | 30.93±6.56(18) | 27.61±6.5(20) | 73.25±1.71(17) | 65.77±5.43(19) | 89.16±5.91(13) | 56.08±3.5(25) | 25.02±6.94(23) | 52.35±21.04(7) | 15.55±2.58(32) | 18.62±12.43(29) | 23.14±13.98(24) | 89.95±1.8(11) | 70.62±4.48(18) | 76.58±1.38(4) | 74.38±1.79(16) |
| hepatitis | 90.42±1.67(12) | 91.69±1.53(10) | 89.63±3.08(12) | 63.49±5.99(22) | 9.41±0.16(29) | 68.38±8.0(18) | 50.54±3.34(23) | 46.31±2.11(24) | 25.18±0.82(28) | 36.09±31.85(27) | 7.46±9.79(31) | 57.53±32.61(19) | 34.91±13.15(32) | 89.48±1.9(14) | 94.39±1.31(8) | 97.56±2.2(4) | 75.04±1.79(25) |
| http | 8.71±0.00(29.5) | 8.71±0.00(29.5) | 9.5±0.67(10) | 9.01±0.00(9) | 52.89±1.61(8) | 8.8±0.1(27) | 10.45±0.46(3) | 9.3±0.0(12) | 8.48±0.0(32) | 9.68±1.47(9) | 8.73±0.39(28) | 9.22±0.0(31.5) | 9.89±0.52(7) | 9.08±0.00(15.5) | 8.96±0.0(22) | 10.17±0.03(5) | 9.21±0.0(2(14) |
| imdb | 46.97±0.02(2.5) | 46.97±0.02(2.5) | 47.57±1.08(17) | 30.79±0.0(30) | 95.38±1.33(17.5) | 93.17±2.64(22) | 30.56±0.37(31) | 61.74+0.0(2) | 61.87+0.0(1) | 75.64±1.71(30) | 39.32±2.28(26) | 31.78±3.63(29) | 43.08±5.72(24) | 49.03±0.00(14) | 47.47±0.0(19) | 58.52±1.04(6) | 47.51±1.92(18) |
| internetads | 91.42±1.37(24) | 90.94±1.25(25) | 97.64±0.86(9) | 64.63±3.76(32) | 37.14±8.33(20) | 31.21±0.83(1) | 94.44±1.96(21) | 78.49±3.06(28) | 98.09±0.81(3) | 85.15±5.63(27) | 85.15±5.63(27) | 77.5±4.94(29) | 71.72±21.88(31) | 97.96±0.46(4) | 96.01±0.78(16) | 97.92±1.11(6) | 86.53±4.28(26) |
| ionosphere | 40.29±7.81(15) | 40.33±0.0(18) | 34.19±0.94(27) | 60.12±0.0(3) | 39.18±0.0(2) | 31.21±0.83(1) | 37.37±0.0(19) | 33.82±0.00(28) | 31.09±0.0(32) | 49.43±2.4(11) | 35.7±5.57(24) | 40.28±2.2(16) | 37.55±1.93(18) | 50.55±0.0(10) | 46.49±0.0(13) | 55.86±0.6(5) | 34.98±0.81(25) |
| landsat | 8.01±0.0(31) | 8.01±0.0(31) | 9.19±0.81(13) | 8.73±0.0(19) | 8.45±0.09(23) | 8.13±0.05(28) | 24.7(+0.01) | 8.85±0.0(18) | 10.65±0.0(7) | 8.93±0.52(16.5) | 8.03±0.0(25.0) | 10.37±1.67(8) | 0.33±0.03(25) | 8.67±0.0(24) | 8.44±0.0(24) | 11.41±0.43(5) | 10.09±0.04(10) |
| letter | 75.27±0.0(26) | 75.2±0.0(26) | 96.24±4.22(19) | 96.55±2.11(18) | 90.53±7.4(24) | 98.76±6.88(11) | 98.1±2.64(16) | 93.9±2.85(23) | 94.38±1.43(21.5) | 96.82±3.69(17) | 24.13±13.29(32) | 73.47±15.88(28) | 30.87±35.63(31) | 96.16±6.43(20) | 98.57±0.65(13) | 100.0+0.0(2.5) | 68.82±5.96(30) |
| lymphography | 41.65±0.0(26) | 41.65±0.0(26) | 18.52±9.52(29) | 77.15±0.0(22) | 65.83±2.36(30) | 76.13±2.4(23) | 65.76±0.0(31) | 72.22±0.0(27) | 67.92±0.0(29) | 69.54±0.73(28) | 75.78±1.08(24) | 64.55±4.62(32) | 83.19±0.66(12) | 85.26±0.0(10) | 84.5±0.0(11) | 80.74±1.03(14) | 77.29±3.52(20) |
| magic.gamma | 64.99±0.0(12.5) | 64.99±0.0(12.5) | 55.22±3.33(21) | 22.21±0.0(29) | 47.97±4.73(23) | 27.82±3.84(22) | 11.17±0.0(31) | 54.63+0.0(2) | 55.2±0.0(4) | 27.54±1.1(4.23) | 43.21±2.04(4) | 22.0±17.15(25) | 59.72±1.98(15) | 72.08±0.0(3) | 39.83±0.0(12) | 28.51±2.57(21) | 28.91±1.92(20) |
| mammography | 100.0+0.08(5.5) | 100.0+0.08(5.5) | 32.68±33.48(31) | 1000.0+0.08(5.5) | 100.0+0.08(5.5) | 69.09±0.57(11) | 72.21±0.0(26) | 96.13±0.0(30) | 98.2±0.0(18) | 99.91±0.17(17) | 90.8±10.84(23) | 70.61±23.94(27) | 15.65±19.61(32) | 100.0+0.08(5.5) | 100.0+0.0(5.5) | 56.56±0.65(17) | 41.15±1.11(26) |
| mnist | 6.01±0.0(25) | 6.02±0.0(24) | 3.93±0.47(31.5) | 42.38±0.0(3) | 11.57±3.94(19) | 7.76±1.15(21) | 5.14±0.0(28) | 5.59±0.0(26.5) | 5.59±0.0(26.5) | 4.53±1.04(30) | 3.93±0.45(31.5) | 4.95±2.39(29) | 9.15±3.94(15) | 27.11±0.0(10) | 21.85±0.0(13) | 41.08±1.34(5) | 62.46±0.0(29) |
| musk | 59.39±0.0(20) | 59.35±0.0(21) | 58.26±5.01(24) | 22.48±0.0(32) | 46.08±16.77(29) | 63.5±1.17(13) | 59.1±0.0(22) | 41.51±0.0(31) | 58.54±0.0(23) | 52.05±3.89(27) | 48.57±3.88(28) | 60.26±12.84(18) | 73.46±2.81(1) | 61.76±0.0(17) | 67.15±0.0(8) | 64.91±1.13(10) | 54.53±1.01(26) |
| optdigits | 39.14±0.0(19) | 38.63±0.0(18) | 14.47±4.88(29) | 42.33±0.0(17) | 14.65±1.76(30) | 33.55±2.85(23) | 22.36±0.0(26) | 30.86±0.0(24) | 41.45±0.0(18) | 9.34±7.78(32) | 37.23±7.94(21) | 11.71±9.8(31) | 14.57±3.43(28) | 93.59+0.0(2) | 82.34±0.0(6) | 58.53±6.17(13) | 29.79±0.96(25) |
| pageblocks | 71.49±3.69(13) | 71.18±3.39(15.5) | 71.23±2.96(14) | 75.88±2.36(6) | 61.66±6.66(27) | 65.15±8.8(24) | 56.78±3.69(30) | 69.07±2.47(19) | 64.77±2.3(25) | 59.75±1.75(28) | 59.36±7.6(29) | 56.48±5.33(31) | 53.42±13.53(32) | 76.68±2.22(5) | 74.01±2.75(9) | 75.31±2.15(8) | 67.61±2.81(23) |
| pendigits | 81.04±0.11(19) | 77.79±0.0(25) | 77.86±2.47(24) | 86.49±0.0(5) | 81.83±0.75(16) | 76.98±6.46(23) | 63.25±0.0(30) | 73.33±4.0(29) | 69.57±0.0(31) | 81.1±1.97(18) | 79.77±0.93(22) | 75.98±3.34(28) | 77.46±6.34(26) | 84.95±0.0(12) | 83.82±0.0(14) | 85.84±2.16(8) | 71.75±2.86(30) |
| pima | 92.94±0.28(13) | 91.92±0.0(16) | 62.47±5.18(29) | 87.68±0.0(20) | 80.25±15.95(23) | 95.89±0.1(8) | 8.0±0.0(32) | 85.27±0.0(21) | 79.66±0.0(24) | 76.28±8.21(26) | 93.72±0.69(12) | 47.48±30.14(30) | 79.32±13.48(25) | 96.92±0.0(3.5) | 97.22+0.0(2) | 71.56±7.98(27) | 43.9±6.54(31) |
| satellite | 96.27±0.0(9.5) | 96.27±0.0(9.5) | 51.66±12.93(30) | 97.49±0.0(16) | 93.94±4.7(24) | 60.16±26.9(28) | 95.2±0.0(27) | 98.05±0.0(19) | 98.03±0.1(22) | 98.03±0.1(22) | 55.74±40.66(29) | 65.98±23.06(27) | 13.35±0.0(32) | 98.84±0.0(7) | 97.38±0.0(17) | 55.29±1.31(27) | 76.73±0.36(26) |
| satimage-2 | 40.14±0.33(27) | 36.39±0.0(28) | 74.74±17.4(8) | 53.37±0.59(20) | 31.88±2.2(30) | 42.18±1.84(26) | 63.01±1.72(15) | 29.69±0.18(32) | 30.49±0.2(31) | 42.99±3.28(25) | 53.04±7.11(21) | 50.37±21.79(23) | 65.62±1.8(13) | 96.37±0.25(3) | 92.71±0.16(5) | 56.71±7.09(19) | 62.54±0.43(16) |
| shuttle | 49.5±6.1(11) | 49.5±6.1(11) | 0.77±0.4(30) | 1.15±0.0(27) | 8.09±1.27(... | 32.4±8.8(19) | 50.5±0.0(32) | 0.99±0.05(29) | 68.0(+5.66(1) | 30.73±22.82(21) | 8.16±5.47(24) | 20.92±26.93(22) | 8.69±19.27(23) | 50.27±5.76(8) | 50.27±5.76(8) | 38.67±3.38(16) | 31.76±4.31(20) |
| skin | 49.38±0.44(13) | 81.84±0.01(5.5) | 85.36±2.6(34) | 78.42±0.0(23) | 83.63±1.4(7) | 82.09±0.21(13) | 50.5±0.0(32) | 73.58±0.0(26) | 71.26±0.0(30) | 75.26±2.44(24) | 80.16±6.45(20) | 74.22±2.55(25) | 79.07±3.1(21) | 83.15±0.0(9) | 82.54±0.0(11) | 83.01±0.5(10) | 73.05±0.13(27) |
| smtp | 2.77±0.0(28.5) | 2.77±0.0(28.5) | 3.26±0.5(7) | 3.21±0.0(30) | 2.76±0.21(30) | 2.81±0.3(24) | 2.97±0.96(16) | 2.79±0.0(26.5) | 2.87±0.0(19) | 3.38±0.38(5.5) | 2.97±0.94(21) | 3.95+0.07(4) | 3.57±0.07(34) | 2.82±0.0(22.5) | 2.75±0.0(23(3) | 3.66±0.23(3) | 73.05±0.13(27) |
| spambase | 59.92±7.99(15) | 58.81±7.82(17) | 52.41±12.56(21) | 52.28±4.56(22) | 33.47±10.36(31) | 49.57±17.72(25) | 49.13±8.19(26) | 56.43±3.1(20) | 49.01±3.86(27) | 42.62±9.94(29) | 57.17±11.21(19) | 46.54±22.11(28) | 28.48±21.94(32) | 72.02±4.82(6) | 65.12±7.1(10) | 66.57±4.58(8) | 50.8±7.62(23) |
| speech | 81.33±0.0(9) | 81.34±0.0(18) | 33.59±6.78(15) | 42.33±0.0(17) | 14.65±1.76(24) | 80.0±0.9(24) | 22.36±0.0(26) | 93.78±2.53(4) | 69.62±4.4(16) | 69.62±4.4(16) | 64.26±6.24(21) | 63.08±15.77(3) | 18.5±14.36(32) | 86.87±5.84(10) | 83.16±0.0(19) | 55.29±1.31(27) | 71.37±4.46(21) |
| stamps | 17.85±1.85(30) | 19.26±1.41(28) | 22.97±5.84(26) | 18.36±2.45(29) | 21.63±1.94(19) | 20.94±2.43(20) | 43.26(+0.01) | 7.06±0.0(31) | 19.93±0.87(26) | 23.42±3.47(17) | 16.72±3.13(31) | 35.09±19.9(29) | 28.38±3.31(11) | 28.3±3.31(11) | 20.42±2.34(25) | 24.91±3.58(15) | 28.14±4.88(12) |
| thyroid | 10.1±0.01(27) | 10.51±0.0(25) | 9.67±2.52(26) | 7.88±0.0(29) | 21.63±1.94(19) | 32.4±8.8(19) | 5.7±0.0(31) | 10.43±2.54(26) | 17.22±0.0(21) | 16.88±1.93(22) | 10.43±2.54(26) | 7.32±3.21(30) | 13.19±9.94(23) | 29.34±0.0(10) | 26.62±0.0(14) | 26.62±0.0(14) | 27.84±1.66(11) |
| vertebral | 8.4±0.0(25) | 8.41±0.0(24) | 25.08±5.76(7) | 9.0±0.0(22) | 13.33±5.54(14) | 8.86±0.64(23) | 5.7±0.0(31) | 9.88±0.0(20) | 7.35±0.0(29) | 11.52±3.39(15) | 7.8±1.04(28) | 6.07±0.89(30) | 20.07±6.96(10) | 25.7±0.0(6) | 24.92±0.0(8) | 28.9+1.5.68(2) | 8.24±0.1.8(26) |
| vowels | 93.22±2.73(11) | 94.33±0.0(8) | 70.99±10.09(25) | 70.1±0.0(11.9) | 73.87±15.14(24) | 79.81±3.26(15) | 79.24±1.88(13) | 93.16±3.26(6) | 93.11±2.88(13) | 56.35±9.82(27) | 56.55±15.9(27) | 23.95±26.86(31) | 22.39±18.04(31) | 23.95±26.86(30) | 31.45±15.97(18) | 93.05±3.70(9) | 49.34±1.43(28) |
| waveform | 83.20±6.35(17) | 82.05±7.51(17) | 45.45±2.24(16) | 72.24±4.26(30) | 46.93±5.57(14) | 96.69±0.18(9) | 70.57±6.66(6) | 38.17±2.2(29) | 35.78±1.83(32) | 84.32±8.89(12) | 7.96±0.92(25) | 84.74±16.75(28) | 12.23±18.04(31) | 83.16±0.0(19) | 43.41±2.44(19) | 64.63±3.4(11) | 49.34±5.19(28) |
| wbc | 7.25±0.0(28) | 6.41±0.0(32) | 17.07±2.4(11) | 7.87±0.0(26) | 8.85±1.26(21) | 10.86±1.37(18) | 11.08±0.0(17) | 6.87±0.0(31) | 7.68±0.0(27) | 7.08±0.1.73(0) | 46.91±3.85(20) | 8.43±1.12(23) | 9.61±2.4(20) | 11.52±0.0(16) | 10.36±0.0(19) | 23.88±1.71(6) | 21.9±0.7(7) |
| wbe | 69.47±6.95(23) | 69.22±6.18(24) | 78.85±9.79(5) | 77.71±9.93(20) | 47.63±29.53(30) | 70.1±6.29(22) | 53.48±10.47(27) | 52.34±5.29(28) | 12.63±0.0(28) | 78.56±9.59(19) | 12.04±1.54(31) | 50.92±36.74(29) | 18.5±14.36(32) | 95.71±0.92(10) | 19.4±0.0(12.5) | 98.38±2.25(6) | 71.37±4.46(21) |
| wilt | 40.26±2.88(25) | 40.03±2.79(26) | 45.45±2.24(16) | 42.61±2.22(20) | 46.93±5.57(14) | 14.93±0.18(15) | 10.75±2.28(30) | 18.17±2.2(29) | 9.52±0.00(31.5) | 74.88±5.58(5) | 12.71±1.55(25) | 11.43±0.99(28) | 35.99±4.15(31) | 56.05±1.44(11) | 15.13±0.0(17) | 80.78±4.52(3) | 52.0±2.61(13) |
| wine | 46.46±0.0(30) | 46.76±0.0(28.5) | 47.04±1.92(26) | 49.78±0.0(10) | 49.04±4.46(17) | 50.77±2.18(5) | 5.7±0.0(31) | 46.82±0.0(27) | 49.21±3.88(16) | 10.02±1.01(28) | 11.77±1.34(23) | 5.71±3.94(28) | 49.76±4.94(11) | 47.65±0.0(24) | 47.24±0.0(25) | 48.61±0.81(20) | 49.58±0.27(13) |
| wpbc | 12.76±0.01(19) | 12.77±0.0(18) | 10.69±0.61(25) | 13.04±0.0(15) | 11.38±0.24(24) | 13.13±0.39(14) | 9.05±0.14(32) | 13.25±0.01(12) | 11.89±0.0(21) | 31.44±3.0(4.25) | 31.44±3.0(4.25) | 29.66±7.59(28) | 10.1±0.59(27) | 15.43±0.0(5) | 15.25±0.0(6) | 9.79±0.08(30) | 11.86±0.62(22) |
| yeast | 40.34±0.0(17) | 40.33±0.0(18) | 34.1±1.28(20) | 21.6±0.0(29) | 43.44±1.57(12) | 41.23±0.35(16) | 11.62±0.84(30) | 9.52±0.0(31.5) | 9.52±0.0(31.5) | 45.1±2.04(22) | 12.04±1.54(31) | 23.41±9.02(28) | 26.92±8.88(26) | 45.51±0.0(9) | 44.49±0.0(10) | 47.83±0.12(4) | 31.85±0.2(24) |
| yelp | 56.16±0.0(15.5) | 56.16±0.0(15.5) | 46.78±1.12(21) | 34.86±0.0(27) | 9.77±0.03(6) | 56.59±0.4(13) | 16.15±1.44(30) | 12.1±0.0(30) | 9.5±0.0(31.5) | 45.1±2.04(22) | 46.91±3.85(20) | 29.66±12.73(29) | 35.99±4.15(31) | 58.64±0.0(9) | 58.26±0.0(10) | 61.22±0.18(4) | 37.25±0.17(26) |
| MNIST-C | 14.86±0.01(16.5) | 19.23±0.01(14.5) | 15.91±0.52(23) | 13.97±0.0(27) | 19.99±0.79(3) | 19.4±0.4(12.5) | 6.15±1.44(30) | 12.63±0.0(28) | 9.52±0.0(31.5) | 14.03±1.08(26) | 14.03±1.08(26) | 12.04±1.54(31) | 12.56±1.69(29) | 19.48±0.0(10) | 44.49±0.0(10) | 16.71±0.19(19) | 15.68±0.07(24) |
| FashionMNIST | 14.86±0.01(16.5) | 14.86±0.01(16.5) | 14.24±0.4(19) | 11.96±0.0(27) | 15.27±0.36(8) | 14.93±0.18(15) | 10.75±2.28(30) | 12.4±0.0(25) | 9.52±0.00(31.5) | 12.4±0.95(26) | 12.71±1.55(25) | 11.43±0.99(28) | 11.17±1.04(29) | 15.21±0.0(9) | 15.13±0.0(17) | 15.16±0.06(20) | 13.16±0.04(23) |
| CIFAR10 | 7.61±2.3(1(22) | 72.05±2.65(20) | 67.85±3.06(25) | 67.62±2.71(26) | 75.56±2.55(13) | 72.62±2.74(19) | 37.83±1.63(31.5) | 37.83±1.63(31.5) | 83.78±3.32(5) | 83.78±3.32(5) | 65.71±3.94(28) | 58.13±6.28(30) | 59.28±10.92(9) | 77.26±0.94(9) | 73.48±0.82(16) | 87.35±0.76(3) | 65.84±0.58(27) |
| SVHN | 11.49±0.65(22.5) | 11.34±0.4(25) | 10.57±1.52(30) | 11.14±0.27(28) | 13.78±1.1(9) | 11.49±0.51(22.5) | 10.44±1.03(31) | 11.09±0.32(29) | 12.9±1.9(16) | 12.9±1.9(16) | 11.23±1.35(27) | 10.17±1.17(32) | 12.01±2.09(19) | 13.66±0.85(10) | 12.64±0.66(17) | 12.98±0.23(15) | 13.29±0.47(14) |
| MVTec-AD | 11.62±0.02(3.5) | 11.62±0.02(3.5) | 9.72±0.37(32) | 11.16±0.0(25) | 12.68±0.53(17) | 12.42±0.31(18) | 12.42±0.31(18) | 11.07±0.0(26) | 10.93±0.0(27) | 10.22±1.85(28) | 12.08±0.99(20) | 10.17±1.33(29.5) | 9.74±0.34(31) | 15.96±0.0(7) | 15.45±0.0(8) | 12.79±0.06(16) | 14.03±0.43(12) |
| 20news | | | | | | | | | | | | | | | | | |
| agnews | | | | | | | | | | | | | | | | | |
| Rank(avg) | 17,482 | 17,904 | 19,132 | 19,316 | 18,561 | 19,061 | 22,325 | 22,044 | 21,500 | 19,114 | 22,939 | 24,009 | 22,158 | 10,035 | 11,719 | 11,184 | 19,965 |
| All | 0.995 | 0.996 | 1.000 | 1.000 | 0.999 | 0.999 | 1.000 | 1.000 | 1.000 | 1.000 | 1.000 | 1.000 | 1.000 | 0.106 | 0.380 | 0.403 | 1.000 |
| d ≤ 20 | 1.000 | 1.000 | 1.000 | 1.000 | 1.000 | 1.000 | 1.000 | 1.000 | 1.000 | 1.000 | 1.000 | 1.000 | 1.000 | 0.868 | 0.963 | 0.990 | 1.000 |
| d ≤ 50 | 1.000 | 1.000 | 1.000 | 1.000 | 1.000 | 1.000 | 1.000 | 1.000 | 1.000 | 1.000 | 1.000 | 1.000 | 1.000 | 0.886 | 0.932 | 0.952 | 1.000 |
| Rank(avg) | 18,149 | 18,57 | 19,956 | 20,149 | 19,281 | 19,816 | 23,202 | 22,939 | 22,342 | 19,939 | 23,816 | 24,939 | 22,982 | 10,579 | 12,228 | 11,746 | 20,825 |
| All | 0.998 | 0.998 | 1.000 | 1.000 | 1.000 | 1.000 | 1.000 | 1.000 | 1.000 | 1.000 | 1.000 | 1.000 | 1.000 | 0.623 | 0.759 | 0.830 | 1.000 |
| d ≤ 100 | 0.999 | 1.000 | 1.000 | 1.000 | 1.000 | 1.000 | 1.000 | 1.000 | 1.000 | 1.000 | 1.000 | 1.000 | 1.000 | 0.876 | 0.902 | 0.967 | 1.000 |
| d ≤ 500 | 0.999 | 0.999 | 0.999 | 1.000 | 1.000 | 1.000 | 1.000 | 1.000 | 1.000 | 1.000 | 1.000 | 1.000 | 1.000 | 0.849 | 0.892 | 0.932 | 1.000 |

Table 14.1: Average F1 score ± standard dev. over five seeds for the semi-supervised setting on ADBench. Rank of each model per dataset is provided (in parentheses) (the lower, the better). We use blue and green respectively to mark the top-1 and the top-2 method.

| Dataset | FoMo(D=100) | FoMo(D=20) | DTE-NP | kNN | ICL | DTE-C | LOF | CBLOF | FeatureBagging | SLAD | DDPM | OCSVM | DTE-IG | IForest | MCD |
|---|---|---|---|---|---|---|---|---|---|---|---|---|---|---|---|
| aloi | 7.82±0.41(4) | 6.49±0.77(12) | 5.82±0.07(17) | 5.9±0.0(15) | 4.91±0.56(22) | 4.2±0.2(26.5) | 8.16±0.0(3) | 6.74±0.08(10) | 8.93±0.57(2) | 5.32±0.11(19) | 6.76±0.19(9) | 7.29±0.0(8) | 5.12±0.68(21) | 4.2±0.26(26.5) | 3.41±0.14(31) |
| amazon | 11.72±3.24(4) | 10.72±1.2(16) | 10.8±0.0(15) | 11.4±0.0(7.5) | 9.4±0.0(23) | 11.8±1.53(2) | 10.0±0.0(23) | 11.52±0.5(6) | 10.0±0.0(14.23) | 10.16±0.22(21) | 11.08±0.11(11) | 12.0±0.0(1) | 10.48±2.37(20) | 11.28±0.64(10) | 11.32±0.23(9) |
| annthyroid | 49.25±5.0(24) | 51.5±0.0(17) | 61.84±1.59(4) | 61.99±0.0(3) | 49.44±3.88(23) | 77.72±0.5(1) | 49.63±0.02(22) | 56.7±3.3(12) | 50.67±5.76(18) | 65.99±0.62(2) | 57.23±2.96(11) | 53.56±0.0(15) | 48.8±7.04(25) | 55.02±4.22(14) | 50.37±0.0(19) |
| backdoor | 45.12±1.91(11) | 12.82±15.17(19) | 51.48±7.26(10) | 52.0±1.92(9) | 87.15±1.1(1) | 82.58±2.44(6) | 72.42±2.18(7) | 7.74±1.1(24) | 58.53±7.63(8) | 0.0±0.0(31) | 9.6±0.62(20) | 7.94±0.85(23) | 84.45±2.16(4) | 4.07±2.4(29) | 19.49±27.36(18) |
| breastw | 95.92±0.79(12) | 96.66±0.71(5.5) | 96.66±0.71(5.5) | 95.77±0.19(17) | 95.91±0.68(13) | 88.18±2.86(26) | 85.4±5.9(27) | 95.78±0.27(15.5) | 60.99±14.7(30) | 96.87±0.65(3) | 95.04±0.75(20) | 96.66±1.1(5.5) | 74.03±10.32(29) | 96.94±0.46(1) | 95.84±0.67(14) |
| campaign | 39.72±0.62(4) | 40.61±9.31(23) | 50.98±0.61(4) | 50.37±0.0(7) | 51.03±0.73(3) | 52.12±0.62(1) | 42.24±0.02(20) | 49.29±0.2(11) | 37.15±6.73(26) | 49.83±0.0(9) | 50.4±0.68(6) | 49.59±0.0(10) | 47.85±2.0(18) | 43.7±0.91(19) | 48.33±1.62(16) |
| cardio | 72.16±0.0(5) | 68.86±2.67(9) | 63.07±0.0(13) | 61.93±0.0(17) | 52.16±5.49(27) | 58.3±0.76(23) | 62.5±0.0(16) | 70.0±5.0(8) | 62.95±3.04(14.5) | 60.8±0.0(19) | 61.7±1.73(18) | 70.45±0.0(6.5) | 36.82±14.2(30) | 67.5±3.32(10) | 59.09±0.0(21) |
| cardiotocography | 56.2±0.0(6) | 48.41±1.52(11.5) | 46.78±1.44(19) | 46.35±0.0(20.5) | 38.93±2.89(23) | 38.37±2.23(25) | 48.2±0.01(15.5) | 51.42±3.49(10) | 48.41±1.51(13.5) | 33.82±0.47(30) | 38.84±2.75(24) | 57.9±4.0(5) | 31.67±2.63(31) | 56.14±2.75(7) | 36.48±1.78(28) |
| celeba | 8.41±0.88(28) | 7.38±1.96(29) | 15.81±0.69(18) | 17.99±0.83(16) | 6.96±1.23(25) | 17.35±3.48(14) | 1.91±0.0(32) | 25.32±7.1(4.5) | 2.92±0.89(31) | 13.69±14.6(20) | 20.02±2.89(4) | 27.37±0.74(1) | 19.11±5.61(10.5) | 17.33±2.29(15) | 25.91±14.9(6) |
| census | 69.92±2.74(6) | 69.59±1.3(7) | 66.84±7.4(8) | 65.1±2.15(9) | 39.96±2.71(13) | 71.04±3.46(5) | 11.0±0.02(23) | 21.42±2.06(20) | 3.42±1.3(30) | 9.14±8.13(27) | 76.86±1.3(4) | 20.65±0.38(21) | 77.79±3.91(3) | 11.61±12.4(25) | 29.45±33.3(17) |
| cover | 57.5±0.08(8) | 55.59±3.22(20) | 56.2±0.4(16) | 55.57±0.0(19) | 57.59±0.56(7) | 56.23±1.73(15) | 50.67±0.0(31) | 56.4±0.87(12) | 50.4±0.85(32) | 60.06±0.24(3) | 58.45±1.15(5) | 55.13±0.0(23) | 55.81±0.56(18) | 53.64±1.34(25.5) | 56.37±1.62(13) |
| donors | 97.85±0.79(1) | 77.32±3.52(10) | 92.9±2.83(6) | 94.91±0.67(3) | 97.22±1.04(2) | 82.17±2.54(8) | 74.47±2.04(11) | 48.48±1.13(14) | 79.41±3.96(9) | 55.86±8.74(13) | 25.01±7.13(27) | 73.16±2.29(1) | 93.11±3.02(5) | 43.46±3.54(18) | 33.32±14.21(25) |
| fault | 61.34±1.8(6) | 44.63±8.94(20) | 24.58±16.83(15) | 25.87±13.76(13) | 57.43±5.97(10) | 37.46±6.39(8) | 59.47±4.58(9) | 34.39±0.62(6) | 67.65±4.19(3) | 47.39±4.57(15) | 86.69±4.26(9) | 14.97±7.84(30) | 55.61±10.96(12) | 16.18±7.01(26) | 16.25±9.09(25) |
| glass | 77.6±6.07(3) | 59.96±10.09(6) | 97.43±3.5(4) | 10.0±0.0(1) | 87.83±9.1(1) | 92.8±3.3(8) | 20.46±8.79(20) | 23.75±14.31(17) | 22.45±7.76(18) | 35.03±7.6(19) | 32.81±13.36(10) | 73.16±4.93(17) | 78.82±42.94(16) | 25.8±22.46(20) | 93.05±1.5(9) |
| hepatitis | 89.24±5.79(13) | 96.0±1.94(7) | 79.03±1.95(14) | 81.29±5.04(11) | 99.64±0.79(2.5) | 68.23±13.1(12) | 41.95±0.62(29) | 66.93±8.85(16) | 41.27±12.6(30) | 88.49±9.3(15) | 99.68±0.3(3) | 99.78±0.3(2) | 78.82±42.94(16) | 25.8±22.46(20) | 93.05±1.5(9) |
| http | 10.12±4.42(6) | 7.6±1.82(18) | 5.2±0.0(29) | 5.4±0.0(26) | 55.87±0.91(4) | 7.24±1.62(12) | 6.4±0.0(18.5) | 6.96±0.09(14.5) | 6.56±0.3(17) | 10.24±0.22(4) | 5.56±0.09(25) | 5.8±0.0(21) | 10.36±5.1(3) | 6.2±0.49(20) | 7.44±0.22(11) |
| imdb | 40.82±7.48(25) | 33.8±3.72(27) | 53.21±3.77(11) | 51.9±0.0(13) | 55.87±0.91(4) | 6.4±0.0(18.5) | 54.62±0.0(8) | 45.76±6.24(21) | 54.35±3.71(9) | 57.83±0.3(2) | 45.87±0.35(20) | 46.2±0.0(18) | 54.95±5.79(7) | 26.41±4.44(32) | 33.42±0.0(28) |
| internetads | 40.0±1.0(14) | 49.93±1.75(9) | 51.22±2.25(7) | 51.46±0.0(6) | 53.82±0.35(2) | 38.29±2.94(23) | 53.64±0.0(3) | 38.33±0.19(22) | 53.97±0.17(1) | 46.93±0.04(12) | 40.23±1.02(19) | 38.56±0.0(21) | 30.26±4.37(32) | 43.27±1.34(14) | 47.7±9.54(11) |
| ionosphere | 92.3±1.3(6) | 89.99±1.04(12) | 90.47±2.17(11) | 90.9±2.0(8) | 94.18±1.62(1) | 89.58±0.9(14) | 87.53±2.54(19) | 91.96±2.1(4) | 87.65±2.26(18) | 92.65±1.2(6) | 88.64±1.58(15.5) | 92.62±1.51(5) | 89.68±4.23(13) | 88.64±1.2(15.5) | 2.4±0.89(17.5) |
| landsat | 52.66±0.04(4) | 49.93±4.32(.) | 51.22±2.55(7) | 51.46±0.0(6) | 53.82±0.35(2) | 38.29±2.94(23) | 53.64±0.0(3) | 38.33±0.19(22) | 53.97±0.17(1) | 46.93±0.04(12) | 40.23±1.02(19) | 38.56±0.0(21) | 30.26±4.37(32) | 43.27±1.34(14) | 3.8±1.11(13.5) |
| letter | 4.0±1.0(11.5) | 0.2±0.4(32) | 1.0±0.0(27.5) | 72.86±1.2(1) | 72.6±0.0(1) | 2.4±1.6(17.5) | 100.0±0.0(1) | 1.0±0.0(27.5) | 8.6±1.3(14.5) | 1.6±0.5(20) | 3.6±0.89(16) | 1.0±0.0(27.5) | 3.4±0.58(16) | 3.8±1.1(13.5) | 2.4±0.89(17.5) |
| lymphography | 96.39±5.14(7) | 100.0±0.0(2.5) | 95.78±5.81(8) | 94.5±6.47(10) | 100.0±0.0(2.5) | 82.01±3.83(26) | 74.87±7.44(27) | 89.28±1.85(19) | 65.34±17.16(29) | 99.47±1.18(5) | 95.19±6.67(9) | 100.0±0.0(2.5) | 97.89±4.71(6) | 85.05±4.72(23) | 83.73±4.92(24) |
| magic.gamma | 77.89±0.22(6) | 78.78±0.07(5) | 76.46±0.81(8) | 76.17±0.0(9) | 69.55±0.47(15) | 80.78±0.6(2) | 76.08±0.0(10) | 69.24±0.01(17) | 76.85±0.7(7) | 63.95±0.02(22) | 78.88±0.95(4) | 68.38±0.0(18) | 79.31±1.11(3) | 69.64±1.25(14) | 67.89±0.12(19) |
| mammography | 37.0±0.6(17) | 45.3±1.0(8.5) | 42.38±1.03(10) | 42.38±0.0(12) | 17.38±3.16(29) | 37.41±3.9(16) | 38.46±0.01(15) | 49.23±0.0(3) | 39.38±0.97(13) | 22.15±1.26(27) | 24.62±4.04(26) | 41.92±0.0(11) | 35.31±7.05(20) | 39.23±2.48(14) | 2.62±0.63(32) |
| mnist | 62.0±0.0(14) | 31.71±4.35(28) | 72.6±1.2(1) | 71.86±0.0(1) | 64.89±1.95(9) | 84.46±2.88(16) | 71.43±0.0(3) | 65.94±0.44(8) | 100.0±0.08(8.5) | 67.0±0.39(7) | 64.37±3.89(15) | 100.0±0.08(8.5) | 88.86±25.36(23) | 52.6±4.64(20) | 53.61±4.3(29) |
| musk | 93.2±3.72(18) | 92.1±6.17(20) | 92.1±6.17(20) | 21.33±0.0(12) | 83.3±8.97(25) | 100.0±0.0(8.5) | 100.0±0.08(8.5) | 1.6±0.37(20) | 100.0±0.08(8.5) | 39.87±0.7(58) | 60.37±3.89(15) | 100.0±0.08(8.5) | 88.86±25.36(23) | 52.6±4.64(20) | 53.61±4.3(29) |
| optdigits | 37.33±0.0(7) | 22.4±7.5(11) | 27.2±0.73(9) | 21.33±0.0(12) | 57.73±8.41(1) | 10.93±3.39(16) | 51.33±0.02(.) | 1.6±0.37(20) | 47.33±4.45(3) | 39.87±0.7(58) | 28.4±7.05(8) | 0.67±0.0(23.5) | 27.07±18.57(10) | 12.8±6.28(15) | 0.0±0.0(30) |
| pageblocks | 61.18±0.0(10) | 64.71±0.0(5) | 59.29±0.26(13.5) | 59.29±0.0(13.5) | 64.9±1.29(4) | 62.12±0.67(8) | 65.88±0.0(2) | 65.29±0.28(3) | 63.45±1.67(6) | 60.2±0.0(11) | 50.31±0.79(22) | 55.69±0.0(18) | 54.59±2.54(20) | 42.63±2.29(29) | 57.57±0.11(17) |
| pendigits | 69.23±0.0(8) | 80.13±0.0(5) | 83.46±6.02(3) | 90.38±0.0(1) | 61.15±5.82(10) | 56.03±8.28(14) | 76.28±0.0(7) | 49.23±0.7(16) | 83.33±3.51(4) | 44.36±1.05(17) | 64.62±2.26(9) | 53.21±0.0(15) | 59.74±9.95(11) | 57.95±4.75(12) | 14.36±0.35(29) |
| pima | 74.65±1.84(2) | 72.13±1.63(5) | 74.69±2.25(1) | 70.56±2.49(8) | 74.99±0.46(5) | 73.54±2.6(3) | 66.75±3.3(18) | 68.78±2.76(14) | 68.43±3.04(16) | 58.87±2.87(27) | 66.62±2.26(19) | 68.59±2.2(15) | 65.17±3.7(22) | 67.12±6.04(21) | 63.15±5.1(29) |
| satellite | 77.75±0.0(3) | 74.28±0.66(6) | 71.91±0.09(12) | 71.81±0.0(13) | 74.99±0.46(5) | 72.33±0.63(11) | 72.64±0.0(9) | 64.02±0.07(26) | 72.6±0.08(10) | 78.24±0.14(2) | 73.74±0.27(7) | 67.34±0.0(20) | 70.57±3.05(15) | 67.12±6.04(21) | 63.15±5.1(29) |
| satimage-2 | 88.73±0.0(1.5) | 88.73±1.26(1.5) | 90.14±0.0(7.5) | 90.14±0.0(7.5) | 88.45±1.54(14) | 66.48±2.71(28) | 81.69±0.0(19) | 92.96±0.0(2) | 84.51±0.0(17) | 88.73±0.0(1.5) | 78.59±5.1(22) | 91.55±0.0(4) | 78.31±5.23(23) | 89.58±1.61(9) | 95.77±0.0(1) |
| shuttle | 97.72±0.2(15) | 98.3±0.0(7) | 98.3±0.0(7) | 98.23±0.0(9) | 99.82±0.0(2) | 97.99±0.02(13) | 98.41±0.0(4) | 96.31±0.16(18) | 30.91±41.52(31) | 98.47±0.05(3) | 98.3±0.0(8) | 96.5±0.0(17) | 98.78±0.0(9.2) | 96.71±0.53(16) | 84.62±0.0(25) |
| smtp | 49.67±1.9(25) | 73.54±2.57(15) | 69.59±4.42(2.5) | 69.5±4.43(7.5) | 6.96±1.23(8.2.5) | 82.23±0.43(6) | 70.8±2.09(18) | 81.39±0.38(7) | 59.02±1.62(19) | 69.59±4.42(2.5) | 73.42±5.33(16) | 80.02±0.42(8) | 93.36±2.92(5) | 78.06±0.72(11) | 76.76±0.35(12) |
| spambase | 74.27±0.0(21) | 72.76±8.49(24) | 80.67±0.48(3) | 80.52±0.0(4) | 6.96±1.23(8.2.5) | 2.62±1.87(24) | 65.82±5.88(14) | 78.78±0.49(12) | 71.52±1.85(26) | 81.5±0.13(2) | 63.55±0.95(31) | 69.5±4.43(7.5) | 75.18±2.49(5) | 3.93±2.49(5.5) | 0.0±0.0(29) |
| speech | 2.95±1.23(20) | 3.61±2.1(17) | 3.61±2.1(17) | 3.28±0.0(13) | 2.62±1.87(24) | 3.93±1.47(5.5) | 3.28±0.0(13) | 1.64±0.09(30.5) | 2.95±0.73(20) | 6.23±1.2(6) | 2.95±1.37(20) | 3.28±0.0(13) | 1.97±1.8(28.5) | 3.62±2.8(8.12) | 2.62±1.47(24) |
| stamps | 18.72±5.7(2) | 63.44±0.02(1) | 74.84±0.96(7) | 75.27±0.65(5) | 77.17±7.83(4) | 75.48±0.9(3.5) | 63.52±13.2(13) | 64.39±1.86(11) | 64.7±12.55(10) | 50.99±12.77(24) | 66.62±2.26(9)? | 63.44±9.99(14) | 70.23±1.13(57) | 63.62±8.81(12) | 30.97±8.32(31) |
| thyroid | 59.1±0.02(3.5) | 63.44±0.0(21) | 21.56±14.78(15) | 75.27±0.65(5) | 56.13±8.72(26) | 42.13±1.49(6) | 33.68±6.2(19) | 74.19±1.08(10.5) | 40.22±1.57(31) | 71.18±1.8(15) | 75.48±2.07(3.5) | 75.27±0.65(5) | 47.74±10.66(30) | 80.43±1.08(1) | 73.12±0.0(14) |
| vertebral | 64.44±2.3(0.7) | 57.49±3.66(5) | 29.6±0.89(8) | 26.0±0.0(13.5) | 6.39±5.17(2) | 37.2±4.15(4) | 33.68±6.2(19) | 25.73±3.79(13) | 33.33±8.67(10) | 14.2±0.67(25) | 37.54±1.2(52) | 20.37±3.6(18) | 46.58±10.24(4) | 15.2±3.63(23) | 17.32±5.22(21) |
| vowels | 28.0±0.0(10.5) | 2.6±0.0(10.5) | 29.6±0.89(8) | 26.0±0.0(13.5) | 24.4±8.29(17) | 37.2±4.15(4) | 34.0±0.0(7) | 19.6±2.6(22) | 35.2±4.3(6) | 38.84±4.38(3) | 41.6±5.37(2) | 28.0±0.0(10.5) | 36.0±2.45(5) | 15.2±3.63(23) | 9.0±0.0(32) |
| waveform | 8.0±0.0(25) | 28.4±2.58(4) | 27.0±0.0(7) | 27.0±0.0(7) | 64.9±1.29(4) | 12.2±2.77(16) | 28.0±0.0(5) | 26.4±1.5(12) | 29.6±2.19(2) | 2.2±1.1(32) | 12.0±1.22(17.5) | 13.0±0.0(15) | 24.8±4.27(13) | 10.2±2.17(19) | 0.0±0.0(30) |
| wbc | 91.1±5.4(13) | 87.4±3.9(10) | 89.35±3.45(5) | 86.41±3.23(13) | 92.88±4.57(1) | 89.28±2.42(5) | 20.27±9.64(31) | 80.68±9.87(18) | 6.29±14.06(32) | 92.2±4.48(2) | 86.03±5.16(15) | 89.84±2.99(4) | 62.64±10.31(24) | 88.25±2.24(19) | 79.13±14.06(20) |
| wdbc | 92.67±5.32(1) | 87.81±6.07(4) | 92.13±1.8(58) | 78.72±3.2(17) | 90.52±9.27(2) | 98.29±2.42(5) | 85.63±6.59(6) | 69.65±7.36(13) | 87.11±5.67(5) | 85.23±6.4(7) | 79.28±7.36(13) | 80.34±2.47(11) | 89.47±7.99(3) | 70.91±1.05(21) | 58.73±7.26(25) |
| wilt | 77.04±0.0(1) | 28.79±0.0(4) | 2.41±1.49(19) | 2.33±0.0(20) | 35.18±2.59(3) | 57.76±6.44(10) | 16.73±0.0(9) | 1.09±0.49(29) | 19.14±13.28(7) | 7.0±0.0(14) | 20.23±0.0(16) | 1.17±0.0(28) | 6.4±1.54(2) | 2.02±0.33(21) | 7.78±0.0(13) |
| wpbc | 58.02±1.22(9) | 98.18±3.64(6) | 92.13±1.18(58) | 87.16±5.61(11) | 9.32±1.52(3) | 57.76±6.44(10) | 41.26±4.76(21) | 75.73±7.62(17) | 38.86±5.39(22) | 87.9±1±3.64(2) | 50.71±4.98(13) | 78.28±3.76(15) | 59.51±9.36(8) | 36.63±1.93(24) | 41.28±3.93(20) |
| yeast | 50.49±0.0(6) | 71.46±0.08(4) | 68.16±14.02(6) | 49.09±2.1(14) | 90.55±0.97(1) | 49.23±1.12(13) | 44.73±3.0(8) | 51.36±0.1(14) | 47.5±1.5(19) | 49.27±0.2(12) | 50.85±2.0(25) | 35.79±1.3(27) | 44.46±0.86(27) | 44.46±0.86(27) | 46.27±0.18(25) |
| yelp | 16.12±3.95(11) | 17.4±1.4(46) | 19.6±0.0(3) | 18.8±0.0(4) | 8.88±0.73(27) | 14.72±1.51(17) | 20.6±0.0(2) | 13.4±0.2(20.5) | 20.72±0.23(1) | 7.12±0.1(31) | 16.08±0.18(12) | 15.2±0.0(16) | 14.6±2.17(18) | 15.8±0.62(14) | 12.52±0.3(23) |
| MNIST-C | 39.11±2.4(19) | 35.71±6.0(20) | 47.51±0.0(7) | 46.3±0.0(8) | 52.07±1.23(3) | 48.68±1.5(5) | 52.96±0.02(2) | 42.94±0.16(13) | 53.21±0.66(1) | 47.64±0.22(8) | 42.32±0.6(15) | 42.25±0.0(15) | 46.21±5.1(9) | 34.7±2.59(21) | 25.69±4.92(7) |
| FashionMNIST | 53.44±2.28(1) | 42.76±0.07(24) | 35.71±0.0(20) | 59.05±0.0(9) | 59.22±0.98(8) | 59.22±1.09(5) | 57.14±0.32(12) | 57.14±0.32(12) | 63.78±0.86(4) | 59.67±2.26(5) | 56.67±0.72(13) | 54.66±0.0(14) | 54.66±0.0(14) | 45.33±2.4(23) | 37.55±5.77(25) |
| CIFAR10 | 20.03±1.05(18) | 17.45±1.08(24) | 23.19±0.0(7) | 22.85±0.0(11) | 20.59±1.37(16) | 23.79±1.09(5) | 27.07±0.0(2) | 23.48±0.53(6) | 27.22±0.71(1) | 24.67±0.48(3) | 22.98±0.48(8) | 22.81±0.12(8) | 19.07±2.49(19) | 18.19±1.18(21) | 17.91±2.06(22) |
| SVHN | 17.73±0.63(18) | 15.9±0.89(22) | 21.21±0.0(5) | 18.95±0.0(8) | 10.55±4.99(2) | 19.49±0.6(13) | 19.23±0.0(4) | 18.65±0.37(12.5) | 19.66±0.45(1) | 19.18±0.24(7) | 18.73±0.23(11) | 18.43±0.0(15) | 17.59±1.78(19) | 16.45±1.06(21) | 14.34±1.94(25) |
| MVTec-AD | 64.2±1.71(21) | 64.04±2.1(22) | 75.69±2.88(7) | 67.37±3.1(12) | 3.263±2.26(1) | 78.98±3.08(4) | 67.29±3.16(13) | 66.48±3.06(15) | 67.48±3.26(11) | 81.77±2.71(2) | 65.02±2.84(18) | 64.6±2.67(19.5) | 75.87±4.33(6) | 64.6±2.65(19.5) | 72.35±3.2(8) |
| 20news | 14.59±2.22(10) | 17.59±2.1(3) | 17.83±1.58(2) | 14.61±1.48(9) | 16.43±1.74(6) | 18.95±4.69(1) | 16.77±1.37(4) | 12.83±2.11(16) | 16.64±1.39(5) | 14.09±4.43(14) | 10.55±1.27(24) | 11.31±1.1(19) | 14.33±4.65(12) | 10.49±1.66(26) | 14.72±1.95(8) |
| agnews | 17.61±4.6(10) | 20.98±1.04) | 20.7±0.0(5) | 20.1±0.0(6) | 17.79±0.73(9) | 23.93±3.44(3) | 30.6±0.0(1) | 15.35±0.25(11) | 30.53±0.73(2) | 13.33±0.014(17) | 12.52±0.18(21) | 13.7±0.0(15) | 14.11±3.85(14) | 12.48±0.65(22) | 13.08±0.32(19) |
| **Rank(avg)** | | | | | | | | | | | | | | | |
| All | - | 11.798 | 11.798 | 10.202 | 11.158 | 11.667 | 12.289 | 13.658 | 13.456 | 11.719 | 12.64 | 13.658 | 13.482 | 17.781 | 18.912 |
| d ≤ 20 | - | - | - | 0.324 | 0.626 | 0.118 | 0.764 | 0.952 | 0.949 | 0.597 | 0.907 | 0.923 | 0.603 | 1.000 | 1.000 |
| d ≤ 50 | - | - | - | 0.905 | 0.958 | 0.910 | 1.000 | 0.999 | 1.000 | 0.979 | 0.990 | 0.996 | 0.868 | 1.000 | 1.000 |
| | | | | 0.955 | 0.993 | 0.666 | 1.000 | 0.999 | 1.000 | 0.963 | 0.993 | 0.996 | 0.956 | 1.000 | 1.000 |
| **Rank(avg)** | | | | | | | | | | | | | | | |
| All | - | 12.456 | 9.193 | 10.728 | 10.263 | 11.667 | 12.781 | 14.272 | 14.009 | 12.211 | 13.228 | 14.193 | 14.114 | 18.57 | 19.702 |
| d ≤ 100 | - | - | 0.143 | 0.555 | 0.550 | 0.790 | 0.736 | 0.973 | 0.966 | 0.809 | 0.962 | 0.963 | 0.979 | 1.000 | 1.000 |
| d ≤ 500 | - | - | 0.640 | 0.889 | 0.868 | 0.986 | 0.975 | 0.991 | 0.999 | 0.912 | 0.967 | 0.982 | 0.996 | 1.000 | 1.000 |
| | | | 0.448 | 0.786 | 0.806 | 0.960 | 0.944 | 0.991 | 0.997 | 0.928 | 0.969 | 0.979 | 0.992 | 1.000 | 1.000 |

Table 14.2: Average F1 score ± standard dev. over five seeds for the semi-supervised setting on ADBench. Rank of each model per dataset is provided (in parentheses) (the lower, the better). We use `blue` and `green` respectively to mark the `top-1` and the `top-2` method.

| Dataset | VAE | PCA | PlanarFlow | HBOS | GANomaly | GOAD | DIF | COPOD | ECOD | DeepSVDD | LODA | DAGMM | DROCC | DTE-NP | KNN | ICL | DTE-C |
|---|---|---|---|---|---|---|---|---|---|---|---|---|---|---|---|---|---|
| aloi | 7.63±0.05(5.5) | 7.63±0.05(5.5) | 3.83±0.72(29) | 7.43±0.0(7) | 9.35±2.27(1) | 5.73±1.45(18) | 3.91±0.28(26) | 4.58±0.0(24) | 4.44±0.0(25) | 5.17±0.92(20) | 6.6±1.58(11) | 5.98±1.67(14) | 0.0±0.0(32) | 5.86±0.0(16) | 6.01±0.0(13) | 4.59±0.07(23) | 3.56±0.03(30) |
| amazon | 11.0±0.0(12.5) | 11.0±0.0(12.5) | 9.48±1.62(26.5) | 10.6±0.0(18.5) | 8.96±0.74(30) | 11.56±0.26(5) | 9.32±0.66(29) | 11.4±0.07(5) | 10.0±0.0(23) | 11.76±2.19(3) | 10.64±1.18(17) | 9.48±1.23(26.5) | 0.0±0.0(32) | 10.6±0.0(18.5) | 10.92±0.0(14) | 9.94±0.32(25) | 7.01±0.72(31) |
| annthyroid | 50.19±0.0(20) | 50.0±0.0(21) | 60.0±1.76(7) | 35.96±0.0(29) | 34.27±2.48(30) | 55.77±4.62(13) | 58.99±0.0(9) | 31.65±0.0(31) | 38.39±0.0(28) | 23.33±5.12(32) | 46.78±5.19(6.5) | 45.66±6.44(27) | 57.42±2.53(10) | 60.56±0.0(6) | 60.56±0.06(6) | 52.56±3.29(16) | 61.63±0.1(2.5) |
| backdoor | 8.5±1.27(21) | 8.3±1.0(22) | 36.73±22.17(14) | 6.93±0.61(25) | 21.92±0.41(16) | 4.79±3.47(27) | 20.28±2.02(17) | 0.0±0.0(31) | 0.0±0.0(31) | 82.96±3.28(5) | 4.64±4.15(28) | 5.25±3.94(26) | 85.44±1.14(3) | 42.86±1.13(12) | 31.87±1.22(15) | 86.76±1.0(2) | 42.75±1.54(13) |
| breastw | 96.12±0.47(10) | 95.78±0.45(15.5) | 94.09±1.69(22) | 96.93±0.34(2) | 90.05±3.37(25) | 95.66±0.34(19) | 56.81±4.58(31) | 94.63±0.53(21) | 94.63±0.53(21) | 91.85±0.77(24) | 95.67±0.47(18) | 83.52±1.11(28) | 45.66±6.44(27) | 96.22±0.43(9) | 95.97±0.31(11) | 96.48±0.28(7) | 92.5±0.79(23) |
| campaign | 48.4±0.0(14) | 48.84±0.0(13) | 42.11±2.88(21) | 47.91±0.0(17) | 40.92±4.2(22) | 22.62±9.05(31) | 27.11±0.0(30) | 49.27±0.0(12) | 48.36±0.0(15) | 37.89±12.9(25) | 30.74±4.5(27) | 34.13±3.43(27) | 0.0±0.0(32) | 50.93±0.0(5) | 65.0±0.0(1) | 50.07±0.49(8) | 31.35±0.44(28) |
| cardio | 76.14±0.0(1.5) | 76.14±0.0(1.5) | 59.77±1.94(20) | 56.25±0.0(24) | 58.75±3.47(22) | 74.89±0.93(3) | 27.27±0.0(32) | 70.45±0.06(5.5) | 73.86±0.0(4) | 38.41±4.19(29) | 63.41±3.82(12) | 53.07±6.55(26) | 46.93±3.41(28) | 62.95±0.01(4.5) | 65.0±0.0(11) | 53.68±2.83(25) | 34.91±0.09(31) |
| cardiotocography | 61.59±0.0(2.5) | 61.59±0.0(2.5) | 17.91±7.49(12) | 41.42±0.0(22) | 46.35±9.46(20.5) | 59.96±1.08(4) | 31.33±0.0(32) | 48.28±0.0(15.5) | 62.88±0.0(1) | 37.08±5.46(26) | 55.11±7.7(8) | 14.19±5.61(19) | 33.95±12.37(29) | 47.0±0.0(18) | 48.76±0.0(12) | 36.88±1.27(27) | 48.23±0.39(17) |
| celeba | 27.04±0.41(3) | 27.17±0.49(2) | 17.91±7.49(12) | 22.68±0.93(9) | 11.07±7.99(23) | 3.98±2.98(30) | 10.79±4.24(24) | 22.78±0.91(8) | 22.81±0.78(7) | 8.43±5.5(27) | 13.26±8.39(21) | 14.19±5.61(19) | 8.62±0.84(26) | 17.87±0.57(13) | 19.11±0.61(10.5) | 16.84±0.88(17) | 10.12±1.04(25) |
| census | 20.82±0.33(9) | 13.86±2.4(22) | 13.86±2.4(22) | 10.77±0.62(24) | 10.77±0.62(24) | 4.96±2.3(29) | 14.44±1.82(20) | 0.0±0.0(31.5) | 0.0±0.0(31.5) | 3.43±3.44(29) | 14.15±3.41(21) | 12.16±11.56(24) | 15.55±1.44(18) | 21.62±0.14(6) | 21.55±0.24(7) | 24.76±0.5(2) | 10.31±0.52(26) |
| cover | 16.21±1.68(22) | 2.59±2.48(30) | 2.59±2.48(30) | 10.75±1.26(26) | 25.69±34.37(16) | 0.0±0.0(32) | 1.19±0.85(31) | 18.82±0.81(20) | 24.46±1.11(18) | 3.43±3.44(29) | 24.19±12.26(19) | 12.16±11.56(24) | 41.87±7.55(12) | 61.65±2.14(10) | 52.19±1.92(11) | 36.61±2.59(15) | 38.39±0.62(14) |
| donors | 37.75±0.93(22) | 47.84±4.58(15) | 2.59±2.48(30) | 24.36±3.68(28) | 18.98±16.47(31) | 55.96±0.68(17) | 37.46±5.83(23) | 41.37±0.99(20) | 44.6±1.04(17) | 4.44±30.4(19) | 21.04±27.83(30) | 21.94±14.54(29) | 29.37±27.54(26) | 94.05±0.64(4) | 86.6±0.98(7) | 78.2±2.6(19) | 47.58±4.61(16) |
| fault | 55.22±0.08(22) | 57.65±4.55(6) | 57.65±4.55(6) | 53.64±0.0(25.5) | 56.76±4.09(10) | 37.28±25.8(24) | 61.96±0.0(2) | 50.82±0.0(30) | 51.56±0.0(29) | 54.92±1.38(24) | 51.59±1.84(28) | 53.22±4.51(27) | 56.71±4.72(11) | 56.32±0.01(4) | 56.85±0.0(9) | 59.13±1.36(4) | 72.0±0.47(1) |
| fraud | 34.46±3.22(25) | 33.26±1.7(27) | 19.56±1.1(22) | 41.51±4.73(22) | 24.83±10.99(14) | 20.15±11.0(21) | 60.41±4.55(5) | 46.16±3.48(16) | 37.81±2.09(23) | 45.39±2.18(7) | 45.06±11.59(19) | 13.72±16.1(32) | 15.48±13.14(29) | 47.78±3.5(14) | 45.61±3.48(17) | 62.3±4.91(5) | 30.48±4.66(28) |
| glass | 18.03±5.11(24) | 15.77±8.65(27.5) | 19.56±1.1(22) | 27.68±10.39(12) | 65.51±6.66(1) | 57.86±7.77(23) | 81.44±4.43(12) | 19.06±8.56(23) | 15.77±8.65(27.5) | 35.39±2.18(7) | 14.6±4.71(31) | 20.88±22.32(30) | 15.48±13.14(29) | 29.42±2.61(11) | 21.09±5.16(19) | 70.87±3.12(4) | 24.15±2.73(16) |
| hepatitis | 60.54±9.21(1) | 60.56±9.59(2) | 60.91±7.76(7) | 58.08±1.26(4) | 18.99±4.13(24) | 57.86±7.77(23) | 14.18±12.07(26) | 21.6±1.15(28) | 37.6±3.88(31) | 93.82±1.2(9) | 46.76±9.9(28) | 47.96±7.54(27) | 29.29±17.28(32) | 84.58±2.24(10) | 66.13±5.71(18) | 66.52±2.11(5) | 70.86±3.4(15) |
| http | 91.94±1.25(11) | 92.71±1.45(10) | 14.43±12.31(25) | 3.64±3.54(27) | 18.99±4.13(24) | 56.39±17.76(18) | 14.18±12.07(26) | 2.16±1.15(28) | 2.05±1.32(29) | 25.0±22.46(21) | 1.05±0.96(30) | 48.95±34.37(19) | 0.0±0.0(31.5) | 88.73±2.54(14) | 95.32±0.75(8) | 97.2±1.22(45) | 24.79±1.28(22) |
| imdb | 5.6±0.0(23.5) | 5.6±0.0(23.5) | 9.24±1.45(8) | 6.4±0.0(18.5) | 6.96±0.55(14.5) | 5.64±0.6(22) | 11.44±0.0(1) | 6.6±0.0(16) | 5.0±0.0(30) | 9.96±2.71(7) | 7.04±1.08(13) | 8.92±1.37(9) | 4.4±0.03(13) | 5.28±0.0(27) | 5.24±0.0(28) | 10.1±0.0(2.3) | 4.47±0.43(31) |
| internetads | 45.65±0.0(22.5) | 45.65±0.0(22.5) | 55.82±1.76(5) | 27.17±0.0(31) | 52.72±0.54(12) | 46.14±0.52(19) | 46.14±1.82(20) | 50.0±0.0(14.5) | 50.0±0.0(14.5) | 54.29±5.92(10) | 41.36±2.56(24) | 31.85±5.57(29) | 38.42±6.06(26) | 49.57±0.0(16) | 46.68±0.0(17) | 55.68±0.94(6) | 56.5±0.87(3) |
| ionosphere | 79.82±2.6(24) | 78.99±2.57(26) | 90.83±1.6(10) | 69.49±1.68(29) | 86.17±2.5(2) | 83.96±1.59(23) | 85.86±1.47(21) | 69.53±2.51(28) | 64.5±2.11(31) | 93.07±2.13(1) | 77.34±4.54(27) | 69.33±4.23(30) | 60.19±21.57(32) | 91.12±1.24(9) | 87.86±1.86(17) | 69.34±1.08(16) | 79.23±4.14(25) |
| landsat | 38.77±4.59(20) | 33.58±0.0(28) | 35.77±1.67(25) | 52.14±0.0(5) | 35.12±1.26(30) | 32.96±1.19(30) | 34.43±0.0(27) | 33.83±0.0(29) | 36.76±0.0(31) | 42.18±2.53(15) | 36.89±4.89(24) | 40.95±4.11(17) | 40.8±3.16(18) | 48.79±0.0(10) | 46.1±0.0(13) | 50.85±0.68(8) | 41.47±2.39(16) |
| letter | 1.0±0.0(27.5) | 1.0±0.0(27.5) | 3.8±2.39(13.5) | 6.0±0.0(8) | 1.2±0.84(22) | 1.2±0.45(22) | 28.0±0.0(1) | 4.0±0.0(11.5) | 9.0±0.0(4) | 5.0±1.58(10) | 1.2±0.45(22) | 8.6±4.39(5.5) | 13.6±8.62(2) | 1.0±0.0(27.5) | 1.0±0.0(27.5) | 5.84±1.11(9) | 1.72±0.37(19) |
| lymphography | 92.96±5.55(14) | 90.86±4.08(17) | 91.07±9.32(16) | 88.49±3.88(20) | 83.31±1.43(25) | 93.13±5.46(13) | 94.47±7.77(11) | 88.47±4.27(21) | 86.67±6.42(22) | 89.82±9.49(18) | 24.05±20.79(32) | 67.58±11.63(28) | 26.15±35.82(31) | 94.11±5.93(12) | 91.77±3.69(15) | 100.0±0.0(2.5) | 58.34±6.75(30) |
| magic.gamma | 65.19±0.0(25) | 65.19±0.0(25) | 67.82±1.62(20) | 67.19±0.0(23) | 66.95±1.27(32) | 62.72±1.48(27) | 62.02±0.0(28) | 62.86±0.0(26) | 59.72±0.0(30) | 59.88±0.89(29) | 65.48±1.31(23) | 57.35±2.82(31) | 72.61±0.67(13) | 75.61±0.01(1) | 74.48±0.0(2) | 69.34±1.08(16) | 85.36±1.69(1) |
| mammography | 45.0±0.0(6) | 44.62±0.0(7) | 22.0±0.88(28) | 16.92±0.0(30) | 36.85±22.78(18) | 35.62±5.7(19) | 16.85±40.43(31) | 52.69±0.0(2) | 53.0±0.0(1) | 31.62±0.21(22) | 47.92±2.01(4) | 26.85±20.2(25) | 32.69±2.68(21) | 43.15±0.08(5) | 43.0±0.0(9) | 29.94±2.59(23) | 28.42±1.4(24) |
| mnist | 63.86±0.0(12.5) | 63.86±0.0(12.5) | 52.14±4.56(21) | 24.14±0.0(29) | 63.91±1.08(4) | 63.91±1.13(11) | 21.29±0.0(30) | 87.63±0.0(24) | 92.78±0.0(19) | 33.31±11.28(24) | 33.8±7.68(27) | 72.09±0.17(17) | 57.29±2.09(17) | 71.37±0.04(4) | 70.49±0.0(5) | 52.79±0.77(19) | 34.87±1.48(26) |
| musk | 100.0±0.0(8.5) | 100.0±0.0(8.5) | 35.05±28.17(31) | 100.0±0.0(8.5) | 100.0±0.0(8.5) | 100.0±0.0(8.5) | 70.1±0.0(27) | 87.63±0.0(24) | 0.0±0.0(31.5) | 99.18±1.13(17) | 90.72±5.41(21) | 70.72±21.53(26) | 12.16±17.44(32) | 100.0±0.0(8.5) | 100.0±0.0(8.5) | 89.36±1.65(22) | 60.0±0.0(28) |
| optdigits | 0.67±0.03(25) | 0.67±0.03(25) | 0.0±0.0(30) | 40.67±0.0(5) | 4.93±4.07(19) | 50.24±1.07(23) | 0.67±0.0(23.5) | 87.63±0.0(24) | 0.0±0.0(30) | 0.0±0.0(30) | 1.07±1.67(21) | 0.27±0.0(27) | 20.13±7.2(13) | 16.0±0.0(14) | 8.93±0.0(17) | 46.6±2.0(4) | 6.11±1.52(18) |
| pageblocks | 46.86±0.0(26.5) | 46.86±0.0(26.5) | 54.35±2.08(21) | 12.35±0.0(30) | 39.8±16.8(30) | 41.54±3.49(22) | 61.96±0.0(9) | 35.26±0.0(24) | 49.22±0.0(25) | 54.71±3.06(19) | 46.82±3.32(28) | 57.92±1.77(16) | 48.43±1.73(1) | 59.02±0.01(3.5) | 58.16±0.0(15) | 62.93±0.29(7) | 49.33±0.37(24) |
| pendigits | 44.23±0.0(18.5) | 44.23±0.0(18.5) | 67.92±2.87(17) | 41.03±0.0(23) | 15.51±20.48(28) | 59.19±1.77(25) | 26.92±0.0(30) | 63.16±2.25(23) | 58.54±1.89(28) | 55.95±2.36(29) | 61.14±5.31(24) | 54.05±5.38(31) | 50.01±13.57(32) | 72.5±1.97(4) | 76.41±0.0(6) | 56.56±5.52(13) | 32.36±1.43(25) |
| pima | 70.45±2.76(9.5) | 69.3±2.92(13) | 62.25±5.49(29) | 69.6±2.76(11) | 58.93±5.42(26) | 63.58±0.48(28) | 65.85±0.0(27) | 60.71±0.0(31) | 56.63±0.0(32) | 67.76±2.88(18) | 65.11±1.75(25) | 65.13±1.63(24) | 67.52±3.93(19) | 73.55±0.0(13) | 71.3±2.0(6) | 69.69±7.51(27) | 81.04±1.96(1) |
| satellite | 66.17±0.11(22) | 62.67±0.0(30) | 62.25±5.49(29) | 75.98±0.0(4) | 76.62±15.83(24) | 56.26±30.2(28) | 68.05±0.0(27) | 60.28±0.0(20) | 78.87±0.0(21) | 73.24±6.68(26) | 88.73±1.0(11.5) | 50.42±3.38(30) | 76.34±13.45(25) | 70.73±0.0(14) | 69.52±2.69(8) | 73.62±2.69(8) | 42.31±7.14(31) |
| satimage-2 | 88.17±0.77(15) | 87.32±0.0(16) | 21.53±8.46(16) | 83.1±0.0(18) | 91.15±8.04(24) | 52.05±1.57(24) | 97.86±0.0(14) | 91.8±0.0(23) | 98.11±0.09(11.5) | 67.9±24.09(27) | 98.1±0.0(10) | 67.9±24.09(27) | 0.0±0.0(32) | 91.27±0.0(5) | 98.11±0.0(10) | 99.8±0.24(5) | 77.31±0.58(26) |
| shuttle | 95.78±0.0(20.5) | 95.78±0.0(20.5) | 78.59±11.12(49) | 58.3±0.29(20) | 31.3±2.79(29) | 48.56±12.43(17) | 72.67±0.95(17) | 20.2±0.58(31) | 22.0±0.36(30) | 43.26±2.59(27) | 55.77±10.14(22) | 55.71±28.67(23) | 78.44±1.35(10) | 95.82±0.14(2) | 95.88±0.13(4) | 98.3±15.0(24) | 28.18±1.79(11) |
| skin | 44.73±0.84(28) | 37.91±0.84(28) | 46.15±11.43(30) | 74.93±0.0(20) | 31.3±2.79(29) | ... | 72.67±0.95(17) | 20.2±0.58(31) | 69.51±0.0(29) | 34.0±23.28(21) | 34.0±23.28(21) | 26.32±34.3(22) | 73.94±3.09(23) | 89.89±0.13(4) | 89.89±0.13(4) | 50.5±3.54(16) | 76.25±0.32(13) |
| smtp | 69.59±4.42(2.5) | 69.5±4.43(7.5) | 0.0±0.0(30) | 0.0±0.0(30) | 0.0±0.0(30) | 78.81±0.63(11) | 0.67±0.0(23.5) | 0.0±0.0(30) | 0.0±0.0(30) | 34.0±23.28(21) | 7.4±2.79(27) | 8.2±2.1(82.8) | 13.75±30.75(23) | 6.773±4.99(13) | 69.59±3.95(2.5) | 50.51±3.54(16) | 43.11±3.67(18) |
| spambase | 78.49±0.03(15) | 78.5±0.0(14) | 77.62±3.62(18) | 74.93±0.0(20) | 79.33±1.09(9) | 2.95±1.3(30) | 50.98±0.0(32) | 71.59±0.0(25) | 49.51±0.09(30) | 69.55±3.79(28) | 71.03±5.1(27) | 68.45±3.55(30) | 73.94±3.09(23) | 80.13±0.0(6) | 79.48±0.0(8) | 77.87±0.46(16) | 83.25±0.07(1) |
| speech | 3.28±0.0(13) | 3.28±0.0(13) | 1.64±1.16(30.5) | 4.92±0.0(2.5) | 1.97±1.37(28.5) | 52.72±15.8(22) | 4.59±1.8(4) | 3.28±0.0(13) | 3.28±0.0(13) | 1.31±1.37(32) | 2.3±1.87(26) | 3.28±1.16(13) | 2.95±2.14(20) | 2.62±0.0(24) | 2.62±0.0(24) | 3.48±0.92(8) | 2.03±0.25(27) |
| stamps | 61.44±6.81(17) | 57.86±0.97(20) | 51.82±17.69(23) | 57.55±6.45(21) | 52.9±20.51(27) | 74.19±1.32(10.5) | 43.72±9.88(28) | 67.23±4.51(8) | 49.48±5.09(26) | 37.07±0.49(74.29) | 60.16±12.64(18) | 47.02±25.21(27) | 69.03±3.68(18) | 74.62±0.0(8) | 66.49±7.15(9) | 57.68±1.74(25) | 50.23±10.89(25) |
| thyroid | 74.19±0.0(10.5) | 74.19±0.0(10.5) | 21.53±8.46(16) | 72.42±0.0(17) | 19.76±4.38(19) | 18.25±10.64(20) | 51.61±0.0(29) | 30.11±0.0(23) | 59.14±4.0(23.5) | 65.59±8.6(19) | 70.75±5.18(16) | 5.68±3.33(16) | 32.11±0.12(8) | 73.55±0.0(13) | 73.55±0.0(13) | 59.1±0.0(13) | 60.69±0.57(22) |
| vertebral | 14.07±1.08(26) | 13.93±1.31(28) | 21.53±8.46(16) | 9.53±5.46(30) | 22.81±5.4(19) | 23.6±2.19(18) | 36.37±4.2(18) | 0.28±0.63(32) | 12.61±2.46(29) | 16.71±5.92(3) | 8.42±4.46(31) | 21.19±13.5(17) | 16.99±22.1(41.2) | 26.68±3.24(12) | 13.99±2.95(27) | 44.91±2.82(5) | 28.18±1.79(11) |
| vowels | 12.0±0.0(26.5) | 12.0±0.0(26.5) | 15.18(24) | 8.0±0.0(29) | 22.81±5.4(19) | 9.8±2.39(20) | 50.0±0.0(1) | 6.0±0.0(30) | 22.0±0.0(20) | 20.8±4.15(21) | 10.4±3.29(28) | 5.6±6.69(31) | 13.6(12.2) | 28.8±0.0(9) | 26.8±0.0(12) | 25.36±2.94(15) | 25.2±1.36(16) |
| waveform | 8.0±0.0(25) | 9.0±0.0(22) | 28.6±4.39(3) | 8.0±0.0(25) | 12.0±7.21(17.5) | 86.45±4.97(12) | 5.0±0.0(30) | 9.0±0.0(22) | 7.0±0.0(29) | 14.6±3.58(14) | 7.4±2.79(27) | 4.6±1.67(31) | 26.0±6.11(10) | 27.0±0.0(7) | 27.0±0.0(7) | 89.1±0.3(8.6) | 7.04±0.23(28) |
| wbc | 88.42±3.21(8) | 87.28±5.1(11) | 55.68±15.83(25) | 80.41±7.38(19) | 12.0±7.21(17.5) | 75.82±5.69(19) | 71.77±11.92(1) | 82.4±5.65(16.5) | 82.4±5.65(16.5) | 83.34±7.3(9) | 68.8±16.05(22) | 46.16±26.61(27) | 26.61±27.57(30) | 88.75±2.48(7) | 86.1±2.43(14) | 88.1±6.3(8.6) | 36.84±4.92(28) |
| wdbc | 78.69±6.98(18) | 78.78±1.28(16) | 75.08±11.73(20) | 67.95±6.63(24) | 57.89±9.0(72) | 12.45±0.0(10) | 3.35±5.65(32) | 9.55±3.91(12) | 13.4±0.0(20.5) | 48.17±1.12(28) | 52.65±20.79(27) | 32.5±29.05(30) | 8.7±19.44(31) | 82.28±4.91(10) | 78.79±5.12(15) | 78.86±2.34(14) | 47.39±3.91(29) |
| wilt | 1.95±0.0(22) | 1.56±0.0(24.5) | 3.27±4.59(18) | 0.0±0.0(32) | 6.15±5.11(15) | 65.52±5.99(25) | 10.51±0.0(14.2) | 1.56±0.0(24.5) | 4.28±0.0(17) | 0.62±0.35(31) | 0.86±0.58(30) | 5.68±3.33(16) | 1.48±1.39(26) | 1.79±0.0(23) | 1.25±0.0(27) | 26.35±2.35(5) | 12.36±2.38(11) |
| wine | 67.96±3.31(22) | 66.01±5.57(24) | 68.31±14.39(21) | 77.67±8.64(16) | 42.72±36.22(30) | 34.22±3.42(28) | 48.58±10.34(28) | 56.13±3.76(26) | 39.27±8.98(31) | 69.81±9.29(20) | 55.33±13.28(27) | 48.54±37.49(29) | 12.78±16.95(32) | 88.37±1.61(10) | 80.99±2.84(13) | 97.76±2.0(3) | 66.58±4.38(23) |
| wpbc | 36.48±3.41(25) | 33.62±3.01(29) | 42.84±4.39(18) | 44.59±3.59(16) | 45.61±5.66(15) | 53.21±2.4(3.2) | 67.99±4.49(7) | 33.52±3.96(30) | 36.21±1.91(26) | 33.18±3.98(31) | 37.28±3.79(23) | 51.99±3.47(3) | 31.9±5.22(32) | 56.11±1.24(11) | 41.99±2.22(19) | 81.35±3.33(3) | 53.91±5.76(12) |
| yeast | 44.26±0.18(29) | 43.39±0.0(31) | 47.77±2.46(17) | 44.38±0.0(28) | 49.51±4.78(10) | 15.36±0.38(15) | 43.79±0.0(30) | 42.6±0.0(32) | 46.35±0.0(24) | 49.47±5.66(11) | 48.05±3.25(16) | 51.99±3.47(3) | 48.8±1.78(31) | 46.39±0.0(23) | 46.71±0.0(21) | 48.41±0.87(15) | 58.93±0.44(1) |
| yelp | 16.2±0.0(9) | 16.2±0.0(9) | 34.67±1.19(22) | 16.2±0.0(9) | ... | 42.11±0.45(16) | 12.0±1.73(26.5) | 16.0±0.0(13) | 13.4±0.0(20.5) | 10.4±1.77(25) | 13.8±2.27(19) | 8.2±2.1(82.8) | 35.09±3.37(6) | 17.72±0.0(5) | 17.36±0.0(37) | 7.7±0.2(30) | 9.11±1.45(26) |
| MNIST-C | 41.11±0.0(17.5) | 41.11±0.0(17.5) | 34.67±1.69(22) | 33.24±0.0(29) | ... | 56.15±3.3(6) | 12.0±1.53(90) | 43.11±0.45(16) | 35.09±3.37(6.5) | 35.45±5.04(24) | 34.48±5.32(23) | 25.17±6.18(28) | 30.54±9.37(25) | 45.8±0.0(10) | 44.72±0.0(11) | 49.22±0.12(4) | 29.35±0.11(26) |
| FashionMNIST | 55.65±0.0(16.5) | 55.62±0.0(14.5) | 34.67±0.43(74.5) | 46.67±1.69(22) | ... | 83.53±3.27(1) | 66.34±4.21(16) | 62.92±4.92(6.5) | 56.98±8.89(9.5) | 68.81±0.58(30) | 48.95±1.39(26) | 48.54±5.78(31) | 47.47±0.26(18) | 57.07±0.0(13) | 57.07±0.0(13) | 49.4±2.05(12.4) | 71.97±0.92(9) |
| CIFAR10 | 22.62±0.04(14.5) | 22.62±0.04(14.5) | 17.53±0.97(20) | 34.07±1.25(2) | ... | ... | 10.12±2.67(28) | 0.89±0.77(20) | 10.76±1.05(22) | 15.2±1.66(24) | 20.42±3.07(17) | 12.99±2.1(28) | 14.97±2.42(28) | 22.88±0.09(5) | 80.99±2.84(13) | 19.62±0.19(20) | 10.44±0.45(27) |
| SVHN | 18.27±0.0(16.5) | 18.27±0.0(16.5) | 9.78±3.33(32) | 14.9±0.0(27) | 19.19±0.89(6) | 18.49±0.42(14) | 11.37±0.78(30) | 9.7±0.0(32) | 10.19±0.0(31) | 15.2±1.66(24) | 15.3±2.34(23) | 9.61±2.52(30) | 13.18±1.68(26) | 18.89±0.0(10) | 57.97±0.0(11) | 80.65±0.56(3) | 11.78±0.08(29) |
| MVTec-AD | 63.11±2.64(25) | 63.41±2.78(24) | ... | 13.1±0.0(27) | 67.28±2.76(14) | 10.7±1.03(23) | 10.12±2.67(28) | 0.89±0.77(20) | 10.76±1.05(22) | 10.64±3.63(15) | 9.95±1.97(29) | 9.61±2.52(30) | 12.55±3.73(18) | 69.62±0.83(10) | 65.23±0.73(17) | 14.26±0.73(13) | 10.44±0.45(27) |
| 20news | 10.79±1.83(21) | 10.5±1.24(25) | ... | 9.38±0.78(31) | 14.54±2.93(11) | ... | 10.12±2.67(28) | 10.89±0.77(20) | 10.76±1.05(22) | 13.66±3.57(30) | 13.52±1.55(16) | 9.61±2.52(30) | 3.72±4.16(32) | 15.0±1.49(7) | 12.82±1.49(17) | 80.65±0.56(3) | 12.53±0.63(20) |
| agnews | 12.25±0.0(23.5) | 12.25±0.0(23.5) | 9.78±1.2(31) | 11.45±0.0(26.5) | 14.44±0.0(26.5) | 13.15±0.69(18) | 10.71±0.01(29) | 11.55±0.0(25) | 11.45±0.0(26.5) | 10.71±0.01(29) | 13.52±1.55(16) | 10.77±3.43(28) | 3.72±4.16(32) | 19.4±0.0(7) | 18.14±0.0(8) | 14.25±0.08(13) | ... |
| **Rank(avg)** | | | | | | | | | | | | | | | | | |
| All | 16.816 | 17.825 | 19.316 | 19.781 | 18.693 | 18.439 | 21.395 | 21.807 | 22.404 | 18.991 | 21.798 | 23.649 | 22.947 | 10.509 | 12.105 | 10.947 | 19.298 |
| d≤20 | 0.997 | 0.998 | 1.000 | 1.000 | 1.000 | 1.000 | 1.000 | 1.000 | 1.000 | 1.000 | 1.000 | 1.000 | 1.000 | 0.371 | 0.731 | 0.403 | |
| d≤50 | 1.000 | 1.000 | 1.000 | 1.000 | 1.000 | 1.000 | 1.000 | 1.000 | 1.000 | 1.000 | 1.000 | 1.000 | 1.000 | 0.924 | 0.988 | 0.995 | 1.000 |
| | 1.000 | 1.000 | 1.000 | 1.000 | 1.000 | 1.000 | 1.000 | 1.000 | 1.000 | 1.000 | 1.000 | 1.000 | 1.000 | 0.949 | 0.993 | 0.961 | 1.000 |
| **Rank(avg)** | | | | | | | | | | | | | | | | | |
| All | 17.439 | 18.456 | 20.14 | 20.579 | 19.465 | 19.132 | 22.167 | 22.675 | 23.237 | 19.763 | 22.684 | 24.544 | 23.798 | 11.053 | 12.649 | 11.474 | 20.088 |
| d≤100 | 0.999 | 0.999 | 1.000 | 1.000 | 1.000 | 1.000 | 1.000 | 1.000 | 1.000 | 1.000 | 1.000 | 1.000 | 1.000 | 0.670 | 0.897 | 0.757 | 1.000 |
| d≤500 | 0.999 | 1.000 | 1.000 | 1.000 | 1.000 | 1.000 | 1.000 | 1.000 | 1.000 | 1.000 | 1.000 | 1.000 | 1.000 | 0.886 | 0.953 | 0.939 | 1.000 |
| | 0.999 | 1.000 | 1.000 | 1.000 | 1.000 | 1.000 | 1.000 | 1.000 | 1.000 | 1.000 | 1.000 | 1.000 | 1.000 | 0.849 | 0.947 | 0.880 | 1.000 |

# H    BENCHMARK OD DATASETS

Table 15: Description of all datasets in ADBench Livernoche et al. (2024).

| Dataset Name | # Samples | # Features | # Anomaly | % Anomaly | Category |
|---|---|---|---|---|---|
| ALOI | 49534 | 27 | 1508 | 3.04 | Image |
| annthyroid | 7200 | 6 | 534 | 7.42 | Healthcare |
| backdoor | 95329 | 196 | 2329 | 2.44 | Network |
| breastw | 683 | 9 | 239 | 34.99 | Healthcare |
| campaign | 41188 | 62 | 4640 | 11.27 | Finance |
| cardio | 1831 | 21 | 176 | 9.61 | Healthcare |
| Cardiotocography | 2114 | 21 | 466 | 22.04 | Healthcare |
| celeba | 202599 | 39 | 4547 | 2.24 | Image |
| census | 299285 | 500 | 18568 | 6.20 | Sociology |
| cover | 286048 | 10 | 2747 | 0.96 | Botany |
| donors | 619326 | 10 | 36710 | 5.93 | Sociology |
| fault | 1941 | 27 | 673 | 34.67 | Physical |
| fraud | 284807 | 29 | 492 | 0.17 | Finance |
| glass | 214 | 7 | 9 | 4.21 | Forensic |
| Hepatitis | 80 | 19 | 13 | 16.25 | Healthcare |
| http | 567498 | 3 | 2211 | 0.39 | Web |
| InternetAds | 1966 | 1555 | 368 | 18.72 | Image |
| Ionosphere | 351 | 32 | 126 | 35.90 | Oryctognosy |
| landsat | 6435 | 36 | 1333 | 20.71 | Astronautics |
| letter | 1600 | 32 | 100 | 6.25 | Image |
| Lymphography | 148 | 18 | 6 | 4.05 | Healthcare |
| magic.gamma | 19020 | 10 | 6688 | 35.16 | Physical |
| mammography | 11183 | 6 | 260 | 2.32 | Healthcare |
| mnist | 7603 | 100 | 700 | 9.21 | Image |
| musk | 3062 | 166 | 97 | 3.17 | Chemistry |
| optdigits | 5216 | 64 | 150 | 2.88 | Image |
| PageBlocks | 5393 | 10 | 510 | 9.46 | Document |
| pendigits | 6870 | 16 | 156 | 2.27 | Image |
| Pima | 768 | 8 | 268 | 34.90 | Healthcare |
| satellite | 6435 | 36 | 2036 | 31.64 | Astronautics |
| satimage-2 | 5803 | 36 | 71 | 1.22 | Astronautics |
| shuttle | 49097 | 9 | 3511 | 7.15 | Astronautics |
| skin | 245057 | 3 | 50859 | 20.75 | Image |
| smtp | 95156 | 3 | 30 | 0.03 | Web |
| SpamBase | 4207 | 57 | 1679 | 39.91 | Document |
| speech | 3686 | 400 | 61 | 1.65 | Linguistics |
| Stamps | 340 | 9 | 31 | 9.12 | Document |
| thyroid | 3772 | 6 | 93 | 2.47 | Healthcare |
| vertebral | 240 | 6 | 30 | 12.50 | Biology |
| vowels | 1456 | 12 | 50 | 3.43 | Linguistics |
| Waveform | 3443 | 21 | 100 | 2.90 | Physics |
| WBC | 223 | 9 | 10 | 4.48 | Healthcare |
| WDBC | 367 | 30 | 10 | 2.72 | Healthcare |
| Wilt | 4819 | 5 | 257 | 5.33 | Botany |
| wine | 129 | 13 | 10 | 7.75 | Chemistry |
| WPBC | 198 | 33 | 47 | 23.74 | Healthcare |
| yeast | 1484 | 8 | 507 | 34.16 | Biology |
| CIFAR10 | 5263 | 512 | 263 | 5.00 | Image |
| FashionMNIST | 6315 | 512 | 315 | 5.00 | Image |
| MNIST-C | 10000 | 512 | 500 | 5.00 | Image |
| MVTec-AD | 5354 | 512 | 1258 | 23.50 | Image |
| SVHN | 5208 | 512 | 260 | 5.00 | Image |
| Agnews | 10000 | 768 | 500 | 5.00 | NLP |
| Amazon | 10000 | 768 | 500 | 5.00 | NLP |
| Imdb | 10000 | 768 | 500 | 5.00 | NLP |
| Yelp | 10000 | 768 | 500 | 5.00 | NLP |
| 20newsgroups | 11905 | 768 | 591 | 4.96 | NLP |

## I DIFFERENCES TO PRIOR WORK ON PFNS FOR TABULAR DATA

There exist applications of PFNs (originally developed by Müller et al. (2022)) that pre-date our proposed FoMo-0D, namely, TabPFN (Hollmann et al., 2023) for supervised classification, LC-PFN (Adriaensen et al., 2024) for learning curve extrapolation, PFN4BO (Müller et al., 2023) for Bayesian optimization, and ForecastPFN (Dooley et al., 2023) for time series forecasting.

Here we highlight the differences of our proposed FoMo-0D from these existing PFNs.

1. **First PFN4OD:** We employ prior-data fitted networks (PFNs) for outlier detection (OD) for the first time.

2. **First large-scale pretrained OD model:** FoMo-0D is the first model for zero-shot OD that is pretrained at large scale on a large collection of (synthetic) datasets, due to the minuscule nature of existing real-world OD benchmark datasets.

3. **New data prior:** Thanks to PFN's reliance on synthetically generated datasets, we establish a new data prior for OD, specifically for outlier synthesis.

4. **Data transformation for scale:** While drawing samples from a data prior may be relatively fast, pretraining a large foundation model requires many such draws for every step of each epoch. To speed up data synthesis on-the-fly, we are the first to leverage a linear transformation.

5. **Router-based attention for scale:** PFNs ingest the entire training dataset as context for in-context learning at inference time. To accommodate larger datasets at both training (for better generalization) and inference (for large-scale real-world datasets), we leveraged a "bottleneck" architecture for scalable self-attention, and in turn, larger context size.

## J RELATED WORK

**Outlier Detection (OD):** Thanks to diverse applications in numerous fields, such as security, finance, manufacturing, to name a few, OD on tabular (or point-cloud) datasets has a vast literature with a long list of techniques. For earlier, shallow approaches preceding the advances in deep learning, we refer to the books by Aggarwal (2013) and Aggarwal and Sathe (2017). The modern, deep learning based techniques are surveyed in Chalapathy and Chawla (2019); Pang et al. (2021); Ruff et al. (2021). Most recent deep OD techniques take advantage of newly emerging paradigms, including self-supervised learning (Hojjati et al., 2022; Yoo et al., 2023) as well as the most recently popularized diffusion-based models (Yoon et al., 2023; Livernoche et al., 2024; Du et al., 2024; He et al., 2024).

**Unsupervised Model Selection for OD:** It is typical of models to exhibit various hyperparameters (HPs) that play a role in the bias-variance trade-off and hence the generalization performance, and OD models are no exception. Many earlier work on OD have showcased the sensitivity of classical (i.e. shallow) OD methods to the choice of their HP(s) (Aggarwal and Sathe, 2015; Campos et al., 2016; Goldstein and Uchida, 2016). Similarly, sensitivity to HPs has also been shown for deep OD models more recently (Zhao et al., 2021; Ding et al., 2022), as well as for those relying on self-supervised learning/data augmentation (Yoo et al., 2023).

While critical, work on unsupervised outlier model selection (UOMS) is slim as compared to the vast literature on detection methods. A handful of existing, mostly heuristic strategies has been studied by Ma et al. (2023) reporting discouraging results; they have shown that existing heuristics are either not significantly different from random selection, or do not outperform iForest (Liu et al., 2008) with its default HPs (an extremely fast ensemble of randomized trees).

More recent, state-of-the-art (SOTA) UOMS approaches go beyond heuristic measures and instead design scalable hyperensembles (Ding et al., 2022; 2024), as well as take advantage of meta-learning on historical real-world OD datasets (Zhao et al., 2021; 2022; Zhao and Akoglu, 2024). These SOTA approaches demonstrate the value of learning from many other OD datasets, and transfer these learnings to a new dataset. While sharing the same spirit on learning from a large collection of (in our case, simulated) datasets, our FoMo-0D differs from these prior art in a key aspect; FoMo-0D is *not* a model selection technique, but rather, a foundation model that abolishes model training and selection altogether and unlocks zero-shot inference on a new dataset.

**Prior-data Fitted Networks:** Based on the seminal work by Müller et al. (2022), Prior-data-fitted Networks (PFNs) establish a new paradigm for machine learning, where a PFN is pretrained on

synthetic datasets generated from a data prior, and the pretrained PFN can then infer the posterior predictive distribution (PPD) for test points in a new dataset in a single forward pass, through in-context learning (Xie et al., 2021; Garg et al., 2022). It is shown that PFNs provably approximate Bayesian inference (Müller et al., 2022). Follow-up TabPFN (Hollmann et al., 2023) achieved SOTA classification performance on small tabular datasets of size up to 1024. Other subsequent works designed LC-PFN (Adriaensen et al., 2024) and ForecastPFN (Dooley et al., 2023), respectively zero-shot learning curve extrapolation and zero-shot time-series forecasting models, trained purely on synthetic data. PFN4BO (Müller et al., 2023) employed PFNs for Bayesian optimization, while Nagler (2023) studied the statistical foundations of PFNs. As training data is passed as context to PFN, others proposed scaling solutions to enable training on larger pretraining datasets for better generalization (Ma et al., 2024; Feuer et al., 2023; 2024).

Our proposed FoMo-0D differs from these in being the first PFN for OD, using a novel inlier/outlier data prior, employing linear transform for fast data synthesis, and incorporating the "router" attention mechanism for linear-time scalability w.r.t. context size. See Appendix I for additional details.

**Zero-Shot Outlier Detection:** Foundation models pretrained on massive text and image corpora, such as large language and/or vision models (L(V)LMs) like OpenAI's GPT-series (Achiam et al., 2023), DALL-E (Ramesh et al., 2021) and Flamingo (Alayrac et al., 2022), CLIP (Radford et al., 2021), and LLaVA (Liu et al., 2024) to name a few, have demonstrated remarkable success on several zero-shot tasks in CV and NLP. Follow-up work extended these models for zero-shot out-of-distribution detection (Esmaeilpour et al., 2022), zero-shot image OD (Liznerski et al., 2022; Jeong et al., 2023) as well as dialogue-based industrial image anomaly detection (Gu et al., 2024).

Foundation models, however, do not exist for tabular data which is widespread across OD applications in the real world, such as detecting credit card fraud, network intrusion, medical anomalies, and any sensor measurement abnormalities, to name a few. The recent ACR model by Li et al. (2023) on zero-shot OD does *not* rely on a pretrained foundation model, but rather is meta-trained on each specific domain using inlier-only datasets from the *same domain*. Concurrent to our work, Li et al. (2024) apply pretrained LLMs for prompt-based OD on tabular data which they serialize to text. Similar to our work, they also use *simulated* labeled OD datasets to fine-tune several existing LLMs to improve their performance. Their work, however, is quite preliminary in several fronts; a key limitation is that they assume independent features and query the LLM one-feature-at-a-time to reach an outlier score. Further, they fine-tune using only 5,000 data batches with up to 100 samples each, subsample 150 points and the first 10 columns of each dataset for evaluation (due to GPU memory constraint), and their testbed includes only two baseline methods. In contrast, FoMo-0D employs and pretrains PFNs at a much larger scale with rigorous evaluation on a much larger testbed.

# K  DISCUSSION

**Summary:**  We introduced FoMo-0D, **the first foundation model for outlier detection** (OD) on tabular data. FoMo-0D is a prior-data fitted network (PFN), pretrained on a large number of *synthetic* datasets generated from a new data prior for OD, which can infer the posterior predictive distribution for test points in a new dataset in a **zero-shot** fashion where the training data is input as context, capitalizing on *in-context learning*.

Zero-shot OD implies no more OD model (parameter) training and **no more model selection**, given a new OD task. That is a revolution for OD (!), for which algorithm and hyperparameter selection are notoriously-hard *without any labeled data*, and also computationally taxing especially for today's modern deep OD models with numerous parameters *and* a long list of hyperparameters. What is more, FoMo-0D provides **extremely fast inference** thanks to a mere *single forward pass*, making it amenable for OD on data streams.

Building on the PFN paradigm (Müller et al., 2022), FoMo-0D breaks new ground not only con-ceptually by abolishing the burden of model training and selection, but also empirically: Against **26** different (both classical and modern) baselines on **57** public benchmark datasets from diverse domains, FoMo-0D performs on par with the top $2nd$ baseline, while significantly outperforming the majority of the baselines. Without the need to train any, let alone multiple models for HP tuning, FoMo-0D takes a mere **7.7 ms** per test sample for inference only.

**Limitations and Future Directions:** FoMo-0D employs a simple straightforward data prior based on GMMs. While it is remarkable to see how far one can go with synthetic data from such a simple prior, future work can design more comprehensive data priors, inclusive of discrete features as well as other possible outlier types. We have also pretrained FoMo-0D solely on synthetic datasets, while future work can augment both synthetic and real-world datasets for pretraining.

Besides the lack of massive real-world datasets for tabular OD, a motivation for a data prior to pretrain purely on synthetic datasets comes from neural scaling laws (Kaplan et al., 2020; Zhai et al., 2022). Interestingly, the scaling laws for large Transformer models have shown that their generalization error tends to drop as a power law with the amount of training data (also, with number of parameters and amount of compute), but the power law exponent is very small—suggesting that acquiring more colossal real-world datasets would be a slow, if not expensive approach to advancing ML/AI. Others have proposed ways to subset-select smaller, non-redundant "foundation datasets" (Sorscher et al., 2022; Paul et al., 2021), and emphasized the importance of task/dataset diversity in pretraining (Raventós et al., 2024). Arguably, synthetic data from a complex and diverse data prior is a potential gateway to obtaining non-redundant and diverse datasets for pretraining large foundation models like FoMo-0D. On the other hand, designing such a data prior requires a level of domain/prior knowledge.

Another improvement could be scaling up to even larger context (i.e. dataset) size and dimensionality. While FoMo-0D generalizes beyond pretrained context sizes and dimensionality, it is limited to and performs particularly well on downstream datasets of similar nature as our experiments showed. A promising direction for size generalization is using PFNs as extremely fast ensemble components at inference; since "*PFNs are quick enough to be used as ensemble members. The size constraints could therefore be overcome by boosting and bagging techniques*" (Nagler, 2023).

Further, our work focused on semi-supervised OD with clean/inlier-only training data. Future work can study the unsupervised OD setting and pretraining with mixed/"contaminated" data in this transductive setting, where the unlabeled test data is the same as training data. In addition, we performed offline evaluation of FoMo-0D on static datasets, while its fast inference lends itself to streaming OD, which future work can explore. Technically, both extensions (unsupervised OD and streaming OD) are straightforward from the implementation perspective.

Our current work is limited to OD for tabular (or point-cloud) data. Our ideas can be extended to other data modalities, such as image, graph, and text outliers, to comprise other domains with critical OD applications such as video surveillance, fraud detection and LLM hallucination detection. To that end, the design of novel inlier/outlier priors would be an open direction. A promising approach here could be the use of pretrained generative models to draw synthesized image/text/etc. datasets for pretraining the PFN, in place of manually-designed data priors.

Finally, our quest here has been mainly experimental. Theoretically understanding why these models work as well as they do and investigating their failure cases are important yet open questions.

As the first foundation model for OD, FoMo-0D inspires many promising directions for future research that could lead to fruition for additional practical applications.

## L    BROADER IMPACT STATEMENT

FoMo-0D is zero-shot, abolishing not only parameter training but also model selection given a new dataset. This is a radical paradigm shift for OD literature, which historically focused on designing new models and recently also effective ways for unsupervised model selection. Obviating the need for either, we expect FoMo-0D to route attention of the community from new OD model design and selection to designing better data priors and gathering datasets for PFN pretraining, along with better and more scalable architectures for PFN.

From the applied perspective, a zero-shot OD model like FoMo-0D is a game-changer for practitioners! Given the plethora of OD algorithms to choose from, which often come with a list of hyperparameters to set, and not having the tools for effective and efficient model selection, the practitioners are burdened with a "choice paralysis". With FoMo-0D, practitioners can not only bypass such dilemmas on one dataset, but thanks to the "train once, use many times" nature of pretrained models, they can do so for any dataset such as those arriving over time. In fact, provided its lightening-fast inference

via a single forward pass, FoMo-0D is amenable to deploy in real time on streaming datasets, such that each (test) sample over a stream can be inferred with the preceding samples passed as context.

## M  REPRODUCIBILITY STATEMENT

We expect that the disruptive nature of FoMo-0D will trigger future innovations in the OD literature, as well as a widespread adoption by practitioners thanks to its key desirable properties. To foster future research and accessibility in practice, we make all resources (our codebase used for prior data synthesis, data transformation, and pretraining as well as our pretrained model checkpoints) publicly available at `https://anonymous.4open.science/r/PFN40D`. Further, full implementation details are provided in Appendix C.

