# OpenReview forum: "Zero-shot Outlier Detection via Synthetically Pretrained Transformers: Model Selection Bygone!"
_ICLR.cc/2025/Conference — Submitted to ICLR 2025_

### Official Review · Reviewer_Kh3r · 2024-10-29

**Soundness:** 3
**Presentation:** 3
**Contribution:** 2
**Rating:** 6
**Confidence:** 3

**Summary:**

The paper proposes to pretrain a foundation model on synthetic data for outlier detection, which utilizes in-context learning for the downstream detection tasks without hyperparameter tuning or post-training. The authors leverage the "router mechanism" to facilitate architecture scaling up and random linear transformation to speed up the synthesis process.

**Strengths:**

1. This paper presents a paradigm shift for outlier detection, training a foundation model for various downstream tasks without model selection and hyperparameter tuning.
2. The paper is well-written, which makes it easy to follow.
3. The experiments are sufficient for elucidating the intrinsic mechanisms of the proposed FoMo-0D.

**Weaknesses:**

1. While FoMo-0D doesn't require training on specific real datasets, it has inferior inference time compared to several baselines in Table 3, due to the forward pass of the whole training dataset during inference. If the training dataset scales up, there will be a high computation load during inference, undermining FoMo-0D's efficiency in time-critical inference scenarios.

**Questions:**

1. As the foundation model already achieves comparative performance compared with baselines, can the performance be continuously improved if we conduct post-training with specific real data, a common practice in NLP or CV?

---

> ### Author Response · Authors · 2024-11-21
>
> ### - **time & efficiency if training dataset scales up**
>
> We’d like to draw attention that in Table 3, time for baselines include **both** training and inference time, as they first need to train on each dataset separately _preceding inference_. FoMo-0D is zero-shot, and therefore bypasses training altogether, on any dataset, and the inference for test samples can be parallelized on GPU. As such, we get at least 7.4x speed up (against DTE-NP) if training and inference time are both considered (see Table 3). If the training dataset scales up, please note that the training time of the baselines will also go up (at least linear in the dataset size). FoMo-0D _inference_ time will grow only linearly (thanks to our router attention mechanism) as a function of training data/context size. In other words, scalability is a concern for the (training of) baselines, too, and linear-time scaling is the best one can hope for.
>
> If the training dataset grows considerably, to an extent that hinders inference time, we can resort to strategies such as dataset distillation, which takes additional training time to learn a small number of “distilled” tokens that are representative of the whole dataset. Dataset distillation as presented in [1] is scalable, also requiring linear time-complexity in the training dataset size. Having learned the small/constant number of distilled tokens as context, inference can then be sped up considerably.
>
> We’d like to also remark that time-critical applications often demand OD over data streams - in which it is often the case that only the most recent data is relevant (due to possible non-stationarity) and hence arguably a context size of 5K would effectively accommodate the window of relevant data in the recent history in such scenarios. While our work focused on designing the first foundation model for static OD, FoMo-0D is readily applicable to operate on streaming data thanks to its zero-shot nature that does not need re-training.
>
> ### - **post-training for performance improvement**
>
> Yes, even though we focused on zero-shot detection and in-context generalization, we can indeed think of a model that does more work at inference time.
>
> A promising direction is to “attune” the model better to a specific input dataset via fine-tuning, but note that the only labels we have at inference are the inliers in $D_{train}$. Having completed pre-training, and given a new dataset D = {(inlier-only) $D_{train}$, (unlabeled) $D_{test}$}, FoMo-0D parameters can be fine-tuned to accurately predict label 0 for all the inliers in $D_{train}$, which in effect could serve as better attuning the model to the given inlier distribution prior to inference.  To take fine-tuning one step further, self-training could be employed, where we can first treat predictions from FoMo-0D on new samples as pseudo-labels, and later use them toward fine-tuning the model parameters.
>
> Of course, these approaches have different trade-offs not only w.r.t. inference time but also access to model parameters. Our work has focused on zero-shot generalization of synthetically-pretrained Transformer models for OD. It would be valuable future work to design FoMo-0D variants with varying trade-offs that could serve OD applications with different needs.
>
> [1] In-Context Data Distillation with TabPFN. Junwei Ma, Valentin Thomas, Guangwei Yu, Anthony L. Caterini. CoRR abs/2402.06971 (2024)

---

> > ### Author Response · Authors · 2024-11-21
> >
> > We hope that we addressed your questions and comments, and that you feel more confident to support our work in achieving a better score and/or confidence. We also look forward to your further feedback, if any. Thank you!

---

> > > ### Comment · Reviewer_Kh3r · 2024-11-26
> > >
> > > Thank you for your active response. I will keep my positive rating.

---

> > > > ### Author Response · Authors · 2024-12-02
> > > >
> > > > Thank you for your support and engaging during the rebuttal!

---

### Official Review · Reviewer_WpVL · 2024-10-29

**Soundness:** 2
**Presentation:** 3
**Contribution:** 3
**Rating:** 5
**Confidence:** 3

**Summary:**

This paper leverages Prior-data Fitted Networks, which can directly predict the (outlier/inlier) label of any test data at inference time, by merely a single forward pass—making obsolete the need for choosing an algorithm/architecture and tuning its associated hyperparameters, besides requiring no training of model parameters when given a new OD dataset.

**Strengths:**

1. The paper is written well and is easy to understand.

2. The studied problem is very important.

3. The results seem to outperform state-of-the-art.

**Weaknesses:**

1. The baselines compared in this paper are not competitive enough. There are much more advanced OD algorithms that are unsupervised or semi-supervised and do not require retraining on the new test set. Please refer to [1] for more advanced algorithms. There are plenty of strong post-hoc algorithms that can be efficient and effective at the same time.

2. The inlier piror and outlier synthesis idea is explored in [2]. Could the authors clarify the differences? For me, I did not see any significant technical differences from the prior selection and the synthesis approach, even though there are naunces in model architecture and the focused problem.

3. Is it possible for the authors to show the actual detection metrics rather than the rank so that it makes the readers easy to see the algorithm performance.



[1] Yang et al., Generalized Out-of-Distribution Detection: A Survey

[2] Du et al., VOS: Learning What You Don't Know by Virtual Outlier Synthesis

**Questions:**

see above

---

> ### Author Response · Authors · 2024-11-21
>
> ### - **choice of baselines**
> Our OD baselines constitute quite an extensive testbed; with 26 different well-established methods – a much larger number than typically seen in current literature. These baselines are diverse; including both well-known shallow methods as well as deep models, and among the latter, include those based on recent techniques such as diffusion-based models from as latest as 2024. In fact, we improve over the SOTA diffusion-based approach DTE (Livernoche et al., 2024) which compared to the same set of 25 baselines.
>
> Algorithms that “do not require retraining on the new test set” would be zero-shot OD algorithms for which the literature is slim. OOD methods do not directly apply to OD.
>
> Please note a key difference between OD vs OOD: OD is typically given a **single**, fully unlabeled dataset with the goal of detecting outliers that do not “fit in” with the inliers, while OOD is provided with **multiple**, labeled (or known) K classes and the goal is to detect novel classes K+1, K+2, etc. In fact, Figure 1 in reviewer-provided citation [1] illustrates the differences: OOD requires ID (in-distribution) classification, while OD does not. Also, OD concerns both covariate shift and semantic shift whereas OOD mainly focuses only on semantic shift (i.e. new classes).  As such, OOD baselines do not directly apply here. And of OD baselines mentioned in Section 4 of [1], many well-established ones are already included among our baselines, such as LOF, OCSVM, GANomaly, DeepSVDD, etc.
>
> ### - **data synthesis**
>
> Our main contribution is not data synthesis, but a foundation model for OD.  In fact we use draws from GMMs for inlier synthesis and borrow ideas from existing literature (see ADBench; cited as (Han et al., 2022)) to synthesize outliers to obtain pretraining data. As such, synthetic data is only a component of our work, on which we train a novel Transformer/foundation model with in-context outlier detection ability.  As far as we can tell, VOS [2] is not a pretraining-focused foundation model with zero-shot outlier detection capability.
>
> ### - **actual detection metrics**
>
> All of those have already been given in the Appendix - please see Table 12 (pages 34-35), Table 13 (pages 36-37), and Table 14 (pages 38-39) respectively presenting AUROC, AUPR and F1 absolute performance values on all 57 datasets for all 30 methods.
>
> [1] Yang et al., Generalized Out-of-Distribution Detection: A Survey
>
> [2] Du et al., VOS: Learning What You Don't Know by Virtual Outlier Synthesis

---

> > ### Author Response · Authors · 2024-11-21
> >
> > We hope that our clarifications and pointers have addressed your concerns, and that you could support our paper in achieving a better score. We also look forward to your further feedback, if any. Thank you!

---

> > > ### Comment · Reviewer_WpVL · 2024-11-22
> > > **discussion**
> > >
> > > Thank the authors for the rebuttal! Here are additional concerns that I have:
> > >
> > > 1. I wouldn't agree with the authors' claim that "As such, OOD baselines do not directly apply here". In fact, there are a couple of OOD detection methods that does not require discriminative training and thus can be applied to OD. For example, those methods that are based on unsupervised contrastive learning, such as [1,2]. I would appreciate if the authors can discuss and compare with these methods.
> > >
> > > [1] Tack et al., CSI: Novelty Detection via Contrastive Learning on Distributionally Shifted Instances
> > >
> > > [2] Sehwag et al., SSD: A Unified Framework for Self-Supervised Outlier Detection
> > >
> > > 2. Can the authos clarify the difference between anomaly detection and OD. I did not see a clear difference here. If there is no difference, shouldn't the authors use the terminology of anomaly detection?
> > >
> > > 3. In addition to this, it would be useful for the authors to clarify clearly the difference between OD, OOD, novelty detection, and anomaly detection in the paper to avoid confusion and give this submission a proper positioning.

---

> > > > ### Author Response · Authors · 2024-11-23
> > > > **Reply to Q1 (Part 1)**
> > > >
> > > > We thank the reviewer for creating the opportunity for us to highlight, once more, the merits of FoMo. Let us start by enumerating these merits, against which we will discuss the baselines in detail.
> > > >
> > > > FoMo-0D is a fundamentally different approach to OD from all baselines, in that, it is **zero-shot**. Unlike the baselines, it is a pre-trained Foundation Model that tackles a new OD task **without any training**. Let us recall and remark what zero-shot offers:
> > > >
> > > > 1) **Performance:** Supervised pretraining (on synthetic labeled data) potentially leads to high performance downstream.
> > > >
> > > > 2) **HP tuning:** Zero-shot or zero-training implies NO hyperparameter (HP) tuning, which is extremely _nontrivial_ for an OD task with no labeled outliers.
> > > > This is what we think is the KEY advantage of FoMo-0D. All baselines struggle with how to choose HP values, to which they are sensitive (see below). Poor HP choices then feed back to poor performance. Recall our title: Zero-shot Outlier Detection …: Model Selection Bygone! FoMo-0D abolishes HP selection altogether, which makes its performance robust.
> > > >
> > > > 3) **Speedy inference:** With zero training and zero hyperparameter tuning, inference becomes very efficient via a mere forward pass over the pretrained FoMo-0D.
> > > >
> > > > Given 1), 2) and 3) above, please notice that performance is not the **only** important factor for OD. In fact, we argue that the KEY factor is bypassing the nontrivial task of choosing HPs.
> > > > Speedy inference comes as “cherry on top” thanks to zero training (and zero HP tuning) at inference time (given a new OD task).
> > > >
> > > > Now, let us deep dive into 2) and 3) in the context of the two baselines you mentioned: SSD [2] and CSI [1], to discuss the advantages of FoMo-0D against them.
> > > >
> > > > **Hyperparameters:**
> > > >
> > > >
> > > > Both SSD and CSI employ contrastive learning based on the SimCLR loss [Chen et al. 2020], which creates positive (image) pairs by augmentation, and treats all other samples in a batch as negatives. As the SimCLR paper states, ''_Contrastive learning benefits from larger batch sizes_'' (i.e. from many negative pairs). Thus, the first critical hyperparameter (HP) is `batch_size`.
> > > >
> > > > **SSD** has two main steps: **1)** embed images by a ResNet trained on contrastive SimCLR loss, and **2)** `k`-means clustering and nearest cluster Mahalanobis distance as outlier score.
> > > >
> > > > An important HP here is `k`, the number of clusters, which is not obvious how to choose. Another HP is `layer`, i.e. which ResNet layer they use as the feature extractor.
> > > > We now refer to Figure 5 in the SSD paper, which shows how AUC changes by `k` and `layer`. The authors sneak-peek into these AUC results, based on which they pick the HP values, and we quote ''_While for the first three blocks, we find an increase in AUROC with number of clusters, the trend is reversed for the last block (Figure 5). Since last block features achieve highest detection performance, we model in-distribution features using a single cluster._'' That is, they choose `k`=1 and `layer`=4 after the sneak-peek into these results.
> > > >
> > > > This is exactly why we ran the top 4 baselines in our paper not only with author-recommended HP values, but also with varying HPs and compared to their _average/expected_ performance. All four $^{avg}$ variants of the models were worse, underscoring HP sensitivity.
> > > >
> > > > FoMo-0D bypasses this HP-selection struggle altogether, and this aspect is important for OD.
> > > >
> > > > On the other hand, **CSI** employs ''distribution shifting transformations'' to construct negative pairs for SimCLR, which are HPs. In their paper (Sec 2.2. pg. 3) they have a subsection titled ''_OOD-ness: How to choose the shifting transformation?_'' in which they refer to Table 5, comparing different shifting transformations. This table shows that `rotation` is better than 5 other HP choices. Then they write in experiments, and we quote ''_For shifting transformations S, we use the random **rotation** …  for natural images, e.g., CIFAR-10 [33]. However, we remark that the best shifting transformation can be different for other datasets, e.g., Gaussian **noise** performs better than rotation for texture datasets..._''
> > > >
> > > > The point we attempt to make here is that all baselines come with HPs. Their performance is sensitive to HP choices, which varies by the dataset. However, HP tuning is either non-obvious (without labeled outliers), or expensive (to search over many candidate HP values, which often necessitates retraining the model multiple times with varying configurations). Then, it becomes nontrivial not only to use them in practice, but also to compare them fairly in the literature.
> > > >
> > > > FoMo-0D shifts the narrative fundamentally, by abolishing HP tuning altogether. That is, **zero-shot** OD fully relieves unsupervised OD from HP tuning.

---

> > > > > ### Author Response · Authors · 2024-11-23
> > > > > **Reply to Q1 (Part 2)**
> > > > >
> > > > > **Inference time:**
> > > > >
> > > > > Another advantage of zero-shot OD that FoMo-0D offers is fast inference. While baselines need to train a model prior to inference, FoMo-0D need not.
> > > > >
> > > > >
> > > > > Now consider SSD and CSI, both of which train a model on a given dataset based on SimCLR. In the SimCLR paper, [footnote 2] states: ''_With 128 TPU v3 cores, it takes **$\approx$1.5 hours** to train our ResNet-50 with a batch size of 4096 for 100 epochs._'' (SSD also requires additional time for clustering, and CSI has the auxiliary shift classification task.)
> > > > > In contrast, FoMo-0D takes 7.7 milliseconds for _inference only_ per test sample on average.
> > > > >
> > > > >
> > > > > Before we conclude, we want to also clarify that FoMo-0D is designed for **tabular** data. To test on image/NLP datasets, we used pretrained encoders to flatten them into vectors. In general, FoMo-0D can be used on images or other modalities as long as a (pretrained) encoder is available.  We plan to update our title as follows to make it clear:
> > > > > _Zero-shot *Tabular* Outlier Detection via Synthetically Pretrained Transformers: Model Selection Bygone!_
> > > > >
> > > > > On the other hand, both SSD and CSI are designed for image data in particular. CSI employs shifting transformations that are image specific, and thus does not apply to tabular datasets in our ADBench directly. We could compare to SSD post embedding, which employs k-means clustering and nearest cluster distance scoring. As such, SSD would be a naive baseline and we have similar baselines in our testbed, such as kNN. The issue with the baselines is the choice of the critical HP `k`, as we discussed earlier. (In fact, even the choice of the k-means clustering algorithm can be seen as a HP, since there exist many other, more sophisticated clustering algorithms beyond k-means).

---

> > > > > > ### Author Response · Authors · 2024-11-23
> > > > > > **Reply to Q2**
> > > > > >
> > > > > > There is confusion in the literature regarding all these terms. In the survey paper you had shared earlier, they consider the transductive setting where $D_{train} = D_{test}$ as OD, while the inductive setting –  where there is a train/test split; s.t. $D_{train}$ is inliers only, and $D_{test}$ is disjoint and unlabeled – as AD.
> > > > > >
> > > > > >
> > > > > > Other literature refer to the former as unsupervised OD and the latter as semi-supervised OD. Even the term semi-supervised is confusing here; for traditional ML semi-supervised problems come with labeled examples from all classes (inlier and outlier), whereas semi-supervised OD concerns with labeled inliers only.
> > > > > >
> > > > > > Yet other literature makes a semantic distinction between OD and AD; they state OD concerns with detecting statistical outliers, which may or may not be anomalies. For example, spiked sales during Thanksgiving is a statistical outlier but is not fraud (semantic anomaly). They state that while all anomalies are outliers, not all outliers are anomalies.
> > > > > >
> > > > > > As you can see, the jargon can be confusing and should be understood carefully from the specific paper in context.
> > > > > >
> > > > > > For our work, we make our setting very clear: We consider the settings in which there is a train/test split; s.t. $D_{train}$ is inliers only, and $D_{test}$ is disjoint and unlabeled – where the goal is to label/rank the samples in $D_{test}$.
> > > > > > One could call it semi-supervised OD/AD or inductive OD/AD, although we are not tied to any particular naming.

---

> > > > > > > ### Author Response · Authors · 2024-11-23
> > > > > > > **Reply to Q3**
> > > > > > >
> > > > > > > We are fully on the same page with the reviewer. However, the jargon confusion is implicit in the literature itself and not driven by our paper. Typically, OOD concerns with K known/labeled classes and the goal is to detect new classes K+1, K+2, …
> > > > > > > SSD paper that you have referenced, which addresses [semi-supervised/inductive] OD (as per their title ''A Unified Framework for Self-Supervised Outlier Detection''), states the problem as and we quote
> > > > > > > ''_Can we design an effective out-of-distribution (OOD) data detector with access to only unlabeled data from training distribution?_''
> > > > > > > OOD with just one class (=inliers), i.e. K=1 _is_  [semi-supervised/inductive] OD.
> > > > > > > We hope that these explanations 1) make the perhaps subtle differences clearer, and more importantly 2) make it clear what setting/problem we are considering in FoMo-0D.
> > > > > > >
> > > > > > > We want to leave with a final remark that FoMo-0D can be pretrained for transductive/unsupervised OD as well, which we already discussed in Section K of our paper in the Appendix. We pull that paragraph up here as-is, for quick reference:
> > > > > > >
> > > > > > > ''_Further, our work focused on semi-supervised OD with clean/inlier-only training data. Future work can study the unsupervised OD setting and pretraining with mixed/“contaminated” data in this transductive setting, where the unlabeled test data is the same as training data._''
> > > > > > >
> > > > > > > Technically, one would pretrain FoMo-OD by passing the unlabeled/contaminated $D_{test}$ as the context. Training on contaminated data may render downstream performance suboptimal, however, which presents additional research challenges for future work.

---

> > > > ### Author Response · Authors · 2024-12-02
> > > >
> > > > Thank you for engaging during the rebuttal. We hope that our new replies have addressed your further questions. Please let us know if you have any other comments or questions.

---

### Official Review · Reviewer_VxxK · 2024-11-02

**Soundness:** 3
**Presentation:** 2
**Contribution:** 3
**Rating:** 6
**Confidence:** 3

**Summary:**

This paper present FoMo-0D, the first foundation model for zero-shot OD on tabular datasets. FoMo-0D is pretrained on many synthetically generated datasets drawn from a novel data prior that the author introduce to capture various inlier and outlier distributions. Feeding the FoMo-0D with a new training set and a text point, it perform zero-shot inference on a new dataset via a single forward pass, fully abolishing the need for both model training on a new dataset and hyperparameter selection.

**Strengths:**

1. The paper presentation is clear. The proposed method is well-motivated.
2. The experimental results are highly sufficient and also demonstrate the effectiveness of the proposed foundation model.

**Weaknesses:**

1. While it is noted that "finding a prior that supports a sufficiently large subset of possible [data generating] functions isn’t trivial," the paper demonstrates that the initial attempts were adequate to achieve remarkable performance even with a simple data prior. Could you provide some in-depth insights into this result?

2. The paper conducted experiments on several benchmark datasets for out-of-distribution (OOD) detection, including CIFAR-10, Fashion-MNIST, MNIST-C, MVTec-AD, and SVHN. I am curious about the performance of FoMo-0D on more complex datasets, such as the ImageNet-level benchmarks recently proposed in OpenOOD v1.5 [1].

[1] OpenOOD v1.5: Enhanced Benchmark for Out-of-Distribution Detection, NeurIPS 2023 Workshop on Distribution Shifts

**Questions:**

See Weaknesses.

---

> ### Author Response · Authors · 2024-11-21
> **Part 1**
>
> ### - **in-depth insights on GMM prior**
>
> This outcome, that a basic data prior derived from GMMs can drive a pre-trained Transformer generalize to real-world datasets, is indeed a curious one. The emergent in-context learning (ICL) ability of pretrained Transformers on many non-trivial downstream tasks is intriguing, and has similarly surprised the NLP and ML communities at large (triggering a series of theoretical work aiming to understand how/why ICL works).
>
> Nevertheless, there could be a simple explanation to GMMs’ prowess in our setting: which is that GMMs are good models for real-world datasets in our testbed. To test this hypothesis, we performed goodness-of-fit tests: We fit GMMs to each real-world dataset D_real with up to 5 components (as with our pre-training datasets), then sampled $D_{syn}$ from the best-parameter-fit GMM and performed a two-sample test on $D_{real}$ and $D_{syn}$ (here we used e-test [1]), with the null hypothesis stating that they come from the same distribution. A smaller p-value of such a test provides evidence toward rejecting the null (i.e. GMM is not a good fit to $D_{real}$).
>
> We provide the results below, depicting the p-value (of the goodness-of-GMM-fit test) vs. FoMo-0D’s performance rank among 30 baselines (lower is better): We observe (Table 1) that performance is good on datasets with relatively large p-value where we cannot reject the null (i.e. GMM is a relatively good fit). This is where arguably FoMo-0D recalls its “past experience” and generalizes to datasets similar to those seen during pretraining. We also see (Table 2) datasets with relatively poor performance where we can reject the null (i.e. GMM is not a good fit). These can be attributed to falling short in generalization to out-of-distribution datasets.
>
> Curiously, on the other hand, we also still observe (Table 3) many datasets where p-value is small (GMM not a good fit) yet the performance is competitive —  those are the datasets for which FoMo-0D seems to have achieved out-of-distribution generalization; a property of ICL. It remains an open (theoretical) question to understand what (algorithm, if any) FoMo-0D might have learned that generalizes to those out-of-distribution datasets. It is also an open (empirical) quest to explore whether a more complex data prior, beyond GMMs, could further push the performance up and by how much.  All in all, our work presents the first foundation model for zero-shot OD in the literature and opens new theoretical and empirical directions for the community to research further.
>
> [1] https://www.rdocumentation.org/packages/energy/versions/1.7-11/topics/eqdist.etest

---

> > ### Author Response · Authors · 2024-11-21
> > **Part 2**
> >
> > ### - **in-depth insights on GMM prior** (tables)
> >
> > **Table 1**:  p>=0.05 and average performance rank<15
> > | dataset      |   p_value |   avg-rank |
> > |:-------------|----------:|-----------:|
> > | fault        |    0.1287 |          4 |
> > | glass        |    0.2376 |          3 |
> > | hepatitis    |    1      |          4 |
> > | http         |    0.1287 |         14 |
> > | ionosphere   |    0.5446 |         11 |
> > | lymphography |    0.901  |          9 |
> > | pima         |    0.6139 |          2 |
> > | speech       |    1      |         11 |
> > | stamps       |    0.5545 |          3 |
> > | vertebral    |    0.8812 |          1 |
> > | vowels       |    0.2871 |         12 |
> > | waveform     |    1      |         13 |
> > | wbc          |    0.5446 |          4 |
> > | wdbc         |    0.9307 |          2 |
> > | wilt         |    0.4653 |          1 |
> > | wine         |    1      |          4 |
> > | wpbc         |    0.9703 |          8 |
> >
> > **Table 2**:  p<0.05 and average performance rank>=15
> > | dataset      |   p_value |   avg-rank |
> > |:-------------|----------:|-----------:|
> > | backdoor     |    0.0099 |         17 |
> > | campaign     |    0.0099 |         25 |
> > | census       |    0.0099 |         20 |
> > | fraud        |    0.0099 |         18 |
> > | internetads  |    0.0099 |         25 |
> > | pageblocks   |    0.0198 |         19 |
> > | satimage-2   |    0.0297 |         20 |
> > | skin         |    0.0099 |         23 |
> > | spambase     |    0.0099 |         18 |
> > | MNIST-C      |    0.0495 |         19 |
> > | FashionMNIST |    0.0297 |         18 |
> >
> > **Table 3**:  p<0.05 and average performance rank<15
> > | dataset          |   p_value |   avg-rank |
> > |:-----------------|----------:|-----------:|
> > | aloi             |    0.0099 |        2   |
> > | breastw          |    0.0396 |       10   |
> > | cardio           |    0.0099 |        6   |
> > | cardiotocography |    0.0099 |        8   |
> > | celeba           |    0.0099 |       13   |
> > | cover            |    0.0099 |        3   |
> > | donors           |    0.0099 |        1   |
> > | imdb             |    0.0396 |       10   |
> > | landsat          |    0.0297 |        2   |
> > | letter           |    0.0396 |        9   |
> > | magic.gamma      |    0.0495 |        5   |
> > | mnist            |    0.0099 |        7   |
> > | optdigits        |    0.0099 |       12   |
> > | pendigits        |    0.0099 |        8.5 |
> > | satellite        |    0.0099 |        4   |
> > | shuttle          |    0.0099 |       12   |
> > | yeast            |    0.0099 |        4   |
> > | yelp             |    0.0198 |       13   |

---

> > > ### Author Response · Authors · 2024-11-21
> > > **Part 3**
> > >
> > > ### - **zero-shot OD on ImageNet-level benchmarks**
> > >
> > > First we’d like to carefully emphasize the difference between OD vs OOD: OD is typically given a **single**, fully unlabeled dataset with the goal of detecting outliers that do not “fit in” with the inliers, while OOD is provided with **multiple**, labeled (or known) K classes and the goal is to detect novel classes K+1, K+2, etc.  As such, OOD baselines are not directly comparable to FoMo-0D.
> > >
> > > Nevertheless, we can obtain an OD dataset from an OOD dataset by treating all the samples from K classes as inliers (without using class label information).
> > >
> > > From OpenOOD v1.5 [1], we use 2 datasets, ImageNet200 and ImageNet1K, as the source datasets for inliers. We then create outliers from 5 OOD datasets, 1. ssh-hard, 2. ninco, 3. inaturalist, 4. textures, 5. openimageo. Overall, we construct 2x5=10 OD datasets in total from ImageNet-level OOD benchmarks.
> > >
> > > Following [2], we create OD datasets for CV tasks containing 10,000 samples with 5% outliers, and use the embedding after the last average pooling layer of ResNet18 [3] as the feature (512) for each sample. Results are shown below, comparing FoMo-0D with the top-4 (on our original testbed) baselines in order of: DTE-NP, kNN, ICL, DTE-C. We follow [4] and report mean (standard dev.) over 5 runs (seed=0/1/2/3/4) on each dataset. We use bold (**text**) to highlight the best model, and italics (_text_)  to denote the second-best model, and discuss the detailed results and average performance on AUROC in the following.
> > >
> > > **Table 1**: AUROC performance on OOD datasets using ImageNet200.
> > > | Dataset      | DTE-NP       | KNN          | ICL              | DTE-C             | FoMo-0D          |
> > > |--------------|--------------|--------------|------------------|-------------------|------------------|
> > > | ssb-hard     | 58.03 (0.00) | 58.14 (0.00) | _60.52 (0.25)_   | **60.74 (1.88)**  | 58.34 (1.55)     |
> > > | ninco        | 53.28 (0.00) | 54.14 (0.00) | **59.56 (0.63)** | _58.83 (1.54)_    | 55.16 (2.19)     |
> > > | inaturalist  | 29.38 (0.00) | 29.51 (0.00) | _35.96 (1.10)_   | **41.77 (2.84)**  | 38.85 (3.29)     |
> > > | textures     | 59.28 (0.00) | 59.91 (0.00) | _66.40 (0.69)_   | **70.33 (3.18)**  | 59.89 (2.07)     |
> > > | openimageo   | 52.82 (0.00) | 53.79 (0.00) | _55.20 (0.69)_   | **59.09 (1.50)**  | 54.77 (1.19)  |
> > > | **Average**  | 50.56        | 51.10        |  55.53          |  58.15         | 53.40            |
> > >
> > >
> > > **Table 2**: AUROC performance on OOD datasets using ImageNet1K.
> > > | Dataset      | DTE-NP       | KNN          | ICL              | DTE-C             | FoMo-0D          |
> > > |--------------|--------------|--------------|------------------|-------------------|------------------|
> > > | ssb-hard     | 55.63 (0.00) | 55.94 (0.00) | _58.79 (1.20)_   | **59.17 (1.82)**  | 56.73 (2.65)     |
> > > | ninco        | 48.23 (0.00) | 49.10 (0.00) | _55.25 (0.87)_   | **57.60 (3.93)**  | 52.70 (2.70)     |
> > > | inaturalist  | 30.24 (0.00) | 30.28 (0.00) | 35.03 (1.42)   | **41.96 (3.13)**  | _38.94 (4.59)_     |
> > > | textures     | 54.38 (0.00) | 55.43 (0.00) | _61.30 (0.95)_   | **63.10 (3.72)**  | 55.18 (2.92)     |
> > > | openimageo   | 54.31 (0.00) | 54.91 (0.00) | 54.02 (0.43)     | **58.71 (2.08)**  | _56.95 (3.89)_  |
> > > | **Average**  | 48.56        | 49.13        | 52.88          |  56.11         | 52.10            |
> > >
> > > We also report the p-value of the Wilcoxon sign test between the baselines and FoMo-0D on these 10 datasets (top), as well as on an expanded testbed combining those 10 with our original ADBench (10+57) (bottom) as follows.
> > >
> > > | method   | DTE-NP | KNN | ICL | DTE-C  |
> > > |----------|----------|------------|---------|------------|
> > > | OpenOOD  | 1        | 0.9951 | 0.1875  | 0.0009 |
> > > | OpenOOD+ADBench | 0.1271 | 0.3308 | 0.3153 | 0.1265 |
> > >
> > > In terms of metric values, FoMo-0D performs 2nd or 3rd best across OOD datasets. P-values show that it significantly outperforms DTE-NP and kNN (p>0.95) and is no different from ICL (2nd best after DTE-C). These results demonstrate that FoMo-0D maintains strong zero-shot OD performance on image-level OOD benchmarks.
> > >
> > > Interestingly, we observe that ICL and DTE-C outperform DTE-NP and kNN on the OpenOOD datasets, whereas on ADBench, DTE-NP and kNN are the top-2 methods outperforming ICL and DTE-C (likely because it is harder for non-parametric methods like DTE-NP and kNN to estimate meaningful decision boundaries in high dimensions). In contrast, FoMo-0D performance is consistently competitive. In fact, p-values on the combined testbed (OpenOOD+ADBench) show that FoMo-0D performance is as competitive as all these top baselines right out of the box (i.e. zero-shot) across 67 diverse datasets.
> > >
> > > [1] OpenOOD v1.5: Enhanced Benchmark for Out-of-Distribution Detection, NeurIPS 2023 Workshop on Distribution Shifts.
> > >
> > > [2] Adbench: Anomaly detection benchmark. NeurIPS 2022.
> > >
> > > [3] Deep residual learning for image recognition. CVPR 2016.
> > >
> > > [4] On Diffusion Modeling for Anomaly Detection. ICLR 2024.

---

> > > > ### Author Response · Authors · 2024-11-21
> > > >
> > > > We hope that we addressed your comments and questions clearly. If you feel more confident with these additional results and explanations, please help us in achieving a better score and confidence. Thank you!

---

> > > > > ### Comment · Reviewer_VxxK · 2024-11-26
> > > > >
> > > > > Thank you for the detailed responses, which have addressed my concerns. I would like to raise my confidence score to 3 and maintain my current rating.

---

> > > > > > ### Author Response · Authors · 2024-12-02
> > > > > >
> > > > > > Thank you for your extended support!

---

### Official Review · Reviewer_UDeE · 2024-11-04

**Soundness:** 2
**Presentation:** 3
**Contribution:** 3
**Rating:** 6
**Confidence:** 3

**Summary:**

This paper introduces FoMo-0D, a novel zero-shot approach for outlier detection (OD) that bypasses the traditional model selection process, a major obstacle in unsupervised OD tasks. Model selection, including algorithm and hyperparameter tuning, has long been a challenge due to the unsupervised nature of OD. FoMo-0D leverages Prior-data Fitted Networks (PFNs), a Transformer-based model trained on a large synthetic dataset from a prior distribution, allowing it to make direct outlier predictions on new datasets without further tuning or training. As a foundation model for zero-shot OD on tabular data, FoMo-0D delivers outlier predictions with a single forward pass, eliminating the need for manual algorithm selection or hyperparameter adjustments. The authors of the paper conduct extensive experiments to demonstrate the effectiveness of the proposed method.

**Strengths:**

- The authors of the paper tackles an important yet very challenging problem of unsupervised anomaly detection.
- The proposed method is technically sound.
- The paper is well written and easy to follow.
- The authors of the paper conduct ample experiments to demonstrate the effectiveness of the proposed method.

**Weaknesses:**

- Despite the optimization done to scale the proposed method up, it is still not immediately clear to me how scalable the proposed method is. Does the proposed method still work if the training data contains millions of samples? Moreover, what is the inference latency and memory overhead? Would the proposed method incur very heavy computational burden if I have to handle a large number of samples during inference? Such a scenarios is arguably common in real world anomaly detection settings.
- How limiting is the use of GMM for synthetic data generation for pre-training. Does such a method scale to more complex datasets?

**Questions:**

- I wonder if it would be possible to do some sort of finetuning when a training data comes in, so that we don't need to feed in the entire training data into the model?

---

> ### Author Response · Authors · 2024-11-21
>
> ### - **scalability of the method, inference latency & memory overhead**
>
> FoMo-0D **scales linearly** with respect to its context size $n$ in both pretraining and inference – thanks to the router mechanism we employed for self-attention, that takes $O(Rn)$ instead of $O(n^2)$ where $R$ is the number of routers. Thus, complexity is effectively $O(n)$ (i.e. linear) for a small/constant number of routers (we used $R$=$500$ in our experiments).
>
> As our paper reports (please see Table 3), **inference latency** is $7.7 ms$ (milli-second) per test sample on average using a context size of $n$=$5K$. As for **memory overhead**, FoMo-0D requires less than $1000 MiB$ GPU memory; please see line 1089, Figure 8 (middle) in Section E.2 Effect of Routers on Cost in the Appendix.
>
> To speed up inference for massive datasets with very large $n$, dataset distillation can be employed as presented in [1]. Here, prior to inference, we’d freeze the pretrained FoMo-0D and distill a given dataset by learning a set of “distilled” input tokens, which produce similar predictions for all training samples when fed to the frozen FoMo-0D, i.e. loss is based on ALL training points (Please see Figure 2(a) in [1]). In our case, training points in D_train are all inliers with label 0, thus dataset distillation in effect would estimate a small number of representative inlier distribution tokens to be used as (now smaller) context.
> Our work’s main focus has been to design a zero-shot OD foundation model, and we agree that scaling improvements would constitute valuable future work.
>
> ### - **GMM for synthetic data generation**
>
> GMMs are relatively simple, yet they can capture increasingly complex distributions with an increasing number of components. We found them to be surprisingly effective as synthetic data prior. Our work left as future work exploring other, more complex data priors, for both inliers and outliers, since GMMs sufficed to achieve SOTA results against numerous baselines in our benchmark testbed.
>
> For scalability to complex input datasets during _inference_, we tested FoMo-0D on 57 **real-world** datasets, including complex, high-dimensional ones, such as 5 **vision/image** datasets (d=512) and 5 **NLP/text** datasets (d=768). The results (Section G Full Results in Appendix) show that FoMo-0D performs competitively on these datasets, despite being pre-trained on relatively smaller dimensional (d=100) prior data.
>
> ### - **finetuning on training data**
>
> This is a good direction to explore to speed up inference in terms of absolute time. While FoMo-0D scales linearly w.r.t. context size (e.g. if context/dataset size doubles, inference time also doubles) (as discussed above), one may still aim to keep the context as small as possible for speedy inference in terms of absolute wall-clock time.
>
> One direction in these lines is dataset distillation (see [1] above): Having completed pre-training, and given a new dataset {(inlier-only) $D_{train}$, (unlabeled) $D_{test}$}, one can learn a small number of “distilled” dataset tokens based on $D_{train}$, toward utilizing a more compact context for test samples in $D_{test}$. Note that this would still utilize in-context learning, but now with a smaller “distilled” context with reduced inference latency.
>
> In the traditional sense of the term fine-tuning, on the other hand, one may consider parameter fine-tuning on labels, but the only labels we have at inference are the inliers in $D_{train}$. That is, FoMo-0D parameters can be fine-tuned to accurately predict label 0 for all the inliers in $D_{train}$, which in effect could serve as better attuning the model to the given inlier distribution prior to inference.  Thus, fine-tuning may help improve performance, as it implicitly aligns (“distills” or “bakes in”) the given dataset to the model parameters. However, note that it does not reduce context size and hence would not help with inference latency.
>
> More inference-time work could help offload some of the learning on a new dataset from fully-in-context to fine-tuning, and in doing so, may reduce the model’s reliance on a large context size input. Of course, these approaches have different trade-offs not only w.r.t. inference time but also access to model parameters. Our work has focused on zero-shot generalization of synthetically-pretrained Transformer models. It would be valuable future work to design FoMo-0D variants with varying trade-offs that could serve applications with different needs. Specifically, more work/compute/learning can indeed be dedicated to inference time with the aforementioned approaches (or other future work) to speed up FoMo-0D further.
>
> [1] In-Context Data Distillation with TabPFN. Junwei Ma, Valentin Thomas, Guangwei Yu, Anthony L. Caterini. CoRR abs/2402.06971 (2024)

---

> > ### Author Response · Authors · 2024-11-21
> >
> > We hope that we answered your question clearly, and that you feel more confident to support our paper in achieving a better score and confidence. We also look forward to your further feedback, if any. Thank you!

---

> ### Author Response · Authors · 2024-12-02
>
> Thank you for your support of our work. Please let us know if you have any further inquiries about our paper and/or rebuttal.

---

### Meta-Review · Area_Chair_jcpN · 2024-12-18

**Metareview:**

This paper introduces FoMo-0D, a novel zero-shot approach for outlier detection (OD) that bypasses the traditional model selection process, a major obstacle in unsupervised OD tasks. Model selection, including algorithm and hyperparameter tuning, has long been a challenge due to the unsupervised nature of OD. FoMo-0D leverages Prior-data Fitted Networks (PFNs), a Transformer-based model trained on a large synthetic dataset from a prior distribution, allowing it to make direct outlier predictions on new datasets without further tuning or training. As a foundation model for zero-shot OD on tabular data, FoMo-0D delivers outlier predictions with a single forward pass, eliminating the need for manual algorithm selection or hyperparameter adjustments. The authors of the paper conduct extensive experiments to demonstrate the effectiveness of the proposed method. However, there are several points to be further improved. For example, the baselines compared in this paper are not competitive enough. There are much more advanced OD algorithms that are unsupervised or semi-supervised and do not require retraining on the new test set. There are plenty of strong post-hoc algorithms that can be efficient and effective at the same time. The inlier piror and outlier synthesis idea is explored. Could the authors clarify the differences? I did not see any significant technical differences from the prior selection and the synthesis approach, even though there are naunces in model architecture and the focused problem. Is it possible for the authors to show the actual detection metrics rather than the rank so that it makes the readers easy to see the algorithm performance? Therefore, this paper cannot be accepted at ICLR this time, but the enhanced version is highly encouraged to submit other top-tier venues.

**Additional Comments On Reviewer Discussion:**

Reviewers keep the score after the rebuttal.

---

### Decision · Program_Chairs · 2025-01-22

Reject